# The general version of Hamilton's rule

**Matthijs van Veelen***

Department of Economics, Universiteit van Amsterdam, Amsterdam, Netherlands

## eLife Assessment

Kin selection and inclusive fitness have generated significant controversy. This paper reconsiders the general form of Hamilton's rule in which benefits and costs are defined as regression coefficients, with higher-order coefficients being added to accommodate non-linear interactions. The paper is a **landmark** contribution to the field with **compelling**, systematic analysis, giving clarity to long-standing debates.

**Abstract** The generality of Hamilton's rule is much debated. In this paper, I show that this debate can be resolved by constructing a general version of Hamilton's rule, which allows for a large variety of ways in which the fitness of an individual can depend on the social behavior of oneself and of others. For this, I first derive the Generalized Price equation, which reconnects the Price equation with the statistics it borrows its terminology from. The Generalized Price equation, moreover, shows that there is not just one Price equation, but there is a Price-like equation for every possible true model. This implies that there are also multiple, nested rules to describe selection. The simplest rule is the rule for selection of non-social traits with linear fitness effects. This rule is nested in the classical version of Hamilton's rule, for which there is consensus that it works for social traits with linear, independent fitness effects. The classical version of Hamilton's rule, in turn, is nested in more general rules that, for instance, allow for nonlinear and/or interdependent fitness effects, like Queller's rule. The general version of Hamilton's rule, therefore, is a constructive solution that allows us to accurately describe when costly cooperation evolves in a wide variety of circumstances. As a byproduct, we also find a hierarchy of nested rules for selection of non-social traits.

**\*For correspondence:**
c.m.vanveelen@uva.nl

**Competing interest:** The author declares that no competing interests exist.

## Introduction

Hamilton's rule (**Hamilton, 1964a**; **Hamilton, 1964b**) is one of the more famous rules in evolutionary biology. The rule states that altruism will evolve if $rb > c$, where $b$ is the fitness benefit to the recipient, $c$ is the fitness cost to the donor, and $r$ is the genetic relatedness between them. The generality of Hamilton's rule, however, has always been a topic of contention (**Karlin and Matessi, 1983**; **Matessi and Karlin, 1984**), and positions range all the way from '*Hamilton's rule almost never holds*' (**Nowak et al., 2010**) to '*Inclusive fitness is as general as the genetical theory of natural selection itself*' (**Abbot et al., 2011**).

The claim of generality is not based on the original derivation of Hamilton's rule (**Hamilton, 1964a**; **van Veelen, 2007**; **van Veelen et al., 2017**), but on later derivations (**Hamilton, 1970**; **Hamilton, 1975**; **Grafen, 1985a**; **Taylor, 1989**; **Taylor and Frank, 1996**; **Grafen, 2006**; **Gardner et al., 2011**; **Marshall, 2015**; **Rousset, 2015**) that use the Price equation (**Price, 1970**). This tradition also started with Hamilton himself (**Hamilton, 1970**; **Hamilton, 1975**). The usefulness of the Price equation as a tool for doing theory, however, is not undisputed (**van Veelen, 2005**; **van Veelen et al., 2012**; **Frank, 2012**; **Allen et al., 2013**; **Nowak et al., 2017**; **van Veelen, 2020b**). One of the remarkable characteristics of the Price equation literature is that it borrows terms from statistics, like regression coefficients,

without also inheriting the natural preoccupations of statistics, such as worrying about model choice or statistical significance.

In this paper, I will show that this debate can be resolved by deriving general versions of the Price equation and of Hamilton's rule. The Generalized Price equation in regression form generates a set of Price-like equations; one for every different choice for a statistical model that may describe how the fitness of an individual depends on its genetic makeup. This makes the original Price equation in regression form a special case; it is the Generalized Price equation in regression form, combined with a linear model. The Generalized Price equation repairs the broken link between the Price equation and statistics; when applied to data, standard statistical considerations concerning model choice now translate one-to-one to considerations concerning which of these Price-like equations describes the population genetic dynamics. I will also consider the application of the Generalized Price equation in the modeling domain, and it will be helpful to always discuss the application to data and to modeling separately.

With the Generalized Price equation, I then derive the general version of Hamilton's rule. Both the limitations of the original Price equation and ways to overcome these limitations by using the generalized version are mirrored in the limitations of Hamilton's rule as we know it and ways to overcome those. Just like there is not just one Price equation, but a multitude of Price-like equations, there is not just one Hamilton's rule, but a multitude of Hamilton-like rules. All of them are correct, and all of them are general, but none of them is generally meaningful. A specific Hamilton's rule, although always correct, is only meaningful if based on the Generalized Price equation, in combination with a model that is appropriately specified for the evolutionary system under study.

The Generalized Price equation puts the debate concerning the validity of Hamilton's rule in perspective by showing that some arguments are not as decisive as they are currently perceived to be. The side of the debate that claims full generality has repeatedly stressed that Hamilton's rule, derived with the Price equation, is an identity that holds generally (*Abbot et al., 2011*; *Gardner et al., 2011*; *Marshall, 2015*; *Rousset, 2015*) (i.e. it holds, whatever the parent and the offspring generations are). This is correct, but the general version of Hamilton's rule shows that being an identity that holds generally cannot be the only relevant characteristic for regarding it as the proper, the right, or the most insightful rule that describes evolution. The reason why this cannot be is that all Price-like equations generated by the Generalized Price equation, and thereby all Hamilton-like rules produced by them, have those properties; they are all identities that hold with exactly the same generality. To decide between the different Hamilton-like rules, we therefore need additional criteria. When doing empirics, we then have to resort to classical statistics to see which model agrees with the data. With sufficiently many data, this will then point to a statistical model, and by doing so, it will also point to one of these Hamilton-like rules. An indication that the approach suggested in this paper does indeed pick a Hamilton-like rule that is well-specified, both with modeling and with empirics, is that the quantities in it that we treat as constants are in fact constant, and that they do not change with the composition of the parent population.

## Results

### The Generalized Price equation in combination with models for non-social traits

I would like to start with describing how generalizing the Price equation works. By applying this to linear and non-linear fitness functions in a non-social context, I would moreover like to illustrate that for the Price-like equations that this generates, and that are meant to describe population genetic dynamics, there is scope for under- and for overspecification in the same way that there is scope for under- and for overspecification in statistics.

Assume that we track a set of genes, summarized by a *p*-score (*Grafen, 1985a*). This *p*-score is a value between 0 and 1. In the simplest possible haploid setup, this reflects the presence or absence of a single gene, in which case an individual's *p*-score would be 1 if the gene is present, and 0 if it is absent. In less simple setups, the *p*-score can, for instance, reflect a set of genes that affect the same trait. The change in average *p*-score between the parent and the offspring generation can then be written as a sum of two terms, in what Price called the 'covariance form' of his equation (*Price, 1970*).

$$\bar{w}\Delta\bar{p} = \mathrm{Cov}\left(w,p\right) + E\left(w\Delta p\right) \tag{PE.C}$$

Here, $\bar{w}$ is the average fitness in the parent generation; $\Delta\bar{p}$ is the change in average $p$-score between parent and offspring generation; $\mathrm{Cov}\left(w,p\right)$ is $\frac{n-1}{n}$ times the sample covariance between fitness and $p$-score (or, in other words, the sample covariance without Bessel's correction; see Section 2 of Appendix 1 for details about the terminology); and $E\left(w\Delta p\right)$ is the fitness-weighted average difference between the $p$-scores of the parents, and, in an asexual model, the $p$-scores of their offspring, or, in a model with sexual reproduction, the $p$-scores of their successful gametes (all details are in Appendix 1).

In order to get to the Generalized Price equation in covariance form, we assume a statistical model that has fitness as a dependent variable, and that contains, at least, (1) a constant term and (2) a linear term for the $p$-score as an explanatory variable. Besides those two terms, the statistical model may contain any number of other terms as well. This statistical model is then combined with the transition we want to apply the Generalized Price equation to. A transition implies that we know what the $p$-score is for all individuals in the parent population, and that we know how many offspring all individuals have, and therefore what their realized fitness is (this is the number of offspring divided by the ploidy). Combining a transition with a statistical model here means that we choose the parameters of the model so that they minimize the sum of squared differences between the model-predicted fitness $\hat{w}_i$ and the realized fitnesses $w_i$, for individuals $i = 1,\ldots,n$. For any model that does indeed include a constant term and a linear term for the $p$-score, we can then replace the realized fitnesses $w_i$ in the Price equation with the estimated fitnesses $\hat{w}_i$ according to the model. This leads to the Generalized Price equation in covariance form:

$$\bar{w}\Delta\bar{p} = \mathrm{Cov}\left(\hat{w},p\right) + E\left(w\Delta p\right) \tag{GPE.C}$$

The algebra to show that this is an identity is in Appendix 1. While replacing $w$ with $\hat{w}$ seems like a minimal change, the important thing to keep in mind is that this equation is an identity for *any* model that contains a constant and a linear term for the $p$-score. In other words, for any given transition (i.e. for any combination of a parent and an offspring population), the model-predicted fitnesses $\hat{w}_i$ that result from minimizing the sum of squared differences will typically differ, depending on what the model is, but $\mathrm{Cov}\left(\hat{w},p\right)$ equals the sample covariance $\mathrm{Cov}\left(w,p\right)$ for all of these models.

In order to get to what Price calls the 'regression form' of the equation, we now choose a set of models for non-social traits, all of which contain a constant and a linear term (which makes it satisfy the condition for the Generalized Price equation to hold). The set of models we choose here is given by

$$w_i = \alpha + \sum_{r=1}^{R} \beta_r p_i^r + \varepsilon_i$$

where $w_i$ is the number of offspring of individual $i$; $p_i$ is the $p$-score of individual $i$; $\alpha$ is a constant term; $\beta_1,\ldots\beta_R$ are the other coefficients of the model; $\varepsilon_i$ is the error term for individual $i$; and where we assume that $R \geq 1$. This gives a different model for every $R$; a linear model for $R = 1$; a quadratic one for $R = 2$; and so on. If we now minimize the sum of squared differences between the model-predicted fitnesses $\alpha + \sum_{r=1}^{R} \beta_r p_i^r$ (the right-hand side of the equation above, but without the error term) and the realized fitnesses (on the left-hand side of the same equation), then we arrive at values $\hat{\beta}_1,\ldots,\hat{\beta}_R$ for the regression coefficients.

In Appendix 1, I show that, for every $R \geq 1$, and thereby for every model in this set, we can rewrite the Generalized Price equation as follows:

$$\bar{w}\Delta\bar{p} = \sum_{r=1}^{R} \hat{\beta}_r \mathrm{Cov}\left(p,p^r\right) + E\left(w\Delta p\right) \tag{GPE.R}$$

In line with Price's terminology (**Price, 1970**), this would be the 'regression form' of the Generalized Price equation. If we take the linear model ($R = 1$), then, using $\mathrm{Cov}\left(p,p\right) = \mathrm{Var}\left(p\right)$, the above equation becomes

$$\bar{w}\Delta\bar{p} = \hat{\beta}_1 \mathrm{Var}\left(p\right) + E\left(w\Delta p\right)$$

Below we will refer to the linear model as Model A. If instead we take the quadratic model ($R = 2$), the equation becomes

$$\bar{w}\Delta\bar{p} = \hat{\beta}_1 \text{Var}\left(p\right) + \hat{\beta}_2 \text{Cov}\left(p, p^2\right) + E\left(w\Delta p\right)$$

Below we will refer to the quadratic model as Model B. Again, it is important to remember that both of these equations are identities for every possible transition from a parent to an offspring generation. It is also important to recognize that the $\hat{\beta}_1$ that we arrive at if we combine Model A with a given transition, and the $\hat{\beta}_1$ that we arrive at if we use Model B for the same transition, may, and generically will, differ, as we will see.

## The Generalized Price equation applied to two artificial datasets

The following example illustrates how the flexibility of the Generalized Price equation in regression form comes with scope for under- and overspecification. The two Price-like equations above – the one with Model A as a reference model, and the one with Model B as a reference model – are combined with two artificial datasets; one where the offspring generation is generated by Model A; and one where the offspring generation is generated by Model B. Because both equations are identities for all possible transitions between parent and offspring generation, they are also both identities for both of these datasets. Depending on the system under study, the terms in these equations, however, are not equally meaningful as a description of the population genetic dynamics.

The Generalized Price equation in regression form for Model A, in combination with the offspring generation generated by Model A, returns a value for $\hat{\beta}_1$ that is very close to the true $\beta_1$; $\hat{\beta}_1 = 0.982$, while $\beta_1 = 1$. This is depicted in *Figure 1a*. All details are in Detailed Calculations 1.6 at the end of Appendix 1.

The Generalized Price equation in regression form for Model B, in combination with the offspring generation generated by Model B, returns values for $\hat{\beta}_1$ and $\hat{\beta}_2$ that are also close to their true values; $\hat{\beta}_1 = 0.040$, while $\beta_1 = 0$; and $\hat{\beta}_2 = 0.944$, while $\beta_2 = 1$. This is depicted in *Figure 1d*.

If we combine the Generalized Price equation in regression form for Model B with the offspring generation generated by Model A, then this returns values for $\hat{\beta}_1$ and $\hat{\beta}_2$ that are also close to their true values; $\hat{\beta}_1 = 0.998$, while $\beta_1 = 1$; and $\hat{\beta}_2 = -0.016$, while $\beta_2 = 0$. This is depicted in *Figure 1b*. Appropriate statistical tests reject that the true $\beta_2$ is different from 0, but the harm done by using the Generalized Price equation in regression form for Model B – which is overspecified for data generated by Model A, because it includes a parameter for $\hat{\beta}_2$, which is really 0 – is therefore limited; the estimated coefficient for $\beta_2$ is very close to 0.

The Generalized Price equation in regression form for Model A, in combination with the offspring generation generated by Model B, however, does not return values for $\hat{\beta}_1$ and $\hat{\beta}_2$ that are close to their true values. For $\hat{\beta}_2$ that is obvious, because the statistical model specification sets it to 0, while $\beta_2 = 1$. Moreover, $\hat{\beta}_1 = 0.984$, and this is also rather different from the true $\beta_1$, which is 0. These differences are the result of underspecification. What is important to note here is that none of this keeps the Generalized Price equation in regression form for Model A from being an identity, also for data that are really generated by Model B. Being an identity, therefore, does not guarantee that the regression coefficients in it have a meaningful interpretation; in this case, the $\hat{\beta}_1$ is *not* the linear effect of the $p$-score on fitness.

This also illustrates a general property of the Generalized Price equation. If the data are generated by the model that matches the model that we combine the Generalized Price equation with, then the $\hat{\beta}_1, \ldots, \hat{\beta}_R$ that feature in the regression form are unbiased estimators of the true $\beta_1, \ldots, \beta_R$. This is exemplified by panel a, where the Generalized Price equation for Model A is applied to an offspring generation that is really generated by Model A, and by panel d, where the Generalized Price equation for Model B is applied to an offspring generation that is really generated by Model B. If, on the other hand, the model that we combine the Generalized Price equation with is underspecified for the underlying model that generated the data – as it is in *Figure 1b* – then the regression coefficients are not unbiased estimators of any parameter of the true model. As a symptom of this, if the offspring population is indeed generated by a richer model, then the expected values of the regression coefficients $\hat{\beta}_1, \ldots, \hat{\beta}_R$ (i.e. what values for $\hat{\beta}_1, \ldots, \hat{\beta}_R$ we will find, on average, if we repeatedly draw an offspring population according to the true model) will depend on the composition of the parent population.

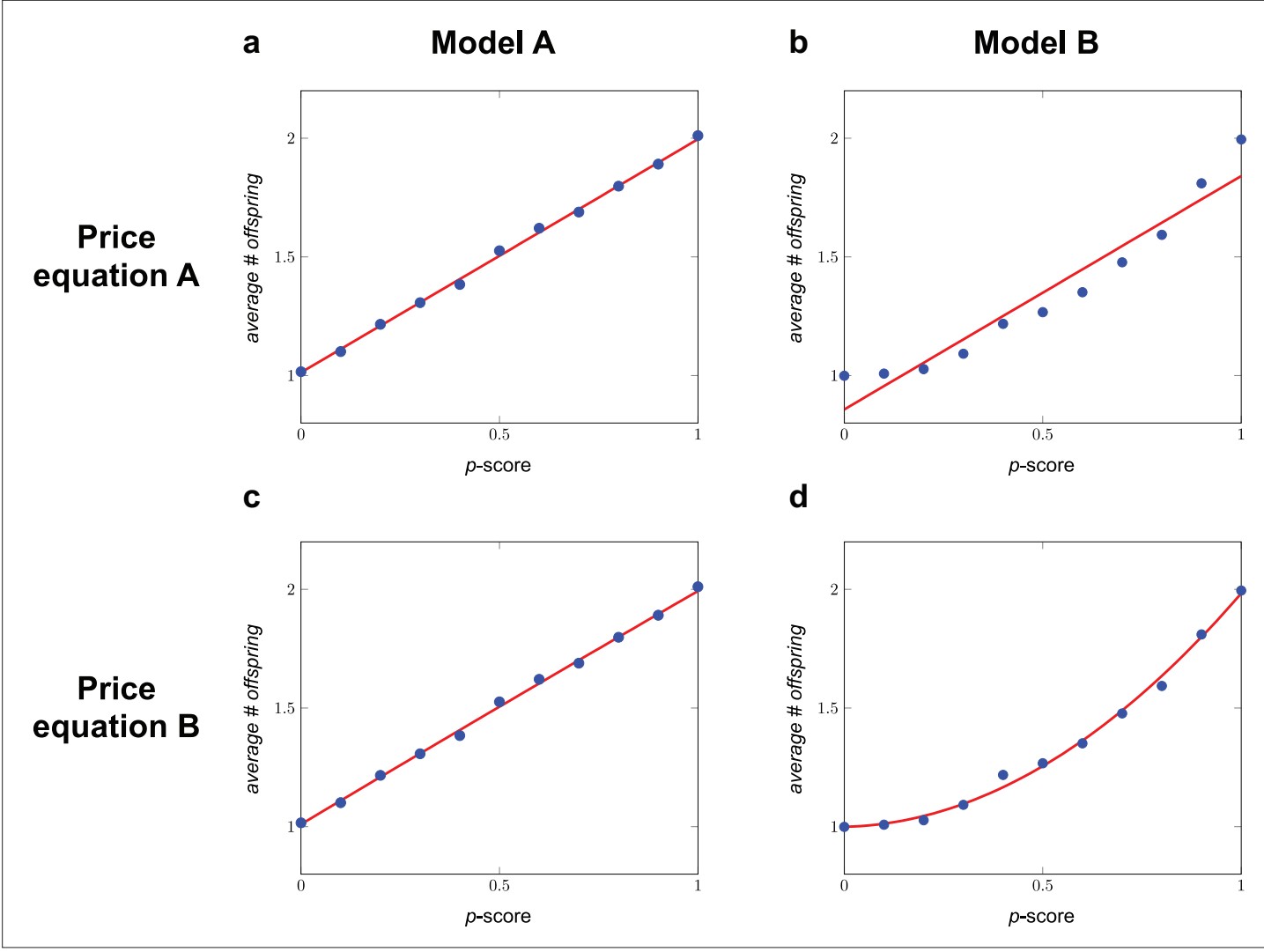

**Figure 1.** Two Price-like equations and two models. In Model A, the number of offspring follows a binomial distribution with an expected number of offspring of $\alpha + \beta_1 p_i$. This means that the model can be summarized as $w_i = \alpha + \beta_1 p_i + \varepsilon_i$ (see Detailed Calculations 1.6 at the end of Appendix 1 for details). For the transition depicted in panels **a** and **c**, we generated an offspring generation using Model A, with $\alpha = 1$ and $\beta_1 = 1$. Combining the Generalized Price equation in regression form with Model A includes choosing $\hat{\alpha}$ and $\hat{\beta}_1$ so that they minimize the sum of squared differences between $w_i$ and $\hat{\alpha} + \hat{\beta}_1 p_i$. In Model B, the number of offspring follows a binomial distribution with an expected number of offspring of $\alpha + \beta_1 p_i + \beta_2 p_i^2$, which means that $w_i = \alpha + \beta_1 p_i + \beta_2 p_i^2 + \varepsilon_i$. For the transition depicted in panels **b** and **d**, we generated an offspring generation using Model B, with $\alpha = 1$, $\beta_1 = 0$, and $\beta_2 = 1$. Combining the Generalized Price equation in regression form with Model B includes choosing $\hat{\alpha}$, $\hat{\beta}_1$, and $\hat{\beta}_2$ so that they minimize the sum of squared differences between $w_i$ and $\hat{\alpha} + \hat{\beta}_1 p_i + \hat{\beta}_2 p_i^2$. Reproduction is asexual, so parents and their offspring are always identical, and $E(w\Delta p) = 0$ by definition. For both of these models, we started with a population consisting of 2500 parents for each $p$-score, ranging from 0 to 1 in increments of 0.1. The four panels represent the four combinations of the two Price-like equations and the two datasets. The red lines in all panels represent the estimated fitnesses as a function of the $p$-score, as implied by the respective Price-like equations. The aim of this example is not to show that the estimated fitnesses from Price-like equation A match the data generated by Model A, and the estimated fitnesses from Price-like equation B match the data generated by Model B, but the estimated fitnesses from Price equation A do not match the data generated by Model B; that part is obvious. The purpose of this example, instead, is to illustrate that both Price-like equations remain identities, also when they are overspecified (panel **c**) or underspecified (panel **b**) with respect to the transition between parent and offspring population they are applied to. With underspecification, it is also visually clear that the estimated fitnesses do *not* match the data, even though Price-like equation A remains an identity, also when combined with data generated by Model B.

The online version of this article includes the following source data for figure 1:

**Source data 1.** Simulation data generated by a simple computer algorithm.

Again, none of this keeps the Generalized Price equation for Model A from being an identity, also when applied to data generated by Model B.

The regression form of the original Price equation coincides with the regression form of the Generalized Price equation, combined with the linear model for $R = 1$ (Price equation A in **Figure 1**). This implies that limitations of the latter are shared with the original Price equation.

In the Price equation literature, the fact that regression coefficients may not be constant is conflated with, and inadequately labeled as, dynamic insufficiency. We will return to this point after introducing the Generalized Price equation in combination with social traits, but it is worthwhile already here to stress that regression coefficients depending on the composition of the parent population are not a sign of dynamic insufficiency, but rather an indication of misspecification.

## The Generalized Price equation applied in a modeling context

The next example shows that the problem of misspecification is not limited to applications of the Generalized Price equation in a statistical setting. Misspecification is also possible when the Generalized Price equation is applied to a theoretical model. Section 5 in Appendix 1 describes the conditions under which one can in fact apply the Generalized Price equation in a modeling setting, so here we will assume that those conditions are satisfied; we assume an infinitely large population, error terms with expected value 0, and, if reproduction is sexual, fair meiosis.

In a modeling setting, it seems almost trivial that there is room for misspecification, if we apply the Generalized Price equation for one model to transitions that follow a different model. It will, however, be useful to acknowledge that this possibility exists. Below, we will therefore consider a linear model, $w_i = \alpha + \beta_1 p_i + \varepsilon_i$, which we will again refer to as Model A, and a quadratic one, $w_i = \alpha + \beta_1 p_i + \beta_2 p_i^2 + \varepsilon_i$, which we will refer to as Model B. With infinite populations, such a model then implies that every parent population produces an offspring generation deterministically. If we minimize the squared differences in this transition, with respect to the model that generated the transition, we recover the coefficients of the model exactly (and not with some noise, as in the statistical example above). With two models to combine the Generalized Price equation with, and two models to apply these Price-like equations to, we again have four possible combinations.

As a preparation for moving on to generalizing Hamilton's rule, I will also point to the rules for selection that the Price-like equations produce.

## 1. The Generalized Price equation for Model A applied to transitions following Model A

The Generalized Price equation in regression form for Model A is

$$\bar{w}\Delta\bar{p} = \hat{\beta}_1 \mathrm{Var}\left(p\right)$$

If this equation is applied to a transition that does indeed follow Model A, then the $\hat{\beta}_1$ in this Price-like equation is equal to the model parameter $\beta_1$. Translating this equation into a criterion for when higher $p$-scores are selected for is straightforward: higher $p$-scores are selected for at all frequencies if and only if $\hat{\beta}_1 > 0$. With transitions that are generated by Model A, this means that higher $p$-scores are selected for if and only if $\beta_1 > 0$.

## 2. The Generalized Price equation for Model B applied to transitions following Model B

The Generalized Price equation in regression form for Model B is

$$\bar{w}\Delta\bar{p} = \hat{\beta}_1 \mathrm{Var}\left(p\right) + \hat{\beta}_2 \mathrm{Cov}\left(p, p^2\right)$$

If this equation is applied to a transition that does indeed follow Model B, $\hat{\beta}_1$ is equal to the model parameter $\beta_1$, and $\hat{\beta}_2$ is equal to the model parameter $\beta_2$. If we now translate this equation into a criterion for when higher $p$-scores are selected for, then this is still straightforward, but less concise:

$$\hat{\beta}_1 \mathrm{Var}\left(p\right) + \hat{\beta}_2 \mathrm{Cov}\left(p, p^2\right) > 0$$

This can be used to describe, for instance, the population genetic dynamics for cases with hetero-zygote advantage (see Example 3.2 in Appendix 1). This is a nice bycatch in the non-social domain of this endeavour, which was meant to solve a long-standing debate in the social domain. If we further-more assume random mating between generations (see Section 5 of Appendix 1), then this simplifies to

$$\hat{\beta}_1 + \hat{\beta}_2 \left( \frac{1}{2} + \bar{p} \right) > 0$$

where $\bar{p}$ is the average $p$-score in the parent population. With transitions that are indeed generated by Model B, this means that higher $p$-scores are selected for if and only if $\beta_1 + \beta_2 \left( \frac{1}{2} + \bar{p} \right) > 0$. The rule for the Generalized Price equation in regression form for Model A, applied to transitions that follow Model A, is nested in this rule here, in the sense that if we choose $\beta_2 = 0$, this rule reverts to $\beta_1 > 0$.

## 3. The Generalized Price equation for Model B applied to transitions following Model A

The Generalized Price equation in regression form for Model B is still

$$\bar{w}\Delta\bar{p} = \hat{\beta}_1 \text{Var}(p) + \hat{\beta}_2 \text{Cov}\left(p, p^2\right)$$

If this equation is applied to a transition that follows Model A, then $\hat{\beta}_1$ is equal to the model param-eter $\beta_1$, and $\hat{\beta}_2$ is equal to the model parameter $\beta_2$, which is 0. The overspecification, therefore, is inconsequential; we end up with the same equation that we arrived at when we applied the General-ized Price equation in regression form for Model A to a transition that follows Model A.

## 4. The Generalized Price equation for Model A applied to transitions following Model B

The Generalized Price equation in regression form for Model A is still

$$\bar{w}\Delta\bar{p} = \hat{\beta}_1 \text{Var}(p)$$

If applied to a transition that follows Model B, then this implies that $\hat{\beta}_1 = \beta_1 + \beta_2 \frac{\text{Cov}(p,p^2)}{\text{Var}(p)}$. What is important to realize is that now $\hat{\beta}_1$ is no longer a constant, as it depends on the population state through the $\frac{\text{Cov}(p,p^2)}{\text{Var}(p)}$-term. Also, it would be wrong to interpret $\hat{\beta}_1$ as the linear effect of the gene on fitness, because here $\hat{\beta}_1 \neq \beta_1$.

If we translate this equation into a criterion for when higher $p$-scores are selected for, then this would be the same rule as the one we arrived at when the Generalized Price equation in regression form for Model A was applied to Model A, which is that higher $p$-scores are selected for if $\hat{\beta}_1 > 0$. This is still a rule, and it gets the direction of selection right, but the $\hat{\beta}_1$ in it does not have a meaningful interpretation anymore. The fact that the regression coefficient $\hat{\beta}_1$ here changes with the population state moreover reflects misspecification, but now in a modeling context, and also here this does not reflect dynamic insufficiency.

### The general version of Hamilton's rule

For the derivation of the general version of Hamilton's rule, we need a richer setup, with two variables instead of one: a $p$-score and a $q$-score. The $p$-score represents a set of genes that code for some social behavior, and the $q$-score of an individual represents the $p$-score of its interaction partner. We assume that both can take values in $[0, 1]$. We will combine the Generalized Price equation in regres-sion form with a general set of models for social traits. This requires some formal notation.

Here, we choose a set of models, in which a model is defined by its non-zero coefficients. That is, a model is a choice for a set of non-zero coefficients $E$, where $(k, l) \in E$ if the term $p_i^k q_i^l$ in the model has a non-zero coefficient $\beta_{k,l}$. This then defines a model as follows:

$$w_i = \sum_{(k,l) \in E} \beta_{k,l} p_i^k q_i^l + \varepsilon_i$$

where $\varepsilon_i$ is a noise term with expected value 0. We assume that coefficients $\beta_{0,0}$ and $\beta_{1,0}$ are included, or, in other words, that the model contains a constant and a linear term for the $p$-score.

We can take any such model as a reference model for the Generalized Price equation in regression form. If we do, we find the following Price-like equation for the population genetic dynamics (see Section 4 of Appendix 1 for details).

$$\bar{w}\Delta\bar{p} = \left( \sum_{(k,l) \in E} \hat{\beta}_{k,l} \frac{\mathrm{Cov}\left(p, p^k q^l\right)}{\mathrm{Var}\left(p\right)} \right) \mathrm{Var}\left(p\right) + E\left(w\Delta p\right)$$

If we define $E'$ as the set of non-zero coefficients of the model, besides the constant term and the linear term for the $p$-score, then the generalized version of Hamilton's rule is that $\Delta\bar{p} > 0$ if and only if

$$\sum_{(k,l) \in E'} r_{k,l} b_{k,l} > c$$

where $c = -\beta_{1,0}$, $b_{k,l} = \beta_{k,l}$, and $r_{k,l} = \frac{\mathrm{Cov}\left(p, p^k q^l\right)}{\mathrm{Var}(p)}$, both for all $(k,l) \in E'$. The linear term for the $p$-score still features in this equation, because that is what $c$ represents. Details are in Section 3 of Appendix 2. In Section 5 of Appendix 2, this is generalized further to include interactions between more than two individuals.

The general version of Hamilton's rule produces a set of rules; one for every model of how traits affect fitnesses. Less general models, that are nested in more general ones, come with rules that are nested in more general rules. This includes rules for non-social traits. We get the rules for non-social traits if the set of model coefficients, besides a constant, only includes coefficients $\beta_{k,0}$, and no coefficients for terms that relate to the $q$-score. The simplest such rule is the linear one that we have seen above, and that we get if the true model only contains a constant and a term for the linear effect of the $p$-score. Above, we referred to this model as Model A. This rule is nested in the rule for the quadratic model, which has a constant, a term for the linear effect of the $p$-score, and a term for the quadratic effect of the $p$-score. We referred to this model as Model B above. This describes selection in case of, for instance, heterozygote advantage (see Appendix 1, Sections 3 and 5). This rule, in turn, is nested in more general rules for selection of non-social traits that include coefficients $\beta_{k,0}$ for $k > 2$.

If we then go back to the rule for non-social traits, but with linear fitness effects only, then this rule is also nested in the classical Hamilton's rule. The classical Hamilton's rule is the rule that we get for a model that includes a constant, a term for the linear effect of the $p$-score, and a term for the linear effect of the $q$-score. This rule, in turn, is nested in the rule that we get when an interaction effect between the $p$- and the $q$-score is added to the model. The classical Hamilton's rule therefore is itself nested in what is best referred to as Queller's rule (**Queller, 1985**). These rules, moreover, allow for further nesting, if the model we are interested in is more general, or if the data suggest that the true model includes additional non-zero coefficients. Below, we will, however, restrict attention to these three nested models in order to illustrate how they produce three rules that are nested, and how using the generalized version of Hamilton's rule allows us to see the debate on the generality of the classical Hamilton's rule in a new light.

## Three models and three rules

In Model 1, the $p$-score relates to a non-social trait that only affects the fitness of its carrier. We also referred to this model as Model A before, where we nested it in a different, also non-social, but more general model.

$$w_i = \alpha + \beta_{1,0} p_i + \varepsilon_i$$

In Model 2, the $p$-score relates to a social trait that affects the fitness of its carrier and the fitness of the partner of its carrier. In the fitness function, this is reflected by the effect of the $p$-score of the partner (the $q$-score) on individual $i$. These effects are moreover assumed to be independent; the effect on the partner is the same, irrespective of the $p$-score of the partner.

$$w_i = \alpha + \beta_{1,0}p_i + \beta_{0,1}q_i + \varepsilon_i$$

In Model 3, the $p$-score also relates to a social trait, but now the effects are not assumed to be independent; the effect on the partner may depend on the $p$-score of the partner. This is reflected by the interaction term.

$$w_i = \alpha + \beta_{1,0}p_i + \beta_{0,1}q_i + \beta_{1,1}p_iq_i + \varepsilon_i$$

## Model 1: selection of a non-social trait

The Generalized Price equation in regression form for Model 1 is

$$\bar{w}\Delta\bar{p} = \hat{\beta}_{1,0}\text{Var}\left(p\right)$$

The rule that this Price-like equation produces is that higher $p$-scores will be selected for if and only if $\hat{\beta}_{1,0} > 0$. This is also the rule that we found above, when we referred to Model 1 as Model A. If the model under consideration is indeed Model 1, then $\hat{\beta}_{1,0} = \beta_{1,0}$, both in the Price-like equation and in the rule for selection.

## Model 2: selection of a linear social trait

The Generalized Price equation in regression form for Model 2 is

$$\bar{w}\Delta\bar{p} = \left(\hat{\beta}_{1,0} + \frac{\text{Cov}\left(p,q\right)}{\text{Var}\left(p\right)}\hat{\beta}_{0,1}\right)\text{Var}\left(p\right)$$

From this, we can see that $\Delta\bar{p} > 0$ if and only if $\hat{\beta}_{1,0} + \frac{\text{Cov}(p,q)}{\text{Var}(p)}\hat{\beta}_{0,1} > 0$. If the model under consideration is indeed Model 2, then, with an infinitely large population, the regression coefficients $\hat{\beta}_{1,0}$ and $\hat{\beta}_{0,1}$ will coincide with the true values of the parameters $\beta_{1,0}$ and $\beta_{0,1}$, respectively, and $\frac{\text{Cov}(p,q)}{\text{Var}(p)}$ will coincide with relatedness $r$ between the individuals that interact. If we then define $c = -\beta_{1,0}$ and $b = \beta_{0,1}$, we can rewrite this as

$$\bar{w}\Delta\bar{p} = \left(-c + rb\right) \cdot \text{Var}\left(p\right)$$

Here, we naturally recognize the classical Hamilton's rule, because this implies that $\Delta\bar{p} > 0$ if and only if $rb > c$.

At this point, it is useful to observe that if this is a social trait, and the true model is Model 2, but we combine it with the Generalized Price equation in regression form for Model 1, the latter still gets the direction of selection right. In that case, we would still use

$$\bar{w}\Delta\bar{p} = \hat{\beta}_{1,0}\text{Var}\left(p\right)$$

but now with $\hat{\beta}_{1,0} = \beta_{1,0} + \frac{\text{Cov}(p,q)}{\text{Var}(p)}\beta_{0,1}$. This, however, is typically not how we describe the evolution of social traits. The reason why we do not describe the selection of a social trait with the rule for the non-social model is, however, *not* that this rule would get the direction of selection wrong; it would not. As a matter of fact, this rule always gets the direction of selection right too. The reason that here we use the classical version of Hamilton's rule instead is that the classical version of Hamilton's rule reveals the actual population genetic dynamics that govern selection; with a social trait for which $rb > c$, the reason why the genes for the trait are selected is not that the behavior they induce is good for the fitness of the individual that carries these genes itself; the reason is that the fitness costs of the associated behavior to the individual itself is outweighed by how much fitness benefits carriers get through related individuals, compared to how much such benefits non-carriers get. In a theory model, this is postulated. With data, and if the sample size is large enough, statistical tests would make the same determination, by pointing to Model 2, and not Model 1, as the true model, if it is in fact the true model.

At this point, we therefore have two Price-like equations: the one for Model 1 and the one for Model 2. Both of these Price-like equations are identities, and both of them are general. When applied to Model 2, however, the Price-like equation for Model 1 is not meaningful, in the sense that

the regression coefficient $\hat{\beta}_{1,0}$ does not reflect the effect of the behavior on the individual itself. As a symptom of this, it is not a constant and varies with the population state. The core point of the original derivation of Hamilton's rule, using the original Price equation, therefore, is to not use an underspecified model. It is useful to keep this in mind when we are faced with an analogous choice between Price-like equations and between rules based on them, below.

## Model 3: selection of a social trait with an interaction effect

The Generalized Price equation in regression form for the third model is

$$\bar{w}\Delta\bar{p} = \hat{\beta}_{1,0}\mathrm{Var}\left(p\right) + \hat{\beta}_{0,1}\mathrm{Cov}\left(p,q\right) + \hat{\beta}_{1,1}\mathrm{Cov}\left(p,pq\right)$$

If the model under consideration is indeed Model 3, and we assume an infinite population, the regression coefficients $\hat{\beta}_{1,0}$, $\hat{\beta}_{0,1}$, and $\hat{\beta}_{1,1}$ will coincide with the true values of the parameters $\beta_{1,0}$, $\beta_{0,1}$, and $\beta_{1,1}$, respectively. Also, in an infinite population model, $\frac{\mathrm{Cov}\left(p,q\right)}{\mathrm{Var}\left(p\right)}$ will coincide with relatedness $r$, which we call $r_{0,1}$ here, in order to distinguish it from $r_{1,1} = \left(1-r\right)\bar{p} + r$. The latter is what $\frac{\mathrm{Cov}\left(p,pq\right)}{\mathrm{Var}\left(p\right)}$ will be in an infinite population (see Appendix 2). If we then define $c = -\beta_{1,0}$, $b_{0,1} = \beta_{0,1}$, and $b_{1,1} = \beta_{1,1}$, we can rewrite the Generalized Price equation in regression form for the third model as

$$\bar{w}\Delta\bar{p} = \left(-c + r_{0,1}b_{0,1} + r_{1,1}b_{1,1}\right)\mathrm{Var}\left(p\right)$$

This does not give us the classical Hamilton's rule that we are familiar with. It does, however, give us a correct criterion for when higher $p$-scores are selected for; $\Delta\bar{p} > 0$ if and only if $r_{0,1}b_{0,1} + r_{1,1}b_{1,1} > c$. This criterion features two benefits and two relatednesses; besides the classical, linear benefit $b$, which we call $b_{0,1}$ here, it also has an interaction benefit $b_{1,1}$, that only fully materializes if both the individual itself and its partner have a $p$-score of 1. If $b_{1,1} = 0$, then we are back in Model 2, and this rule reverts to the classical Hamilton's rule. The rule with a non-zero interaction effect is the same as the rule suggested by Queller (*Queller, 1985*), if we assume that genotype and phenotype correlate perfectly (see also Section 2 of Appendix 2).

**Table 1.** Three rules and three models.
This table gives all combinations of the three rules and the three models discussed in the example. All rules indicate the direction of selection correctly for all models. Yellow indicates a combination of a rule and a model, where the rule is more general than is needed for the model. This leads to one or more $b$'s being 0. These are relatively harmless overspecifications. Red indicates a combination of a rule and a model, where the rule is not general enough (underspecified) for the model. This leads to $b$'s and $c$'s that depend on the population state. Terms that depend on the population state are abbreviated as follows: $r = r_{0,1} = \frac{\mathrm{Cov}\left(p,q\right)}{\mathrm{Var}\left(p\right)}$, $r_{1,1} = \frac{\mathrm{Cov}\left(p,pq\right)}{\mathrm{Var}\left(p\right)}$, $s_b = \frac{\mathrm{Cov}\left(p,q\right)\mathrm{Cov}\left(p,pq\right) - \mathrm{Cov}\left(q,pq\right)\mathrm{Var}\left(p\right)}{\left(\mathrm{Cov}\left(p,q\right)\right)^2 - \mathrm{Var}\left(p\right)\mathrm{Var}\left(q\right)}$, and $s_c = \frac{\mathrm{Cov}\left(p,pq\right)\mathrm{Var}\left(q\right) - \mathrm{Cov}\left(p,q\right)\mathrm{Cov}\left(q,pq\right)}{\left(\mathrm{Cov}\left(p,q\right)\right)^2 - \mathrm{Var}\left(p\right)\mathrm{Var}\left(q\right)}$ (see Detailed Calculations 2.3 at the end of Appendix 2 for calculations). Rule 1 is the standard rule for non-social evolution for linear non-social traits. Rule 2 is the classical Hamilton's rule. Rule 3 is Queller's rule (*Queller, 1985*), which is a rule that allows for an interaction effect. Rule 1 is nested in Rule 2, which is nested in Rule 3, which can be nested in more general rules as well.

| | | Model 1: $\alpha + \beta_{1,0}p_i$ | Model 2: $\alpha + \beta_{1,0}p_i + \beta_{0,1}q_i$ | Model 3: $\alpha + \beta_{1,0}p_i + \beta_{0,1}q_i + \beta_{1,1}p_iq_i$ |
|---|---|---|---|---|
| **Rule 1:** $0 > c$ | $c =$ | $-\beta_{1,0}$ | $-\left(\beta_{1,0} + r_{0,1}\beta_{0,1}\right)$ | $-\left(\delta_{1,0} + r_{0,1}\beta_{0,1} + r_{1,1}\beta_{1,1}\right)$ |
| **Rule 2:** $rb > c$ | $c =$ | $-\beta_{1,0}$ | $-\beta_{1,0}$ | $-\beta_{1,0} + s_c\beta_{1,1}$ |
| | $b =$ | $0$ | $\beta_{0,1}$ | $\beta_{0,1} + s_b\beta_{1,1}$ |
| **Rule 3:** $r_{0,1}b_{0,1} + r_{1,1}b_{1,1} > c$ | $c =$ | $-\beta_{1,0}$ | $-\beta_{1,0}$ | $-\beta_{1,0}$ |
| | $b_{0,1} =$ | $0$ | $\beta_{0,1}$ | $\beta_{0,1}$ |
| | $b_{1,1} =$ | $0$ | $0$ | $\beta_{1,1}$ |

We can, of course, still apply the Generalized Price equation in regression form for Model 1, or the Generalized Price equation for Model 2, to Model 3 in the same way that we can apply the Generalized Price equation for Model 1 to Model 2, as we did above. We do this in detail in Section 2 of Appendix 2 to make sure that there can be no misunderstanding concerning the details, and the results are shown in *Table 1*. The important observation to make here is that both of these rules *also* get the direction of selection right. The argument for not using these rules for Model 3, even though they do get the direction of selection right, is identical to the argument for not using the rule for selection of non-social traits for social traits: the rule for non-social traits just does not describe the population genetic dynamics for a social trait accurately. This is why we use the conventional version of Hamilton's rule when modeling social behavior, and not the rule that we would get if we applied the Generalized Price equation in regression form for the non-social model (Model 1). The Price-like equation for the non-social model also gets the direction of selection right, but we know that this version is incomplete, precisely because it leaves out the true social effects we care about unveiling when we use the conventional Hamilton's rule. For exactly analogous reasons, we should not rely on underspecified versions of the Price equation in other instances, including the case where the conventional Hamilton's rule itself, in turn, is based on an underspecified Price-like equation.

As a symptom of underspecification, if we apply the Generalized Price equation in regression form for Model 1 to Model 3, then the value for $\hat{\beta}_{1,0}$ is not a constant, and depends on the population state. Similarly, if we apply the Generalized Price equation for Model 2 to Model 3, then the values for $\hat{\beta}_{1,0}$ and $\hat{\beta}_{0,1}$ are not constants and depend on the population state. The part of the literature on Hamilton's rule that has emphasized its full generality has, however, insisted that we always use one and the same equation, even if it is underspecified, and incorporate more general models, such as Model 3, within the classical version of Hamilton's rule (*Grafen, 1985a*; *Queller, 1992*; *Grafen, 2006*; *Abbot et al., 2011*; *Gardner et al., 2011*; *Marshall, 2015*; *Rousset, 2015*).

The relation between these examples can be summarized in a table that, for the three models and the three rules, represents all nine combinations of them (see *Table 1*). This matrix of combinations of models and rules also helps to understand what causes the contentiousness of the debate on the generality of Hamilton's rule. Rule 2, which is the rule that we get from applying the standard Price equation (also known as using the regression method), is a completely general rule, in the sense that whatever the true model is, it always gets the direction of selection right. This is true, but that fact is not a good argument for singling this rule out as more helpful, meaningful, or insightful than other rules. In the example above, we have seen that if we apply the Generalized Price equation in regression form (which one could also describe as using the regression method, but now with a richer menu of alternative underlying statistical models) then this can also give us Rule 1 or Rule 3, depending on the statistical reference model we use. These rules are equally correct, in the sense that they also always get the direction of selection right. They are also equally general, in the sense that they also get it right for every possible model. Being a general rule, that always gets the direction of selection right, therefore, cannot be a criterion for elevating Rule 2 (which is the classical Hamilton's rule), above the other ones, because Rules 1 and 3 are also general rules, that always get the direction of selection right.

Since being completely general and always getting the direction of selection right does not single out any of the possible rules, we need additional criteria. A natural criterion would be that besides being correct, the terms in the rule would have to be meaningful. More precisely, we think the rule should do what Rule 1 (the rule for evolution of non-social traits) does for Model 1, and what Rule 2 (the classical Hamilton's rule) does for Model 2, and that is to separate model parameters from properties of the population state, in order to produce an equation that describes the population genetic dynamics. That is what Rule 3 does for Model 3. Ever more general models would moreover require ever more general rules to accurately capture the population dynamics.

The debate in the literature on the generality of Hamilton's rule is so long-lasting because it mostly focuses on whether or not rules are correct, and not on whether the rules are (also) meaningful. One side of the debate tends to return to the argument that Rule 2 is general and correct (*Abbot et al., 2011*; *Gardner et al., 2011*; *Marshall, 2015*; *Rousset, 2015*). The other side of the debate tends to return to models that do not fit Model 2 (*Karlin and Matessi, 1983*; *Matessi and Karlin, 1984*; *van Veelen, 2009*). Sometimes arguments on this side take a completely different approach, and rather than describing selection with the Price equation, and then worry about whether or not one can

interpret regression coefficients in it as benefits and costs, they start with models with a priori interpretable definitions for $b$ and $c$. This is called the counterfactual approach to defining costs and benefits (*van Veelen et al., 2017*), as opposed to the regression approach, and because it has a different definition of the costs and benefits, it can result in rules that end up getting the direction of selection wrong (*Karlin and Matessi, 1983*; *Matessi and Karlin, 1984*; *van Veelen et al., 2017*).

Another recurrent point is that when regression coefficients in the Price equation are found to vary with the state of the parent population, this is often claimed to be the result of the Price equation being dynamically insufficient. In order to point out why that is inaccurate, we can first point to ways in which an underlying model *can* be dynamically insufficient. If we have a model that describes how the fitness of an individual depends on its $p$- and $q$-score, and we are given a parent population, then this model would produce the $p$-scores in the new generation. It would, however, not generate which individuals are partnered up with whom in the new generation, and therefore it would not identify who has which $q$-score in the new generation. In this case, a full model would have to include more than just the fitness function, but in the absence of assumptions about the matching in the new generation, such a model would be dynamically insufficient. It is, however, important to notice that this is *not* the same as the regression coefficients $\hat{\beta}_{k,l}$, depending on the population state, which is regularly also referred to as dynamical insufficiency. Describing this as dynamical insufficiency is incorrect, and the dependence on the population state is really a symptom of misspecification. Section 6 of Appendix 1 elaborates on this point in a more precise way (see also *van Veelen, 2018b*).

## Discussion

The general version of Hamilton's rule provides a positive resolution of a long-standing, heated debate. I derived the Generalized Price equation, which reconnects the Price equation with statistics and allows us to base rules for selection properly on dynamical models for population genetics that offer the flexibility needed to accommodate a variety of ways in which fitness can depend on traits of individuals and their interaction partners. While previous work focused on the limitations of the original Price equation (*van Veelen, 2005*; *van Veelen et al., 2012*; *van Veelen, 2020b*), and on limitations of the classical version of Hamilton's rule, when there are interaction effects (*Karlin and Matessi, 1983*; *Matessi and Karlin, 1984*; *Nowak et al., 2010*; *Allen et al., 2013*; *Nowak et al., 2017*; *van Veelen et al., 2017*), the Generalized Price equation, and the general version of Hamilton's rule that is derived with it, is a constructive contribution that offers alternatives for a large variety of forms the fitness function can have. The insight that comes with it has major implications, also for empirical research on kin selection. It implies that whether or not Hamilton's rule holds is not a meaningful empirical question. As a field, we have nonetheless spent quite some time and energy on trying to answer this question (*Bourke, 2014*). Because the question turns out to be ill-posed, we can stop trying to answer it. The meaningful empirical question that we should focus on instead is what form the fitness function has, and which version of Hamilton's rule therefore applies to which social trait.

A video presentation of the results reported in this article can be found here.

## Additional information

### Funding
No external funding was received for this work.

### Author contributions
Matthijs van Veelen, Conceptualization, Formal analysis, Visualization, Writing – original draft, Writing – review and editing

### Author ORCIDs
Matthijs van Veelen https://orcid.org/0000-0002-8290-9212

Joint Public Review: https://doi.org/10.7554/eLife.105065.3.sa1
Author response https://doi.org/10.7554/eLife.105065.3.sa2

# Additional files

**Supplementary files**
MDAR checklist

**Data availability**
The manuscript is a theoretical study, and does not use data, other than the artificial data generated for *Figure 1* in the main text which is provided as *Figure 1—source data 1*.

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

## Appendix 1

### The Generalized Price Equation

The main ingredient of this appendix is the derivation of the Generalized Price equation. This generalizes the original Price equation in regression form in the sense that it produces a set of Price-like equations, one for every different underlying model that one could assume has generated the data. All of these different Price-like equations are identities, and all of them only have a meaningful interpretation if the data are indeed generated by the model they presuppose. The criteria for choosing between these different Price-like equations are the exact same as the criteria that standard statistics uses when choosing the right statistical model, based on the data. The original Price equation in regression form is the generalized Price equation that goes with a specific linear model. The problem with the widespread misuse of the Price equation is caused by the fact that it loses its meaning if the data are not generated by this model – in the same way that any of the other Price-like equations lose their meaning if the data are not generated by the models they presuppose.

### 1. Introduction

Right from the beginning, it has been unclear if the Price equation (*Price, 1970*; *Price, 1972*) was intended as a tool for modeling, or if it was supposed to be applied to data. The terminology of regression coefficients suggests that the Price equation is to be applied to data, but the use in the population genetics literature mostly suggests that it is a tool for modeling. Here, we will do both, but we will begin with the first option. The reason to start there is that this offers a way in which we can give a meaningful interpretation to the 'regression coefficients' in what Price (*Price, 1970*) called the 'regression form' of his equation. Both the paper in which the equation was presented and the subsequent literature suggest there is a link between statistics and the regression form of the Price equation, without establishing what that link is. Appendix 1 provides that link, and in doing so, it formulates a generalized version of the Price equation that gives it the flexibility that a proper link with statistics requires. This Generalized Price equation also turns out to provide a solution for the problems with the Price equation identified in *van Veelen, 2005*; *van Veelen et al., 2012*; *van Veelen, 2020a*. The insight that it provides also sheds light on exactly what the source of the controversy in the Price equation literature is.

I will begin with the derivation of the general version of the Price equation. Besides the introduction of a bit of matrix notation, the first part of that derivation is totally standard and leads to the Price equation as we know it, in what Price (*Price, 1970*; *Price, 1972*) called the covariance form.

$$\bar{w}\Delta\bar{p} = \mathrm{Cov}\left(w, p\right) + E\left(w\Delta p\right) \tag{PE.C}$$

Then we combine this with a set of statistical models, all of which include (1) a constant term, and (2) a linear term for the set of genes, the selection of which is tracked by the variable $\bar{p}$, or the average $p$-score. For any model in this set, we find that in the covariance term, we can replace the realized fitnesses $w_i$ with the estimated fitnesses $\hat{w}_i$ according to the model. This leads to the generalized Price equation in covariance form that, for any model in this set, reads

$$\bar{w}\Delta\bar{p} = \mathrm{Cov}\left(\hat{w}, p\right) + E\left(w\Delta p\right) \tag{GPE.C}$$

If we then unpack these covariances for any model in this set, we arrive at the generalized Price equation in regression form. We can, for instance, consider the set of models given by

$$w_i = \alpha + \sum_{r=1}^{R} \beta_r p_i^r + \varepsilon_i$$

The $R$ is the largest exponent that is included in the polynomial of the model. Different choices for $R$ therefore imply different models; for $R = 1$ we get a linear model; for $R = 2$ we get a quadratic model; for $R = 3$ a cubic one, and so on. For every $R$, and thereby for every model in this set, we get a Price-like equation:

$$\bar{w}\Delta\bar{p} = \sum_{r=1}^{R} \hat{\beta}_r \mathrm{Cov}\left(p, p^r\right) + E\left(w\Delta p\right) \tag{GPE.R1}$$

The central point here is that there is not just one Price equation in regression form. There is a Price-like equation for every different model, as long as these models meet the minimal requirement of including a constant and a linear term. For rich enough settings (i.e. for settings with sufficiently many different values that the $p$-scores can take), these Price-like equations, moreover, are generically proper different. That is to say, if we take any given transition from a parent population to an offspring population, and first combine this, for instance, with the Price equation in regression form for the linear model ($R = 1$), and then with the Price equation in regression form for the quadratic model ($R = 2$), then the $\hat{\beta}_1$ in the Price equation for $R = 1$ will almost always be different from the $\hat{\beta}_1$ in the Price equation for $R = 2$, unless the $p$-score can only be 0 or 1.

The original Price equation in regression form is the generalized Price equation in regression form for the linear model. That implies that if the data are in fact generated by the standard linear model, writing *the* Price equation is a meaningful exercise. It also implies that if the data are not generated by the standard linear model, then writing *the* Price equation is not a meaningful exercise. For any set of data, one can write many Price-like equations, and all of them are identities. A meaningful Price equation exercise pertaining to data should really consist of using standard statistical tools for choosing between different statistical models. Any choice for a statistical model then automatically implies a choice for one of these Price-like equations in regression form. The confidence with which the data allow us to pick one of the models is then by definition equal to the confidence we can have that we picked the right Price-like equation to describe the population genetic dynamics.

The fact that there is more than one Price-like equation requires us to reflect on what the original Price equation is, and what we want to use it for. The Price equation is (i) an identity. That means that the left-hand side is equal to the right-hand side, whatever the parent population is and whatever the offspring population is. This is true for the original Price equation, and it is also true for all Price-like equations that we get by combining the Generalized Price equation with some model (or, in other words, if we choose some $R$). The being an identity implies that (ii) it 'always gets the direction of selection right'; if the left-hand side is negative, then so is the right-hand side. This is a somewhat riskier way of looking at the equation because it makes it tempting to see one side of the identity as an explanation of the other. Again, this property is shared by all Price-like equations. The main ingredient that I would like to add to this, and that the abundance of Price-like equations necessitates, is that we would like to pick one of those equations on the basis of whether or not it accurately describes the underlying population genetical dynamics. If it does, then I use the word 'meaningful'.

The main insight from the derivation of the generalized Price equation in regression form is therefore first and foremost positive and constructive, in the sense that it provides a recipe for finding an equation that is meaningful in the same way that the original Price equation is meaningful for the linear model. Besides that, it also helps us pinpoint what drives the misuse of the Price equation in the literature. Misapplying the Price equation comes down to applying the original Price equation in regression form (which is the generalized Price equation in regression form for the linear model) in a setting with a statistical model that is not linear, or that is general enough to allow for models other than the standard linear model. The misuse of the Price equation, therefore, is a special case of the chosen Price-like equation not matching the statistical model.

Besides this main result, this appendix also serves as a guide to the Price equation. We will start simple, by introducing a haploid setting with asexual reproduction. The reason to start there is that it is the easiest. We then derive the normal, standard Price equation and consider a simple example, where it tracks the absence or presence of a single gene. Then we move on to allow for a genetic measure that is not binary and derive the general version of the Price equation, both in covariance form and, for a specific set of models, in regression form. This derivation helps establish that there is a variety of Price-like equations. It also shows that the answer to the question of which one allows for a meaningful interpretation is given by standard statistical considerations involving sample size and model specification. All of this happens in Section 2.

In Section 3, we repeat this for diploid, sexually reproducing species. This is more interesting and allows us to illustrate the richness of possible underlying true models better. This section contains an example in which we assume heterozygote advantage, or dominance, and an example in which the genes we consider are sex-determining genes.

In Section 4, we further enrich the set of possible models by allowing the model to not only include genes, the selection of which we track with the average *p*-score, but also take other genes into consideration, reflected in an average *q*-score. This introduces additional scope for misspecification.

In Section 5, we will describe how this carries over to the setting in which the Price equation is used for modeling.

In Section 6, we conclude and reflect on a few related issues, including dynamic sufficiency.

In Appendix 2, we discuss how the use of the original Price equation, instead of its generalized version, has led to a longstanding disagreement concerning the generality of Hamilton's rule, and how it can be resolved using the Generalized Price equation.

## 2. The Price equation for asexual reproduction

The original Price equation allows for any ploidy. Here we begin simple, by choosing a ploidy of 1. That keeps the introduction of the matrix notation as well as the first examples easy to follow.

We assume that the parent generation consists of $n$ individuals, and the offspring generation consists of $m$ individuals. We can represent who is whose offspring with an offspring matrix $A$. Let $A_{ij} = 1$ if $i$ is $j$'s parent, and $A_{ij} = 0$ if not.

If we fix a member of the offspring generation and sum over the parent generation, we get the number of parents per child. With asexual reproduction, all kids have one parent, which implies that this sum must be 1; $\sum_{i=1}^{n} A_{ij} = 1$ for all offspring $j$. Any offspring matrix must have this property.

If we fix a member of the parent generation and sum over the offspring generation, we get the number of children for that parent; $a_i = \sum_{j=1}^{m} A_{ij}$ for all parents $i$. With asexual reproduction, this is also the fitness of the parent; $w_i = a_i$.

These two observations together have a straightforward implication. We can sum all elements of this matrix in two ways. The first is $\sum_{j=1}^{m} \left( \sum_{i=1}^{n} A_{ij} \right) = \sum_{j=1}^{m} 1 = m$. The second is $\sum_{i=1}^{n} \left( \sum_{j=1}^{m} A_{ij} \right) = \sum_{i=1}^{n} w_i$. Because these must be equal, the sum of all fitnesses must equal the number of individuals in the offspring generation; $\sum_{i=1}^{n} w_i = m$.

### Genes in the parent generation

Suppose that we know the 'dose' of a gene (*Price, 1970*), or the *p*-score (*Grafen, 1985a*), for every individual in the parent population, and denote it by $p_i, i = 1, ..., n$ for all parents. This dose may be restricted to be either 0 or 1, as it will in our first example, where this represents the presence or absence of a single gene. The idea of a dose, or a *p*-score, however, is that we also allow for this to be any measure for the presence or absence of a set of genes, and in particular genes that all contribute to a certain trait value. If we have two genes, both of which independently raise a certain trait value by the same amount, then a natural choice would be for the *p*-scores to be 0 if both are absent, $\frac{1}{2}$ if one is present, and 1 if both are present. In general, this means that $p_i$ can take values on some subset of the interval $[0, 1]$.

### Which genes are passed on

With sexual reproduction, there is randomness in which genes a parent passes on to their offspring. With asexual reproduction, offspring are just copies of their parents. That makes the matrix $P$, that represents the *p*-scores of the genes that are passed on, relatively easy. With asexual reproduction, $P_{ij} = 0$ if individual $i$ from the parent population is just not a parent of individual $j$ in the offspring population (in other words $P_{ij} = 0$ if $A_{ij} = 0$). If individual $i$ from the parent population is in fact the parent of individual $j$ in the offspring population ($A_{ij} = 1$), then $P_{ij}$ is the *p*-score of the offspring. Here we assume that there is no mutation, which means that this is also the *p*-score of the parent. This implies a simple, straightforward relation between the *p*-scores in the offspring generation and the matrix $P$. If we fix an offspring and sum over everyone in the parent generation, then all but one are not the parent of individual $j$, and for those, $P_{ij} = 0$. For the actual parent, the matrix returns the *p*-score of the offspring, and hence $\sum_{i=1}^{n} P_{ij} = p_j'$ for all offspring $j$.

### Change

The average *p*-score in the parent generation is $\bar{p} = \frac{\sum_{i=1}^{n} p_i}{n}$. Some of the older papers on the Price equation do not use upper bars to indicate averages. Since it is natural for $p$ to represent the vector of *p*-scores in the parent generation, $p = [p_1, \ldots, p_n]$, we add an upper bar to make sure there is no

ambiguity regarding what this is: $\bar{p}$ is the average of the $p_i$'s in the parent population. Similarly, the average $p$-score in the offspring generation is $\bar{p}' = \frac{\sum_{j=1}^{m} p'_j}{m}$.

## Derivation of the Price equation

The key ingredient in the derivation of the Price equation is that one can calculate the sum of the $p$-scores by going over the individuals in the offspring generation in two ways. With sexual reproduction, this is a bit more complex, but with asexual reproduction, this is really quite simple.

1. One can go over all offspring, starting at offspring number 1 and ending at number $m$, and just add up their $p$-scores. With the relation between matrix $P$ and vector $p'$, that amounts to $\sum_{j=1}^{m} p'_j = \sum_{j=1}^{m} \left( \sum_{i=1}^{n} P_{ij} \right)$.

2. For every parent, we can add up the $p$-scores of their offspring. Then we can go over all parents and add those numbers up. This amounts to a switch in summation order from $\sum_{j=1}^{m} \left( \sum_{i=1}^{n} P_{ij} \right)$ to $\sum_{i=1}^{n} \left( \sum_{j=1}^{m} P_{ij} \right)$.

This switch allows us to write the average $p$-score in the offspring generation as

$$\bar{p}' = \frac{\sum_{j=1}^{m} p'_j}{m} = \frac{\sum_{j=1}^{m} \sum_{i=1}^{n} P_{ij}}{m} = \frac{\sum_{i=1}^{n} \sum_{j=1}^{m} P_{ij}}{m}$$

Then we can divide $\sum_{j=1}^{m} P_{ij}$ by $w_i$, if we undo that by also multiplying by $w_i$, and we can subtract $\sum_{i=1}^{n} w_i p_i$, if we undo that by also adding $\sum_{i=1}^{n} w_i p_i$. This way we get

$$\bar{p}' = \frac{\sum_{i=1}^{n} w_i \left( \frac{\sum_{j=1}^{m} P_{ij}}{w_i} \right)}{m} = \frac{\sum_{i=1}^{n} w_i p_i + \sum_{i=1}^{n} w_i \left( \frac{\sum_{j=1}^{m} P_{ij}}{w_i} - p_i \right)}{m}$$

Then we add and subtract $\frac{1}{n} \sum_{i=1}^{n} w_i \sum_{i=1}^{n} p_i$

$$\bar{p}' = \frac{\sum_{i=1}^{n} w_i p_i + \frac{1}{n} \sum_{i=1}^{n} w_i \sum_{i=1}^{n} p_i - \frac{1}{n} \sum_{i=1}^{n} w_i \sum_{i=1}^{n} p_i}{m} + \frac{\sum_{i=1}^{n} w_i \left( \frac{\sum_{j=1}^{m} P_{ij}}{w_i} - p_i \right)}{m}$$

Now if we look at the first term on the right-hand side, and then the second term in the numerator, we can use $\sum_{i=1}^{n} w_i = m$ to simplify it. Because $\frac{1}{n} \sum_{i=1}^{n} p_i$ is moreover the definition of the average $p$-score in the parent population, we can rewrite this whole equation as

$$\bar{p}' = \bar{p} + \frac{\sum_{i=1}^{n} w_i p_i - \frac{1}{n} \sum_{i=1}^{n} w_i \sum_{i=1}^{n} p_i}{m} + \frac{\sum_{i=1}^{n} w_i \left( \frac{\sum_{j=1}^{m} P_{ij}}{w_i} - p_i \right)}{m}$$

We can subtract $\bar{p}$ on both sides and multiply this left and right by the average fitness $\bar{w} = \frac{m}{n}$. If we do that, we get

$$\bar{w} \left( \bar{p}' - \bar{p} \right) = \frac{1}{n} \sum_{i=1}^{n} w_i p_i - \frac{1}{n^2} \sum_{i=1}^{n} w_i \sum_{i=1}^{n} p_i + \frac{1}{n} \sum_{i=1}^{n} w_i \left( \frac{\sum_{j=1}^{m} P_{ij}}{w_i} - p_i \right)$$

A short way to write this is

$$\bar{w} \Delta \bar{p} = \text{Cov}\left( w, p \right) + E\left( w \Delta p \right) \tag{PE.C}$$

where

$$\text{Cov}\left( w, p \right) = \frac{1}{n} \sum_{i=1}^{n} w_i p_i - \frac{1}{n^2} \sum_{i=1}^{n} w_i \sum_{i=1}^{n} p_i$$

and

$$E\left(w\Delta p\right) = \frac{1}{n}\sum_{i=1}^{n} w_i \left(\frac{\sum_{j=1}^{m} P_{ij}}{w_i} - p_i\right)$$

This is what Price (**Price, 1970**; **Price, 1972**) calls the *covariance form*. Here, it is important to realize that the term that is abbreviated as $\mathrm{Cov}\left(w,p\right)$ is *not* an actual covariance. I made this point elsewhere (see page 414 of **van Veelen, 2005**, and Box 1, page 66, of **van Veelen et al., 2012**), and while it seems nit-picking, it pays to be precise conceptually. A covariance is a property of a random variable, and that is not what the $\mathrm{Cov}\left(w,p\right)$-term is here. If $p_i$ and $P_{ij}$ are data, as we assume they are here, then the term abbreviated as $\mathrm{Cov}\left(w,p\right)$ is $\frac{n-1}{n}$ times the *sample* covariance, or, in other words, the sample covariance without Bessel's correction. In Section 5, we switch to a modeling context. In that case, the vector $p = \left[p_1,\ldots,p_n\right]$ represents a population state in a model, and the model is a statement about the probabilities of different transitions. Also here, the term abbreviated as $\mathrm{Cov}\left(w,p\right)$ is not a covariance; it is a summary statistic describing an aspect of the transition from one population state to another.

Similarly, the second term on the right-hand side is *not* an expected value. An expected value is a property of a random variable, and that is not what $E\left(w\Delta p\right)$ is. If $p_i$ and $P_{ij}$ are data, it is the average difference between, on the one hand, the average *p*-score of the offspring of a parent and, on the other, the parent's own *p*-score. Without mutation, this term is 0 for asexual reproduction by definition. Also in a modeling context, the term abbreviated as $E\left(w\Delta p\right)$ is not an expectation; this is a summary statistic describing an aspect of the transition from one population state to another. The model can make assumptions about this. For instance, if we consider a model with asexual reproduction, then that implies that this term is always 0 by definition. If we consider a model with sexual selection, but assume fair meiosis, then the expected value of this term (that is: the expected value of what is denoted here as $E\left(w\Delta p\right)$, but what really is an average) is 0. This point is also made on page 418 of **van Veelen, 2005**.

When we derive the Generalized Price equation in Section 2.2, we will come back to this point in the derivation and proceed from there. In this section, on the other hand, we will simply follow Price (**Price, 1970**; **Price, 1972**) in substituting $\beta \cdot \mathrm{Var}\left(p\right)$ for $\mathrm{Cov}\left(w,p\right)$, where

$$\mathrm{Var}\left(p\right) = \frac{1}{n}\sum_{i=1}^{n} p_i^2 - \frac{1}{n^2}\left(\sum_{i=1}^{n} p_i\right)^2$$

and

$$\beta = \frac{\mathrm{Cov}\left(w,p\right)}{\mathrm{Var}\left(p\right)}$$

If we do, then we get the following equation.

$$\bar{w}\Delta\bar{p} = \beta \cdot \mathrm{Var}\left(p\right) + E\left(w\Delta p\right) \tag{PE.R}$$

This is what Price (**Price, 1970**; **Price, 1972**) calls *the regression form*. $\mathrm{Var}\left(p\right)$ is not really a variance, but if $p_i$ and $P_{ij}$ are data, it is $\frac{n-1}{n}$ times the *sample* variance, or, in other words, the sample variance without Bessel's correction. The $\beta$ is typically referred to as a regression coefficient, which it is if $p_i$ and $P_{ij}$ are data. We will get back to all of this below, but for now, this is the final equation.

Our next step is to look at two specific examples. After that, we will return to the *covariance form* of the Price equation (PE.C) above and derive the generalized version starting from there.

### Example 2.1
We start by assuming that the model captures one single gene – so *p*-scores are either 0 or 1 – and that this one gene has some (unknown) effect on fitness. In other words, we assume that the true model that describes how fitnesses depend on *p*-scores is

$$w_i = \alpha + \beta p_i + \varepsilon_i \tag{A}$$

where $\alpha$ is the baseline fitness, $p_i = 0$ if the gene is absent, $p_i = 1$ if it is present, $\beta_1$ is the linear effect of the gene on fitness, and $\varepsilon_i$ is a noise term.

We can, for example, take a parent population that consists of two individuals. The first has the gene and two offspring. The second does not have the gene and has no offspring. In other words, $p = [1,0]$, $w = [2,0]$ and

$$A = P = \begin{bmatrix} 1 & 1 \\ 0 & 0 \end{bmatrix}$$

The ingredients of the Price equation then are: $\bar{w} = 1$, as there are 2 parents and 2 offspring; $\Delta\bar{p} = \frac{1}{2}$, because the average $p$-score went from $\frac{1}{2}$ to 1; $E(w\Delta p) = 0$, by definition with asexual reproduction without mutations; $\mathrm{Cov}(w,p) = \frac{1}{2}$, $\mathrm{Var}(p) = \frac{1}{4}$ and $\beta = 2$. The calculations are at Detailed Calculations 1.1, at the end of Appendix 1.

The Price equation in regression form therefore is:

$$\bar{w}\Delta\bar{p} = \beta \cdot \mathrm{Var}(p) + E(w\Delta p)$$

$$1 \cdot \frac{1}{2} = 2 \cdot \frac{1}{4} + 0$$

In this case, the sample size is clearly not large enough for the $\beta$ from the Price equation to estimate the true $\beta$ from the model with any significance. The $\beta$ in the Price equation therefore cannot be interpreted as the fitness effect of the gene.

## Being careful with labels: populations, samples, and data-generating processes

At this point, there is some room for confusion, caused by the use of the word 'population'. This is discussed on page 416 of *van Veelen, 2005*, but it can be helpful to also elaborate a little on this here. Part of our intuition about statistics is shaped by polling or surveys. The setting for a poll or a survey is that there is a large population, of which we would like to know, for instance, what they would vote, if the election were held today; or how much time the average EU citizen spends behind a screen; or what fraction of the people in the UK believes in evolution. The idea is then to randomly draw a sample from the population as a whole, and the larger the sample, and the more balanced it is, the larger the confidence that the average within the sample is close to the average of the population as a whole. On election day, we then find out the thing that the polls were trying to get at.

This shapes our intuition, in the sense that we are tempted to think that if the two individuals in our sample *are* the whole population (as they are in the example above with the Price equation), then there is no statistical uncertainty left, and we know everything there is to know. That would be the case if the question at hand were '*how many offspring did the carriers of the gene get within this population, and how many did the non-carriers get?*' In our case, however, the question that we would be interested in, in order to be able to describe the population genetic dynamics, is '*how does the p-score affect fitnesses?*' For answering the second question, having a parent population of size 2 will not be sufficient.

In order to illustrate the difference with a thought experiment, we can think of a relatively common versus an extremely rare genetic disorder. Suppose there is a genetic disorder that one in 1000 people have. Assume also that we have data on everyone that was born with that genetic disorder in the UK in the years 2000 to 2005. In that case, we would have a few thousand observations, which is more than enough to calculate the probability that people born with that genetic disorder survive into adulthood. If, within this thought experiment, we furthermore can assume that there are no changes in medical treatment, or other factors that could make being born in 2024 be different from being born in 2000, then for someone born with that genetic disorder in the UK today, the probability of surviving until the age of 18 is estimated with high accuracy by the realized survival rate within this population.

Now imagine another genetic disorder that only 1 in a million people are born with, and again assume that we have data for everyone that was born with this genetic disorder in the UK between

2000 and 2005. With the low incidence of this particular genetic disorder, this can easily leave us with less than a handful of observations only. Now even though in both cases we know the entire population within the bracket, for the extremely rare genetic disorder, we would not make predictions concerning the chances of survival until adulthood with the same confidence as we would for the much more common genetic disorder. For a doctor, the relevant question is what the probabilities are for her patient, and for this, knowing the whole population is very useful with the common genetic disorder, but much less so with the rare genetic disorder.

This serves to illustrate that, while with polling, the aim is to know the composition of the population, there are many settings in which that is not the underlying question. Statisticians therefore sometimes like to use the term 'data generating process' instead of 'population'. For polling, or market research, the question at hand is: if I were to pick a random member of the population, what would this person vote, or like to purchase? The data-generating process then would be that we ask a subset of people about their preferences. For an extremely rare genetic disease, however, the data generating process is that every once in a blue moon, someone with that genetic disorder is born. This can render so few individuals that have the genetic disorder, that even knowing everyone in the population that does, will not give an answer to the question of what, for instance, someone's life expectancy is. For an example where the statistical population and the biological population are confused, see page 1017 of *Frank, 2012*.

## Back to example 2.1

If we now go back to the Price equation in regression form, then with a small population, even if the Price equation encompasses the whole population, it may not always be right to interpret the $\beta$ in it as the effect of the gene. It is a regression coefficient, and as such, it may be a noisy, and therefore unreliable, estimator of the true fitness effect of the gene, even if it is unbiased.

One can, however, also imagine a vector $p$, and matrices $A$ and $P$, that together represent a much larger population, and therefore a much larger dataset. If this results in an estimate of the true $\beta$ that is statistically significant (that is: the true $\beta$ is most likely to be different from 0) then the $\beta$ in the Price equation can be interpreted as the estimated effect of the gene in question on fitness. After the next example, from the derivation of the Generalized Price equation onwards, we will therefore use notation that differentiates between the true effect $\beta$ and the estimator $\hat{\beta}$ of the true effect.

All of this is under the assumption that model (A), $w_i = \alpha + \beta p_i + \varepsilon_i$, is accurately specified. In this example, we assumed that the $p$-score only reflects the absence or presence of one single gene, and therefore the $p$-score can only be 0 or 1. That implies that in this case, there is no room for underspecification, and this assumption does not need to be tested.

The main observation here is that the extent to which we can give a meaningful interpretation of the right-hand side of the Price equation is determined by standard statistical considerations. With small sample sizes, the $\beta$ from the Price equation will, as an estimator of the true $\beta$, typically not be significant, even if the true $\beta$ would really be different from 0. The low power of statistical testing then implies that there is really nothing we can say about the true $\beta$. Larger sample sizes allow us to infer more from the data, but even then, the interpretation is bound by the rules of statistics. If the $\beta$ from the Price equation is statistically significant, it is still the (unbiased) *estimator* of the effect of the gene on fitness, and not the effect of the gene on fitness itself.

## Summary

If the $p$-score only reflects the presence or absence of a single gene, and $\beta$ is statistically significant, then the right-hand side of the Price equation in regression form has a meaningful interpretation; the $\beta$ from the regression form of the Price equation is the estimated linear effect of this gene on fitness.

## Example 2.2

We now assume that the model captures two genes, both of which contribute to one and the same trait value. The possible $p$-scores, therefore, are 0 if both are absent; $\frac{1}{2}$ if one is present; or 1 if both are present. If the $p$-scores translate to fitnesses in a linear way, then we are back in the setting of Example 2.1, just with one additional value that the $p$-score can take. We can, however, also allow for the possibility that the relation between $p$-scores and fitnesses is not linear. There are two steps between $p$-scores and fitnesses, so the non-linearity might arise in either of the two. It might be that $p$-scores translate to trait values in a non-linear way; or it might be that the relation between trait

value and fitness is not linear. Here, we assume that, for either of these two reasons, fitnesses follow a quadratic model:

$$w_i = \alpha + \beta_1 p_i + \beta_2 p_i^2 + \varepsilon_i \qquad (B)$$

where $\alpha$ is the baseline fitness, $\beta_1$ and $\beta_2$ are the linear and the quadratic effect of the $p$-score on fitness, and $\varepsilon_i$ is a noise term. As an aside: if we define $w^0$, $w^{\frac{1}{2}}$, $w^1$ and as the expected fitnesses of an individual with $p$-score 0, $\frac{1}{2}$ and 1, then $w^0 = \alpha$, $w^{\frac{1}{2}} = \alpha + \frac{1}{2}\beta_1 + \frac{1}{4}\beta_2$, and $w^1 = \alpha + \beta_1 + \beta_2$. This we can rewrite as $\alpha = w^0$, $\beta_1 = -3w^0 + 4w^{\frac{1}{2}} - w^1$ and $\beta_2 = 2\left(w^0 - 2w^{\frac{1}{2}} + w^1\right)$.

Then we can, for example, take a parent population that consists of two individuals. The first has a $p$-score of 1 and two offspring, and the second has a $p$-score of $\frac{1}{2}$, and no offspring. In other words, $p = \left[1, \frac{1}{2}\right]$, $w = [2, 0]$ and

$$A = P = \begin{bmatrix} 1 & 1 \\ 0 & 0 \end{bmatrix}$$

The ingredients of the Price equation in regression form then are: $\bar{w} = 1$, as there are 2 parents and 2 offspring; $\Delta\bar{p} = \frac{1}{4}$, because the average $p$-score went from $\frac{3}{4}$ to 1; $E\left(w\Delta p\right) = 0$, by definition with asexual reproduction without mutations; $\text{Cov}\left(w, p\right) = \frac{1}{4}$, $\text{Var}\left(p\right) = \frac{1}{16}$ and $\beta = 4$. The calculations are in Detailed Calculations 1.2, at the end of Appendix 1.

The Price equation therefore is:

$$\bar{w}\Delta\bar{p} = \beta \cdot \text{Var}\left(p\right) + E\left(w\Delta p\right)$$

$$1 \cdot \frac{1}{4} = 4 \cdot \frac{1}{16} + 0$$

As was the case in the first example, the sample size here is too small to estimate anything with significance. In this case, there is moreover an additional problem. The $\beta$ from the Price equation would be the right estimator to consider if the true model were to be model (A) from the previous example. If the true model is model (B), however, then with ever larger sample sizes, the data would get ever likelier to reveal that, which would imply that the $\beta$ from the Price equation in regression form cannot be interpreted meaningfully for any population size.

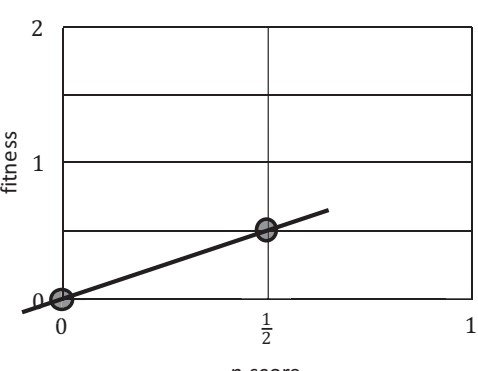 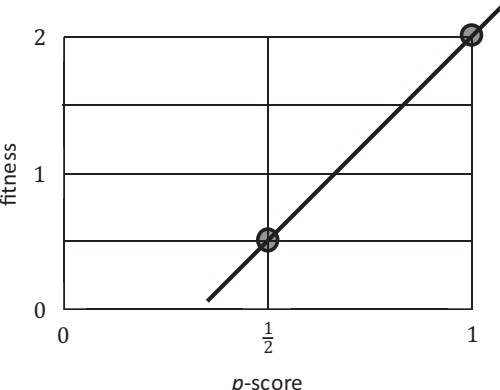

**Appendix 1—figure 1.** Dependence on the parent population. We assume that the population is infinitely large, so that the average fitness matches the expected values. On the left, a parent population, all members of which either have a $p$-score of 0 or a $p$-score of $\frac{1}{2}$. With expected fitnesses belonging to these $p$-scores of 0 and $\frac{1}{2}$, respectively, that results in a $\beta$ of 1. On the right, a parent population, all members of which either have a $p$-score of $\frac{1}{2}$, or a $p$-score of 1. With expected fitnesses belonging to these $p$-scores of $\frac{1}{2}$ and 2, respectively, that results in a $\beta$ of 3. In the literature, the dependence of the $\beta$ on the parent population is sometimes referred to as *dynamical insufficiency*. That, however, is not what this is in this case, as here, this is really a symptom of misspecification. If the model that generated the data would have been linear (model A), then, absent the noise, the $\beta$ would have been the same, regardless of the composition of the parent population. Dynamic (in)sufficiency is discussed in Section 6 of this appendix.

A symptom of this is that even in a situation without noise, or with an infinitely large population, the $\beta$ would vary, depending on the composition of the parent population. If $\beta_1 = 0$ and $\beta_2 = 2$, and the parent population would only contain individuals with a $p$-score of 0 or a $p$-score of $\frac{1}{2}$, then the $\beta$ in the Price equation in regression form would be 1 (see **Appendix 1—figure 1** on the left). This is true, regardless of the relative shares of individuals with a $p$-score of 0 and a $p$-score of $\frac{1}{2}$; all that is required is that the parent population contains individuals with a $p$-score of 0 and individuals with a $p$-score of $\frac{1}{2}$, and no individuals with a $p$-score of 1. If, on the other hand, the parent population would only contain individuals with a $p$-score of $\frac{1}{2}$ or a $p$-score of 1, then the $\beta$ in the Price equation in regression form would be 3 (see **Figure 1** on the right).

More generally, if we apply what we find at the very end of Section 5 to this example, then we find that the $\beta$ in the original Price equation in regression form equals $\beta = 2 \cdot \frac{\text{Cov}\left(p, p^2\right)}{\text{Var}\left(p\right)}$. This implies that the $\beta$ depends on the composition of the population. If we furthermore assume random mating, this simplifies to $\beta = 2 \cdot \left(\frac{1}{2} + \bar{p}\right)$, where $\bar{p}$ is the average $p$-score in the parent population (see Section 5). With $\bar{p}$ ranging from 0 to 1, this implies that the $\beta$ in the Price equation then ranges from 1 to 3.

This implies that if the true underlying model is model (B), and not model (A), we cannot interpret $\beta$ as the (linear) effect of these genes on fitness. There is, however, an alternative equation in which the regression coefficients would allow for a meaningful interpretation. In order to see what that would be, we go back to the derivation of the Price equation we went through earlier.

## Derivation of the generalized Price equation

In the standard derivation of the Price equation, we arrived at

$$\bar{w}\Delta\bar{p} = \text{Cov}\left(w, p\right) + E\left(w\Delta p\right) \tag{PE.C}$$

where

$$\text{Cov}\left(w, p\right) = \frac{1}{n}\sum_{i=1}^{n} w_i p_i - \frac{1}{n^2}\sum_{i=1}^{n} w_i \sum_{i=1}^{n} p_i$$

and

$$E\left(w\Delta p\right) = \frac{1}{n}\sum_{i=1}^{n} w_i \left(\frac{\sum_{j=1}^{m} P_{ij}}{w_i} - p_i\right)$$

Now assume that we know that the fitnesses are generated by a model that can be written as follows

$$w_i = \alpha + \sum_{r=1}^{R} \beta_r p_i^r + \varepsilon_i$$

where $\varepsilon_i$ is a noise term with expected value 0. If $R = 1$, this is the standard linear model we considered in Example 1.1; $w_i = \alpha + \beta_1 p_i + \varepsilon_i$. If $R = 2$, that is the quadratic model we are considering in Example 2.1; $w_i = \alpha + \beta_1 \cdot p_i + \beta_2 \cdot p_i^2 + \varepsilon_i$. Later, we will expand the set of models to all models that include a constant $\alpha$ and a linear term $\beta_1$, but for the derivation, it works perfectly fine to focus on this set of models.

If we have confidence in a model for some $R \geq 1$, we think that the true fitnesses are best described by

$$\hat{w}_i = \alpha + \sum_{r=1}^{R} \beta_r p_i^r = w_i - \varepsilon_i$$

for some choice of $\alpha$ and the $\beta_r$'s. Therefore, it will be useful to think of this as the vector of estimated fitnesses, which we will indicate by

$$\hat{w} = \left[\hat{w}_1, \ldots, \hat{w}_n\right]$$

Now for any $R \geq 1$, minimizing the sum of squared errors,

$$\sum_{i=1}^{n} \varepsilon_i^2 = \sum_{i=1}^{n} \left( w_i - \left( \alpha + \sum_{r=1}^{R} \beta_r p_i^r \right) \right)^2$$

will imply that the derivatives to $\alpha$ and to $\beta_1$ are set to 0 (along with all other derivatives to the parameters of the model). This has two implications.

Setting the derivative to $\alpha$ to 0 implies that we choose α such that

$$-2 \sum_{i=1}^{n} \left( w_i - \left( \alpha + \sum_{r=1}^{R} \beta_r p_i^r \right) \right) = 0$$

and therefore also such that

$$\sum_{i=1}^{n} w_i = \sum_{i=1}^{n} \left( \alpha + \sum_{r=1}^{R} \beta_r p_i^r \right) = \sum_{i=1}^{n} \hat{w}_i$$

In other words, although for each individual $i$ its actual number of offspring $w_i$ and its estimated fitness $\hat{w}_i$ may, and typically will, differ, if we arrive at $\hat{w}_i$ by minimizing least squares with a model that includes $\alpha$, then they do add up to the same total.

Setting the derivative to $\beta_1$ to 0 implies that we choose $\beta_1$ such that

$$\sum_{i=1}^{n} -2p_i \left( w_i - \left( \alpha + \sum_{r=1}^{R} \beta_r p_i^r \right) \right) = 0$$

and therefore

$$\sum_{i=1}^{n} p_i w_i = \sum_{i=1}^{n} p_i \left( \alpha + \sum_{r=1}^{R} \beta_r p_i^r \right) = \sum_{i=1}^{n} p_i \hat{w}_i$$

In other words, for each individual the $w_i$ and $\hat{w}_i$ may differ (and if they do, then also $p_i w_i$ and $p_i \hat{w}_i$ will be different), but if we arrive at $\hat{w}_i$ by minimizing least squares with a model that includes $\beta_1$, then the weighted sum $\sum_{i=1}^{n} p_i w_i$ will nonetheless equal $\sum_{i=1}^{n} p_i \hat{w}_i$.

These two observations imply that if we use the data to estimate such a model with ordinary least squares (OLS), then we can also write

$$\mathrm{Cov}\,(w,p) = \frac{1}{n} \sum_{i=1}^{n} w_i p_i - \frac{1}{n^2} \sum_{i=1}^{n} w_i \sum_{i=1}^{n} p_i = \frac{1}{n} \sum_{i=1}^{n} \hat{w}_i p_i - \frac{1}{n^2} \sum_{i=1}^{n} \hat{w}_i \sum_{i=1}^{n} p_i = Cov\,(\hat{w},p)$$

The Generalized Price equation in covariance form then becomes

$$\bar{w} \Delta \bar{p} = \mathrm{Cov}\,(\hat{w},p) + E\,(w \Delta p) \tag{GPE.C}$$

Before we move from the covariance form to the regression form, I would like to make two remarks. The first is that we can also replace $E\,(w\Delta p)$ with $E\,(\hat{w}\Delta p)$, because

$$E\,(w\Delta p) = \frac{1}{n} \sum_{i=1}^{n} \sum_{j=1}^{m} P_{ij} - \frac{1}{n} \sum_{i=1}^{n} w_i p_i = \frac{1}{n} \sum_{i=1}^{n} \sum_{j=1}^{m} P_{ij} - \frac{1}{n} \sum_{i=1}^{n} \hat{w}_i p_i = E\,(\hat{w}\Delta p)\,.$$

For the remainder of the appendix, it does not matter whether we replace $E\,(w\Delta p)$ with $E\,(\hat{w}\Delta p)$ or not.

The second point is that what I chose to call the 'Generalized Price equation in covariance form' (GPE.C) is not really a generalization of the Price equation in covariance form (PE.C). For any given dataset, $\mathrm{Cov}\,(w,p)$ is just a number, and it just so happens to be the case that $\mathrm{Cov}\,(\hat{w},p)$ returns the same number, whatever statistical model we use for determining what the model-estimated fitnesses $\hat{w}_i$ are (as long as the statistical model includes a constant and a linear term for the $p$-score). In other words, (PE.C) is not really nested in (GPE.C), so (GPE.C) is not a proper generalization.

There are two reasons why I chose this label nonetheless. The first is that equation (GPE.C) is general, in the sense that we can change the statistical model, and thereby change the vector of

model-estimated fitnesses $\hat{w} = [\hat{w}_1, \ldots, \hat{w}_n]$, but as long as we keep the constant and the linear term in the statistical model, the equation still applies. The second reason is that this is Step 1 in a sequence of three steps, the other two of which do produce proper generalizations. Step 2 goes from this equation in covariance form to the Generalized Price Equation in regression form, which is a proper generalization of the traditional Price equation in regression form. I will take this step immediately below. Step 3 goes from the Generalized Price Equation in regression form to the general version of Hamilton's rule, which is also a proper generalization of the classical Hamilton's rule. I will take this step in Appendix 2. By lack of a better term that captures the subtle way in which GPE.C is general, and the way in which it opens the door for the other two proper generalizations, I therefore stuck with this imperfect terminology.

Now for the derivation of the Generalized Price equation in regression form, we can focus on the term summarized as $\text{Cov}(\hat{w}, p)$, and fill in $\hat{\alpha} + \sum_{r=1}^{R} \hat{\beta}_r p_i^r$ for $\hat{w}_i$. We write $\hat{\alpha}$ and $\hat{\beta}_r$ to indicate that these are not just any $\alpha$ and $\beta_r$, but the ones we find by minimizing the sum of squared errors.

$$\frac{1}{n}\sum_{i=1}^{n}\left(\hat{\alpha} + \sum_{r=1}^{R}\hat{\beta}_r p_i^r\right)p_i - \frac{1}{n}\sum_{i=1}^{n}\left(\hat{\alpha} + \sum_{r=1}^{R}\hat{\beta}_r p_i^r\right)\frac{1}{n}\sum_{i=1}^{n}p_i$$

This can be shortened to

$$\frac{1}{n}\sum_{i=1}^{n}\left(\sum_{r=1}^{R}\hat{\beta}_r p_i^r\right)p_i - \frac{1}{n}\sum_{i=1}^{n}\left(\sum_{r=1}^{R}\hat{\beta}_r p_i^r\right)\frac{1}{n}\sum_{i=1}^{n}p_i$$

and if we change the summation order, this becomes

$$\sum_{r=1}^{R}\hat{\beta}_r\left(\frac{1}{n}\sum_{i=1}^{n}p_i^{r+1} - \frac{1}{n}\sum_{i=1}^{n}p_i^r\frac{1}{n}\sum_{i=1}^{n}p_i\right) = \sum_{r=1}^{R}\hat{\beta}_r\text{Cov}\left(p, p^r\right)$$

Therefore, if there is a model $w_i = \alpha + \sum_{r=1}^{R}\beta_r p_i^r$ that we believe generated the data, and for which we estimate the parameters using ordinary least squares (OLS), we can always write the following equation, which we will call the Generalized Price equation in regression form. It is general, in the sense that it produces different equations for different choices of $R$, and therefore for every model within this set of models, where fitnesses are polynomials. In Section 3, and in Section 5 of Appendix 2, we will see that by combining the Generalized Price equation in covariance form with an even richer set of models, we can generalize the Generalized Price equation in regression form even further.

$$\bar{w}\Delta\bar{p} = \sum_{r=1}^{R}\hat{\beta}_r\text{Cov}\left(p, p^r\right) + E\left(w\Delta p\right) \tag{GPE.R1}$$

If $R = 1$ we have the linear model, $w_i = \alpha + \beta_1 p_i + \varepsilon_i$, and since $\text{Cov}(p, p) = \text{Var}(p)$ the above equation then becomes

$$\bar{w}\Delta\bar{p} = \hat{\beta}_1\text{Var}\left(p\right) + E\left(w\Delta p\right) \tag{2.1}$$

If $R = 2$ we have the quadratic model, $w_i = \alpha + \beta_1 p_i + \beta_2 p_i^2 + \varepsilon_i$, and the above equation becomes

$$\bar{w}\Delta\bar{p} = \hat{\beta}_1\text{Var}\left(p\right) + \hat{\beta}_2\text{Cov}\left(p, p^2\right) + E\left(w\Delta p\right) \tag{2.2}$$

This moreover works for all models; the only thing that we need for this to work is that the statistical model includes a fixed term (for which we set the derivative of $\alpha$ to 0) and a linear term (for which we set the derivative of $\beta_1$ to 0). What is important to keep in mind is that if we apply the linear model to a dataset, and then we apply the quadratic model to the same dataset, then the $\hat{\beta}_1$'s will be different between them. This is also what we encounter in our everyday statistics; if we estimate the same parameters in one model versus the other, the estimates will change.

Which of these equations has a meaningful interpretation, and/or to what degree is perfectly in step with standard statistics. We may not have enough data to pick any model with any confidence,

nor to estimate any parameter with significance. In that case, none of these equations has a meaningful interpretation. We may, on the other hand, have an extremely large dataset that allows us to say with large confidence that the true model that generated them is, for instance, quadratic, and the sample size may also allow us to estimate the parameters with high accuracy. In that case, *Equation 2.2* above has a meaningful interpretation, but not *Equation 2.1*. Equations for $R \geq 3$ will then typically only differ marginally from *Equation 2.2*, with estimates for $\beta_i$, $i \geq 2$, close to 0.

The standard Price equation in regression form is *Equation 2.1* above. This equation only has a meaningful interpretation if we have confidence that the data are generated by a linear model ($R = 1$). Anything that one would be tempted to infer from the standard Price equation, without doing the statistics that confirms that the data are indeed generated by the linear model, is unwarranted. If a statistical test rejects model (A), then one cannot interpret the $\beta$ in the standard Price equation as the effect of these genes on fitness.

Many researchers are enthusiastic about the fact that the Price equation is an identity. This regularly leads to the belief that the Price equation 'cannot be wrong'. While it is correct to say that the Price equation is not wrong, in the sense that the left-hand side is equal to the right-hand side, it is important to also observe that this property is shared with other identities. In fact, we have found a range of different equations, one for every $R \geq 1$, and all of them are identities. If the 'not being wrong' would just be about whether the left and the right-hand side are equal, then these equations can be (very) different, while none of them are wrong. If the 'not being wrong' is to pertain to inferences one would draw from the Price equation, or interpretations of terms in it, then the multiplicity of equations implies that there *must* be some scope for being wrong, since these equations can differ from each other, leading to conclusions or interpretations that are at odds with each other.

## Summary

The Price equation in regression form can be generalized, in the sense that one can write a variety of Price-like equations for a variety of possible true models that may have generated the data. Which one we can interpret meaningfully with how much confidence depends on completely standard statistical considerations concerning model specification and significance of parameter estimates.

## Restrictions on the models imposed by the *p*-scores we consider

Before we go on to diploid species, there are a few more remarks that are worth making. The first is that I would like to reiterate that the genes that we are considering, and the genetic architecture, can limit which models make sense. If the *p*-score can only be 0 or 1, for instance, as it is in Section 2.1, then all models from the set we are considering here can be reduced to a model with $R = 1$. If we take

$$w_i = \alpha + \sum_{r=1}^{R} \beta_r p_i^r + \varepsilon_i$$

in combination with a binary *p*-score, then, since $p_i^r = p_i$ for all $r \geq 1$, we can rewrite this as

$$w_i = \alpha + \left( \sum_{r=1}^{R} \beta_r \right) p_i + \varepsilon_i$$

This brings us back to a model with $R = 1$, if we choose $\sum_{r=1}^{R} \beta_r$ as the coefficient for the linear term. Equivalently, if we look at the Generalized Price equation in regression form, then the fact that $p^r = p$ also implies that $\mathrm{Cov}\left(p, p^r\right) = \mathrm{Cov}\left(p, p\right) = \mathrm{Var}\left(p\right)$ for all $r$, and hence the Generalized Price equation in regression form can be written as

$$\bar{w}\Delta\bar{p} = \sum_{r=1}^{R} \hat{\beta}_r \mathrm{Cov}\left(p, p^r\right) + E\left(w\Delta p\right) = \left( \sum_{r=1}^{R} \hat{\beta}_r \right) \mathrm{Var}\left(p\right) + E\left(w\Delta p\right)$$

This is the Generalized Price equation in regression form for the linear model, if we, again, choose $\sum_{r=1}^{R} \beta_r$ as the coefficient for the linear term.

All of this is a formal way of saying that in this case, the regression coefficients only have meaning for the linear model, which is the model with $R = 1$.

## Alternative formulation and relation to the Gauss-Markov theorem

The second remark is that we can also write the general regression form as follows.

$$\bar{w}\Delta\bar{p} = \left(\sum_{r=1}^{R} \hat{\beta}_r \frac{\text{Cov}\left(p, p^r\right)}{\text{Var}\left(p\right)}\right) \text{Var}\left(p\right) + E\left(w\Delta p\right) \tag{GPE.R2}$$

The third remark is that for any model in the set of models we are considering here, if the error terms have a constant distribution with expectation 0 (that is, the distribution is the same for all values of the $p$-score), and the data are indeed generated by that model, then the Gauss-Markov theorem implies that the regression coefficients $\hat{\beta}_r$ for that model are Best Linear Unbiased Estimators, which means that they have minimal variance within the set of unbiased estimators. It is however quite possible that the distribution of numbers of offspring depends on the $p$-score in ways that do not satisfy this.

One possibility is that the distribution of the error terms does not have a constant variance. An example would be if the number of offspring is drawn from a binomial distribution with $N$ trials and success probability $\frac{\alpha + \beta_1 p_i}{N}$. This makes $\alpha + \beta_1 p_i$ the expected number of offspring, and it makes $w_i = \alpha + \beta_1 p_i + \varepsilon_i$ a well-specified model, but the variance of the error term now depends on $p_i$ (see also *Figure 1* in the Main Text, and *Figure 1* in Appendix 2). The Gauss-Markov theorem then no longer applies, and the regression coefficients $\hat{\beta}_r$ no longer have minimal variance. If the error terms still have expectation 0 – as they do in these examples – then they are however still unbiased. This implies that the setup of the Generalized Price equation is still useful. Applying OLS will still produce regression coefficients $\hat{\beta}_r$ that are unbiased, even though there may be other unbiased estimation procedures that reduce the variance of the estimators. Heteroskedasticity (the variance of the error term not being constant) also makes a difference for what appropriate statistical tests are.

It is, however, also possible that the data are generated by a model that does not fit the set of models specified above. This would imply a departure from the setup that allows us to use the Generalized Price equation, which is predicated on the statistical model including a constant term, and a term that is linear in the $p$-score.

## Higher moments and correlation coefficients

Finally, we can also consider all parameters included in the model in one go. If we write $\beta_0$ for $\alpha$, then the model can be written as

$$w_i = \sum_{r=0}^{R} \beta_r p_i^r + \varepsilon_i$$

Minimizing the sum of squared errors

$$\sum_{i=1}^{n} \varepsilon_i^2 = \sum_{i=1}^{n} \left(w_i - \sum_{r=0}^{R} \beta_r p_i^r\right)^2$$

means that we set the derivative to $\beta_r$ to 0 for $r = 0, 1, \ldots, R$. This implies that we choose $\beta_r$ such that

$$\sum_{i=1}^{n} -2p_i^r \left(w_i - \sum_{s=0}^{R} \beta_s p_i^s\right) = 0$$

and therefore

$$\sum_{i=1}^{n} p_i^r w_i = \sum_{i=1}^{n} p_i \sum_{s=0}^{R} \beta_s p_i^s = \sum_{i=1}^{n} p_i^r \hat{w}_i$$

This includes what we found above. If we take $r = 0$, this implies that $\sum_{i=1}^{n} w_i = \sum_{i=1}^{n} \hat{w}_i$ and if we take $r = 1$, this implies that $\sum_{i=1}^{n} p_i w_i = \sum_{i=1}^{n} p_i \hat{w}_i$.

Moreover, this straightforwardly implies that the following quantity is 0 for all $r = 0, 1, \ldots, R$.

$$\sum_{i=1}^{n} \varepsilon_i p_i^r - \frac{1}{n} \sum_{i=1}^{n} \varepsilon_i \sum_{i=1}^{n} p_i^r = \sum_{i=1}^{n} (w_i - \hat{w}_i) p_i^r - \frac{1}{n} \sum_{i=1}^{n} (w_i - \hat{w}_i) \sum_{i=1}^{n} p_i^r = 0$$

If we scale this expression to account for the number of observations and for the variance in $\varepsilon$ and the variance in $p^r$, then this becomes the *sample correlation coefficient* between $\varepsilon$ and $p^r$. This scaling is not important here, because, by construction of the minimization, we have just seen that the unscaled expression is already 0 for $r = 0, 1, \ldots, R$. If we are choosing between a model with exponents up to $R$ and exponents up to $R + 1$, then we can first minimize the sum of squared errors for the model with exponents up to $R$, and then consider whether the sample correlation coefficient between $\varepsilon$ and $p^{R+1}$ is a number consistent with what one would expect if the true model is the model with exponents up to $R$. This is where statistical tests come in. We will come back to this at the end of Section 4 of this appendix and at the end of Section 2 of Appendix 2.

## 3. The Price equation for diploid species

We will now repeat this for a diploid, sexually reproducing species. This is also a special case of the original Price equation, which allows for any possible ploidy. What we add to the original Price equation, besides a bit of matrix notation, is that we separate both generations into females and males. This will be useful for the examples.

### Parents: mothers and fathers

Let the parent generation consist of $k$ females, numbered from $i = 1$ to $i = k$, and $n - k$ males, numbered from $i = k + 1$ to $i = n$. This makes for a total of $n$ individuals in the parent generation.

### Offspring: girls and boys

Let the offspring generation consist of $l$ females, numbered from $j = 1$ to $j = l$, and $m - l$ males, numbered from $j = l + 1$ to $j = m$. This makes for a total of $m$ individuals in the parent generation.

### Which kids belong to which parents?

Everyone in the offspring generation is the offspring of one female and one male parent in the parent generation. Who are the parents of which offspring, we represent, again, with an offspring matrix $A$; let $A_{ij} = 1$ if $i$ is $j$'s parent, and $A_{ij} = 0$ if not. Every offspring having one mother and one father means that $\sum_{i=1}^{k} A_{ij} = 1$ for all $j$ (one mother per child), and $\sum_{i=k+1}^{n} A_{ij} = 1$ for all $j$ (one father per child). This also implies $\sum_{i=1}^{n} A_{ij} = 2$ for all $j$ (two parents per child).

### Parent fitnesses

The number of offspring of parent $i$, denoted by $a_i$, is found by summing over the offspring; $a_i = \sum_{j=1}^{m} A_{ij}$. The fitness of parent $i$ we get by dividing this by the ploidy; $w_i = \frac{1}{2} \sum_{j=1}^{m} A_{ij}$. Because everyone in the offspring generation has two parents in the parent population, the sum of these fitnesses of the parents must equal the number of individuals in the offspring generation: $\sum_{i=1}^{n} w_i = m$.

### Genes in the parent generation

The 'dose' of a gene (**Price, 1970**), or the *p*-score (**Grafen, 1985b**) per individual in the parent population, is denoted by $p_i, i = 1, \ldots, n$ for all parents. This dose may be restricted to be 0, $\frac{1}{2}$, or 1, if this is a counter for a specific allele, of which one can have 0, 1, or 2 copies. The idea of a dose or a *p*-score, however, is that this allows for any measure for the presence or absence of alleles all over the genome, and in particular genes that all contribute to a certain trait value. This means that $p_i$ can take values on some subset of the interval $[0, 1]$.

### Which genes are passed on

The matrix $P$ represents the *p*-score of the successful gametes that went into the offspring. This works as follows: $P_{ij} = 0$ if individual $i$ from the parent population is just not a parent of individual $j$ in the offspring population ($A_{ij} = 0$). If individual $i$ from the parent population is a parent of individual $j$ in the offspring population ($A_{ij} = 1$), then $P_{ij}$ is the *p*-score of the successful gamete. Because

every individual is composed of two gametes, this is a value in $\left[0, \frac{1}{2}\right]$. There may be restrictions on what $P_{ij}$ can be depending on what $p_i \in [0, 1]$ is; for instance, if $p_i$ counts alleles at one locus, as suggested above, then $p_i$ can be either 0, if the gamete does not carry it, or $\frac{1}{2}$ if it does. In this case, that would mean that if $A_{ij} = 1$ ($i$ is $j$'s parent), and the parent has 0 copies of the allele ($p_i = 0$), then the gametes cannot contain any copies either ($P_{ij} = 0$); if the parent has 1 copy of the allele ($p_i = \frac{1}{2}$), then the gametes can either contain 0 or 1 copy ($P_{ij} = 0$ or $P_{ij} = \frac{1}{2}$); and if the parent has 2 copies of the allele ($p_i = 1$), then the gametes must contain a copy as well ($P_{ij} = \frac{1}{2}$). More generally, without mutations, the genetic details would impose restrictions that will typically include that $p_i = 0$ will imply $P_{ij} = 0$ for all of its offspring, and that $p_i = 1$ implies $P_{ij} = \frac{1}{2}$ for all of its offspring (i.e. for all $j$ for which $A_{ij} = 1$).

### Genes in the offspring population

The $p$-score of individual $j$ in the offspring generation is the sum of the $p$-scores of its gametes; $p_j' = \sum_{i=1}^{n} P_{ij}$.

### Change

The average $p$-score in the parent generation is $\bar{p} = \frac{\sum_{i=1}^{n} p_i}{n}$. The average $p$-score in the offspring generation is $\bar{p}' = \frac{\sum_{j=1}^{m} p_j'}{m}$.

### Derivation of the Price equation

The key ingredient in the derivation of the Price equation is that one can calculate the sum of the $p$-scores in the offspring generation in two ways.

1. One can first calculate the $p$-score of every individual in the offspring generation by adding the $p$-scores of the incoming gametes of each offspring individual ($p_j' = \sum_{i=1}^{n} P_{ij}$ for every offspring $j$), and then sum these over all the members of the offspring generation. That amounts to $\sum_{j=1}^{m} p_j' = \sum_{j=1}^{m} \sum_{i=1}^{n} P_{ij}$.
2. One can also first add up the $p$-scores of the successful outgoing gametes of the parents ($\sum_{j=1}^{m} P_{ij}$ for parent $i$) and then sum over the parents. This amounts to a switch in summation order from $\sum_{j=1}^{m} \sum_{i=1}^{n} P_{ij}$ to $\sum_{i=1}^{n} \sum_{j=1}^{m} P_{ij}$. This we can always do, because $n$ and $m$ are finite.

Despite the differences (combinations of $A$ and $P$ that fit the asexual model setup of Section 2 do not fit the sexual setup in this section, and vice versa), the remainder of the derivation is the exact same as the derivation for asexual reproduction in Section 2. Also, the derivation of the generalized Price equation is the same.

In order to illustrate the Price equation for a diploid, sexually reproducing species, we have three more examples. These examples, moreover, further illustrate that the interpretation of the Price equation is bound by the exact same considerations that confine interpreting parameter estimates in normal statistics.

### Example 3.1: an allele with a fixed effect on fitness

We begin with the equivalent of Example 1.1. The only difference between the setting with asexual reproduction from Example 1.1 and the setting with sexual reproduction is that in the latter, there is scope for model misspecification. We nonetheless start by assuming that we know what the underlying model of fitness is, and that it is linear:

$$w_i = \alpha + \beta p_i + \varepsilon_i \tag{A}$$

Here, the $p$-score $p_i$ can be 0, $\frac{1}{2}$, or 1, depending on whether the individual has 0, 1, or 2 copies of a certain allele.

Then we take a parent population consisting of two mothers and two fathers. The first mother has two copies of the allele, the second has none, the first father has one copy of the allele, the second has none. In other words, $p = \left[1, 0, \frac{1}{2}, 0\right]$. Mother 1 has two kids with father 1 and one with father 2, mother 2 has one kid with father 2, which makes $w = \left[\frac{3}{2}, \frac{1}{2}, 1, 1\right]$ and

$$A = \begin{bmatrix} 1 & 1 & 1 & 0 \\ 0 & 0 & 0 & 1 \\ 1 & 1 & 0 & 0 \\ 0 & 0 & 1 & 1 \end{bmatrix}$$

The transmission matrix is

$$P = \begin{bmatrix} \frac{1}{2} & \frac{1}{2} & \frac{1}{2} & 0 \\ 0 & 0 & 0 & 0 \\ 0 & \frac{1}{2} & 0 & 0 \\ 0 & 0 & 0 & 0 \end{bmatrix}$$

This makes $p' = \left[\frac{1}{2}, 1, \frac{1}{2}, 0\right]$, which we get by summing over the columns. The ingredients of the Price equation in regression form then are: $\bar{w} = 1$, as there are 4 parents and 4 offspring; $\Delta\bar{p} = \frac{1}{8}$, because the average $p$-score went up from $\frac{3}{8}$ to $\frac{1}{2}$; $E\left(w\Delta p\right) = 0$, because the only parent with a non-binary $p$-score had a $p$-score of $\frac{1}{2}$, and this parent had one successful gamete without, and one with the allele; $\text{Cov}\left(w,p\right) = \frac{1}{8}$; and $\text{Var}\left(p\right) = \frac{11}{64}$. In line with the insight from the Generalized Price equation, we give the $\beta$ a hat, indicating that this is what the estimator would be for the true $\beta$, under the assumption that the true model is in fact linear. The $\beta$ in the original Price equation is defined as $\frac{\text{Cov}\left(w,p\right)}{\text{Var}\left(p\right)}$, which is indeed the estimator of $\beta$, under the assumption of the data are generated by the linear model, so for the original Price equation, this does not change how the $\beta$ is calculated. For this example, this is $\hat{\beta} = \frac{8}{11}$. The calculations are in Detailed Calculations 1.3 at the end of Appendix 1.

The Price equation in regression form, and the Generalized Price equation in regression form under the linear model, therefore, is:

$$\bar{w}\Delta\bar{p} = \hat{\beta} \cdot \text{Var}\left(p\right) + E\left(w\Delta p\right)$$

$$1 \cdot \frac{1}{8} = \frac{8}{11} \cdot \frac{11}{64} - 0$$

As in previous examples, this sample size is not large enough to estimate the true $\beta$ with any significance. Also, the sample size is not large enough to test if the model $w_i = \alpha + \beta p_i + \varepsilon_i$ is accurately specified. In this case, the $\hat{\beta}$ therefore cannot be interpreted as the fitness effect of the gene.

One can, however, also imagine a vector $p$, and matrices $A$ and $P$, that together represent a much larger population, and therefore a much larger dataset. If this results in an estimate of the true $\beta$ that is significantly different from 0; if a statistical test with sufficient power does not reject model (A) in favor of richer models; and if the data are consistent with the expectation of the noise term $\varepsilon_i$ being zero, then the $\beta$ in the Price equation does have a meaningful interpretation; it is an unbiased estimator of the true $\beta$, which is the effect of the $p$-score on fitness.

The main observation here is the same as with Examples 2.1 and 2.2. The extent to which we can give a meaningful interpretation to the right-hand side of the Price equation is determined completely by standard statistical considerations. With small sample sizes, $\hat{\beta}$ will typically not be significant, even if the true $\beta$ would be different from 0. Also, tests for linearity will have low power and will not reject linearity, even if the true model would not be linear. Larger sample sizes allow us to infer more from the data, but also then, the interpretation is bound by the rules of statistics. Any interpretation will have to be done with the exact same reservations that one would have with standard statistics.

## Summary

If the $\hat{\beta}$ is significant, and statistical tests with sufficient power do not reject model (A), then the right-hand side of the original Price equation in regression form (which is the right-hand side of the

Generalized Price equation in regression form for the model $w_i = \alpha + \beta p_i + \varepsilon_i$) has a meaningful interpretation; $\hat{\beta}$ is the estimated (linear) effect of the $p$-score on fitness.

## Example 3.2: heterozygote advantage / dominance

In this example, we begin with a description of a life cycle in which heterozygotes have an advantage. The quadratic model that describes this can, however, also be used to describe dominance for alternative choices of the parameters.

Suppose the $p$-score can be 0, $\frac{1}{2}$, or 1, and assume that for any of those $p$-scores there are equally many females as there are males. The life cycle is as follows. Those with a $p$-score of 0 all die. Those with a $p$-score of $\frac{1}{2}$ all survive. Half of individuals with a $p$-score of 1 survive, the other half die. The female and male survivors then are randomly matched in pairs, and all of those pairs produce 3 kids.

The fitnesses for these three $p$-scores therefore are 0, $\frac{3}{2}$, and $\frac{3}{4}$. Another way to describe that is:

$$w_i = \alpha + \beta_1 p_i + \beta_2 p_i^2 + \varepsilon_i \tag{B}$$

where $\varepsilon_i$ is a noise term, and $\alpha = 0$, $\beta_1 = \frac{21}{4}$, and $\beta_2 = -\frac{9}{2}$.

In general $w^0 = \alpha$, $w^{\frac{1}{2}} = \alpha + \frac{1}{2}\beta_1 + \frac{1}{4}\beta_2$ and $w^1 = \alpha + \beta_1 + \beta_2$. If fitness is linear, that would mean $\beta_2 = 0$. Heterozygote advantage would be $-2\beta_1 < \beta_2 < -\frac{2}{3}\beta_1$ (which can only hold for $\beta_1 > 0$). A dominant gene would make $w^{\frac{1}{2}}$ and $w^1$ equal, and therefore $\beta_2 = -\frac{2}{3}\beta_1$; a recessive gene would meke $w^0$ and $w^{\frac{1}{2}}$ equal, and therefore $\beta_2 = -2\beta_1$.

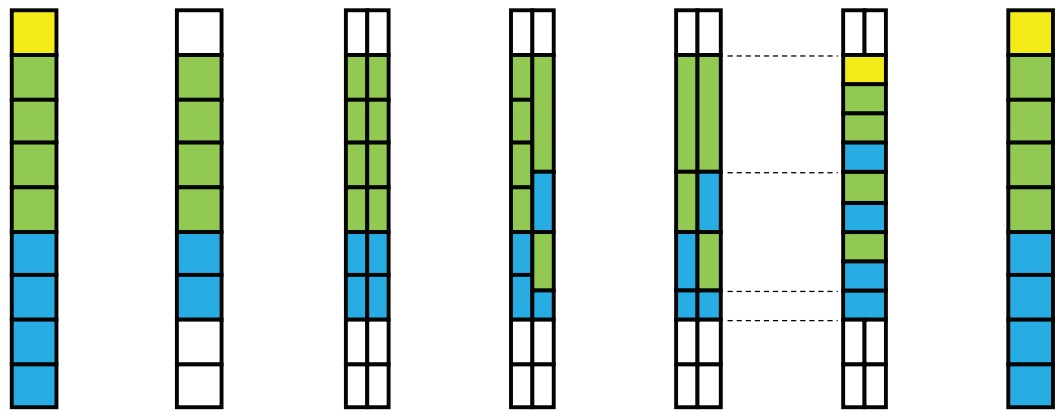

**Appendix 1—figure 2.** A graphical representation of the life cycle for the equilibrium state of the population. *Step 1: differential survival.* All individuals in the parent generation with a $p$-score of 0 die, and half of those with a $p$-score of 1 do. None of the parents with a $p$-score of $\frac{1}{2}$ die, so the heterozygote has the highest fitness. *Step 2: mothers and fathers.* Half of all surviving parents are female, and half are male. *Step 3 and 4: random matching.* The parents are randomly matched. *Step 5: fair meiosis.* In each pair, the offspring inherits one allele from either parent. *Step 6: scaling up.* Parent pairs have on average three kids.

Then we can assume an infinite population, and we can assume that the shares in the parent population are as follows: 1 out of 9 parents has a $p$-score of 0; 4 out of 9 parents have a $p$-score of $\frac{1}{2}$; and 4 out of 9 parents have a $p$-score of 1. These frequencies are chosen to be stable; the life cycle as described above creates an offspring generation that is identical to the parent population. *Appendix 1—figure 2* above illustrates all steps in the calculations, which are in A.4 at the end of the Appendices, and that confirm that this is indeed a fixed point of the dynamics.

One can do this for non-equilibrium parent populations too, in which case the numbers will change, but it might be instructive to first focus on this equilibrium population state. For this state, the ingredients of the Price equation in regression form are: $\bar{w} = 1$, as the number of offspring equals the number of parents; $\Delta\bar{p} = 0$, because the frequencies do not change; $E(w\Delta p) = 0$, because of fair meiosis in an infinite population; $\mathrm{Cov}(w, p) = 0$, $\mathrm{Var}(p) = \frac{1}{9}$ and $\hat{\beta} = 0$. The calculations are in Detailed Calculations 1.4 at the end of Appendix 1.

The Price equation, and the Generalized Price equation in regression form under the linear model, for the equilibrium population state therefore is:

$$\bar{w}\Delta\bar{p} = \hat{\beta} \cdot \mathrm{Var}\,(w,p) + E\,(w\Delta p)$$

$$1 \cdot 0 = 0 \cdot \frac{1}{9} + 0$$

If we were to interpret the $\hat{\beta}$ as the effect of the gene on fitness, then the fact that $\hat{\beta} = 0$ would suggest that the gene has no effect on fitness. This would be wrong because which $p$-score an individual has clearly matters for fitness; it is just that the effect is not linear. At this particular frequency, it happens to be the case that the advantageous effect of an additional allele compared to the homozygote with a $p$-score of 0, and the disadvantageous effect of an additional allele compared to the homozygote with a $p$-score of $\frac{1}{2}$ balance out. If the data are generated by model (B), as we assume they are in this example, then one cannot interpret the $\hat{\beta}$ from the Price equation as the effect of the gene on fitness, which one can do if the data were generated by model (A). Moreover, as in example 2.2, the $\hat{\beta}$ from the Price equation in regression form will depend on the composition of the parent population; for all parent populations that are not the equilibrium shares given above, the $\hat{\beta}$ will be non-zero and can be positive or negative. This is a symptom of misspecification. (It can also be a reflection of frequency dependence, as we will see in the next example, but here fitnesses are not frequency dependent).

With an infinitely large population, the Price equation in regression form would have a meaningful interpretation, if the true model were model (A). If the true model is model (B), however, even if the parent population is assumed to be infinitely large (which removes statistical considerations concerning sample size), the $\hat{\beta}$ in the Price equation does not have a meaningful interpretation, due to misspecification of the model.

## Summary

If the fitnesses depend on the $p$-scores in a quadratic way, as they do in this example, then, provided that we have sufficiently much data, a standard statistical exercise is likely to uncover the true model and estimate its parameters with some accuracy and confidence. In this model, the fitnesses do not depend on the composition of the parent population, which means that for instance the expected values of the estimators for $\beta_1$ and $\beta_2$ do not depend on the composition of the parent population. The $\hat{\beta}$ that we are likely to find in the original Price equation in regression form, which is the Generalized Price equation in regression form for the model $w_i = \alpha + \beta p_i + \varepsilon_i$, on the other hand, does depend on the composition of the parent population. This is caused by misspecification, because in this example, we assume that the true model is $w_i = \alpha + \beta_1 p_i + \beta_2 p_i^2 + \varepsilon_i$.

## Example 3.3: sex-determining genes

In this example, the $p$-score will represent sex-determining genes. If we think of females as $XX$ and males as $XY$, and we count the number of $X$'es, then the $p$-score for females is 1 and for males it is $\frac{1}{2}$ (everything would also work if the $p$-score would count the number of $Y$'s, in which case the $p$-score would be 0 for females and $\frac{1}{2}$ for males). We can then write the Price equation, just based on the numbers of females and males in both generations.

The ingredients of the Price equation then are: $\bar{w} = \frac{m}{n}$, which is the size of the offspring generation over the size of the parent population; $\Delta\bar{p} = \frac{1}{2}\left(\frac{l}{m} - \frac{k}{n}\right)$, which is one half times the difference between the frequency of females in the offspring generation and the parent generation; $E\,(w\Delta p) = \frac{1}{2n}\left(l - \frac{1}{2}m\right)$, which does not depend on the sex ratio in the parent population; $\mathrm{Cov}\,(w,p) = \frac{m}{2n}\left(\frac{1}{2} - \frac{k}{n}\right)$, $\mathrm{Var}\,(p) = \frac{k}{4n}\left(1 - \frac{k}{n}\right)$ and $\hat{\beta} = m \cdot \frac{n-2k}{k(n-k)}$. The calculations are in Detailed Calculations 1.5 at the end of Appendix 1.

The Price equation in regression form therefore is:

$$\bar{w}\Delta\bar{p} = \hat{\beta} \cdot \mathrm{Var}\,(p) + E\,(w\Delta p)$$

$$\frac{m}{n} \cdot \frac{1}{2}\left(\frac{l}{m} - \frac{k}{n}\right) = m \cdot \frac{n-2k}{k(n-k)} \cdot \frac{1}{4}\frac{k}{n}\left(1 - \frac{k}{n}\right) + \frac{1}{2n}\left(l - \frac{1}{2}m\right)$$

The most interesting observation here is that the $\hat{\beta}$ is determined entirely by the composition of the parent population (the $k$ and the $n$) and the number of offspring $m$. Just by the mechanics of sexual reproduction, if the number of females is smaller than the number of males ($k < n - k \rightarrow 2k < n$), the $\hat{\beta}$ will be positive, whereas it will be negative if the number of females is larger than the number of males.

### Summary

In this example, the fitnesses are frequency dependent. Recovering this in a statistical exercise requires observations for different compositions of the parent population, and writing the Price equation in regression form for one composition of the parent population only is not informative.

These examples for a diploid, sexually reproducing species explore a bit more what the scope is for possible ways in which fitnesses can depend on $p$-scores. The message however remains the same. The standard Price equation in regression form is just one out of a range of Price-like equations. There is one for every model, and the standard Price equation is the one for the linear model. Which of those equations has a meaningful interpretation, and to what extent, depends completely on what statistics has to say about the possible underlying true models that could have generated the data.

## 4. The Price equation with a *p*-score and a *q*-score

Here we take the same setup as in Section 2, but now we add information about genes that are not included in the $p$-score, while they may have an effect on fitness. Besides a vector $p$ of $p$-scores for the parents, a vector $p'$ of $p$-scores for the offspring, and a transmission matrix $P$, we now also have a vector $q$ of $q$-scores for the parents, a vector $q'$ of $q$-scores for the offspring, and a transmission matrix $Q$, all of which satisfy the same properties that a genetic system of transmission imposes (such as, for example, that if $A_{ij} = 0$, then $Q_{ij} = 0$).

In this setting, the Price equation tracks changes in the average $p$-score, while these $p$-scores are not the only genetic determinant of fitness. This enriches the set of possible true models, and therefore the set of possible Price equations, and it allows us to give additional illustrations of how there is no way around classical statistics if our aim is to find the right Price-like equation in this multitude of options.

### Derivation of the Generalized Price equation in regression form for this richer set of models

With this richer setup, allowing for $p$-scores and $q$-scores, we can choose a richer set of models that we consider. Assume, therefore, that we know that the fitnesses are generated by some model that can be written as follows:

$$w_i = \sum_{(k,l) \in E} \beta_{k,l} p_i^k q_i^l + \varepsilon_i$$

where $\varepsilon_i$ is a noise term with expected value 0, and $E$ is some set that indicates which coefficients $\beta_{k,l}$ are nonzero. We assume that $(0,0)$ and $(1,0)$ are included. If those are the only ones in $E$, then this is the standard linear model we considered in Example 2.1; $w_i = \alpha + \beta p_i + \varepsilon_i$, with $\beta_{0,0} = \alpha$ and $\beta_{1,0} = \beta$. If $E = \{(0,0),(1,0),(2,0)\}$, that is the quadratic model we are considering in Example 2.2; $w_i = \alpha + \beta_1 p_i + \beta_2 p_i^2 + \varepsilon_i$, with $\beta_{0,0} = \alpha$, $\beta_{1,0} = \beta_1$, and $\beta_{2,0} = \beta_2$. If $E = \{(0,0),(1,0),(0,1),(1,1)\}$, that is a model in which both the $p$-score and the $q$-score matter, and in which there is an interaction term; $w_i = \beta_{0,0} + \beta_{1,0} p_i + \beta_{0,1} q_i + \beta_{1,1} p_i q_i + \varepsilon_i$.

If we repeat what we did in Section 2, choosing parameter values that minimize the sum of squared errors, where $(0,0)$ and $(1,0)$ are included, we arrive at the same generalized Price equation in covariance form.

$$\bar{w}\Delta\bar{p} = \text{Cov}\left(\hat{w}, p\right) + E\left(w\Delta p\right) \tag{GPE.C}$$

Also, the continuation from the covariance form to the regression form is almost the same. For this, we focus on the term summarized as $\text{Cov}\left(\hat{w}, p\right)$.

$$\text{Cov}\left(\hat{w}, p\right) = \frac{1}{n} \sum_{i=1}^{n} \hat{w}_i p_i - \frac{1}{n^2} \sum_{i=1}^{n} \hat{w}_i \sum_{i=1}^{n} p_i$$

Here we fill in $\sum_{(k,l) \in E} \hat{\beta}_{k,l} p_i^k q_i^l$ for $\hat{w}_i$

$$\frac{1}{n} \sum_{i=1}^{n} \left( \sum_{(k,l) \in E} \hat{\beta}_{k,l} p_i^k q_i^k \right) p_i - \frac{1}{n} \sum_{i=1}^{n} \left( \sum_{(k,l) \in E} \hat{\beta}_{k,l} p_i^k q_i^l \right) \frac{1}{n} \sum_{i=1}^{n} p_i$$

Because $p_i^0 q_i^0 = 1$, which makes $\frac{1}{n} \sum_{i=1}^{n} p_i^0 q_i^0 p_i = \frac{1}{n} \sum_{i=1}^{n} p_i = p_i^0 q_i^0 \frac{1}{n} \sum_{i=1}^{n} p_i$, we can leave out the first term, which means this is also

$$\frac{1}{n} \sum_{i=1}^{n} \left( \sum_{(k,l) \in E \backslash (0,0)} \hat{\beta}_{k,l} p_i^k q_i^l \right) p_i - \frac{1}{n} \sum_{i=1}^{n} \left( \sum_{(k,l) \in E \backslash (0,0)} \hat{\beta}_{k,l} p_i^k q_i^l \right) \frac{1}{n} \sum_{i=1}^{n} p_i$$

and if we change the summation order, this becomes

$$\sum_{(k,l) \in E \backslash (0,0)} \hat{\beta}_{k,l} \left( \frac{1}{n} \sum_{i=1}^{n} p_i^{k+1} q_i^l - \frac{1}{n} \sum_{i=1}^{n} p_i^k q_i^l \frac{1}{n} \sum_{i=1}^{n} p_i \right) = \sum_{(k,l) \in E \backslash (0,0)} \hat{\beta}_{k,l} \text{Cov}\left(p, p^k q^l\right)$$

Therefore, if there is any model $w_i = \sum_{(k,l) \in E} \beta_{k,l} p_i^k q_i^l$ that we believe generated the data, and for which we estimate the parameters using ordinary least squares (OLS), we can always write the following equation, which we call the Generalized Price equation in regression form, for this larger set of models.

$$\bar{w} \Delta \bar{p} = \sum_{(k,l) \in E} \hat{\beta}_{k,l} \text{Cov}\left(p, p^k q^l\right) + E\left(w \Delta p\right) \tag{GPE.R3}$$

Because $\text{Cov}\left(p, p^0 q^0\right) = \text{Cov}\left(p, 1\right) = 0$, it, again, does not matter if we leave $(0,0)$ in or take it out. This, first of all, encompasses the models that were included in Section 2.

Choosing $E = \left\{ (0,0), (1,0) \right\}$ here is the same as choosing $R = 1$ in Section 2; both give us the linear model, $w_i = \alpha + \beta_{1,0} p_i + \varepsilon_i$, and since $\text{Cov}\left(p, p\right) = \text{Var}\left(p\right)$ the above equation becomes

$$\bar{w} \Delta \bar{p} = \hat{\beta}_{1,0} \text{Var}\left(p\right) + E\left(w \Delta p\right) \tag{1.4.1}$$

Choosing $E = \left\{ (0,0), (1,0), (2,0) \right\}$ here is the same as choosing $R = 2$ in Section 2; both give us the quadratic model, $w_i = \beta_{0,0} + \beta_{1,0} p_i + \beta_{2,0} p_i^2 + \varepsilon_i$, and the above equation becomes

$$\bar{w} \Delta \bar{p} = \hat{\beta}_{1,0} \text{Var}\left(p\right) + \hat{\beta}_{2,0} \text{Cov}\left(p, p^2\right) + E\left(w \Delta p\right) \tag{1.4.2}$$

This setup also allows for models that are not included in Section 2.

Choosing $E = \left\{ (0,0), (1,0), (0,1), (1,1) \right\}$, for instance, gives us a model with an interaction term between the p-score and the q-score, $w_i = \beta_{0,0} + \beta_{1,0} p_i + \beta_{0,1} q_i + \beta_{1,1} p_i q_i + \varepsilon_i$, and the above equation becomes

$$\bar{w} \Delta \bar{p} = \hat{\beta}_{1,0} Var\left(p\right) + \hat{\beta}_{0,1} \text{Cov}\left(p, q\right) + \hat{\beta}_{1,1} \text{Cov}\left(p, pq\right) + E\left(w \Delta p\right) \tag{1.4.3}$$

This works for all models; the only thing that we need for this to work is that the statistical model includes a fixed term (for which we set the derivative to its coefficient to 0) and a linear term (for which we set the derivative to its coefficient to 0).

Also here, we can rewrite the regression form:

$$\bar{w}\Delta\bar{p} = \left( \sum_{(k,l)\in E} \hat{\beta}_{k,l} \frac{\mathrm{Cov}\left(p, p^k q^l\right)}{\mathrm{Var}\left(p\right)} \right) \mathrm{Var}\left(p\right) + E\left(w\Delta p\right) \tag{GPE.R4}$$

## Summary

The Price equation in regression form can be generalized, also in a setting with two sets of genes, one captured by the *p*-score, the changes in which we track, and one by a *q*-score, the changes in which we do not track. The larger variety of possible true models, compared to models with a *p*-score only, is reflected in the larger variety of Price-like equations. Which one we can interpret meaningfully with how much confidence depends, as before, on completely standard statistical considerations concerning model specification and significance of parameter estimates.

## Minimizing the sum of squared errors and correlation coefficients
### Minimizing the sum of squared errors

$$\sum_{i=1}^{n} \varepsilon_i^2 = \sum_{i=1}^{n} \left( w_i - \sum_{(k,l)\in E} \beta_{k,l} p_i^k q_i^l \right)^2$$

means that we set the derivative to $\beta_{k,l}$ to 0 for all $(k,l)\in E$. This implies that we choose $\beta_{k,l}$ such that

$$\sum_{i=1}^{n} -2 p_i^k q_i^l \left( w_i - \sum_{(r,s)\in E} \beta_{r,s} p_i^r q_i^s \right) = 0$$

and therefore we find that

$$\sum_{i=1}^{n} p_i^k q_i^l w_i = \sum_{i=1}^{n} p_i^k q_i^l \sum_{(r,s)\in E} \beta_{r,s} p_i^r q_i^s = \sum_{i=1}^{n} p_i^k q_i^l \hat{w}_i$$

This also implies that the following quantity is 0 for all $(k,l)\in E$.

$$\sum_{i=1}^{n} \varepsilon_i p_i^k q_i^l - \frac{1}{n} \sum_{i=1}^{n} \varepsilon_i \sum_{i=1}^{n} p_i^k q_i^l = \sum_{i=1}^{n} \left( w_i - \hat{w}_i \right) p_i^k q_i^l - \frac{1}{n} \sum_{i=1}^{n} \left( w_i - \hat{w}_i \right) \sum_{i=1}^{n} p_i^k q_i^l = 0$$

If we scale the last expression, to account for the number of observations, and for the variance in $\varepsilon$ and the variance in $p^k q^l$, then this becomes the *sample correlation coefficient* between $\varepsilon$ and $p^k q^l$. This scaling is not important here, because, by construction of the minimization, we have just seen that the unscaled expression is already 0 for all $(k,l)\in E$. If we are choosing between including a term $(k,l)$ and excluding it from the statistical model, then we can first minimize the sum of squared errors for the model without the term $(k,l)$, and then consider whether the sample correlation coefficient between $\varepsilon$ and $p^k q^l$ is a number consistent with what one would expect if the true model is the model without $p^k p^l$. If it is, then we do not include the term $(k,l)$. If it is not, then we do include the term $(k,l)$. Again, this is the domain of statistical testing. We will return to this at the end of Section 2 in Appendix 2.

## 5. Modeling

Part of statistics is having to choose between statistical models, based on data. That means that the best we can do is to maximize the probability that we pick the right model. Modeling, on the other hand, relieves us of that uncertainty, as the model is just whatever we choose it to be. That makes it even more clear that there is something to be gained from matching the model that we happen to have chosen with the right Price-like equation.

While we have seen that the literature is somewhat sloppy on how to use the Price equation on data, it is also not always very specific or precise on how the Price equation is to be used on models. In this section, we will first circumvent all complications by assuming that we have a deterministic model, without noise. This is of course totally unrealistic, but it allows us to make a point about over- and underspecification of Price-like equations in a modeling context. This point is similar to the point about over- or underspecification in a statistical context. Moreover, this point also carries over to a setting where we assume an infinitely large population. This assumption allows us to treat the population dynamics as deterministic, even if there is uncertainty at the individual level about the number of offspring. In later subsections, we will drop those assumptions and show how even well-specified Price-like equations have limitations in describing properties of dynamics in stochastic models that do not assume that the population size is large enough for models with infinite population size to be a good approximation.

For simplicity, we begin assuming asexual reproduction. All points made below carry over to a setting with sexual reproduction but are easier made without. At the end of this section, we will revisit heterozygote advantage, for which we do switch to sexual reproduction.

## Model A

The first model is $w_i = \alpha + \beta_{1,0}p_i$. The absence of a noise term severely limits what values $\alpha$ and $\beta_{1,0}$ can take. If $p_i$ is binary, then the fact that numbers of offspring are integers means that we can also only choose integers for $\alpha$ and $\beta_{1,0}$. If $p_i$ is restricted to be 0, $\frac{1}{2}$ or 1, then $\alpha$ can still only be an integer, while $\beta_{1,0}$ needs to be an even number. Larger sets of possible $p$-scores further reduce what values for $\beta_{1,0}$ we can choose. If we assume infinite populations, with uncertainty at the individual level, but deterministic population dynamics, these restrictions are lifted, but for now, we just assume that they are satisfied.

A deterministic model implies that for any given parent population, represented by a vector of $p$-scores $p$, there is only one possible transition to an offspring population. For that one transition, we can use the Generalized Price equation in regression form, where the sum of squared errors is minimized and equal to 0, at $\hat{\alpha} = \alpha$ and $\hat{\beta}_{1,0} = \beta_{1,0}$. Here we simply use the matching statistical model, which in this case only needs to include $\alpha$ and $\beta_{1,0}$. The matching Price-like equation for this model is

$$\bar{w}\Delta\bar{p} = \hat{\beta}_{1,0}\text{Var}\left(p\right)$$

where $\hat{\beta}_{1,0}$ does not depend on which $p$ we choose and equals $\beta_{1,0}$ for any composition of the parent population. This equation can therefore serve as a useful summary of the dynamics induced by the model.

## Model B

The second model is $w_i = \alpha + \beta_{1,0}p_i + \beta_{0,1}q_i$. Again, the deterministic nature of the model puts restrictions on the values we can choose for $\alpha$, $\beta_{1,0}$, and $\beta_{0,1}$, but those restrictions would be lifted if we assume infinite populations, with uncertainty at the individual level, but deterministic dynamics at the population level.

As before, we can use the Generalized Price equation in regression form, where the sum of squared errors is minimized and equal to 0, at $\hat{\alpha} = \alpha$, $\hat{\beta}_{1,0} = \beta_{1,0}$, and $\hat{\beta}_{0,1} = \beta_{0,1}$. The matching Price-like equation for this model is

$$\bar{w}\Delta\bar{p} = \hat{\beta}_{1,0}\text{Var}\left(p\right) + \hat{\beta}_{0,1}\text{Cov}\left(p,q\right) \tag{5.B}$$

in which $\hat{\beta}_{1,0}$ and $\hat{\beta}_{0,1}$ do not depend on the choice of $p$ and $q$, and are equal to $\beta_{1,0}$ and $\beta_{1,0}$, respectively, for any composition of the parent population.

## Model C

The third model is $w_i = \alpha + \beta_{1,0}p_i + \beta_{0,1}q_i + \beta_{1,1}p_iq_i$. Here we jump over the details and go straight to the matching Price-like equation for this model, which is

$$\bar{w}\Delta\bar{p} = \hat{\beta}_{1,0}\text{Var}\left(p\right) + \hat{\beta}_{0,1}\text{Cov}\left(p,q\right) + \hat{\beta}_{1,1}\text{Cov}\left(p,pq\right) \tag{5.C}$$

These three examples together illustrate that, also when modeling, there is a multitude of Price-like equations that capture properties of the population dynamics implied by the model.

## Mismatching models and Price-like equations

It is important to note that also in this modeling setting, all of these Price-like equations remain identities, and they can be combined, not just with the model with respect to which they minimize the sum of squared errors, but also with all other, possibly richer models, as long as they have a constant term, and a linear term for $p_i$. As a symptom of such a mismatch, the coefficients in the Price-like equation stop being constants; they will vary with the composition of the parent population (the $p$ and, if present, the $q$ vector) if they are applied to other models. Another symptom is that the coefficients in the Price-like equation stop matching the according coefficients in the model.

### Applying the Generalized Price equation for Model A to Model B

If we apply the Generalized Price equation in regression form for Model A to Model B, then the sum of squared errors can no longer be reduced to be 0 and is minimized at $\hat{\beta}_{1,0} = \beta_{1,0} + \beta_{0,1}\frac{\text{Cov}(p,q)}{\text{Var}(p)}$. The fact that the sum of squared errors is positive is noteworthy. The model is deterministic and has no errors, and yet the sum of squared errors in the Price-like equation is larger than 0. By lack of actual errors, this can only be caused by misspecification of the model. The value for $\hat{\beta}_{1,0}$ at which the sum of squared errors is minimized if we apply the Generalized Price equation in regression form for Model A to Model B follows from the fact that both $\bar{w}\Delta\bar{p} = \hat{\beta}_{1,0}\text{Var}\left(p\right)$ and $\bar{w}\Delta\bar{p} = \beta_{1,0}\text{Var}\left(p\right) + \beta_{0,1}\text{Cov}\left(p,q\right)$ are identities. The first follows from the derivation of the Generalized Price equation, combined with Model A, and the second was established above for transitions generated by Model B. We therefore end up with the following identity:

$$\bar{w}\Delta\bar{p} = \hat{\beta}_{1,0}\text{Var}\left(p\right)$$

where $\hat{\beta}_{1,0} = \beta_{1,0} + \beta_{0,1}\frac{\text{Cov}(p,q)}{\text{Var}(p)}$. This implies that $\hat{\beta}_{1,0}$ no longer is a constant, as it varies with $p$ and $q$ through the $\frac{\text{Cov}(p,q)}{\text{Var}(p)}$ – term. It also implies that $\hat{\beta}_{1,0}$ is not the linear effect for this model, which would be $\beta_{1,0}$.

### Applying the Generalized Price equation for Model A to Model C

If we apply the Generalized Price equation in regression form for Model A to Model C, then the sum of squared errors can also no longer be reduced to be 0 and is minimized at $\beta_{1,0} + \beta_{0,1}\frac{\text{Cov}(p,q)}{\text{Var}(p)} + \beta_{1,1}\frac{\text{Cov}(p,pq)}{\text{Var}(p)}$. We therefore end up with the following identity:

$$\bar{w}\Delta\bar{p} = \hat{\beta}_{1,0}\text{Var}\left(p\right)$$

where $\hat{\beta}_{1,0} = \beta_{1,0} + \beta_{0,1}\frac{\text{Cov}(p,q)}{\text{Var}(p)} + \beta_{1,1}\frac{\text{Cov}(p,pq)}{\text{Var}(p)}$. This implies that $\hat{\beta}_{1,0}$ is no longer a constant, as it varies with $p$ and $q$ through the $\frac{\text{Cov}(p,q)}{\text{Var}(p)}$ – term and the $\frac{\text{Cov}(p,pq)}{\text{Var}(p)}$ – term. It also implies that $\hat{\beta}_{1,0}$ is not the linear effect for this model, which would be $\beta_{1,0}$.

### Applying the Generalized Price equation for Model B to Model C

This would lead to more complex relations between, on the one hand, $\hat{\beta}_{1,0}$ and $\hat{\beta}_{0,1}$, and, on the other, the true $\beta_{1,0}$, $\beta_{0,1}$, and $\beta_{1,1}$, but the idea is the same: misspecification leads to a positive sum of squared errors, when it is being minimized, and coefficients that change with the state of the parent population.

### Summary

Also with modeling, misspecification is possible. This leads to Price-like equations in which the coefficients are not constant.

## Expected changes

If we do away with the assumption of a deterministic model, one could still hope for the (Generalized) Price equation to be correct in expectation. In order to explore that possibility, we go back to Model A, but now with noise. That means that we go back to writing the model as $w_i = \alpha + \beta_{1,0}p_i + \epsilon_i$. Explicitly or implicitly, the noise is typically assumed to have mean zero, which makes $\alpha + \beta_{1,0}p_i$ the expected number of offspring of someone with $p$-score of $p_i$. Numbers of realized offspring are obviously integers, limiting what values $w_i$ can take, but the fact that we assume (continuous) probabilities with which an individual has these integer numbers of offspring implies that $\alpha$ and $\beta_{1,0}$ can be any number. The model then captures how the $p$-score affects the distribution of the number of offspring, summarized by $\alpha + \beta p_i$ being the expected value of that distribution.

We begin by fixing a parent generation characterized by a vector of $p$-scores; $p = [p_1, \ldots, p_n]$. Any choice for $p$ also implies an average $p$-score $\bar{p}$ and a 'variance' in $p$-scores $\text{Var}(p)$. The new generation, then, is a combination of $n$ independent random variables, one for every individual, as given by the model. For any realization of these random variables, one could write the Generalized Price equation that belongs to Model A. With asexual reproduction and no mutation – which implies that $E(w\Delta p) = 0$ for every realization of these random variables – this would be

$$\bar{w}\Delta\bar{p} = \hat{\beta}_{1,0}\text{Var}(p)$$

It is worth stressing once more that this equation holds for every realization of the random variables. The average fitness $\bar{w}$ and the value of $\hat{\beta}_{1,0}$ do, of course, depend on the realization, while the expected values of $\bar{w}$ and $\hat{\beta}_{1,0}$, obviously, do not (notice that these are proper expected values, which are a property of a random variable, as opposed to the $E(w\Delta p)$-term in the Price equation, which is really just an average). The expected value of $\bar{w}$ is the expected average fitness $\frac{1}{n}\sum_{1=1}^{n}(\alpha + \beta_{1,0}p_i)$. The expected value of $\hat{\beta}_{1,0}$ is $\beta_{1,0}$, because we have assumed that the data are generated by Model A, and that is what it means for $\hat{\beta}_{1,0}$ to be an unbiased estimator of $\beta_{1,0}$. All of this could make one hope that maybe the (Generalized) Price equation in regression form could reflect a property of the *expected* change in average $p$-score implied by the model. This hope is unfounded, as the following simple example shows.

Assume a population of two parents, one without and one with a certain gene; $p = [0, 1]$. Without the gene, the probability of having 0 offspring and the probability of having 2 is both $\frac{1}{2}$. With the gene, the probability of having 1 offspring and the probability of having 3 is both $\frac{1}{2}$. That implies that the model fits $w_i = \alpha + \beta_{1,0}p_i + \epsilon_i$, where $\alpha = 1$, $\beta_{1,0} = 1$, and $\epsilon_i$ is 1 or -1, both with probability $\frac{1}{2}$.

This super simple model implies that there are four possible transitions, each with probability $\frac{1}{4}$. $\text{Var}(p)$ is a property of the parent population, so $\text{Var}(p) = \frac{1}{4}$ for all four transitions. Below we list the ingredients of the Price equation in regression form for the 4 transitions, all of which happen with probability $\frac{1}{4}$.

If the parents have 0 and 1 offspring, respectively, then $\bar{w} = \frac{1}{2}$, $\Delta\bar{p} = \frac{1}{2}$, and $\hat{\beta}_{1,0} = 1$.

If the parents have 0 and 3 offspring, respectively, then $\bar{w} = \frac{3}{2}$, $\Delta\bar{p} = \frac{1}{2}$, and $\hat{\beta}_{1,0} = 3$.

If the parents have 2 and 1 offspring, respectively, then $\bar{w} = \frac{3}{2}$, $\Delta\bar{p} = -\frac{1}{6}$, and $\hat{\beta}_{1,0} = -1$.

If the parents have 2 and 3 offspring, respectively, then $\bar{w} = \frac{5}{2}$, $\Delta\bar{p} = \frac{1}{10}$, and $\hat{\beta}_{1,0} = 1$.

This helps verify that the expected value of $\hat{\beta}_{1,0}$ is indeed $\beta_{1,0} = 1$, as $E\left[\hat{\beta}_{1,0}\right] = \frac{1}{4}(1 + 3 - 1 + 1) = 1$. Also, that the expected value of $\bar{w}$ is indeed $\frac{1}{n}\sum_{1=1}^{n}(\alpha + \beta_{1,0}p_i) = \frac{1}{2}(1 + 2) = \frac{3}{2}$, as $E[\bar{w}] = \frac{1}{4}\left(\frac{1}{2} + \frac{3}{2} + \frac{3}{2} + \frac{5}{2}\right) = \frac{3}{2}$. The expected value of $\Delta\bar{p}$, on the other hand, is $E[\Delta\bar{p}] = \frac{1}{4}\left(\frac{1}{2} + \frac{1}{2} - \frac{1}{6} + \frac{1}{10}\right) = \frac{7}{30}$. That implies that, while the (Generalized) Price equation in regression form holds for every realization, we cannot replace all terms in it by their expected values, as

$$\frac{3}{2} \cdot \frac{7}{30} \neq 1 \cdot \frac{1}{4}$$

In other words, it is ***not*** generally true that

$$E[\bar{w}]\, E[\Delta\bar{p}] = E\left[\hat{\beta}_{1,0}\right]\text{Var}(p)$$

with expectations taken over the random variables regarding the reproduction.

In models with a fixed population size, all transitions have the same average realized fitness, because for all transitions $\bar{w} = 1$. Examples are the Moran process, the Wright-Fisher process, and all models included in *Allen and Tarnita, 2014*. If we consider one and the same parent population, $\text{Var}(p)$ is a constant, and for every transition, the Price equation in regression form tells us that

$$\bar{w}\Delta\bar{p} = \hat{\beta}_{1,0}\text{Var}(p)$$

This implies that, since $\bar{w} = 1$ for every transition, in this special case, we do have

$$E\left[\bar{w}\right]E\left[\Delta\bar{p}\right] = E\left[\Delta\bar{p}\right] = E\left[\hat{\beta}_{1,0}\right]\text{Var}(p)$$

### Relation to *Grafen, 2000*

As a side note, we can try to reconcile this with (*Grafen, 2000*), results in which one might, at first sight, think contradict this. In this paper with the title "*Developments of the Price equation and natural selection under uncertainty*", the Price equation is extended to cover uncertainty. Below, we will apply the steps taken in *Grafen, 2000* to our example in order to illustrate the differences.

There are two ways in which the Price equation is used differently in *Grafen, 2000*. The first is that it uses the covariance form, and not the regression form. The second is that it does not use realized fitnesses but defines a new variable $v_i = \frac{w_i}{\bar{w}}$ for every realization. It then uses this relative realized fitness instead of the (absolute) realized fitnesses. Both of these changes are needed for the Price equation to remain an identity after taking the expected value on both sides, and below we will show that neither one of the two is enough on its own. It is also important to observe that if we were to go from the Price equation in covariance form to the Price equation in regression form, the change in variable from $w_i$ to $v_i$ would obstruct a meaningful interpretation of the regression coefficients that is stable across realizations. It takes some imagination to envisage scenarios in which a gene increases the expected number of offspring of an individual by 1 if that individual happens to find itself in a population with an average number of offspring of 1, while it increases the expected number of offspring by 2 if that same individual happens to find itself in a population with an average fitness of 2. One could perhaps think of this as a reflection of the effect of the carrying capacity, where the death rate increases as the population size grows, but even then the total number of offspring is more relevant than the average number of offspring per parent. The claim in *Grafen, 2000* is therefore true, but also not very useful for describing population genetic dynamics, which would require the Price equation in regression form.

**Appendix 1—table 1.** Four transitions and the values for the terms in the Price equation. This is an overview of all terms that are in the different versions of the Price equation.

| | Parent 1 | Parent 2 | $\bar{w}$ | $\Delta\bar{p}$ | $\hat{\beta}_{1,0}$ | $\text{Cov}(p,w)$ | $\hat{\vartheta}_{1,0}$ | $\text{Cov}(p,v)$ |
|---|---|---|---|---|---|---|---|---|
| Transition 1 | 0 | 1 | $\frac{1}{2}$ | $\frac{1}{2}$ | 1 | $\frac{1}{4}$ | 1 | $\frac{1}{2}$ |
| Transition 2 | 0 | 3 | $\frac{3}{2}$ | $\frac{1}{2}$ | 3 | $\frac{3}{4}$ | $\frac{1}{3}$ | $\frac{1}{2}$ |
| Transition 3 | 2 | 1 | $\frac{3}{2}$ | $-\frac{1}{6}$ | $-1$ | $-\frac{1}{4}$ | $-\frac{1}{9}$ | $-\frac{1}{6}$ |
| Transition 4 | 2 | 3 | $\frac{5}{2}$ | $\frac{1}{10}$ | 1 | $\frac{1}{4}$ | $\frac{1}{25}$ | $\frac{1}{10}$ |
| Average | 1 | 2 | $\frac{3}{2}$ | $\frac{7}{30}$ | 1 | $\frac{1}{4}$ | $\frac{71}{225}$ | $\frac{7}{30}$ |

In the table above, we first listed $\bar{w}$, $\Delta\bar{p}$, and $\hat{\beta}_{1,0}$ for all 4 realizations, as calculated above. Then we calculated $\text{Cov}(p,w)$. The regression coefficient for $v$ instead of $w$, which we denote by $\hat{\vartheta}_{1,0}$

here, is simply $\hat{\beta}_{1,0}$ divided by $\bar{w}$, as the variable $v_i$ is defined as $w_i$ divided by $\bar{w}$. Finally, $\mathrm{Cov}\,(p,v)$ is $\mathrm{Cov}\,(p,w)$ divided by $\bar{w}$ for the same reason.

From this table, we can see, as before, that it is **not** generally true that $E\left[\bar{w}\right]E\left[\Delta\bar{p}\right] = E\left[\hat{\beta}_{1,0}\right]\mathrm{Var}\,(p)$, as $\frac{3}{2}\cdot\frac{7}{30}\neq 1\cdot\frac{1}{4}$

Also, we can see that it is **not** generally true that $E\left[\bar{w}\right]E\left[\Delta\bar{p}\right] = E\left[\mathrm{Cov}\,(p,w)\right]$, as

$$\frac{3}{2}\cdot\frac{7}{30}\neq\frac{1}{4}$$

Also, we can see that it is **not** generally true that $E\left[\Delta\bar{p}\right] = E\left[\hat{\vartheta}_{1,0}\right]\mathrm{Var}\,(p)$, as

$$\frac{7}{30}\neq\frac{71}{225}\cdot\frac{1}{4}$$

The only thing that is true is that $E\left[\Delta\bar{p}\right] = E\left[\mathrm{Cov}\,(p,v)\right]$. This suggests that both the switch from the regression to the covariance form and the switch from $w$ to $v$ are necessary.

## Infinite populations

With ever-increasing populations, however, the variance in $\bar{w}$, $p'$, and $\hat{\beta}_{1,0}$ decreases ever more. In the limit of infinitely large populations, the dynamics become deterministic, and the (Generalized) Price equation in regression form will also hold again if we take expectations.

To illustrate this, we assume the same model as above, but now not with one parent without the gene and one with, but with infinitely many parents, while we still assume that there are equally many parents with and without the gene. Under that assumption, the dynamics become deterministic. That implies that $\bar{w} = \frac{3}{2}$, which coincides with the expected value for the example when there were only two members in the parent population. It also implies that $\hat{\beta}_{1,0} = 1$, which also matches the expected value in the earlier setting, and $\mathrm{Var}\,(p)$ is also still $\frac{1}{4}$. The only term that changes is $\Delta\bar{p}$; with deterministic dynamics, $p' = \frac{1\cdot 0 + 2\cdot 1}{1\cdot 1 + 2\cdot 1} = \frac{2}{3}$, which makes $\Delta\bar{p} = \frac{1}{6}$.

$$\frac{3}{2}\cdot\frac{1}{6} = 1\cdot\frac{1}{4}$$

## Notation

With infinitely large populations, it is useful to change the notation. Instead of going over individuals and representing what p-scores they have, one would have to go over p-scores and indicate the relative size of the population that has that p-score.

To keep it simple, we can think of a finite set of possible p-scores an individual can have, for instance $\left\{0,\frac{1}{L},\frac{2}{L},\ldots,1\right\}$. An infinite population can then be characterized by a vector $x = [x_0, x_1, \ldots, x_L]$, in which $x_l$ measures the (relative) quantity of individuals with p-score $\frac{l}{L}$. One can assume that these are shares of the total population, in which case $\sum_{l=0}^{L} x_l = 1$. Alternatively, one can imagine that also with infinite populations, the size of the offspring generation as a whole may be larger or smaller than the parent population. This would require the possibility that population vectors do not add up to 1, which would make these numbers not only reflect the relative sizes of the set of individuals with different p-scores within one population, but also allow for a comparison of the size of the parent population and the offspring population.

To get comfortable with this notation, I will first write the (ungeneralized) Price equation in covariance form in this notation. The average p-score in the parent population would then be $\bar{p} = \frac{\sum_{l=0}^{L} x_l\cdot\frac{l}{L}}{\sum_{l=0}^{L} x_l}$. The quantities in the offspring generation can be denoted by $y = [y_0, y_1, \ldots, y_L]$, and these would be given by $y_l = x_l\cdot w_l$, where $w_l$ is the expected fitness of an individual with p-score of $\frac{l}{L}$. With infinite populations, the randomness disappears, and we can treat $w_l$ as the average realized fitness for individuals with p-score of $\frac{l}{L}$. The average p-score in the offspring generation is $\bar{p}' = \frac{\sum_{l=0}^{L} y_l\cdot\frac{l}{L}}{\sum_{l=0}^{L} y_l} = \frac{\sum_{l=0}^{L} x_l w_l\cdot\frac{l}{L}}{\sum_{l=0}^{L} x_l w_l}$, and the average fitness is $\bar{w} = \frac{\sum_{l=0}^{L} x_l w_l}{\sum_{l=0}^{L} x_l}$.

We can use this to write

$$\bar{w}\Delta\bar{p} = \bar{w}\left(\bar{p}' - \bar{p}\right) = \frac{\sum_{l=0}^{L} x_l w_l}{\sum_{l=0}^{L} x_l}\left(\frac{\sum_{l=0}^{L} x_l w_l \frac{l}{L}}{\sum_{l=0}^{L} x_l w_l} - \frac{\sum_{l=0}^{L} x_l \frac{l}{L}}{\sum_{l=0}^{L} x_l}\right)$$

$$= \frac{\sum_{l=0}^{L} x_l w_l \frac{l}{L}}{\sum_{l=0}^{L} x_l} - \frac{\sum_{l=0}^{L} x_l w_l}{\sum_{l=0}^{L} x_l}\frac{\sum_{l=0}^{L} x_l \frac{l}{L}}{\sum_{l=0}^{L} x_l}$$

$$= \sum_{l=0}^{L} \frac{x_l}{\sum_{k=0}^{L} x_k} w_l \frac{l}{L} - \sum_{l=0}^{L} \frac{x_l}{\sum_{k=0}^{L} x_k} w_l \sum_{l=0}^{L} \frac{x_l}{\sum_{k=0}^{L} x_k}\frac{l}{L}$$

With $\frac{x_l}{\sum_{k=0}^{L} x_k}$ being the relative share of individuals with a $p$-score of $\frac{l}{L}$, in this new notation, the last expression is $\mathrm{Cov}\left(w, p\right)$. This gives us the standard, ungeneralized Price equation in covariance form.

$$\bar{w}\Delta\bar{p} = \mathrm{Cov}\left(w, p\right)$$

Here, we left out the $E\left(w\Delta p\right)$–term, because with asexual reproduction, this is always 0. With sexual reproduction, this would also be 0 in the limit of infinitely large populations if we assume fair meiosis.

In this new notation, we can also derive the Generalized Price equation in covariance form, and also here the randomness disappears.

$$\bar{w}\Delta\bar{p} = \mathrm{Cov}\left(\hat{w}, p\right)$$

If we assume Model A, then in this notation, that means that $w_l = \alpha + \beta_{1,0}\frac{l}{L}$. If we fill this in in the formula for $\mathrm{Cov}\left(w, p\right)$, we get

$$\mathrm{Cov}\left(w, p\right) = \sum_{l=0}^{L} \frac{x_l}{\sum_{k=0}^{L} x_k} w_l \frac{l}{L} - \sum_{l=0}^{L} \frac{x_l}{\sum_{k=0}^{L} x_k} w_l \sum_{l=0}^{L} \frac{x_l}{\sum_{k=0}^{L} x_k}\frac{l}{L}$$

$$= \sum_{l=0}^{L} \frac{x_l}{\sum_{k=0}^{L} x_k}\left(\alpha + \beta_{1,0}\frac{l}{L}\right)\frac{l}{L} - \sum_{l=0}^{L} \frac{x_l}{\sum_{k=0}^{L} x_k}\left(\alpha + \beta_{1,0}\frac{l}{L}\right)\sum_{l=0}^{L} \frac{x_l}{\sum_{k=0}^{L} x_k}\frac{l}{L}$$

$$= \beta_{1,0}\left(\sum_{l=0}^{L} \frac{x_l}{\sum_{k=0}^{L} x_k}\left(\frac{l}{L}\right)^2 - \sum_{l=0}^{L} \frac{x_l}{\sum_{k=0}^{L} x_k}\frac{l}{L}\sum_{l=0}^{L} \frac{x_l}{\sum_{k=0}^{L} x_k}\frac{l}{L}\right) = \beta_{1,0}\mathrm{Var}\left(p\right)$$

The (ungeneralized) Price equation in regression form, applied to Model A in an infinite population therefore is

$$\bar{w}\Delta\bar{p} = \beta_{1,0} \cdot \mathrm{Var}\left(p\right)$$

The Generalized Price equation, with Model A as a reference model, is

$$\bar{w}\Delta\bar{p} = \hat{\beta}_{1,0} \cdot \mathrm{Var}\left(p\right)$$

and if the model that this is applied to is Model A, then $\hat{\beta}_{1,0} = \beta_{1,0}$.

## Summary

In stochastic models, the (Generalized) Price equation in regression form holds for every realization. That does not imply that it holds in expectation. However, if we assume that the population is large enough for the infinite population model to be a good approximation, then the Generalized Price equation in regression form will be informative about the dynamics. Misspecification concerns, of course, still apply.

## Heterozygote advantage revisited

With the relation between modeling and the Price equation sorted out, we can revisit selection with and without heterozygote advantage. This means that we go back to a diploid setting, in which the $p$-scores can be 0, $\frac{1}{2}$, and 1.

## Model A

First, we start with the linear model, which is $w_i = \alpha + \beta_1 p_i + \varepsilon_i$. We called this Model A at the beginning of this section, and we will also call it Model A here. We furthermore assume an infinite population and fair meiosis. Under those assumptions, the Generalized Price equation for the linear model, applied to the linear model, is

$$\bar{w}\Delta\bar{p} = \hat{\beta}_1 \text{Var}\left(p\right)$$

Here, $\hat{\beta}_1$ is equal to the true $\beta_1$, and if we now would like to translate this equation into a criterion for whether or not (or, possibly: when) this gene is selected for, then this is really straightforward: at all frequencies, the gene is selected for if

$$\beta_1 > 0$$

In the Main Text, we will refer to this as the rule for selection of a non-social trait with linear fitness effects.

## Model B

If we take the quadratic model, then the fitness function is $w_i = \alpha + \beta_1 p_i + \beta_2 p_i^2 + \varepsilon_i$. In the beginning of this section, we compared Model A to a different Model B, but here we will nonetheless refer to the quadratic model as Model B too, as it will be clear what this refers to here. The Generalized Price equation for the quadratic model, applied to the quadratic model, is

$$\bar{w}\Delta\bar{p} = \hat{\beta}_1 \text{Var}\left(p\right) + \hat{\beta}_2 \text{Cov}\left(p, p^2\right)$$

Here, $\hat{\beta}_1$ is equal to the true $\beta_1$, and $\hat{\beta}_2$ is equal to the true $\beta_2$. If we now would like to translate this equation into a criterion for when this gene is selected for, then this is still straightforward, but less concise: the gene is selected for if

$$\beta_1 \text{Var}\left(p\right) + \beta_2 \text{Cov}\left(p, p^2\right) > 0$$

If we now furthermore assume random mating between generations, then at an allele frequency of $p$, the shares of individuals with a $p$-score of 0, $\frac{1}{2}$, and 1 are: $(1-p)^2$, $2p(1-p)$, and $p^2$, respectively. That makes

$$\text{Var}\left(p\right) = E\left[p^2\right] - E^2\left[p\right] = \frac{1}{4}2p\left(1-p\right) + p^2 - \left(\frac{1}{2}2p\left(1-p\right) + p^2\right)^2$$

$$= \frac{1}{2}p\left(1-p\right) + p^2 - p^2 = \frac{1}{2}p\left(1-p\right)$$

and

$$\text{Cov}\left(p, p^2\right) = E\left[p^3\right] - E\left[p\right]E\left[p^2\right] = \left(\frac{1}{8}2p\left(1-p\right) + p^2\right) - p\left(\frac{1}{2}p\left(1-p\right) + p^2\right)$$

$$= \frac{1}{4}p + \frac{3}{4}p^2 - p\left(\frac{1}{2}p + \frac{1}{2}p^2\right) = \frac{1}{4}p + \frac{1}{4}p^2 - \frac{1}{2}p^3 = \frac{1}{2}p\left(1-p\right)\left(\frac{1}{2} + p\right)$$

With random mating, the criterion for when this gene is selected for then becomes

$$\beta_1 + \beta_2 \left( \frac{1}{2} + p \right) > 0$$

Heterozygote advantage would mean that $\beta_1 > 0$ and $\beta_2 < 0$, and if we for instance take $\beta_1 = 1$ and $\beta_2 = -1$, then that simplifies to

$$p < \frac{1}{2}$$

An important feature is that the rule for selection now depends on the population state, as $\mathrm{Var}\,(p)$ and $\mathrm{Cov}\left(p, p^2\right)$ are not multiples of each other. This implies that $\bar{w}\Delta\bar{p}$ may be positive for some and negative for other population states. With additional assumptions, such as random mating, this reduces to frequency dependence – which is something we should expect with heterozygote advantage. The $\hat{\beta}_1$ and $\hat{\beta}_2$, however, are not dependent on the population state, as they are equal to the true $\beta_1$ and $\beta_2$, and therefore constant.

### The Generalized Price equation for Model A applied to Model B

The Generalized Price equation for the linear model, applied to the quadratic model, is

$$\bar{w}\Delta\bar{p} = \hat{\beta}_1 \mathrm{Var}\,(p)$$

with $\hat{\beta}_1 = \beta_1 + \beta_2 \frac{\mathrm{Cov}\left(p, p^2\right)}{\mathrm{Var}\,(p)}$. This also gets the direction of selection right for every population state, but here the $\hat{\beta}_1$ is not a parameter in the quadratic model. As a symptom of this, $\hat{\beta}_1$ depends on the population state. With the additional assumption of random mating, $\frac{\mathrm{Cov}\left(p, p^2\right)}{\mathrm{Var}\,(p)} = \left( \frac{1}{2} + p \right)$, and therefore, under that assumption, $\hat{\beta}_1 = \beta_1 + \beta_2 \left( \frac{1}{2} + p \right)$, which makes the $\hat{\beta}_1$ vary with $p$.

### Taking expectations with the Price equation in covariance form

For a given state of the parent population, we can also take the expectation of the $\mathrm{Cov}\,(w, p)$ term in the Price equation in covariance form. This is relatively straightforward. The expectation of a sum is the sum of the expectations; and since the expected value is a weighted sum, where the weights are the probabilities of all possible transitions, we can change the summation order in both terms. If we write $E\left[w_i \mid p\right]$ for the expected value of $w_i$ given the state of the population, characterized by the vector $p$, we can write

$$E\left[\mathrm{Cov}\,(w, p)\right] = E\left[ \frac{1}{n} \sum_{i=1}^{n} w_i p_i - \frac{1}{n^2} \sum_{i=1}^{n} w_i \sum_{i=1}^{n} p_i \right]$$

$$= \frac{1}{n} \sum_{i=1}^{n} E\left[w_i \mid p\right] p_i - \frac{1}{n^2} \sum_{i=1}^{n} E\left[w_i \mid p\right] \sum_{i=1}^{n} p_i$$

It is still **not** generally true that $E\left[\bar{w}\right] E\left[\Delta\bar{p}\right] = E\left[\mathrm{Cov}\,(p, w)\right]$, as we have seen in the subsection above on the relation with (**Grafen, 2000**). However, if $\bar{w} = 1$ for every transition, as it is in models with a fixed population size, the following holds:

$$E\left[\bar{w}\right] E\left[\Delta\bar{p}\right] = E\left[\Delta\bar{p}\right] = \frac{1}{n} \sum_{i=1}^{n} E\left[w_i \mid p\right] p_i - \frac{1}{n^2} \sum_{i=1}^{n} E\left[w_i \mid p\right] \sum_{i=1}^{n} p_i$$

This is for instance very close to Equation 5.3 in **Allen and Tarnita, 2014**, who write $w_i\,(\mathbf{s})$ for what I denote as $E\left[w_i \mid p\right]$. Their $w_i$ therefore is the expected fitness, not the realized fitness. Their population state $s$ moreover, is a vector, like our $p$, where $s_i = 1$ if individual $i$ (or the individual at position $i$ in a network) is type 1, and $s_i = 0$ if individual $i$ is type 0. This implies that $s_i$ can be interpreted as a $p$-score.

## 6. Discussion

One of the contributions of the Generalized Price equation is that it helps explain the lack of convergence in the debate on the Price equation, as well as the lack of convergence in the debate on the results that are derived with it (which we will come back to in Appendix 2). If we draw the positions with somewhat broad strokes, then on the one hand, there is a majority position that has confidence in the method of using the original Price equation to derive results, and that believes those results to be correct and meaningful (see for instance *Grafen, 1985a*; *Taylor, 1989*; *Taylor and Frank, 1996*; *Frank, 1998*; *Grafen, 2000*; *Rice, 2004*; *Okasha, 2006*; *Grafen, 2006*; *Gardner, 2008*; *Gardner et al., 2011*; *Rousset, 2015*; *Marshall, 2015*; *Queller, 2017*; *Luque, 2017*). On the other hand, there is a minority position that disagrees with the results as well as the method (see for instance *van Veelen, 2005*; *van Veelen et al., 2010*; *Allen et al., 2013*; *Allen and Nowak, 2015*; *van Veelen et al., 2017*; *van Veelen, 2018a*; *van Veelen, 2020a*; *van Veelen, 2020b*). I claim that the lack of convergence in the debate is the result of looking at everything through the lens of the original Price equation, instead of the generalized version. The original Price equation in regression form, which we understand to be the Generalized Price equation in regression form applied to the standard linear model, is mismatched when paired with models that are different, or more general, or with data that do not support the standard linear model. Empirical or theoretical claims that would be correct and meaningful under the standard linear model, or for data that would support the standard linear model, are easily interpreted to also be correct and meaningful in general. This is facilitated by the fact that the original Price equation is an identity, which does not cease to hold when it is combined with models other than the standard linear model, or with data that are generated by other models. If not paired with statistics, the original Price equation does not differentiate between data that support the standard linear model and data that do not, when applied to data – and neither do the other Price-like equations we arrive at when we combine the Generalized Price equation in regression form with other statistical models. While most of the papers from the minority position, including mine, are negative, in the sense that they point to where this can be problematic, the Generalized Price equation not only points to the problem, but also gives a solution, as it helps formulate correct and meaningful alternatives to accurately describe the population genetic dynamics, when the way in which fitnesses depend on $p$-scores does not fit the standard linear model.

### It cannot be wrong because it is an identity

The arguments in defense of the way the Price equation is used in the literature typically include the fact that it is an identity (and therefore that it cannot be wrong) as well as references to its generality. In order to counterbalance this, it is worthwhile, first of all, to point out that neither being an identity nor being general is all that special. There are literally infinitely many equations, all of which are identities, and all of which are completely general in the same way that the Price equation in regression form is general. In a failed attempt to be funny, I pointed out in an earlier paper (*van Veelen, 2020b*) that if we take the original Price equation in covariance form, and divide $\mathrm{Cov}\,(w,p)$, not by $\mathrm{Var}\,(p)$, but by the Planck constant $h$ times the number of times Denmark won the Eurovision Song Contest (denoted by $DK$), then we can rewrite the Price equation as

$$\bar{w}\Delta\bar{p} = \frac{\mathrm{Cov}\,(w,p)}{h \cdot DK} \cdot h \cdot DK + E\,(w\Delta p)$$

As a next step, we can define $\gamma$ as $\frac{\mathrm{Cov}(w,p)}{h \cdot DK}$, and write

$$\bar{w}\Delta\bar{p} = \gamma \cdot h \cdot DK + E\,(w\Delta p)$$

This equation is every bit as much an identity as the original Price equation in regression form, and it is also every bit as general. The equation as a whole, and the coefficient $\gamma$ in particular, however, is clearly devoid of any use or meaning. What this example illustrates is that it is not enough to be an identity. What we need is an identity with a meaningful interpretation. The original Price equation in regression form does have a meaningful interpretation for data that are generated by the standard linear model. We find that the Generalized Price equation, combined with different statistical models, produces other equations that also are identities and that also are general and that have a meaningful interpretation for data that are indeed generated by their presupposed statistical models. Having a meaningful interpretation, therefore, is limited to a *subset* of the models,

or of the possible datasets, that the equations can be applied to. When dealing with data, one would have to resort to standard statistics to see if the data can inform us about the model that has generated them. Having a meaningful interpretation then is limited to those datasets that make us decide in favor of the model associated with any specific Price-like equation.

## What can we use the Generalized Price equation for?

In a modeling context, we either directly assume a relation between $p$-scores and fitnesses, or we have a model that implies such a relation. For instance, we can assume a model $w_i = \alpha + \sum_{r=1}^{R} \beta_r p_i^r$ for some $R$, in combination with the assumption of unbiased transmission. Alternatively, we can make modeling assumptions that imply such a relation between $p$-scores and fitnesses. The corresponding Price-like equation in regression form then is

$$\bar{w}\Delta\bar{p} = \sum_{r=1}^{R} \beta_r \mathrm{Cov}\left(p, p^r\right)$$

This implies that all terms on the right-hand side are a combination of a model parameter, $\beta_r$, that is independent of the population state, and a term that reflects a property of a population state, $\mathrm{Cov}\left(p, p^r\right)$. The equation then helps state how in infinitely large populations, the change in average $p$-score, corrected for the growth or shrinking of the overall population, on the left-hand side, depends on the combination of the current population state and the model parameters, which we see on the right-hand side. This now describes the population genetic dynamics for all population states, as the model parameters $\beta_r$ are constants that do not change with the population state. This may or may not be helpful for finding out properties of the model that we are interested in. In Section 4, we have also seen that we can choose richer sets of models too, as long as models in it include a constant and a term that is linear in the $p$-score.

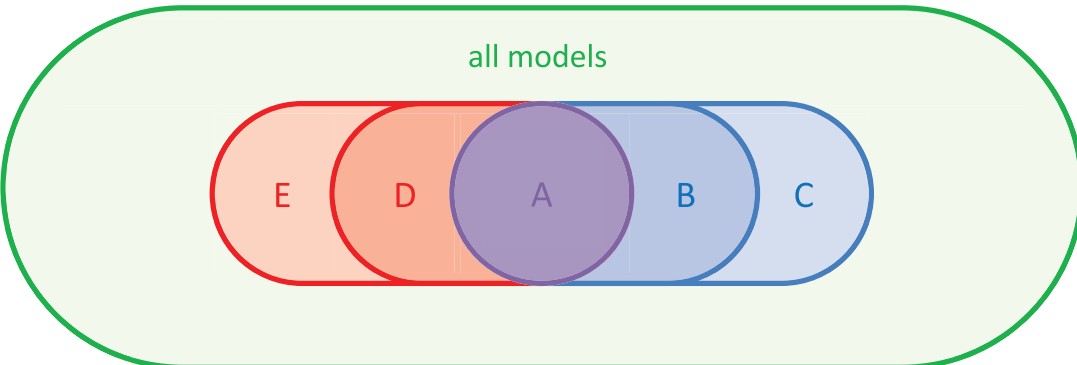

**Appendix 1—figure 3.** Nested models and their Price-like equations. There are different models, and each model has its own Price-like equation. These Price-like equations are general, in the sense that they can be written for *any* dataset, regardless of the underlying data generating process or, in a theory context, for *any* model. The terms in it, however, only have a meaningful interpretation if the data are generated by a model in the set that the Price-like equation belongs to. In line with the setup in this appendix, set A would represent models that are linear in the $p$-score ($w_i = \alpha + \beta_{1,0} p_i$). This is the set of models for which the regression coefficient in the original Price equation in regression form has a meaningful interpretation. Set B could consist of models that are quadratic in the $p$-score ($w_i = \alpha + \beta_{1,0} p_i + \beta_{2,0} p_i^2$), set C could consist of models that also include a coefficient for the $p$-score to the power 3 ($w_i = \alpha + \beta_{1,0} p_i + +\beta_{2,0} p_i^2 + \beta_{3,0} p_i^3$), set D models that are linear in the $p$-score and the $q$-score ($w_i = \alpha + \beta_{1,0} p_i + \beta_{0,1} q_i$), and set E models that include an interaction term between the $p$-score and the $q$-score ($w_i = \alpha + \beta_{1,0} p_i + \beta_{0,1} q_i + \beta_{1,1} p_i q_i$). The Price-like equations for these different models are all different.

## Dynamic (in)sufficiency revisited

One possibility for a full model is that the way fitnesses depend on the genes that an individual carries is all there is to it. In that case, a model $w_i = \alpha + \sum_{r=1}^{R} \beta_r p_i^r$ maps any parent population state $p$ to an offspring population state $p'$. For simplicity, we assume that this is a deterministic model, or an infinitely large population, which also makes this transition deterministic. (The numbering of the offspring generation, that is, which individual is labeled individual 1, and so on, is irrelevant; all

covariances are invariant to permutations of the population vector). We can then iterate this update step, which makes this a dynamically sufficient population dynamics.

One reasonable observation to make here is that the Price equation itself is not something of which it is useful to describe it as dynamically sufficient or not; dynamic sufficiency is a relevant property of the model it is applied to, and not of the Price equation itself (*van Veelen et al., 2012*). If, on the other hand, one sees the Price equation as a tool to compute properties of the offspring generation, then it is tempting to describe it as dynamically insufficient. The Price equation gives the new average $p$-score and uses the old $p$-score and the covariances $\mathrm{Cov}\,(p, p^r)$ pertaining to the first generation to get there. If we then want to repeat this update step, and go from the second population state to the third one, we would need more than the average $p$-score in the second state, because now we also need the covariances $\mathrm{Cov}\,(p, p^r)$ pertaining to the second generation. These covariances in the second state we can get from the model itself, but we cannot get those from applying the Price equation to the first population state. In that sense, one could be tempted to say that the Price equation is not dynamically sufficient; once we have reduced what we know about a population state to its average $p$-score, we cannot apply it again. For more on the subtle difference between dynamic sufficiency and the necessity of higher moments, see sections 7–9 in *van Veelen, 2018b*.

In Appendix 2, we will use the Price equation in regression form to derive Hamilton's rule, and there, we have a setting in which the $q$-score does not reflect the dose of a gene in the individual itself, but in its partner. The model can then describe how the fitness of an individual depends on its $p$- and $q$-score, and this would produce the $p$-scores in the new generation. It would, however, not generate which individuals are partnered up with whom in the new generation, and therefore it would not identify who has which $q$-score in the new generation. In this case, a full model would have to include more than the fitness function. In the absence of assumptions about the matching in the new generation, such a model would be dynamically insufficient.

Notice that this is **not** the same as the regression coefficients $\hat{\beta}_r$, or $\hat{\beta}_{k,l}$, depending on the population state, which is regularly also referred to as dynamical insufficiency. This is incorrect and really a symptom of misspecification. Describing the dependency of the regression coefficients on the population state as dynamical insufficiency is just not correct, and it can have the effect of not recognizing this as a red flag.

## Summarizing

The paper in which the Price equation was presented was ambiguous about whether this was meant for statistics or modeling, and it was a bit loose with using terms from statistics and probability theory. That seemed harmless, but it has led to lasting damage in this field. Compared to the derivation of the original Price equation, it only takes a few extra lines of algebra to derive the Generalized Price equation. Behind those extra lines, however, hides an understanding that the original Price equation in regression form is really just one out of a variety of possible equations, and that the terms in it only have meaning if, when applied to data, the data for this transition justify the conclusion that they are generated by the linear model. When applied to a model, the terms in it are only meaningful if the reference model with respect to which the differences are squared, summed, and minimized coincides with the model under consideration. The original Price equation in regression form can, however, be written for *any* change between a parent and an offspring generation, also changes that are very unlikely to have been generated by the linear model that would give the terms in the classical Price equation in regression form meaning. For alternative models, one can write alternative Price-like equations, and those are equally general, in the sense that these can be written for any transition too. These other Price-like equations come with the same limitations on the interpretation of the terms in it; when applied to data, if the data do not justify concluding that the model the Price-like equation in regression form is built off of is accurate, the terms in it lose their meaning. When applied to a model, again, the terms in it are only meaningful if the reference model with respect to which the differences are squared, summed, and minimized coincides with the model under consideration. The Generalized Price equation does, however, offer a Price-like equation for every reference model. The classical Price equation in regression form is just one of them; the one for the linear model.

## Detailed calculations for Appendix 1

### 1.1: Calculations for Example 2.1

$$p = [1, 0], w = [2, 0] \text{ and } A = P = \begin{bmatrix} 1 & 1 \\ 0 & 0 \end{bmatrix}$$

$$\bar{w} = \frac{1}{2}(2 + 0) = 1$$

$$\Delta\bar{p} = \bar{p}' - \bar{p} = \frac{1}{2}(1 + 1) - \frac{1}{2}(1 + 0) = 1 - \frac{1}{2} = \frac{1}{2}$$

$$E(w\Delta p) = \frac{1}{n}\sum_{i=1}^{n} w_i \left( \frac{\sum_{j=1}^{m} P_{ij}}{w_i} - p_i \right) = \frac{1}{n}\sum_{i=1}^{n} \left( \sum_{j=1}^{m} P_{ij} - w_i p_i \right) = \frac{1}{2}(2) - \frac{1}{2}(2) = 0$$

$$\text{Cov}(w, p) = \frac{1}{n}\sum_{i=1}^{n} w_i p_i - \frac{1}{n^2}\sum_{i=1}^{n} w_i \sum_{i=1}^{n} p_i = \frac{1}{2}(2 \cdot 1 + 0 \cdot 0) - \frac{1}{4}(2 + 0)(1 + 0) = 1 - \frac{1}{2} = \frac{1}{2}$$

$$\text{Var}(p) = \frac{1}{n}\sum_{i=1}^{n} p_i^2 - \frac{1}{n^2}\left( \sum_{i=1}^{n} p_i \right)^2 = \frac{1}{2}(1^2 + 0^2) - \frac{1}{4}(1 + 0)^2 = \frac{1}{2} - \frac{1}{4} = \frac{1}{4}$$

$$\beta = \frac{\text{Cov}(w, p)}{\text{Var}(p)} = \frac{\frac{1}{2}}{\frac{1}{4}} = 2$$

### 1.2: Calculations for Example 2.2

$$p = \left[1, \frac{1}{2}\right], w = [2, 0] \text{ and } A = P = \begin{bmatrix} 1 & 1 \\ 0 & 0 \end{bmatrix}$$

$$\bar{w} = \frac{1}{2}(2 + 0) = 1$$

$$\Delta\bar{p} = \bar{p}' - \bar{p} = \frac{1}{2}(1 + 1) - \frac{1}{2}\left(1 + \frac{1}{2}\right) = 1 - \frac{3}{4} = \frac{1}{4}$$

$$E(w\Delta p) = \frac{1}{n}\sum_{i=1}^{n} w_i \left( \frac{\sum_{j=1}^{m} P_{ij}}{w_i} - p_i \right) = \frac{1}{n}\sum_{i=1}^{n} \left( \sum_{j=1}^{m} P_{ij} - w_i p_i \right) = \frac{1}{2}(2) - \frac{1}{2}(2) = 0$$

$$\text{Cov}(w, p) = \frac{1}{n}\sum_{i=1}^{n} w_i p_i - \frac{1}{n^2}\sum_{i=1}^{n} w_i \sum_{i=1}^{n} p_i = \frac{1}{2}\left(2 \cdot 1 + 0 \cdot \frac{1}{2}\right) - \frac{1}{4}(2 + 0)\left(1 + \frac{1}{2}\right) = 1 - \frac{3}{4} = \frac{1}{4}$$

$$\text{Var}(p) = \frac{1}{n}\sum_{i=1}^{n} p_i^2 - \frac{1}{n^2}\left( \sum_{i=1}^{n} p_i \right)^2 = \frac{1}{2}\left(1^2 + \left(\frac{1}{2}\right)^2\right) - \frac{1}{4}\left(1 + \frac{1}{2}\right)^2 = \frac{5}{8} - \frac{9}{16} = \frac{1}{16}$$

$$\beta = \frac{\text{Cov}(w, p)}{\text{Var}(p)} = \frac{\frac{1}{4}}{\frac{1}{16}} = 4$$

## 1.3: Calculations for Example 3.1

$$p = \left[1, 0, \tfrac{1}{2}, 0\right], \, w = \left[\tfrac{3}{2}, \tfrac{1}{2}, 1, 1\right], \, P = \begin{bmatrix} 1 & 1 & 1 & 0 \\ 0 & 0 & 0 & 0 \\ 1 & 1 & 0 & 0 \\ 0 & 0 & 1 & 1 \end{bmatrix} \text{ and } P = \begin{bmatrix} \tfrac{1}{2} & \tfrac{1}{2} & \tfrac{1}{2} & 0 \\ 0 & 0 & 0 & 0 \\ 0 & \tfrac{1}{2} & 0 & 0 \\ 0 & 0 & 0 & 0 \end{bmatrix}$$

$$\bar{w} = \frac{1}{4}\left(\frac{3}{2} + \frac{1}{2} + 1 + 1\right) = 1$$

$$\Delta\bar{p} = \bar{p}' - \bar{p} = \frac{1}{4}\left(\frac{1}{2} + 1 + \frac{1}{2} + 0\right) - \frac{1}{4}\left(1 + 0 + \frac{1}{2} + 0\right) = \frac{1}{8}$$

$$E\left(w\Delta p\right) = \frac{1}{n}\sum_{i=1}^{n} w_i \left(\frac{\sum_{j=1}^{m} P_{ij}}{w_i} - p_i\right) = \frac{1}{n}\sum_{i=1}^{n}\left(\sum_{j=1}^{m} P_{ij} - w_i p_i\right)$$

$$= \frac{1}{4}\left(\frac{3}{2} - \frac{3}{2} + 0 - 0 + \frac{1}{2} - \frac{1}{2} + 0 - 0\right) = 0$$

$$\text{Cov}\left(w, p\right) = \frac{1}{n}\sum_{i=1}^{n} w_i p_i - \frac{1}{n^2}\sum_{i=1}^{n} w_i \sum_{i=1}^{n} p_i$$

$$= \frac{1}{4}\left(\frac{3}{2}\cdot 1 + \frac{1}{2}\cdot 0 + 1\cdot\frac{1}{2} + 1\cdot 0\right) - \frac{1}{16}\left(\frac{3}{2} + \frac{1}{2} + 1 + 1\right)\left(1 + 0 + \frac{1}{2} + 0\right)$$

$$= \frac{1}{2} - \frac{1}{16}\left(4\right)\left(\frac{3}{2}\right) = \frac{1}{2} - \frac{3}{8} = \frac{1}{8}$$

$$\text{Var}\left(p\right) = \frac{1}{n}\sum_{i=1}^{n} p_i^2 - \frac{1}{n^2}\left(\sum_{i=1}^{n} p_i\right)^2 = \frac{1}{4}\left(1^2 + 0^2 + \left(\frac{1}{2}\right)^2 + 0^2\right) - \frac{1}{16}\left(1 + 0 + \frac{1}{2} + 0\right)^2$$

$$= \frac{1}{4}\cdot\frac{5}{4} - \frac{1}{16}\cdot\frac{9}{4} = \frac{11}{64}$$

$$\hat{\beta} = \frac{\text{Cov}\left(w, p\right)}{\text{Var}\left(p\right)} = \frac{\frac{1}{8}}{\frac{11}{64}} = \frac{8}{11}$$

## 1.4: Calculations for Example 3.2

### The stationary distribution

In order to be able to calculate the frequencies of the different $p$-scores in the offspring generation, we first establish that from the parent generation, all 4 parents with a $p$-score of $\frac{1}{2}$ survive, as well as 2 (out of the 4) parents with a $p$-score of $\frac{1}{2}$ (step 1 in **Appendix 1—figure 2**). Half of those are female and half of those are male, and therefore we can also think of this as a population where 12 out of 18 survive, 4 females and 4 males with a $p$-score of $\frac{1}{2}$, and 2 females and 2 males with a $p$-score of 1 (step 2).

These are randomly matched, which means that two thirds of the 4 females with a $p$-score of $\frac{1}{2}$ (out of 18 parents) are matched with a male that also has a $p$-score of $\frac{1}{2}$, and one third is matched with a male that has a $p$-score of $\frac{1}{2}$, resulting in $\frac{2}{3}\frac{4}{18} = \frac{4}{27}$ parent pairs in which both parents have a $p$-score of $\frac{1}{2}$, and $\frac{1}{3}\frac{4}{18} = \frac{2}{27}$ parent pairs with a female with a $p$-score of $\frac{1}{2}$ and a male with a $p$-score of 1.

Similarly, there will be $\frac{1}{3}\frac{2}{18} = \frac{1}{27}$ parent pairs in which both parents have a *p*-score of 1, and $\frac{2}{3}\frac{2}{18} = \frac{2}{27}$ parent pairs with a female with a *p*-score of 1 and a male with a *p*-score of $\frac{1}{2}$ (step 3 and 4).

If both parents have a *p*-score of $\frac{1}{2}$, 1 out of 4 kids has a *p*-score of 0; 2 have a *p*-score of $\frac{1}{2}$; and 1 has a *p*-score of 1. If one parent has a *p*-score of $\frac{1}{2}$ and the other has a *p*-score of 1, then half of their offspring will have a *p*-score of $\frac{1}{2}$, and the other half will have a *p*-score of 1. Finally, if both parents have a *p*-score of 1, all of their offspring will have a *p*-score of 1 too (step 5).

All parent pairs have 3 kids on average, which then scales this back to the frequencies in the parent population. For example, that results in $3 \cdot \frac{1}{4} \cdot \frac{4}{27} = \frac{1}{9}$ kids with a *p*-score of 0, where 3 is the number of kids per parent pair, $\frac{1}{4}$ is the proportion of kids with a *p*-score of 0 that a parent pair gets when both parents have a *p*-score of $\frac{1}{2}$, and $\frac{4}{27}$ measures the number of such parent pairs (step 6).

## The Price equation

Elements of the Price equation in regression form for the stationary population state:

$$\bar{w} = \frac{1}{9} \cdot 0 + \frac{4}{9} \cdot \frac{3}{2} + \frac{4}{9} \cdot \frac{1}{2} \cdot \frac{3}{2} = 0 + \frac{6}{9} + \frac{3}{9} = 1$$

$$\Delta\bar{p} = \bar{p}' - \bar{p} = \frac{2}{3} - \frac{2}{3} = 0$$

$$E\left(w\Delta p\right) = 0$$

$$\text{Cov}\left(w, p\right) = \frac{1}{n}\sum_{i=1}^{n} w_i p_i - \frac{1}{n^2}\sum_{i=1}^{n} w_i \sum_{i=1}^{n} p_i$$

$$= \left(\frac{1}{9} \cdot 0 + \frac{4}{9} \cdot \frac{3}{2} \cdot \frac{1}{2} + \frac{4}{9} \cdot \frac{1}{2} \cdot \frac{3}{2} \cdot 1\right) - \left(\frac{1}{9} \cdot 0 + \frac{4}{9} \cdot \frac{3}{2} + \frac{4}{9} \cdot \frac{1}{2} \cdot \frac{3}{2}\right)\left(\frac{1}{9} \cdot 0 + \frac{4}{9} \cdot \frac{1}{2} + \frac{4}{9} \cdot 1\right)$$

$$= \frac{6}{9} - 1 \cdot \frac{6}{9} = 0$$

$$\text{Var}\left(p\right) = \frac{1}{n}\sum_{i=1}^{n} p_i^2 - \frac{1}{n^2}\left(\sum_{i=1}^{n} p_i\right)^2 = \left(\frac{1}{9} \cdot 0^2 + \frac{4}{9} \cdot \left(\frac{1}{2}\right)^2 + \frac{4}{9} \cdot 1^2\right) - \left(\frac{1}{9} \cdot 0 + \frac{4}{9} \cdot \frac{1}{2} + \frac{4}{9} \cdot 1\right)^2$$

$$= \frac{5}{9} - \frac{36}{81} = \frac{1}{9}$$

$$\hat{\beta} = \frac{\text{Cov}\left(w, p\right)}{\text{Var}\left(p\right)} = 0$$

In general, we can assume *k* individuals with a *p*-score of 0, *l* individuals with a *p*-score of $\frac{1}{2}$, and $n - k - l$ individuals with a *p*-score of 1. The ingredients of the Price equation without noise then become:

$$\bar{w} = \frac{1}{n}\left(k \cdot 0 \cdot \frac{3}{2} + l \cdot 1 \cdot \frac{3}{2} + (n - k - l) \cdot \frac{1}{2} \cdot \frac{3}{2}\right) = \frac{1}{n} \cdot \frac{3}{2}\left(l + (n - k - l) \cdot \frac{1}{2}\right) = \frac{3}{4} \cdot \frac{n - k + l}{n}$$

$$\Delta\bar{p} = \bar{p}' - \bar{p} = \frac{4}{3} \cdot \frac{1}{n - k + l}\left(k \cdot 0 \cdot 0 \cdot \frac{3}{2} + l \cdot 1 \cdot \frac{1}{2} \cdot \frac{3}{2} + (n - k - l) \cdot \frac{1}{2} \cdot 1 \cdot \frac{3}{2}\right)$$

$$- \frac{1}{n}\left(k \cdot 0 + l \cdot \frac{1}{2} + (n - k - l) \cdot 1\right) = \frac{n - k}{n - k + l} - \frac{n - k - \frac{1}{2}l}{n}$$

$$= \frac{n(n - k)}{(n - k + l)n} - \frac{(n - k)^2 + \frac{1}{2}l(n - k) - \frac{1}{2}l^2}{(n - k + l)n} = \frac{k(n - k) - \frac{1}{2}l(n - k) + \frac{1}{2}l^2}{(n - k + l)n}$$

$$= \frac{k(n - k) - \frac{1}{2}l(n - k - l)}{(n - k + l)n}$$

$$E\left(w\Delta p\right) = 0$$

$$\mathrm{Cov}\left(w,p\right) = \frac{1}{n}\sum_{i=1}^{n}w_ip_i - \frac{1}{n^2}\sum_{i=1}^{n}w_i\sum_{i=1}^{n}p_i = \frac{1}{n}\sum_{i=1}^{n}w_ip_i - \bar{w}\bar{p}$$

$$= \frac{1}{n}\left(k\cdot 0\cdot\frac{3}{2}\cdot 0 + l\cdot 1\cdot\frac{3}{2}\cdot\frac{1}{2} + \left(n-k-l\right)\cdot\frac{1}{2}\cdot\frac{3}{2}\cdot 1\right) - \frac{3}{4}\cdot\frac{n-k+l}{n}\cdot\frac{n-k-\frac{1}{2}l}{n}$$

$$= \frac{3}{4}\cdot\frac{n-k}{n} - \frac{3}{4}\cdot\frac{n-k+l}{n}\cdot\frac{n-k-\frac{1}{2}l}{n} = \frac{3}{4}\cdot\left(\frac{n\left(n-k\right)}{n^2} - \frac{\left(n-k\right)^2 + \frac{1}{2}l\left(n-k\right) - \frac{1}{2}l^2}{n^2}\right)$$

$$= \frac{3}{4}\cdot\frac{k\left(n-k\right) - \frac{1}{2}l\left(n-k-l\right)}{n^2}$$

$$\mathrm{Var}\left(p\right) = \frac{1}{n}\sum_{i=1}^{n}p_i^2 - \frac{1}{n^2}\left(\sum_{i=1}^{n}p_i\right)^2 = \frac{1}{n}\left(k\cdot 0 + l\cdot\left(\frac{1}{2}\right)^2 + \left(n-k-l\right)\cdot 1^2\right) - \left(\frac{n-k-\frac{1}{2}l}{n}\right)^2$$

$$= \frac{n-k-\frac{3}{4}l}{n} - \left(\frac{n-k-\frac{1}{2}l}{n}\right)^2 = \frac{n^2 - nk - \frac{3}{4}nl}{n^2} - \frac{\left(n-k\right)^2 - l\left(n-k\right) + \frac{1}{4}l^2}{n^2}$$

$$= \frac{\left(k+l\right)\left(n-k\right) - \frac{3}{4}nl - \frac{1}{4}l^2}{n^2}$$

$$\hat{\beta} = \frac{\mathrm{Cov}\left(w,p\right)}{\mathrm{Var}\left(p\right)} = \frac{3}{4}\cdot\frac{k\left(n-k\right) - \frac{1}{2}l\left(n-k-l\right)}{\left(k+l\right)\left(n-k\right) - \frac{3}{4}nl - \frac{1}{4}l^2}$$

When there are only individuals with a $p$-score of 0, or with a $p$-score of $\frac{1}{2}$, we have $k+l = n$, and the $\beta$ is $\frac{kl-0}{nl-\frac{3}{4}nl-\frac{1}{4}l^2} = \frac{kl}{\frac{1}{4}nl-\frac{1}{4}l^2} = \frac{k}{\frac{1}{4}(n-l)} = \frac{k}{\frac{1}{4}k} = 4$.

When there are only individuals with a $p$-score of $\frac{1}{2}$, or with a $p$-score of 1, we have $l = 0$, and the $\beta$ is $\frac{k(n-k)-0}{(k+0)(n-k)-0-0} = \frac{k(n-k)}{k(n-k)} = 1$.

When there are only individuals with a $p$-score of 0, or with a $p$-score of 1, we have $k = 0$, and the $\beta$ is $\frac{0-\frac{1}{2}l(n-l)}{ln-\frac{3}{4}nl-\frac{1}{4}l^2} = \frac{\frac{-1}{2}l(n-l)}{\frac{1}{4}nl-\frac{1}{4}l^2} = \frac{\frac{-1}{2}(n-l)}{\frac{1}{4}n-\frac{1}{4}l} = -2$.

This illustrates the range of possible $\hat{\beta}$'s.

Just as a sanity check, if we choose $k = 1$, $l = 4$ and $n = 9$, as in the stationary population state, we do indeed get

$$\bar{w} = \frac{3}{4}\cdot\frac{n-k+l}{n} = \frac{3}{4}\cdot\frac{12}{9} = 1$$

$$\Delta\bar{p} = \frac{k\left(n-k\right) - \frac{1}{2}l\left(n-k-l\right)}{\left(n-k+l\right)n} = \frac{8-8}{12\cdot 9} = 0$$

$$E\left(w\Delta p\right) = 0$$

$$\mathrm{Cov}\left(w,p\right) = \frac{3}{4}\cdot\frac{k\left(n-k\right) - \frac{1}{2}l\left(n-k-l\right)}{n^2} = \frac{3}{4}\cdot\frac{8-8}{9^2} = 0$$

$$\text{Var}\left(p\right) = \frac{\left(k+l\right)\left(n-k\right) - \frac{3}{4}nl - \frac{1}{4}l^2}{n^2} = \frac{5 \cdot 8 - \frac{3}{4}9 \cdot 4 - \frac{1}{4}4^2}{9^2} = \frac{40 - 27 - 4}{9^2} = \frac{9}{9^2} = \frac{1}{9}$$

$$\hat{\beta} = \frac{\text{Cov}\left(w,p\right)}{\text{Var}\left(p\right)} = 0$$

## 1.5: Calculations for Example 3.3

The average fitness is just the size of the offspring generation over the size of the parent population;

$$\bar{w} = \frac{m}{n}$$

The change in $p$-score between the generations is one half times the difference between the frequency of females in the offspring generation and the frequency of females in the parent population;

$$\Delta\bar{p} = \frac{l + \frac{1}{2}\left(m-l\right)}{m} - \frac{k + \frac{1}{2}\left(n-k\right)}{n} = \frac{1}{2}\left(\frac{l}{m} - \frac{k}{n}\right)$$

If reproduction happens with equal probabilities on girls and boys, one would expect $\frac{k}{n}$ to be close to $\frac{1}{2}$ for large populations.

When calculating the $E\left(w\Delta p\right)$ term, we make use of the fact that all mothers have a $p$-score of 1, and therefore their successful gametes must all have $p$-score $\frac{1}{2}$. This implies that the sum of the $p$-scores of their successful gametes must be $\frac{1}{2}$ times the number of individuals in the offspring generation; $\sum_{i=1}^{k} \sum_{j=1}^{m} P_{ij} = \frac{1}{2}m$. Also, it implies that $w_i p_i = w_i$ for all mothers, and therefore $\sum_{i=1}^{k} w_i p_i = \sum_{i=1}^{k} w_i = \frac{1}{2}m$.

All fathers of daughters must have had a successful gamete with a $p$-score of $\frac{1}{2}$, while all fathers of sons must have had a successful gamete with p-score 0. Therefore, the sum of their successful gametes must be $\frac{1}{2}$ times the total number of daughters; $\sum_{i=k+1}^{n} \sum_{j=1}^{m} P_{ij} = \frac{1}{2}l$. Also, it implies that $w_i p_i = \frac{1}{2}w_i$ for all fathers, and therefore $\sum_{i=k+1}^{n} w_i p_i = \sum_{i=k+1}^{n} \frac{1}{2}w_i = \frac{1}{4}m$.

For mothers and fathers together, therefore, $\sum_{i=1}^{n} \sum_{j=1}^{m} P_{ij} = \frac{1}{2}\left(m+l\right)$, and $\sum_{i=1}^{n} w_i p_i = \frac{1}{2}m + \frac{1}{4}m = \frac{3}{4}m$. Therefore

$$E\left(w\Delta p\right) = \frac{1}{n}\sum_{i=1}^{n}\left(\sum_{j=1}^{m}P_{ij} - w_i p_i\right) = \frac{1}{n}\left(\frac{1}{2}l - \frac{1}{4}m\right) = \frac{1}{2n}\left(l - \frac{1}{2}m\right)$$

When calculating the $\text{Cov}\left(w,p\right)$ term, we use $\sum_{i=1}^{n} w_i p_i = \frac{3}{4}m$ again.

$$\text{Cov}\left(w,p\right) = \frac{1}{n}\left(\sum_{i=1}^{n}w_i p_i\right) - \frac{1}{n^2}\left(\sum_{i=1}^{k}w_i + \sum_{i=k+1}^{n}w_i\right)\left(\sum_{i=1}^{k}p_i + \sum_{i=k+1}^{n}p_i\right)$$

$$= \frac{1}{n}\left(\frac{3}{4}m\right) - \frac{1}{n^2}\left(\frac{1}{2}m + \frac{1}{2}m\right)\left(k + \frac{1}{2}\left(n-k\right)\right) = \frac{3m}{4n} - \frac{1}{n^2}\left(m\right)\frac{1}{2}\left(n+k\right)$$

$$= \frac{3m}{4n} - \frac{m}{2n} - \frac{mk}{2n^2} = \frac{m}{4n} - \frac{mk}{2n^2} = \frac{m}{2n}\left(\frac{1}{2} - \frac{k}{n}\right)$$

$$\text{Var}(p) = \frac{1}{n} \sum_{i=1}^{n} p_i^2 - \frac{1}{n^2} \left( \sum_{i=1}^{n} p_i \right)^2$$

$$= \frac{1}{n} \left( k \cdot 1^2 + (n-k) \left( \frac{1}{2} \right)^2 \right) - \frac{1}{n^2} \left( k + (n-k) \frac{1}{2} \right)^2$$

$$= \frac{1}{n} \cdot \frac{1}{4} (3k+n) - \frac{1}{n^2} \cdot \frac{1}{4} (k+n)^2 = \frac{3}{4} \frac{k}{n} + \frac{1}{4} - \left( \frac{1}{4} \frac{k^2}{n^2} + \frac{1}{2} \frac{k}{n} + \frac{1}{4} \right) = \frac{k}{4n} \left( 1 - \frac{k}{n} \right)$$

$$\hat{\beta} = \frac{\text{Cov}(w,p)}{\text{Var}(p)} = \frac{\frac{m}{2n} \left( \frac{1}{2} - \frac{k}{n} \right)}{\frac{1}{4} \frac{k}{n} \left( 1 - \frac{k}{n} \right)} = \frac{2m \left( \frac{n}{2} - k \right)}{k(n-k)} = \frac{m(n-2k)}{k(n-k)}$$

This implies that $\hat{\beta} > 0$ if $k < \frac{1}{2}n$.

## 1.6: Calculations for *Appendix 1—figure 1* from the Main Text

In Model A from *Appendix 1—figure 1*, the number of offspring follows a binomial distribution with 10 trials and success probability $\frac{1}{10}(1+p_i)$, where $p_i$ is the p-score of individual $i$. This results in an expected number of offspring of $1+p_i$, which means that the model can be summarized as $w_i = 1 + p_i + \varepsilon_i$.

In Model B from *Appendix 1—figure 1*, the number of offspring follows a binomial distribution with 10 trials and success probability $\frac{1}{10}\left(1+p_i^2\right)$, where $p_i$ is the p-score of individual $i$. This results in an expected number of offspring of $1+p_i^2$, which means that the model can be summarized as $w_i = 1 + p_i^2 + \varepsilon_i$.

The variance of the error term is $\frac{(1+p_i)(9-p_i)}{10}$ for Model A, and $\frac{(1+p_i^2)(9-p_i^2)}{10}$ for Model B. These variances are not constant, and therefore the Gauss-Markov theorem does not apply. OLS estimators in the well-specified statistical model, therefore, are not guaranteed to have minimum variance. The error terms do, however, all have expected value 0, and that is a sufficient condition for OLS estimators in well-specified models to be unbiased.

For a parent population, in which all p-scores from 0 to 1, with increments of 0.1, have the same frequencies, as we have in *Appendix 1—figure 1* in the main text, $\text{Var}(p)$ and $\text{Cov}\left(p, p^2\right)$ are calculated as follows.

$$\text{Var}(p) = E\left[p^2\right] - E^2[p] = \frac{1}{11} \sum_{i=0}^{10} \left( \frac{i}{10} \right)^2 - \left( \frac{1}{11} \sum_{i=0}^{10} \frac{i}{10} \right)^2 = 0.35 - 0.25 = 0.1$$

$$\text{Cov}\left(p, p^2\right) = E\left[p^3\right] - E[p] E\left[p^2\right] = \frac{1}{11} \left( \sum_{i=0}^{10} \left( \frac{i}{10} \right)^3 \right) - \left( \frac{1}{11} \sum_{i=0}^{10} \frac{i}{10} \right) \left( \frac{1}{11} \sum_{i=0}^{10} \left( \frac{i}{10} \right)^2 \right)$$

$$= 0.275 - 0.5 \cdot 0.35 = 0.1$$

The simulated numbers of offspring according to Model A result in an average number of offspring per parent of $\bar{w} = 1.50411$ – which is close to 1.5, as it should be, given the fitness function, and the state of the parent population. Also, $p' = 0.565329$, which makes the change in average p-score equal to $\Delta \bar{p} = 0.065329$, as the average p-score in the parent population is 0.5. This amounts to $\bar{w} \Delta \bar{p} = 0.0982618$, which is the left-hand side, both of the Generalized Price equation in regression form assuming Model A and of Generalized Price equation in regression form assuming Model B.

If we apply the Generalized Price equation for Model A to these data to generate the righ-hand side, we find $\hat{\alpha} = 1.01280$ and $\hat{\beta}_1 = 0.982618$, which results in

$$\bar{w} \Delta \bar{p} = 0.0982618 = 0.982618 \cdot 0.1 = \hat{\beta}_1 \cdot \text{Var}(p)$$

That means that the $\hat{\beta}_1$ is close to the true value of $\beta_1$, which is 1. Notice that also $\hat{\alpha}$ is close to its true value, which is also 1, but this does not feature in the Price equation. Here we use $\text{Var}(p) = 0.1$, as calculated above.

If we apply the Generalized Price equation for Model B to the same data, we find $\hat{\alpha} = 1.01039$; $\hat{\beta}_1 = 0.998655$; and $\hat{\beta}_2 = 0.016037$. This results in

$$\bar{w}\Delta\bar{p} = 0.0982618 = 0.998655 \cdot 0.1 - 0.016037 \cdot 0.1 = \hat{\beta}_1 \text{Var}(p) + \hat{\beta}_2 \text{Cov}(p, p^2)$$

Also here, $\hat{\alpha}$, $\hat{\beta}_1$, and $\hat{\beta}_2$ are close to their true values, which are 1, 1, and 0. Here we use $\text{Var}(p) = 0.1$ and $\text{Cov}(p, p^2) = 0.1$, both as calculated above.

The simulated numbers of offspring according to Model B result in an average number of offspring per parent of $\bar{w} = 1.34945$ – which is close to 1.35, as it should be, given the fitness function, and the state of the parent population. Also, $p' = 0.572945$, which makes the change in average $p$-score equal to $\Delta\bar{p} = 0.072945$, as the average $p$-score in the parent population is 0.5. This amounts to $\bar{w}\Delta\bar{p} = 0.0984364$.

If we apply the Generalized Price equation for Model A to these data, we find $\hat{\alpha} = 0.857273$ and $\hat{\beta}_1 = 0.984364$, and

$$\bar{w}\Delta\bar{p} = 0.0984364 = 0.984364 \cdot 0.1 = \hat{\beta}_1 \text{Var}(p)$$

Here, $\hat{\alpha}$ and $\hat{\beta}_1$ are not close to their true values, which are 1 and 0.

If we apply the Generalized Price equation for Model B to the same data, we find $\hat{\alpha} = 0.998853$; $\hat{\beta}_1 = 0.040494$; and $\hat{\beta}_2 = 0.943870$. This implies that

$$\bar{w}\Delta\bar{p} = 0.0984364 = 0.040494 \cdot 0.1 + 0.943870 \cdot 0.1 = \hat{\beta}_1 \text{Var}(p) + \hat{\beta}_2 \text{Cov}(p, p^2)$$

Now $\hat{\alpha}$, $\hat{\beta}_1$, and $\hat{\beta}_2$ are close to their true values again, which are 1, 0, and 1.

## Appendix 2

### The generalized version of Hamilton's rule

The main ingredient of this appendix is the derivation of the generalized version of Hamilton's rule. This version is derived with the Generalized Price equation. The generalized version of Hamilton's rule generalizes the original rule, in the sense that it produces a set of rules; one rule for every different model of how social interactions affect fitnesses. Every such Hamilton-like rule is generally valid; they all correctly determine when altruism, or costly cooperation, will be selected for, whatever model they are combined with. Every such rule, however, only has a meaningful interpretation in combination with the model it belongs to. The classic Hamilton's rule is the generalized Hamilton's rule that goes with the standard linear model. The insight that there are many Hamilton-like rules, all of which are generally valid, but none of which is generally meaningful, helps understand the controversy surrounding Hamilton's rule and provides a constructive way to always find a rule that both gets the direction of selection right and has a meaningful interpretation.

### 1. Introduction

Hamilton's rule is one of the more famous rules in evolutionary biology. The rule states that altruism will evolve if $rb > c$, where $b$ is the fitness benefit to the recipient, $c$ is the fitness cost to the donor, and $r$ is the genetic relatedness between them. The paper in which it was presented has about half as many citations as Darwin's '*On the Origin of Species*' (*Darwin, 1859*) and the rule is one of the core ingredients of '*The Selfish Gene*' by Richard Dawkins (*Dawkins, 1976*), which is the most popular of all popular science books. The generality of Hamilton's rule, however, has always been a topic of contention, and positions range all the way from '*Hamilton's rule almost never holds*' (*Nowak et al., 2010*) to '*Inclusive fitness is as general as the genetical theory of natural selection itself*' (*Abbot et al., 2011*). This raises a range of questions. One of them, obviously, is how generally Hamilton's rule applies. Other questions concern the debate itself and include what explains the lack of convergence.

The crucial ingredient for answering all of these questions has to do with the Price equation. This equation was not used in the original derivation of Hamilton's rule (*Hamilton, 1964b*). The 1964 paper contains a model, and the results in it follow from its assumptions (see also *van Veelen, 2007*, for a missing step in the derivation, and *van Veelen et al., 2017*, for the relation with other well-known results at the time). Later versions do use the Price equation. Hamilton himself started using it in *Hamilton, 1970* and *Hamilton, 1975*, and also later in the literature, the Price equation approach (a.k.a. the regression method) has been used for deriving Hamilton's rule. Examples include *Grafen, 1985a*; *Taylor, 1989*; *Taylor and Frank, 1996*; *Grafen, 2000*; *Grafen, 2006*; *Gardner et al., 2011*; *Rousset, 2015*; *Marshall, 2015*. Along with the Price equation came the claim of generality.

In Appendix 1, I show that what is missing in *Price, 1970* is a few lines of essential algebra. Without these, the Price equation loses the link to normal statistics and modeling. With them, the link is restored, and the Generalized Price Equation is derived. This produces a Price-like equation for every (statistical) model, as long as it includes a constant and a linear term. All of these equations are general, in the sense that they are identities, also for all models other than the ones that they belong to, or for datasets that give us no reason to think they are generated by the statistical model they are formulated for. The terms in these Price equations, however, are only meaningful within the confines of their own model.

Both the shortcomings of the original Price equation and the ways to mend them by using the generalized version are mirrored in the shortcomings of Hamilton's rule as we know it and ways to address those. Just like there is not just one Price equation, but a multitude of Price-like equations, there is not just one Hamilton's rule, but a multitude of Hamilton-like rules. All of them are correct, and all of them are general, in the sense that they always indicate the direction of selection correctly, but none of them is generally meaningful; all of them only have a meaningful interpretation when applied to the model they are associated with.

In Section 2, we look at three models and three Hamilton-like rules. If the genetic setup is binary, in the sense that it features the presence or absence of a (possibly social) gene, then these three models exhaust all possibilities. The Hamilton-like rules for all three are derived using the Generalized Price equation. This follows Section 5 of Appendix 1, but with a social interpretation of the $p$-scores and the $q$-scores featuring there. (The notation here will also deviate a bit for clarity of exposition; instead of using $\beta$'s for all coefficient in all models, we will use one Greek letter, with subscripts, per

model). This illustrates the possibility that there can be multiple rules, all of which are general, in the sense that they all get the direction of selection right for all models, while the terms in them only have a meaningful interpretation in a restricted set of models. In other words, this illustrates that rules, like statistical models, can be over- or underspecified.

In Section 3, I derive the Generalized Hamilton's rule, allowing for a richer genetic setup, where $p$-scores and $q$-scores are not restricted to be binary. This comes with a richer set of Hamilton-like rules implied by the Generalized Price equation.

## 2. Three models, three Price-like equations, three Hamilton-like rules

There are a few things that we keep simple. Reproduction is asexual. That means that we cannot think of pairs of interacting individuals as siblings or cousins, because that would require sexual reproduction. Hamilton's rule, however, is not confined by what causes relatedness between interacting individuals. The literature abounds with models with asexual selection and, for instance, local interaction structures, which induce relatedness (see for instance the overview by *Kay et al., 2020*). Asexual reproduction is easier to model, and what we find will carry over to models with sexual reproduction and kin recognition.

The classical setup with the Price equation assumes a parent population consisting of $n$ individuals. Individual $i$ is characterized by a $p$-score $p_i$ and a $q$-score $q_i$. Something else that we keep simple is that in this section, we assume that $p_i$ is binary; it is 1 if a certain gene is present, and 0 otherwise. In Section 4 of Appendix 1, we have assumed that $q$-scores represent the presence or absence of a gene or a set of genes other than those represented by the $p$-score in person $i$ herself. Here, we switch to $q$-scores representing the presence or absence of the same gene that is represented by the $p$-score, but in the individual that individual $i$ is interacting with. With sexual reproduction, we could think of this as a sibling or a cousin. With asexual reproduction, this could, for instance, be an individual linked to individual $i$ in homogeneous graph (*Taylor et al., 2007*), or an individual within the same group (*Akdeniz and van Veelen, 2020*). The derivation of the Generalized Price equation is not affected by the interpretation or the mode of reproduction (see Appendix 1).

The different models below represent different ways in which an individual's fitness can depend on its $p$-score and on its $q$-score (which is the $p$-score of its relative). In Model 1, the $q$-score does not affect individual $i$'s fitness. In Models 2 and 3, it does, and there, the $p$-score of individual $i$ will represent $i$'s level of cooperation. In evolutionary game theory terms, $p_i = 0$ implies that individual $i$ plays $D$, and $p_i = 1$ implies that individual $i$ plays $C$.

The number of offspring that any individual can have must be an integer. A model would specify or imply a random variable over these integers, and the fitness of an individual is the expected value of this random variable. The models below will therefore include an error term to reflect the randomness. We will, however, assume that the population is sufficiently large for us to approximate it with an infinitely large population. In this infinite population, the number of offspring of any individual with a certain $p$-score and a certain $q$-score is still a random variable, but the average of all individuals with the same combination of $p$-score and $q$-score coincides with the expected value that the model specifies. Assuming an infinite population therefore allows us to apply the Generalized Price equation to the model, which, at the population level, has become deterministic. Not all papers in the literature are explicit or precise about this, but here we would like to not skip over this; see Section 5 in Appendix 1.

The simple setup, with asexual reproduction and binary $p$-scores and $q$-scores, limits the scope of possibilities to three models. This is, however, enough to illustrate that there can be different rules, all of which are equally valid and equally general, while none of these rules, at the same time, is generally meaningful. They are meaningful for their own model but are over- or underspecified when applied to other models.

### Model 1, Price equation 1, rule 1

The first is the linear model, in which the $p$-score of the individual only affects her own fitness, and nobody else's. Consequently, the $q$-score, or the $p$-score of the individual it interacts with, has no effect on her fitness:

$$w_i = \beta_0 + \beta_1 p_i + \varepsilon_i$$

In an infinite population, the Generalized Price equation in regression form for the first model is

$$\bar{w}\Delta\bar{p} = \hat{\beta}_1 \mathrm{Var}\left(p\right)$$

where $\hat{\beta}_1$ is independent of the composition of the parent population, and equal to $\beta_1$. This is straightforward, but just to be sure, we do this step-by-step in Detailed Calculations 2.1, at the end of Appendix 2, also to stress the role of the assumption of large populations.

From this, we can see that $\Delta\bar{p} > 0$ if and only if $\hat{\beta}_1 > 0$. This is clearly a rule, and it is so straightforward that everyone in evolutionary biology uses it for non-social traits with linear fitness effects, often without even referring to it as a rule. The coefficient $\hat{\beta}_1$ has a meaningful interpretation; it is the effect of the gene on its carrier, and absent any effect on or from interaction partners, this is all that matters for whether this gene will be selected.

Later, when we compare the different models and the different rules, it may help to define $c = -\hat{\beta}_1$. One could write a payoff matrix for this model as follows:

$$\begin{bmatrix} -c & -c \\ 0 & 0 \end{bmatrix}$$

The payoff to an individual is $\beta_1 = -c$ with a $p$-score of 1, and 0 with a $p$-score of 0, and the fitness would then be the baseline fitness $\beta_0$ plus the payoff.

### Model 2, Price equation 2, rule 2

The second is also a linear model, but one in which the cooperation level of the individual itself not only has an effect on her own fitness (typically thought of as negative), but also on the fitness of the relative (typically thought of as positive):

$$w_i = \gamma_{0,0} + \gamma_{1,0}p_i + \gamma_{0,1}q_i + \varepsilon_i$$

In an infinite population, the Generalized Price equation in regression form for the second model is

$$\bar{w}\Delta\bar{p} = \hat{\gamma}_{1,0}\mathrm{Var}\left(p\right) + \hat{\gamma}_{0,1}\mathrm{Cov}\left(p,q\right)$$

where $\hat{\gamma}_{1,0}$ and $\hat{\gamma}_{0,1}$ are independent of the composition of the parent population (represented by $p$ and $q$) and are equal to $\gamma_{1,0}$ and $\gamma_{0,1}$, respectively. This can also be rewritten as

$$\bar{w}\Delta\bar{p} = \left(\hat{\gamma}_{1,0} + \frac{\mathrm{Cov}\left(p,q\right)}{\mathrm{Var}\left(p\right)}\hat{\gamma}_{0,1}\right)\mathrm{Var}\left(p\right)$$

From this, we can see that $\Delta\bar{p} > 0$ if and only if $\hat{\gamma}_{1,0} + \frac{\mathrm{Cov}(p,q)}{\mathrm{Var}(p)}\hat{\gamma}_{0,1} > 0$. In infinite population models, $\frac{\mathrm{Cov}(p,q)}{\mathrm{Var}(p)}$ will coincide with relatedness $r$ between the individuals that interact, and if we then define $c = -\gamma_{1,0}$ and $b = \gamma_{0,1}$, we can rewrite this as

$$\bar{w}\Delta\bar{p} = \left(-c + rb\right)\mathrm{Var}\left(p\right)$$

Here, we naturally recognize Hamilton's rule. The payoff matrix for model 2, moreover, makes this a prisoner's dilemma with equal gains from switching:

$$\begin{bmatrix} b - c & -c \\ b & 0 \end{bmatrix}$$

### Model 2, Price equation 1

It is worth noticing that we can also apply the first Price-like equation to the second model. If we do, we get

$$\bar{w}\Delta\bar{p} = \hat{\beta}_1 \mathrm{Var}\left(p\right)$$

Detailed Calculations 2.2, at the end of Appendix 2, shows that $\hat{\beta}_0 = \gamma_{0,0} + \left(1 - \frac{\text{Cov}(p,q)}{\text{Var}(p)}\right)\bar{p}\gamma_{0,1}$ and $\hat{\beta}_1 = \gamma_{1,0} + \frac{\text{Cov}(p,q)}{\text{Var}(p)}\gamma_{0,1}$ minimize the sum of squared errors for Price-like equation 1 in combination with Model 2. (One can also derive $\hat{\beta}_1$ in a more direct way than is done in Detailed Calculations 2.2. Given that both Price-like Equation 1 and Price-like equation 2 are identities, and both have $\bar{w}\Delta\bar{p}$ on the left-hand side, it must be that their right-hand sides are also equal. This implies that $\hat{\beta}_1 \text{Var}(p) = \left(\gamma_{1,0} + \frac{\text{Cov}(p,q)}{\text{Var}(p)}\gamma_{0,1}\right)\text{Var}(p)$, and therefore that $\hat{\beta}_1 = \left(\gamma_{1,0} + \frac{\text{Cov}(p,q)}{\text{Var}(p)}\gamma_{0,1}\right)$. This direct way is possible here because there is only one variable, $\hat{\beta}_1$, and one equation. For some later combinations of models and Price-like equations, there is still one equation, but more degrees of freedom. There, this shortcut is not possible.) The first, $\hat{\beta}_0$, does not feature in the Price equation, but $\hat{\beta}_1$ does. Here, I write $\hat{\beta}_1$, because it minimizes the least squares of the errors relative to Model 1. That means it would be equal to $\beta_1$, when applied to Model 1, but now that it is applied to Model 2, it does not coincide with any parameter. This Price-like equation is, however, still an identity, also when applied to the second model, and it still gets the direction of selection right; $\Delta\bar{p} > 0$ if and only if $\hat{\beta}_1 > 0$ (in fact, it would get the direction of selection right for any model that has a constant and a linear term).

The natural interpretation of this rule would be that the effect of having the gene on the individual herself is $\hat{\beta}_1$, which is equal to $\gamma_{1,0} + r\gamma_{0,1}$. This is clearly not the case, unless $\gamma_{0,1} = 0$, in which case we are back in the situation of Model 1. It is however important to note that we now have *two* rules, both of which we applied to Model 2, and both of which get the direction of selection right. There is a silly one, in which we would choose $c = -\left(\gamma_{1,0} + r\gamma_{0,1}\right)$, and, if one would insist on making it look similar to Hamilton's rule, $b = 0$; and a more sensible one, with $c = -\gamma_{1,0}$ and $b = \gamma_{0,1}$, that everyone would agree is the right one for this model. We will get back to the reasons why we choose Price-like equation 2 here, and not Price-like equation 1, when we compare Models 2 and 3.

## Symptoms of underspecification

If we combine the Generalized Price equation in regression form for Model 1 with Model 1, in an infinite population, then the sum of squared errors is equal to the variance of the noise term $\text{Var}(\varepsilon)$. If we combine the Generalized Price equation in regression form for Model 1 with Model 2, in an infinite population, then the sum of squared errors becomes larger than the variance of the noise term $\text{Var}(\varepsilon)$. This is the result of under-specification.

In *Appendix 1—figure 1*, we see a very simple example of how underspecification can increase the sum of squared errors over the unavoidable level induced by the noise. The example does not illustrate the specific form of underspecification that we get if we leave out the *q*-score as an explanatory variable. Instead, it illustrates the principle by considering an even simpler form of underspecification, in which the *p*-score is left out as an explanatory variable. Both symptoms of underspecification show up here; the sum of squared errors is higher than is justified by the error term of the true model; and the estimator of the coefficient depends on the composition of the parent population. Both of these disappear when we use the correctly specified model instead.

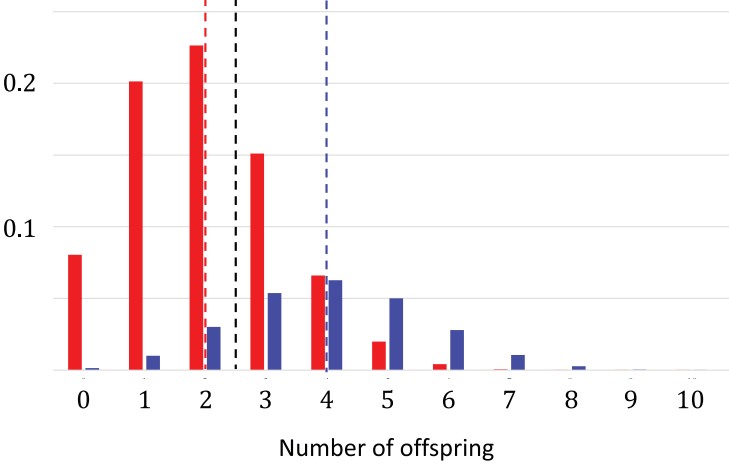

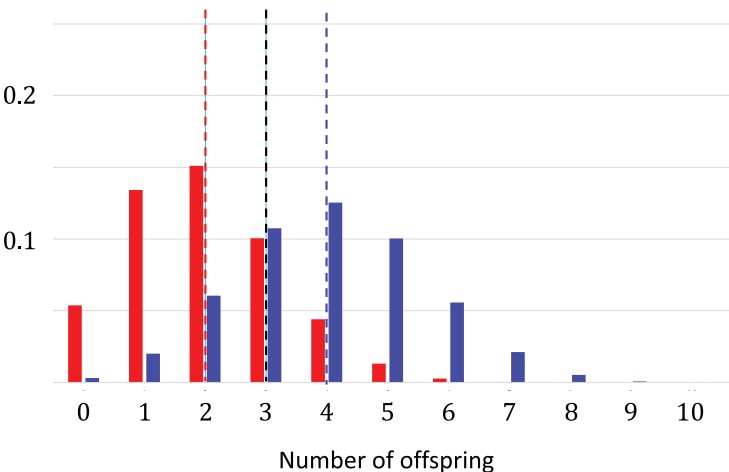

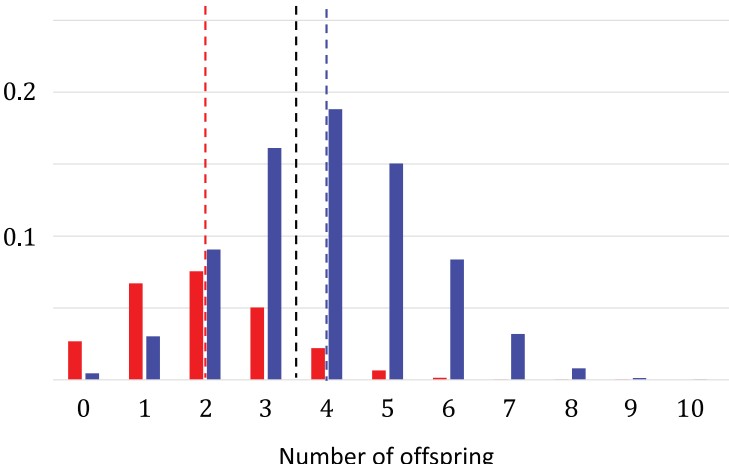

**Appendix 2—figure 1.** Symptoms of underspecification. The red bars indicate frequencies of parents with a *p*-score of 0, and with 0, 1, …, 10 offspring for the example described in the text. The blue bars indicate frequencies

*Appendix 2—figure 1 continued on next page*

*Appendix 2—figure 1 continued*

of parents with a $p$-score of 1, and with 0, 1, …, 10 offspring. The composition of the parent population differs between the panels; in the top panel, 1 out of 4 parents has a $p$-score of 1; in the middle panel, this is 2 out of 4 parents; and in the bottom panel, it is 3 out of 4 parents. For the correctly specified model $w_i = \alpha + \beta p_i + \varepsilon_i$, the red and the blue dotted lines indicate the values for $\alpha$ and for $\alpha + \beta$, if those are chosen so as to minimize the sum of squared errors relative to the correct model. The $\alpha$ is the expected value of the number of offspring for parents with a $p$-score of 0, while $\alpha + \beta$ is the expected value of the number of offspring for parents with a $p$-score of and 1. The $\alpha$ and $\beta$ that minimize the sum of squared errors do not depend on the composition of the population. For the mis-specified model $w_i = \alpha + \varepsilon_i$, the black dotted lines indicate the $\alpha$'s for which the sum of squared errors is minimized. These do move around. The sum of squared errors when using the mis-specified model is also much larger at the minimum.

In the example, parents with a $p$-score of 0 draw their offspring from a binomial distribution with 10 trials and a success probability of $\frac{1}{5}$, while parents with a $p$-score of 1 draw their offspring from a binomial distribution, also with 10 trials, but with a success probability of $\frac{2}{5}$. Their $p$-score, therefore, determines the expected value of their number of offspring; the expected number of offspring is $10 \cdot \frac{1}{5} = 2$ for parents with a $p$-score of 0, and $10 \cdot \frac{2}{5} = 4$ for parents with a $p$-score of 1. If we ignore this and mis-specify the model by leaving out the $p$-score as an explanatory variable, then the model becomes $w_i = \alpha + \varepsilon_i$. Which value of the $\alpha$ minimizes the sum of squared errors will now depend on the composition of the population. In the first panel, 1 out of 4 parents has a $p$-score of 1, and the sum of squared errors is minimized at $\alpha = 2\frac{1}{2}$; in the second panel, 1 out of 2 parents has a $p$-score of 1, and the sum of squared errors is minimized at $\alpha = 3$; and in the third panel, 3 out of 4 parents has a $p$-score of 1, and the sum of squared errors is minimized at $\alpha = 3\frac{1}{2}$. These are the black dotted lines.

The sum of squared errors is significantly reduced if we do include the $p$-score as an explanatory variable, so the model becomes $w_i = \alpha + \beta p_i + \varepsilon_i$. In that case, the expected number of offspring of parents with a $p$-score of 0 is 2, and the expected number of offspring of parents with a $p$-score of 1 is 4, both regardless of the composition of the parent population.

## Model 3, Price equation 3, rule 3

The third model allows for an interaction effect between the $p$-score and the $q$-score. This could, for instance, reflect an efficiency gain from joint cooperation, if positive:

$$w_i = \delta_{0,0} + \delta_{1,0}p_i + \delta_{0,1}q_i + \delta_{1,1}p_iq_i + \varepsilon_i$$

In an infinite population, the Generalized Price equation in regression form for the third model is

$$\bar{w}\Delta\bar{p} = \hat{\delta}_{1,0}\mathrm{Var}\left(p\right) + \hat{\delta}_{0,1}\mathrm{Cov}\left(p,q\right) + \hat{\delta}_{1,1}\mathrm{Cov}\left(p,pq\right)$$

where $\hat{\delta}_{1,0}$, $\hat{\delta}_{0,1}$, and $\hat{\delta}_{1,1}$ are independent of the composition of the parent population, represented by $p$ and $q$, and they are equal to $\delta_{1,0}$, $\delta_{0,1}$, and $\delta_{1,1}$, respectively. This can be rewritten as

$$\bar{w}\Delta\bar{p} = \left(\hat{\delta}_{1,0} + \frac{\mathrm{Cov}\left(p,q\right)}{\mathrm{Var}\left(p\right)}\hat{\delta}_{0,1} + \frac{\mathrm{Cov}\left(p,pq\right)}{\mathrm{Var}\left(p\right)}\hat{\delta}_{1,1}\right)\mathrm{Var}\left(p\right)$$

From this, we can see that $\Delta\bar{p} > 0$ if and only if $\hat{\delta}_{1,0} + \frac{\mathrm{Cov}(p,q)}{\mathrm{Var}(p)}\hat{\delta}_{0,1} + \frac{\mathrm{Cov}(p,pq)}{\mathrm{Var}(p)}\hat{\delta}_{1,1} > 0$. If we then define $c = -\delta_{1,0}$, $r_{0,1} = \frac{\mathrm{Cov}(p,q)}{\mathrm{Var}(p)}$, $b_{0,1} = \delta_{0,1}$, $r_{1,1} = \frac{\mathrm{Cov}(p,pq)}{\mathrm{Var}(p)}$, and $b_{1,1} = \delta_{1,1}$, then we can rewrite this as

$$\bar{w}\Delta\bar{p} = \left(-c + r_{0,1}b_{0,1} + r_{1,1}b_{1,1}\right)\mathrm{Var}\left(p\right)$$

This does not give us the Hamilton's rule we are familiar with, but it does give us a correct criterion for when higher $p$-scores are selected for, in the same way that the derivation of the familiar Hamilton's rule did for Model 2, and the derivation of the first rule did for Model 1; $\Delta\bar{p} > 0$ if and only if $r_{0,1}b_{0,1} + r_{1,1}b_{1,1} > c$. This is the rule suggested by Queller (*Queller, 1985*), if we assume that genotype and phenotype correlate perfectly.

The payoff matrix for this model is a prisoner's dilemma without equal from switching:

$$\begin{bmatrix} b_{0,1} + b_{1,1} - c & -c \\ b_{0,1} & 0 \end{bmatrix}$$

As before, it is worth noticing that we can apply the earlier Price-like equations to the third model too.

## Model 3, Price equation 1

If we apply the Price-like equation that goes with the first model to the third model, we get

$$\bar{w}\Delta\bar{p} = \hat{\beta}_1 \text{Var}(p)$$

with $\hat{\beta}_1 = \delta_{1,0} + \frac{\text{Cov}(p,q)}{\text{Var}(p)}\delta_{0,1} + \frac{\text{Cov}(p,pq)}{\text{Var}(p)}\delta_{1,1}$. Here we use the more direct calculation, suggested above, when Price equation 1 was applied to Model 2. This Price-like equation is still an identity, also when applied to the third model, and it still gets the direction of selection right; $\Delta\bar{p} > 0$ if and only if $\hat{\beta}_1 > 0$.

The natural interpretation of this rule would be that the effect of having the gene on the individual herself is $\hat{\beta}_1$, which is equal to $\delta_{1,0} + \frac{\text{Cov}(p,q)}{\text{Var}(p)}\delta_{0,1} + \frac{\text{Cov}(p,pq)}{\text{Var}(p)}\delta_{1,1}$ here. This is clearly not the case, unless $\delta_{0,1} = \delta_{1,1} = 0$, in which case we are back in Model 1.

## Model 3 with Price equation 2

If we use the Price-like equation that goes with the second model, and apply it to the third model, we get

$$\bar{w}\Delta\bar{p} = \left( \hat{\gamma}_{1,0} + \frac{\text{Cov}(p,q)}{\text{Var}(p)}\hat{\gamma}_{0,1} \right) \text{Var}(p)$$

$$\hat{\gamma}_{1,0} = \delta_{1,0} - \delta_{1,1} \left( \frac{\text{Cov}(p,pq)\text{Var}(q) - \text{Cov}(p,q)\text{Cov}(q,pq)}{(\text{Cov}(p,q))^2 - \text{Var}(p)\text{Var}(q)} \right)$$

$$\hat{\gamma}_{0,1} = \delta_{0,1} + \delta_{1,1} \left( \frac{\text{Cov}(p,q)\text{Cov}(p,pq) - \text{Cov}(q,pq)\text{Var}(p)}{(\text{Cov}(p,q))^2 - \text{Var}(p)\text{Var}(q)} \right)$$

The derivation is a bit long and boring and can be found in Detailed Calculations 2.3 at the end of Appendix 2. This Price-like equation is still an identity, also when applied to the third model, and it still gets the direction of selection right; $\Delta\bar{p} > 0$ if and only if $\hat{\gamma}_{1,0} + \frac{\text{Cov}(p,q)}{\text{Var}(p)}\hat{\gamma}_{0,1} > 0$.

The natural interpretation of this rule would be that the effect of having the gene on the individual herself is $\hat{\gamma}_{1,0}$, of which the first formula above describes how it depends on $\delta_{0,1}$ and $\delta_{1,1}$, while the effect of the $q$-score is $\hat{\gamma}_{0,1}$, of which the first formula above describes how it depends on $\delta_{0,1}$ and $\delta_{1,1}$. This is clearly not the case, unless $\delta_{1,1} = 0$, in which case we are back in the situation of Model 2.

The last expression looks like the traditional Hamilton's rule, with $c = -\hat{\gamma}_{1,0}$ and $b = \hat{\gamma}_{0,1}$. It should be noted, though, that the expressions for the $b$ and the $c$ are not constants, and while they depend on both model parameters, which could in principle be fine, they also depend on population state properties, which is not.

For Model 2, we chose rule 2 (the traditional Hamilton's rule), and not rule 1 (the standard rule for non-social traits). The reason to do this is not that rule 1 gets the direction of selection wrong; it does not. The reason is that the latter mistakenly suggests that having the trait comes with a fitness benefit to oneself, which is not true; it comes with a cost to oneself and a benefit to the other. The reason it can nonetheless go up in frequency is that because of relatedness, those that bear the costs are also disproportionately often on the receiving end. Rule 2 does, and rule 1 does not, reflect that. A symptom of the misspecification, if we nonetheless use rule 1, is that $\hat{\beta}_1$ (the estimator of the effect on oneself) depends on the composition of the population.

If we follow the same logic regarding choosing rules for model 3, then we should go with rule 3, that is, $r_{0,1}b_{0,1} + r_{1,1}b_{1,1} > c$, or Queller's rule, and not rule 2, which is the traditional Hamilton's rule. The latter would still get the direction of selection right, but it would mistakenly suggest that

the effect of having the trait on the other is independent of whether or not the other has it too (or in other words: it would be blind to the fact that there is an interaction term). A symptom of this misspecification is that $\hat{\gamma}_{1,0}$ and $\hat{\gamma}_{0,1}$ (the estimator of the effect on oneself and the other, respectively) both depend on the composition of the population.

The relation between these examples can be summarized in a table that, for three models and three rules, represents all nine combinations of them.

For the combinations in yellow, the rules are more general than the models need them to be. We did not discuss those combinations above, but they are relatively easy to check. Since the rules are nested, in the sense that rule 1 is a special case of rule 2, and rule 2 is a special case of rule 3, all we need to do is choose zeros for the unused variables. In that sense, these combinations do not really produce rules that are different, or that invite different interpretations; they are only written a bit more inefficiently, by adding terms that end up being 0. All of these are stable across population states.

That is not true for the combinations in red. There the $c$ and the $b$'s vary with the population state, and the rules are not general enough for their model. For the combinations in green, none of the $b$'s and $c$'s in the rule is 0 (which implies that the rule is not overspecified for the model) and none of them depend on population properties (which implies that they are also not underspecified).

**Appendix 2—table 1.** Three rules and three models.

This table shows all combinations of the three rules and the three models. All rules indicate the direction of selection correctly for all models. Yellow indicates a combination of a rule and a model, where the rule is more general than is needed for the model. This leads to one or more $b$'s being 0. Red indicates a combination of a rule and a model, where the rule is not general enough for the model. This leads to one or more $b$'s and $c$'s that depend on the population state. Terms that depend on the population state are abbreviated as follows: $r = r_{0,1} = \frac{\text{Cov}(p,q)}{\text{Var}(p)}$, $r_{1,1} = \frac{\text{Cov}(p,pq)}{\text{Var}(p)}$, $s_b = \frac{\text{Cov}(p,q)\,\text{Cov}(p,pq)-\text{Cov}(q,pq)\,\text{Var}(p)}{(\text{Cov}(p,q))^2-\text{Var}(p)\,\text{Var}(q)}$, and $s_c = \frac{\text{Cov}(p,pq)\,\text{Var}(q)-\text{Cov}(p,q)\,\text{Cov}(q,pq)}{(\text{Cov}(p,q))^2-\text{Var}(p)\,\text{Var}(q)}$. Rule 1 is the standard rule for non-social traits. Rule 2 is the classical Hamilton's rule. Rule 3 is a rule that allows for an interaction effect. An appropriate name for this rule would be Queller's rule (**Queller, 1985**). This rule can in turn be nested in a sequence of ever more general rules if we allow for $p$- and $q$-scores that are not restricted to be binary.

| | | Model 1: $\alpha + \beta p_i$ | Model 2: $\alpha + \gamma_{1,0} p_i + \gamma_{0,1} q_i$ | Model 3: $\alpha + \delta_{1,0} p_i + \delta_{0,1} q_i + \delta_{1,1} p_i q_i$ |
|---|---|---|---|---|
| **Rule 1:** | | | | |
| $0 > c$ | $c =$ | $-\beta$ | $-(\gamma_{1,0} + r_{0,1}\gamma_{0,1})$ | $-(\delta_{1,0} + r_{0,1}\delta_{0,1} + r_{1,1}\delta_{1,1})$ |
| **Rule 2:** | | | | |
| $rb > c$ | $c =$ | $-\beta$ | $-\gamma_{1,0}$ | $-\delta_{1,0} + s_c\delta_{1,1}$ |
| | $b =$ | $0$ | $\gamma_{0,1}$ | $\delta_{0,1} + s_b\delta_{1,1}$ |
| **Rule 3:** | | | | |
| $r_{0,1}b_{0,1} + r_{1,1}b_{1,1} > c$ | $c =$ | $-\beta$ | $-\gamma_{1,0}$ | $-\delta_{1,0}$ |
| | $b_{0,1} =$ | $0$ | $\gamma_{0,1}$ | $\delta_{0,1}$ |
| | $b_{1,1} =$ | $0$ | $0$ | $\delta_{1,1}$ |

This matrix of combinations of models and rules also helps understand what causes the contentiousness of the debate on the generality of Hamilton's rule, and why there are no signs of convergence. Rule 2, which is the rule that we get from applying the standard Price equation in regression form (also known as using the regression method), is a completely general rule, in the sense that whatever the true model is, it always gets the direction of selection right. This is true, but that fact is not a good argument for singling this rule out as more helpful, meaningful, or

insightful than other rules. In the example above, we have seen that if we apply the Generalized Price equation in regression form (which is also using the regression method, but now with a richer menu of alternative underlying statistical models), then this can also give us Rule 1 or Rule 3, depending on the statistical model we use. These rules are equally correct, in the sense that they also always get the direction of selection right, and they are also equally general, in the sense that they get it right for every possible model. Being a general rule, that always gets the direction of selection right, therefore, cannot be a criterion for elevating Rule 2 (which is the classical Hamilton's rule), above the other ones, because Rules 1 and 3 are also general rules, that always get the direction of selection right.

Since being completely general and always getting the direction of selection right does not single out any of the possible rules, we need additional criteria. A natural criterion would be that besides being correct, the terms in the rule would have to be meaningful. More precisely, we think the rule should do what Rule 1 does for Model 1, and what Rule 2 does for Model 2, and that is to separate model parameters from properties of the population state. As described above, we do not apply Rule 1 – the rule for non-social traits – to a social trait with fitness effects described by Model 2, and the reason why we do not do this is not that it does not get the prediction right. It does. The reason why we do not apply Rule 1 to Model 2 is that Rule 1 is mis-specified for Model 2, as it does not reflect the fact that Model 2 allows for a trait with a fitness cost to the individual itself, that is nonetheless selected, because of the positive fitness effects between related individuals that outweigh the negative fitness effect on oneself. In other words, it would be wrong to interpret a negative $c$ in Rule 1, applied to Model 2, as the fitness effect on oneself.

The exact same reason would suggest not to apply Rule 2, which is the traditional Hamilton's rule, to models of social interactions that do not fit Model 2, such as for instance Model 3. The argument against interpreting the $c$ in Rule 1, applied to Model 2, as the fitness effect on oneself carries over to this combination of rule and model, as it implies that we can also not interpret the $c$ and $b$ in Rule 2, when applied to Model 3, as the costs and benefits of the social behaviour.

One of the reasons why the debate in the literature on the generality of Hamilton's rule is so long-lasting is that it focuses on whether or not rules are correct, and not on whether they are (also) meaningful. One side of the debate has always returned to the argument that Rule 2 is general and correct – whatever the model or the data (see for instance *Grafen, 1985b*; *Abbot et al., 2011*; *Gardner et al., 2011*; *Marshall, 2015*; *Rousset, 2015*). This is true, but it is not an argument for elevating Rule 2 above other rules. The other side of the debate keeps bringing up models that do not fit Model 2 (see for instance *Karlin and Matessi, 1983*; *Matessi and Karlin, 1984*; *Queller, 1985*; *van Veelen, 2009*). Sometimes arguments on this side take a completely different approach, and rather than describing selection with the Price equation, and then worry about whether or not one can interpret regression coefficients in it as benefits and costs, they start with models with a priori interpretable definitions for $b$ and $c$, for instance by using what is called the counterfactual approach, rather than the regression approach, to define costs and benefits (*van Veelen et al., 2017*; *van Veelen, 2018a*). Unlike the regression approach, the counterfactual approach to defining costs and benefits can result in rules that end up getting the direction of selection wrong (*Karlin and Matessi, 1983*; *Matessi and Karlin, 1984*; *van Veelen et al., 2017*; *van Veelen, 2018a*).

### Grafen's (1985) News and Views on Queller's rule

When Queller's rule (*Queller, 1985*) was published, this opened the door to the idea that Hamilton's rule was nested in a more general rule, or a more general set of rules. A News and Views responding to it (*Grafen, 1985b*), however, immediately closed that door again, by claiming that

"*the third, synergistic, term in Queller's form can be made to disappear by agreeing to define benefit and cost as the average effects on individual's fitnesses, rather than as arbitrary terms in a model of fitness. So in Queller's simple model, Hamilton's rule, with costs and benefits correctly understood, is perfectly adequate for deciding the direction of change in gene frequency.*"

This is first of all a representative example where Hamilton's rule getting the direction of selection right is treated as a decisive argument in its favor, and as an argument against a different rule, even though this other rule also gets the direction of selection right. Regarding the interpretation of benefit and cost as the 'average effects on individual's fitnesses', Detailed Calculations 2.4.5 at the very end of Appendix 2 shows that this claim is simply incorrect; the benefits and costs that make Hamilton's rule work are the ones that we get by applying the Generalized Price equation for Model

2 to Model 3, and these are **not** the average effects on the individual's fitnesses. Also, there is nothing about the terms in Queller's rule (**Queller, 1985**) that makes them conceptually any different from the terms in Hamilton's rule. Suggesting that Queller's rule (**Queller, 1985**) includes terms that are arbitrary in ways that terms in Hamilton's rule are not, therefore, is also incorrect. The literature has nonetheless effectively abandoned this path forward immediately after Queller (**Queller, 1985**) pointed to it.

## Rousset's (2015) claims regarding misspecification

In a paper about the regression method, (**Rousset, 2015**) claims that there is no misspecification.

"*In the presence of synergies, the residuals have zero mean and are uncorrelated to the predictors. No further assumption is made about the distribution of the residuals. Thus, there is no sense in which the regression is misspecified.*"

This is remarkable in a few ways. What is being discussed in this part of **Rousset, 2015** is what is described above under **Model 3 with Price equation 2**, where Model 3 ($w_i = \alpha + \beta_{1,0}p_i + \beta_{0,1}q_i + \beta_{1,1}p_iq_i + \varepsilon_i$) is replaced with Model 2 ($w_i = \alpha + \beta_{1,0}p_i + \beta_{0,1}q_i + \varepsilon_i$), and where the parameters in Model 2 are chosen so that they minimize the sum of squared differences between Model 2 and Model 3. This is just the definition of misspecification.

The second remarkable ingredient is the criterion used in order to determine whether or not a model is misspecified. While **Rousset, 2015** finds itself in the modeling domain, it does point to the field of statistics here, by stating that "*the residuals have zero mean and are uncorrelated to the predictors*". From this, the paper concludes that "*there is no sense in which the regression is misspecified*". The criterion that **Rousset, 2015** uses is that the model is well-specified if there is no correlation between the residuals (here: $\varepsilon_i = \hat{w}_i - w_i$) and the variables included in the reference model (here: $p_i$ and $q_i$). According to this criterion, however, all models would always be well-specified. As we have seen at the end of Sections 2 and 4 in Appendix 1, the residuals always have mean zero, and they are always uncorrelated with the predictors included in the reference model. This, therefore, cannot be the right criterion. The correct criterion, however, is not just that there should be no correlation between the errors and the variables included in the model; the correct criterion is that there should also be no correlation between the errors and variables *not included in the reference model* – over and above what one would expect as a result of the randomness in the model. In this case, we are moreover in a modeling context, so we do not even have to worry about whether or not we can identify the underlying model, and we can forget about the 'over and above' part; there should just also not be any correlation between the errors and the variables not included in the model. Here, the residuals are correlated with $p_iq_i$, which is the variable that is included in Model 3, but not in Model 2. Therefore, according to the correct version of this criterion, this model is in fact misspecified – as it should be, because getting the model wrong is the definition of misspecification. For more details, I would refer to Section 1b (viii) in **van Veelen, 2020b**.

## 3. The general version of Hamilton's rule

If $p$- and $q$-scores can only be 0 or 1, there are only four possibilities combinations of a $p$- and a $q$-score that one can have. If we denote the expected number of offspring of an individual with a $p$-score of $p$ and a $q$-score of $q$ as $E\left(w^{p,q}\right)$, then there are also only four of those. In Model 3 above, fitness depends on $p$- and a $q$-scores as follows:

$$w_i = \delta_{0,0} + \delta_{1,0}p_i + \delta_{0,1}q_i + \delta_{1,1}p_iq_i + \varepsilon_i$$

That implies that $E\left(w^{0,0}\right) = \delta_{0,0}$, $E\left(w^{1,0}\right) = \delta_{0,0} + \delta_{1,0}$, $E\left(w^{0,1}\right) = \delta_{0,0} + \delta_{0,1}$, and $E\left(w^{1,1}\right) = \delta_{0,0} + \delta_{1,0} + \delta_{0,1} + \delta_{1,1}$. With only four values for $E\left(w^{p,q}\right)$ and four $\delta$'s, there is no scope for introducing more coefficients in the model.

If we also allow for $p$- and $q$-scores other than 0 and 1, fitnesses may depend on those in richer ways than the three models above allow for. Below, we will consider a general set of models, which are described by the coefficients that are included. The set of (non-zero) coefficients is denoted by $E$, and if $(r,s) \in E$, then the model includes the term $\beta_{r,s}p_i^rq_i^s$. Given a set of coefficients $E$, the model would be

$$w_i = \sum_{(k,l) \in E} \beta_{k,l} p_i^k q_i^l + \varepsilon_i$$

where $\varepsilon_i$ is a noise term with expected value 0. Whether including a term in the model would be of added value in describing actual fitness effects in a real-life example is, of course, a matter of statistics. For every behavior, there is a point where additional coefficients are not going to be statistically significant, even with a lot of data. Depending on the behavior that the model aims to describe, it is an empirical question for what set $E$ – or, in other words, for what model – that happens. For non-social behaviors, Model 1 above could be enough, which would imply that only the coefficients for $p_i^0 q_i^0 = 1$ and $p_i^1 q_i^0 = p_i$ are needed. Alternatively, it could be that the behavior is non-social, but the fitness is not linear in the $p$-score. This does, of course, require non-binary $p$-scores, and if we allow for those, then higher order terms could be included. Those would be terms $p_i^i q_i^0 = p_i^i$. If the $q$-score represents the $p$-score of the interaction partner, and the behavior is non-social, then the $q$-score does not affect fitness, and hence only coefficients where the exponent of $q_i$ is 0 are included.

For social behaviors, Model 2 could describe the fitness accurately, and for those, the set of nonzero coefficients would be $E = \{(0,0),(1,0),(0,1)\}$. Model 3 is also an option, and this would amount to choosing $E = \{(0,0),(1,0),(0,1),(1,1)\}$. This setup, however, also allows for larger sets $E$ that include coefficients that are not included in Model 3.

The Generalized Price equation in regression form can now be stated for any statistical model, specified by the set of coefficients $E$ we include. If we choose a set $E$, and we choose the coefficients $\hat{\beta}_{r,s}$ so as to minimize the sum of squared errors relative to the model given by $w_i = \sum_{(k,l) \in E} \beta_{k,l} p_i^k q_i^l + \varepsilon_i$, then the Generalized Price equation in regression form for this choice of $E$ reads

$$\bar{w}\Delta\bar{p} = \left( \sum_{(k,l) \in E} \hat{\beta}_{k,l} \frac{\mathrm{Cov}\left(p, p^k q^l\right)}{\mathrm{Var}\left(p\right)} \right) \mathrm{Var}\left(p\right) + E\left(w\Delta p\right)$$

Here it is worth emphasizing that this produces a Price-like equation for every choice of the set of coefficients $E$. That implies that one can make any combination of a set of coefficients $E$ that is used for writing a Price-like equation, and a set of coefficients $E'$ for the model that one would want to consider. Even if these sets differ, the Price-like equation for $E$, applied to a model specified by $E'$, remains an identity. If $E$ is a subset of $E'$, then the Price-like equation is underspecified. If $E'$ is a subset of $E$, then the Price-like equation is overspecified, and the coefficients that are in $E$, but not in $E'$, will be 0. It is also possible that $E$ is not a subset of $E'$, and $E'$ is also not a subset of $E$. For such a combination, one could say that the Price-like equation based on the set of coefficients $E$, and applied to a model specified by $E'$, is both over- and underspecified. If $E$ and $E'$ coincide – that is, if the model used as a basis for the Price-like equation and the model we are studying are the same – then $\hat{\beta}_{k,l} = \beta_{k,l}$ for all coefficients, or, in other words, the coefficients in the Price-like equation coincide with the coefficients of the model we are considering. This generalizes the point made earlier, which is that all of these Price-like equations are identities, but the regression coefficients $\hat{\beta}_{k,l}$ in it only have a meaningful interpretation if the model is correctly specified.

If we define $r_{k,l} = \frac{\mathrm{Cov}\left(p, p^k q^l\right)}{\mathrm{Var}\left(p\right)}$, then the Generalized Hamilton's rule is that $\Delta\bar{p} > 0$ if

$$\sum_{(k,l) \in E} r_{k,l} b_{k,l} > 0$$

where $b_{k,l} = \beta_{k,l}$, and $r_{k,l} = \frac{\mathrm{Cov}\left(p, p^k q^l\right)}{\mathrm{Var}\left(p\right)}$, both for all $(k,l) \in E'$.

Because $r_{0,0} = \frac{\mathrm{Cov}\left(p, p^0 q^0\right)}{\mathrm{Var}\left(p\right)} = \frac{\mathrm{Cov}\left(p, 1\right)}{\mathrm{Var}\left(p\right)} = 0$, it does not matter if we leave the coefficient for $(0,0)$ in or take it out. In the main text, we moreover write $c$ for $-b_{1,0} = -\hat{\beta}_{1,0}$, and put it on the right-hand side of this inequality.

The rule for selection of non-social traits with linear effects is a special case, if we choose the model that only contains $\beta_{1,0}$. Because $r_{1,0} = \frac{\mathrm{Cov}\left(p, p^1 q^0\right)}{\mathrm{Var}\left(p\right)} = \frac{\mathrm{Var}\left(p\right)}{\mathrm{Var}\left(p\right)} = 1$, this produces the rule that a trait is selected if it increases an individual's fitness, or, in other words, if $\hat{\beta}_{1,0} > 0$. This still requires that the model is correctly specified, because only then do we have $\hat{\beta}_{1,0} = \beta_{1,0}$.

For non-social traits with (also) quadratic fitness effects, we would have to allow $\beta_{1,0}$ and $\beta_{2,0}$ to be non-zero. The rule then is $b_{1,0} + b_{2,0} \frac{\text{Cov}(p,p^2)}{\text{Var}(p)} > 0$. If we additionally assume random mating, then this simplifies to $b_{1,0} + b_{2,0} \left( \frac{1}{2} + p \right)$, as we have seen in Section 5 of Appendix 1.

The original Hamilton's rule is also a special case, if we choose the model that contains only $\beta_{1,0}$ and $\beta_{0,1}$. Because $r_{0,1} = \frac{\text{Cov}(p,p^0 q^1)}{\text{Var}(p)} = \frac{\text{Cov}(p,q)}{\text{Var}(p)}$, the rule then becomes that $\Delta \bar{p} > 0$ if $r_{1,0} b_{1,0} + r_{0,1} b_{0,1} > 0$, and with $r_{1,0} = 1$, $c = -\hat{\beta}_{1,0}$, $r = r_{0,1}$, and $b = \hat{\beta}_{0,1}$, this gives us the Hamilton's rule we are familiar with. This also still requires that the model is correctly specified, because only then do we have $\hat{\beta}_{1,0} = \beta_{1,0}$ and $\hat{\beta}_{0,1} = \beta_{0,1}$.

The message from the Generalized Price equation now carries over to the generalized version of Hamilton's rule. There is, in fact, a Hamilton-like rule for every choice of $E$, and they all get the direction of selection right, but only if the set of coefficients matches the set of coefficients of the model we are considering, do they have a meaningful interpretation.

## 4. Relatedness

There are two remarks to be made regarding relatedness.

The first is that in models, both the classic relatedness $r$ from Hamilton's original rule and the general relatednesses $r_{k,l} = \frac{\text{Cov}(p,p^r q^s)}{\text{Var}(p)}$, of which the classical relatedness is a special case ($r = r_{0,1}$), are properties of the population structure. That means that in models where we assume infinitely large populations, the law of large numbers guarantees that the randomness disappears, and probabilities and frequencies coincide. For instance, if we consider full siblings, and assume random mating, then that implies that the probability that one sibling has a gene if the other has it is $(1 - r) \bar{p} + r = \frac{1}{2}\bar{p} + \frac{1}{2}$, where $\bar{p}$ is the share of the population that has it (see Detailed Calculations 2.4 at the end of Appendix 2). The model assumptions, including the assumption of an infinitely large population, then imply that the share of individuals whose sibling has the gene, out of those that have it themselves, is in fact $\frac{1}{2}\bar{p} + \frac{1}{2}$.

With data, on the other hand, the $\frac{\text{Cov}(p,p^r q^s)}{\text{Var}(p)}$ term is an *estimator* of a property of the true, underlying population structure. With few observations, this estimator will be off, on average, more than with many observations. In the first part, where we looked at the Generalized Price equation, we stressed that this is true for the coefficients in it (the $\beta$'s, $\gamma$'s, and $\delta$'s), when considering data, but it is worth stressing that this is also true for the terms $\frac{\text{Cov}(p,p^r q^s)}{\text{Var}(p)}$, when these relate to properties of the population structure. In subsection 4.1 below, I will elaborate on the importance of not mixing up relatedness and estimators of relatedness.

The second remark is that in many infinite population models, relatedness $r = r_{0,1}$ is frequency-independent (although one can think of models in which this is not the case; see for instance **van Veelen et al., 2014**). This is not true for other properties of the population structure, which do determine relevant properties of population states at different frequencies. In Detailed Calculations 1.4.1 at the end of Appendix 2, we for instance calculate that

$$r_{1,1} = (1 - r) \bar{p} + r$$

which changes with $\bar{p}$. That implies that, while the rule for selection of non-social traits and the classical Hamilton's rule either predict one of the other to go to fixation, if relatedness $r = r_{0,1}$ is indeed frequency-independent, richer models allow for richer equilibrium properties, such as coexistence or bistability. For coefficients that are actual constants, that requires the relatedness coefficients to vary with $\bar{p}$, which is not the case for $r_{0,1}$, but it is for the others.

### 4.1 The difference between relatedness and estimators of relatedness

## Lotteries and probabilities

As a preparation, we can consider an analogy. Suppose someone organizes a lottery, where in the morning she sells 10 tickets with the numbers 1–10 on them, and then in the afternoon she draws one of those numbers, with equal probabilities, and gives 10 euros to the person that has purchased the ticket with the number on it that matches the number she draws (aka. the winner). This implies that the probability of winning this lottery is $\frac{1}{10}$.

Now suppose I do not know what the winning probability is. What I can do to get closer to finding out is buy a lottery ticket, and see if I win. Obviously, if I win, it would be wrong to conclude that the probability of winning this lottery is 100%. Similarly, if I don't win, it would be wrong to conclude that the chances of winning are 0%. Basing estimations on a single observation is an obvious mistake, and in order to estimate that probability properly, I should, of course, sample more. However, if we dress it up in a different way, the mistake is less obvious.

Let's say these lottery tickets sell for 2 euros, and let's assume that people are risk neutral. In this case, buying a lottery ticket is a bad idea; the expected amount of money that one would win with a ticket is $\frac{1}{10}$ times 10 is 1 euro, and that is less than 2 euros. However, after the uncertainty is resolved, if it turns out that I had bought a winning ticket, and I ask myself whether it was a good or a bad idea to buy a lottery ticket, I would be tempted to say that it was a good idea, since I gained 8 euros. Of course, this hinges on the discrepancy between before and after the fact, but we know from experience that it is inherently hard to understand that it was nonetheless a bad idea to buy the lottery ticket in the first place, even if I ended up winning, the same way it is a bad idea to drive home drunk, even if one ends up causing no accidents. The problem here is that for the relevant question at hand (should or should I not have bought a lottery ticket?) the relevant variable is not the realization of the lottery (in which case I win or I don't), but the a priori probability of winning.

What I want to say with this analogy is: there is no relevant question regarding the average effect of lotteries on people's finances, for which one would want to use anything other than the actual winning probability. The realizations of the random variable when you buy lottery tickets are only useful for answering relevant questions to the extent that they help us estimate (and therefore get closer to) the true winning probability. The true winning probability is the "underlying property" of the chance experiment.

## Relatedness

Relatedness is a property of a pair of individuals in a population structure. I will try to avoid getting too technical and resort to a simple example. Suppose we are considering full siblings. The relatedness between them is a property of the random process by which full siblings are generated. At a certain locus, if one of the parents is $AA$ and the other is $Aa$, then the probability that this parent pair produces two siblings that are both $AA$ is $\frac{1}{4}$; the probability that both of them are $Aa$ is $\frac{1}{4}$; and the probability that one of them is $AA$ and the other is $Aa$ is $\frac{1}{2}$. Now assume that allele $a$ is very rare. In that case, we can consider the probability that, if I am $Aa$, my sibling is also $Aa$. This probability is (very close to) $\frac{1}{2}$. There are different ways to arrive at that number; one of them is to say that, if I am $Aa$, then one of my parents must have had the rare allele $a$. Because this allele is very rare, it is highly unlikely that both my parents have it. With probability close to 1, my parents therefore are $AA$ and $Aa$. The probability that such a parent pair produces an $Aa$ type is $\frac{1}{2}$. For pairs of random, unrelated individuals, on the other hand, the probability that, if I am $Aa$, the other individual is also $Aa$, on the other hand, is close to 0. These calculations can be extended to situations in which allele $a$ is not rare (*van Veelen et al., 2014*), and also then, the appropriately generalized formulas will result in a relatedness between full siblings of $\frac{1}{2}$, and a relatedness between random individuals of 0.

Now, if I want to estimate the relatedness between two individuals, of whom I do not know what their genealogical relation is, and I want to use sequencing for that, it is not enough to consider one SNP only. Looking at one SNP reveals either an $(AA, AA)$-pair, or an $(AA, Aa)$-pair, or an $(Aa, Aa)$-pair, and even an unbiased estimator of relatedness based on 1 SNP would lead to estimations that in expectation are wildly off (that is: the variance of this estimator with only 1 observation is going to be very high). In order to get a better estimator of their relatedness, we will have to get more observations. The way to do that with one and the same pair of individuals is to use many SNPs.

Now suppose we have a population of full siblings, but we go back to considering one locus only. Again, if I confine myself to one of those pairs of siblings, and I base my estimator of their relatedness on the combination of genotypes, I am going to be wildly wrong. To hammer this point home, it is worth observing that I would reach diverging conclusions for different pairs of siblings in the population; pairs that happen to be $(AA, Aa)$ will have an estimated relatedness that is very different from pairs that happen to be $(Aa, Aa)$. However, if the population is not small and I treat all of the pairs as independent observations, then I can get an estimator of their relatedness with much lower variance. If the population does indeed consist of pairs of full siblings, then, by the law of large numbers, the estimator of relatedness will converge to $\frac{1}{2}$ if the population size goes to infinity.

It would, however, be wrong to confuse the estimator of relatedness with relatedness itself. Relatedness is a property of a population structure. In this example, relatedness between full siblings is the result of the process by which pairs of siblings are created. Relatedness is calculated as a function of certain (conditional) probabilities pertaining to that process, and for full siblings, that calculation produces the number $\frac{1}{2}$. If we have a population of pairs of individuals, I can construct an estimator for this relatedness, and if those pairs are in fact siblings, the estimator will have an expected value of $\frac{1}{2}$ (or in other words, I can construct an unbiased estimator of it). For specific realizations of my population of siblings, however, the estimator will typically not be $\frac{1}{2}$, and the smaller the population size (which here amounts to a smaller sample size), the more variance the estimator will have. For very small population sizes, the estimator will be all over the place and useless as an estimator of relatedness. The estimator, therefore, is not the relatedness itself. The relatedness between siblings does not depend on the outcome of the random draw that produced a specific pair of them; it is a property of the process of making siblings.

To reiterate this in a different way, suppose we just wrote the Price equation for a transition between a very small parent and a very small offspring population. Suppose, moreover, that we then turn the offspring generation into the new parent population and repeat the random draw that the formation of a new generation is. With a small population size in the original parent population, the $\frac{\text{Cov}(p,q)}{\text{Var}(p)}$ that we computed for the first transition now is not going to be very informative about the probabilities with which different possible new generations will be generated in the second transition.

Also, just to be sure, everything that applies to $\frac{\text{Cov}(p,q)}{\text{Var}(p)}$ applies more generally to $\frac{\text{Cov}(p,p^r q^s)}{\text{Var}(p)}$; they are all estimators of some underlying property of the random process that generated the offspring population. The only difference is that the underlying property that $\frac{\text{Cov}(p,q)}{\text{Var}(p)}$ estimates is typically not frequency dependent, while the properties that $\frac{\text{Cov}(p,p^r q^s)}{\text{Var}(p)}$ estimate typically are.

## 5. Extension to interactions between more than two individuals

In this section, I will illustrate that this approach allows for further generalizations and is only limited by the requirement that models need to include a constant and a linear term for the $p$-score of the individual itself. In order to do this, I would like to indicate how this works if we extend the model to include interactions between more than two individuals. This requires a bit of notation. After deriving the Generalized Price equation, allowing for situations that go beyond one type of dyadic interactions in a variety of ways, we will consider two examples that are included in this larger set of models.

The model for dyadic interactions requires a $p$-score of the individual itself, and a $p$-score of its interaction partner. The latter is denoted as the $q$-score of the individual itself. If we extend this to include interactions with $k$ partners, then the $q$-score becomes a vector $q_i = [q_{i1}, \ldots, q_{ik}]$, where $k$ is the number of partners that the individual is interacting with, and $q_{il}, l = 1, \ldots, k$ is the $p$-score of partner $l$ to individual $i$.

### The Generalized Price equation in regression form, allowing for more than two individuals

With this richer setup, that allows for $q$-scores that are vectors, we can choose a richer set of models that we consider. Assume, therefore, that fitnesses are generated by some model that can be written as follows:

$$w_i = \sum_{j \in J} \beta_j q_{i1}^{j_1} \cdots q_{ik}^{j_k} p_i^{j_{k+1}} + \varepsilon_i$$

Here, $\varepsilon_i$ is a noise term with expected value 0, and $J$ is a set of vectors that indicates which non-zero coefficients $\beta_j$ the model allows for. A vector $j \in J$ corresponds to a term in this sum, where $j = [j_1, \ldots, j_k, j_{k+1}]$, and where all elements of this vector are non-negative integers. The coefficient $\beta_j$ belongs to the term $q_{i1}^{j_1} \cdots q_{ik}^{j_k} p_i^{j_{k+1}}$. Because it is more elegant to have $j_l$ be the exponent for $q_{il}$, here we choose for the exponent of the $p$-score to be the last element of the vector $j$.

We assume that $(0, \ldots, 0, 0)$ and $(0, \ldots, 0, 1)$ are always included in the set $J$. The first vector represents a constant; the second represents the linear effect of the $p$-score of the individual itself

on its fitness. Minimizing the sum of squared errors with respect to a model that includes these two terms guarantees that the Generalized Price equation in covariance form holds (see Appendix 1, sections 2 and 4).

$$\bar{w}\Delta\bar{p} = \mathrm{Cov}\left(\hat{w}, p\right) + E\left(w\Delta p\right) \tag{GPE.C}$$

In order to arrive at the Generalized Price equation in regression form for this larger set of models, we focus on the term summarized as $\mathrm{Cov}\left(\hat{w}, p\right)$.

$$\mathrm{Cov}\left(\hat{w}, p\right) = \frac{1}{n}\sum_{i=1}^{n}\hat{w}_i p_i - \frac{1}{n^2}\sum_{i=1}^{n}\hat{w}_i \sum_{i=1}^{n}p_i$$

Here we fill in $\sum_{j\in J}\hat{\beta}_j q_{i1}^{j_1}\cdots q_{ik}^{j_k}p_i^{j_{k+1}}$ for $\hat{w}_i$

$$\frac{1}{n}\sum_{i=1}^{n}\left(\sum_{j\in J}\hat{\beta}_j q_{i1}^{j_1}\cdots q_{ik}^{j_k}p_i^{j_{k+1}}\right)p_i - \frac{1}{n}\sum_{i=1}^{n}\left(\sum_{j\in J}\hat{\beta}_j q_{i1}^{j_1}\cdots q_{ik}^{j_k}p_i^{j_{k+1}}\right)\frac{1}{n}\sum_{i=1}^{n}p_i$$

We can leave out the constant, because $q_{i1}^0\cdots q_{in}^0 p_i^0 = 1$, which makes

$$\frac{1}{n}\sum_{i=1}^{n}q_{i1}^0\cdots q_{in}^0 p_i^0 p_i = \frac{1}{n}\sum_{i=1}^{n}p_i = \frac{1}{n}\sum_{i=1}^{n}1\cdot\frac{1}{n}\sum_{i=1}^{n}p_i = \frac{1}{n}\sum_{i=1}^{n}q_{i1}^0\cdots q_{in}^0 p_i^0\sum_{i=1}^{n}p_i$$

Therefore, this is also

$$\frac{1}{n}\sum_{i=1}^{n}\left(\sum_{j\in \Lambda(0,\ldots,0)}\hat{\beta}_j q_{i1}^{j_1}\cdots q_{ik}^{j_k}p_i^{j_{k+1}}\right)p_i - \frac{1}{n}\sum_{i=1}^{n}\left(\sum_{j\in \Lambda(0,\ldots,0)}\hat{\beta}_j q_{i1}^{j_1}\cdots q_{ik}^{j_k}p_i^{j_{k+1}}\right)\frac{1}{n}\sum_{i=1}^{n}p_i$$

If we change the summation order, this becomes

$$\sum_{j\in \Lambda(0,\ldots,0)}\hat{\beta}_j\left(\frac{1}{n}\sum_{i=1}^{n}q_{i1}^{j_1}\cdots q_{ik}^{j_k}p_i^{j_{k+1}+1} - \frac{1}{n}\sum_{i=1}^{n}q_{i1}^{j_1}\cdots q_{ik}^{j_k}p_i^{j_{k+1}}\frac{1}{n}\sum_{i=1}^{n}p_i\right)$$

$$= \sum_{j\in \Lambda(0,\ldots,0)}\hat{\beta}_{k,l}\mathrm{Cov}\left(p, q_1^{j_1}\cdots q_k^{j_k}p^{j_{k+1}}\right)$$

For any choice of a model that this framework allows for – that is, for any choice of a set $J$ – the Generalized Price equation in regression form therefore is

$$\bar{w}\Delta\bar{p} = \sum_{j\in \Lambda(0,\ldots,0)}\hat{\beta}_j\mathrm{Cov}\left(p, q_1^{j_1}\cdots q_k^{j_k}p^{j_{k+1}}\right) + E\left(w\Delta p\right) \tag{GPE.R5}$$

Because $\mathrm{Cov}\left(p, q_1^0\cdots q_n^0 p^0\right) = \mathrm{Cov}\left(p, 1\right) = 0$, it does not matter if we leave $(0,\ldots,0)$ in or not.

## Example B5.1: a linear model with more than one type of relative

If we choose $J = \left\{(0,\ldots,0), (1,0,\ldots,0),\ldots,(0,0,\ldots,0,1)\right\}$, then this set consists of a constant; linear terms for all of the $p$-scores of the individuals it interacts with; and a linear term for the $p$-scores of the individual itself. With this as a reference model, the Price equation in regression form becomes.

$$\bar{w}\Delta\bar{p} = \hat{\beta}_{(1,0,\ldots,0)}\mathrm{Cov}\left(p, q_1\right) + \ldots + \hat{\beta}_{(0,\ldots,0,1,0)}\mathrm{Cov}\left(p, q_k\right) + \hat{\beta}_{(0,\ldots,0,1)}\mathrm{Cov}\left(p, p\right) + E\left(w\Delta p\right)$$

If we rename the coefficients for brevity, so that $\hat{\beta}_{(1,0,\ldots,0)}$ becomes $\hat{\beta}_1$; $\hat{\beta}_{(0,\ldots,0,1,0)}$ becomes $\hat{\beta}_k$; and $\hat{\beta}_{(0,\ldots,0,1)}$ becomes $\hat{\beta}_{k+1}$, then we can write this as

$$\bar{w}\Delta\bar{p} = \sum_{l=1}^{k} \hat{\beta}_l \text{Cov}\,(p, q_l) + \hat{\beta}_{k+1}\text{Cov}\,(p, p) + E\,(w\Delta p)$$

or, because $\text{Cov}\,(p, p) = \text{Var}\,(p)$, as

$$\bar{w}\Delta\bar{p} = \left( \sum_{l=1}^{k} \hat{\beta}_l \frac{\text{Cov}\,(p, q_l)}{\text{Var}\,(p)} + \hat{\beta}_{k+1} \right) \text{Var}\,(p) + E\,(w\Delta p)$$

If we are studying a model in which there is a number of types of relatives (such as siblings, cousins, etc.) and in which the *p*-scores of those relatives affect the fitness of the individual linearly – or, in other words, if the model that we are studying does indeed match the set $J$ of this example – and if the population is infinitely large, then the coefficients in the Generalized Price equation, using the same model as a reference model, will coincide with the model coefficients – that is, $\hat{\beta}_l = \beta_l$ for $l = 1, \ldots, k + 1$. Moreover, in an infinitely large population $\frac{\text{Cov}\,(p, q_l)}{\text{Var}\,(p)}$ will coincide with the relatednesses with these different types of relatives, for $l = 1, \ldots, k$, and, if we also assume fair meiosis, then $E\,(w\Delta p) = 0$. Therefore, if we denote the relatedness with relatives of type $l$ as $r_l$, and write $b_l = \beta_l$ for $l = 1, \ldots, k$ and $c = -\beta_{k+1}$, this gives us a rule for selection for the linear model that coincides with what we would have expected; $\Delta\bar{p} > 0$ if and only if

$$\sum_{l=1}^{k} r_l b_l - c > 0$$

Here $b_l = \beta_l$ is the linear effect on relatives of type $l = 1, \ldots, k$ (these are the benefits of cooperation, assuming that they are positive), and $c = -\beta_{k+1}$ is minus the linear effect on the individual itself (which makes it the cost of cooperation). This gives us a vector of fitness costs and benefits that is similar to the vector of fitness effects postulated in **Hamilton, 1964a**.

### Example B5.2: models with multiple interaction partners of the same type of relative

If interaction partners 1 to $k$ are all the same type of relative (for instance, if they are all full siblings) who are involved in the same type of interaction, then it would be strange to treat them separately, as the setup would impose a certain symmetry between them. For instance, it would be strange, when minimizing the sum of squared errors, to allow for the linear effect of the *p*-score of sibling 1 to be different from the linear effect of the *p*-score of sibling $k$ or of any other sibling. In other words, if there is no reason to treat those different siblings differently, we would have to restrict the coefficients in $J$ so that $\hat{\beta}_{(1,0,\ldots,0)} = \ldots = \hat{\beta}_{(0,\ldots,0,1,0)}$. More generally, the symmetry would dictate that all vectors in which the first $k$ elements are permutations of each other should be restricted to be the same when the sum of squared errors is minimized. Just to give a few more examples of the restrictions that symmetry imposes, for $k = 3$, besides $\hat{\beta}_{(1,0,0,0)} = \hat{\beta}_{(0,1,0,0)} = \hat{\beta}_{(0,0,1,0)}$, symmetry would also suggest that $\hat{\beta}_{(1,0,0,1)} = \hat{\beta}_{(0,1,0,1)} = \hat{\beta}_{(0,0,1,1)}$; and $\hat{\beta}_{(1,1,0,0)} = \hat{\beta}_{(1,0,1,0)} = \hat{\beta}_{(0,1,1,0)}$; and $\hat{\beta}_{(2,0,0,0)} = \hat{\beta}_{(0,2,0,0)} = \hat{\beta}_{(0,0,2,0)}$; and so on.

### Linear effects (and siblings)

If we consider a situation in which all individuals interact with $k$ partners, all of whom are related in the same way to the individual (for instance, they could all be full siblings), then one possibility would be the following linear model.

$$w_i = \beta_{(0,\ldots,0)} + \beta_{(1,0,\ldots,0)}\,q_{i1} + \beta_{(0,1,0,\ldots,0)}\,q_{i2} + \ldots + \beta_{(0,\ldots,0,1,0)}\,q_{ik} + \beta_{(0,\ldots,0,1)}\,p_i + \varepsilon_i$$

This amounts to a choice for $J = \left\{ (0, \ldots, 0), (1, 0, \ldots, 0), \ldots, (0, \ldots, 0, 1) \right\}$. For this linear model, the symmetry implies that we impose that $\beta_{(1,0,\ldots,0)} = \beta_{(0,1,0,\ldots,0)} = \ldots = \beta_{(0,\ldots,0,1,0)}$. If we do impose this, the fitness function can be rewritten as

$$w_i = \beta_{(0,0,0)} + \beta_{(1,0,\ldots,0)} \sum_{l=1}^{k} q_{il} + \beta_{(0,\ldots,0,1)}\,p_i + \varepsilon_i$$

The Generalized Price equation in regression form for this model then becomes

$$\bar{w}\Delta\bar{p} = \left( \hat{\beta}_{(1,0,\ldots,0)} \frac{\text{Cov}\left(p, \sum_{l=1}^{k} q_{il}\right)}{\text{Var}\left(p\right)} + \hat{\beta}_{(0,\ldots,0,1)} \right) \text{Var}\left(p\right) + E\left(w\Delta p\right)$$

In a modeling exercise, besides a fitness function, we would also need a model for the population structure. This would be an explicit or implicit statement about who interacts with whom. In an infinitely large population, that would have implications for how $\frac{\text{Cov}\left(p, \sum_{l=1}^{k} q_{il}\right)}{\text{Var}(p)}$ depends on the average $p$-score in the population. If we think of full siblings, such a model of the population structure for instance would have to imply that $\frac{\text{Cov}\left(p, \sum_{l=1}^{k} q_{il}\right)}{\text{Var}(p)} = k \frac{\text{Cov}(p,q_{i1})}{\text{Var}(p)} = \frac{k}{2}$. The rule for selection in the case of siblings then becomes that $\Delta\bar{p} > 0$ if and only if $\frac{k}{2}\beta_{(1,0,\ldots,0)} > -\beta_{(0,\ldots,0,1)}$.

## Weakest link game (and random matching)

Here we consider a situation in which all individuals interact in groups of 3, and for simplicity, we assume that the $p$-scores are binary. We can then imagine a situation in which it requires all three group members to cooperate (have a $p$-core of 1) for a benefit for all of them to be created. In other words, we could imagine that the fitness function is

$$w_i = \beta_{(0,0,0)} + \beta_{(1,1,1)} q_{1i} q_{2i} p_i + \beta_{(0,0,1)} p_i + \varepsilon_i$$

with $\hat{\beta}_{(1,1,1)} > 0$ and $\hat{\beta}_{(0,0,1)} < 0$. Cooperation is potentially good for fitness if $\hat{\beta}_{(1,1,1)} > -\hat{\beta}_{(0,0,1)}$. This model would come down to a choice for a set of nonzero coefficients

$$J = \left\{ (0,0,0), (0,0,1), (1,1,1) \right\}$$

The Generalized Price equation in regression form for this model then becomes

$$\bar{w}\Delta\bar{p} = \left( \hat{\beta}_{(1,1,1)} \frac{\text{Cov}\left(p, pq_1q_2\right)}{\text{Var}\left(p\right)} + \hat{\beta}_{(0,0,1)} \right) \text{Var}\left(p\right) + E\left(w\Delta p\right)$$

If we assume an infinitely large population in which this model does indeed correctly describe the fitness function, then $\frac{\text{Cov}(p,pq_1q_2)}{\text{Var}(p)}$ will typically be frequency dependent. For instance, if we assume random group formation, then that implies that the share of groups of 3 that consist of $m$ cooperators and $3 - m$ defectors will depend on the average share $\bar{p}$ of cooperators in the population as follows:

$$f_m\left(\bar{p}\right) = \begin{cases} \left(1 - \bar{p}\right)^3 & if\, m = 0 \\ 3\left(1 - \bar{p}\right)^2 \bar{p} & if\, m = 1 \\ 3\left(1 - \bar{p}\right) \bar{p}^2 & if\, m = 2 \\ \bar{p}^3 & if\, m = 3 \end{cases}$$

That implies that, if we draw a random individual from the population, and then consider its own $p$-score, and the product of its own $p$-score and the $p$-score of the other two members, this results in

$$\text{Cov}\left(p, pq_1q_2\right) = E\left[p^2 q_1 q_2\right] - E\left[p\right] E\left[pq_1q_2\right]$$

With binary variables, that is equal to

$$\text{Cov}\left(p, pq_1q_2\right) = E\left[pq_1q_2\right] - E\left[p\right] E\left[pq_1q_2\right] = E\left[pq_1q_2\right]\left(1 - E\left[p\right]\right) = p^3\left(1 - p\right)$$

With $\text{Var}\left(p\right) = p\left(1 - p\right)$, that implies that

$$\frac{\text{Cov}\left(p, pq_1q_2\right)}{\text{Var}\left(p\right)} = p^2$$

The rule for selection with random group formation then becomes that $\Delta\bar{p} > 0$ if and only if $p^2\hat{\beta}_{(1,1,1)} > -\hat{\beta}_{(0,0,1)}$. This describes the dynamics in the following way. Assuming that $\hat{\beta}_{(1,1,1)} > 0$ and $\hat{\beta}_{(0,0,1)} < 0$, that means $p = 0$ and $p = 1$ are both attractors, separated by $p = \sqrt{\dfrac{-\hat{\beta}_{(0,0,1)}}{\hat{\beta}_{(1,1,1)}}}$.

## Relation to *Smith et al., 2010*

In this section, I will discuss the relation between what I did in this paper and what (*Smith et al., 2010*) do. *Smith et al., 2010* is an empirical study that tries to capture non-linearity. A first point of order is: linear or non-linear in what? For that, I would like to go back to Model 2 from Section 1 of this appendix.

$$w_i = \alpha + \beta_{1,0}p_i + \beta_{0,1}q_i + \varepsilon_i$$

Here we assume that $\alpha = 1$, $\beta_{1,0} = -1$, and $\beta_{0,1} = 2$. We assume that $p$-scores can only be 0 or 1, or, in other words, we assume that there are only cooperators and defectors in the population (or, in terms of *Smith et al., 2010*: cooperators and cheaters).

For a well-mixed population, we can now plot the fitnesses of cooperators (red) and defectors (blue) as a function of the frequency of cooperators (*Appendix 2—figure 2*, left panel). We can do the same for a population with relatedness $\frac{1}{4}$ (*Appendix 2—figure 2*, right panel). For relatedness $r = 0$ and $r = \frac{1}{4}$, cooperation is selected against at every frequency.

Increasing relatedness further, we would find that for $r = \frac{1}{2}$ the lines coincide, and cooperation is neither selected for nor against. For $r > \frac{1}{2}$ cooperation will be selected for at every frequency. This pattern is consistent with the fact that, as we have seen, the classical Hamilton's rule works perfectly fine for Model 2; with $c = -\beta_{1,0} = 1$ and $b = \beta_{0,1} = 2$, cooperation is selected for if and only if $rb > c$. The important thing to notice here is that the fitness of cooperators and the fitness of defectors, both as a function of the frequency of cooperators, are always parallel lines, regardless of relatedness.

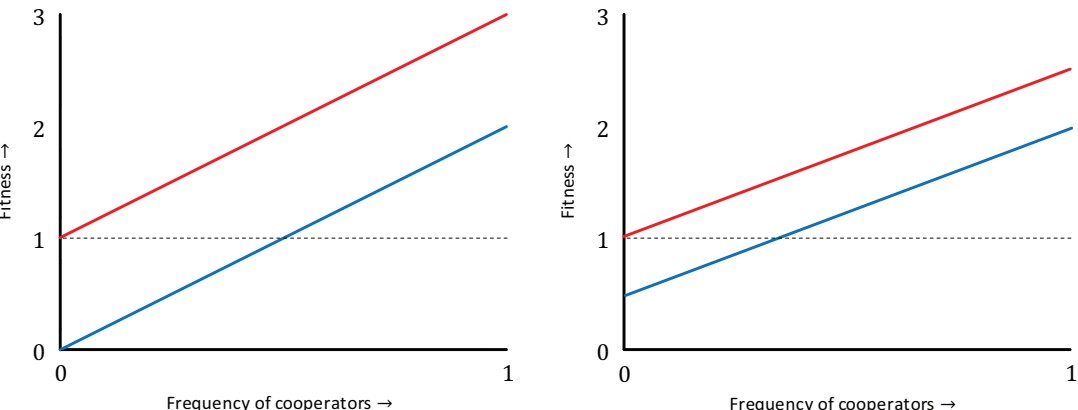

**Appendix 2—figure 2.** Fitnesses for Model 2. On the left, $r = 0$, and the likelihood of being matched with a cooperator is the same for cooperators and defectors, and equal to the frequency of cooperators. On the right, $r = \frac{1}{4}$, where the probability of being matched with a cooperator is $\frac{1}{4} + \frac{3}{4}f_C$ for cooperators, and $\frac{3}{4}f_C$ for defectors, where $f_C$ is the frequency of cooperators.

Model 3 from Section 1 extends Model 2 by adding an interaction term:

$$w_i = \alpha + \beta_{1,0}p_i + \beta_{0,1}q_i + \beta_{1,1}p_iq_i + \varepsilon_i$$

Now we choose $\beta_{1,0} = -1$, $\beta_{0,1} = 1$, and $\beta_{1,1} = 1$. We again draw the fitnesses of cooperators and defectors, both at relatedness $r = 0$ (*Appendix 2—figure 3*, left panel) and at relatedness $r = \frac{1}{4}$ (*Appendix 2—figure 3*, right panel). We have seen that the appropriate version of Hamilton's rule here is Queller's rule; $r_{0,1}b_{0,1} + r_{1,1}b_{1,1} > c$ with $c = -\beta_{1,0} = 1$, $b_{0,1} = \beta_{0,1} = 1$, and $b_{1,1} = \beta_{1,1} = 1$. The fitnesses of cooperators and defectors as functions of the frequency of cooperators are still straight lines, but they are no longer parallel.

The first thing to observe, therefore, is that a model with synergy, in which the classic version of Hamilton's rule would be misspecified, and Queller's rule would be well-specified, does not require

the fitnesses as functions of the frequencies of cooperators to be non-linear. All that changes with the addition of the interaction term is that they stop being parallel.

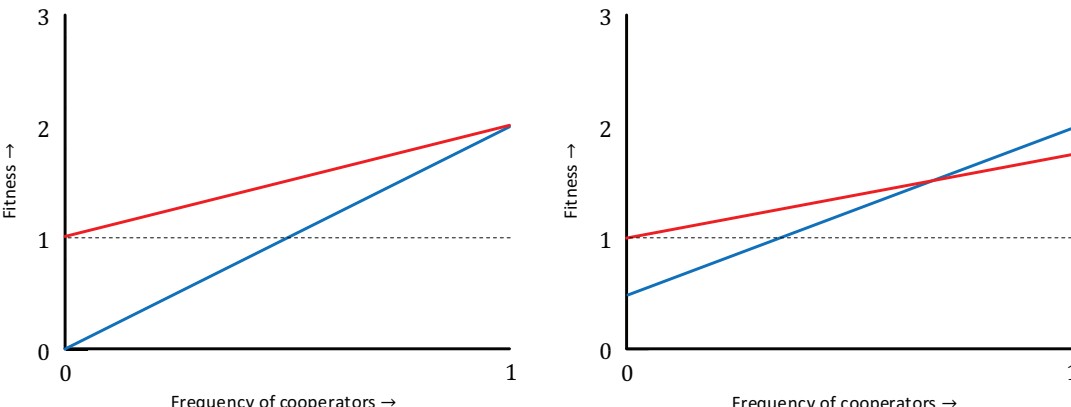

**Appendix 2—figure 3.** Fitnesses for Model 3. On the left,, $r = 0$ and on the right, $r = \frac{1}{4}$. The lines are still straight, but no longer parallel.

*Smith et al., 2010* is an effort to capture non-linearities in the way fitnesses depend on the frequency of cooperators. That, therefore, goes beyond the step from Model 2 to Model 3. Whether it uses the right method to capture those non-linearities, we will come back to in a second, but it is important to realize that also without these non-linearities, the classic version of Hamilton's rule can be too limiting to accurately describe selection. Here, I should add that this implies that we were wrong in *Wu et al., 2013*, when we suggested that

> "*for this experiment, it seems unnecessary to use the generalized Hamilton's rule, if instead the Malthusian fitness is adopted. In other words, the Wrightian fitness approach calls for a generalization of Hamilton's rule, whereas the Malthusian fitness approach does not (or at least not in a drastic way, as Malthusian fitnesses are almost linear in the frequency of cooperators).*"

Using Malthusian fitnesses, the functions were close to linear, but not close to parallel, and therefore also here, Hamilton's rule needs generalizing - albeit in a different way than *Smith et al., 2010* did.

The cooperation that is observed in the *Myxococcus xanthus* studied in *Smith et al., 2010* is not a good match with a model where individuals are matched in pairs for an interaction that determines their fitnesses. These microbes cooperate in large groups, and a better match would therefore be the *n*-player public goods games studied in *van Veelen, 2018a*. There, we see that simple, straightforward ways to describe synergies (or anti-synergies) can easily lead to fitnesses not being linear in the frequency of cooperators.

The way *Smith et al., 2010* try to capture those, however, is not free of complications. We addressed those in *Wu et al., 2013*, and I summarized them, shortly, in *van Veelen, 2018a*. One of the issues is that most of the non-linearity *Smith et al., 2010* pick up is the result of considering Wrightian fitness rather than Malthusian fitness. In a continuous time model with a constant growth rate, the population size at time $t$ is $N(t) = e^{mt}N(0)$, where $m$ is the Malthusian fitness. In a discrete time model with a constant average number of offspring per individual, the population at time $t$ is $N(t) = w^{t}N(0)$, where $w$ is the Wrightian fitness. If we take $m = \ln w$, these are the same, and if $w$ is close to 1, then $m$ can be approximated by $w - 1$. That also implies that if $w$ is close to 1 (or, equivalently, if $m$ is close to 0), one is locally linear if the other is too. However, in the experiment in *Smith et al., 2010* the aggregate fitness effects are not small, and what is highly nonlinear in terms of Wrightian fitness is close to linear in Malthusian fitness.

Another complication is that the Taylor coefficients that *Smith et al., 2010* find are the result of a combination of the data and the choice of a functional form they choose to first apply to their data.

That means that a different choice of a functional form would have given different Taylor coefficients, while the in-between transformation can also be skipped. Also, the number of Taylor coefficients is larger than the dimensionality of the data, which are based on averages for 6 frequencies. For more details on these complications, I would like to refer to *Wu et al., 2013* and *van Veelen, 2018a*. A nice detail is that if we consider the way the fitnesses of cooperators and defectors compare when using Malthusian fitnesses, then a comparison of the slopes suggests anti-synergies, which leads to a stable mix of cooperators and cheaters, already in the absence of population structure, much like what is suggested by *Archetti and Scheuring, 2011*; *Archetti and Scheuring, 2012*; *Archetti, 2018*

Besides these technical complications, *Smith et al., 2010* is also different, in the sense that it is an empirical paper. It does not contain the Generalized Price equation, it contains no insights regarding how to derive population genetic dynamics from the Generalized Price equation, or how to derive the appropriate rules from those, and it has a very different approach to separating fitness effects and population structure.

## Detailed calculations for Appendix 2

### 2.1 Price-like equation 1, applied to Model 1

As a finger exercise, we first apply Price-like equation 1 to Model 1. That helps make a few observations that are relevant when we minimize squared errors, not in a statistical setting, but in a setting with a model.

We get the Price-like equation 1 from minimizing the sum of squared errors, relative to Model 1. The true parameters are indicated by $\beta_0$ and $\beta_1$, their estimates by $\hat{\beta}_0$ and $\hat{\beta}_1$. The actual errors are indicated by $\varepsilon_i$, and the errors, the squared sum of which we minimize, are indicated by $\hat{\varepsilon}_i$.

The sum of squared errors is

$$\sum_{i=1}^{n} \hat{\varepsilon}_i^2 = \sum_{i=1}^{n} \left( w_i - \left( \hat{\beta}_0 + \hat{\beta}_1 p_i \right) \right)^2 = \sum_{i=1}^{n} \left( \beta_0 + \beta_1 p_i + \varepsilon_i - \left( \hat{\beta}_0 + \hat{\beta}_1 p_i \right) \right)^2$$

If we minimize this, we begin with setting the derivative to $\hat{\beta}_0$ to 0.

$$-2 \sum_{i=1}^{n} \left( \beta_0 + \beta_1 p_i + \varepsilon_i - \left( \hat{\beta}_0 + \hat{\beta}_1 p_i \right) \right) = 0$$

$$\sum_{i=1}^{n} \left( \beta_0 - \hat{\beta}_0 \right) + \sum_{i=1}^{n} \left( \beta_1 - \hat{\beta}_1 \right) p_i + \sum_{i=1}^{n} \varepsilon_i = 0$$

$$\beta_0 - \hat{\beta}_0 + \left( \beta_1 - \hat{\beta}_1 \right) \bar{p} + \frac{1}{n} \sum_{i=1}^{n} \varepsilon_i = 0$$

Then we set the derivative to $\hat{\beta}_1$ to 0

$$-2 \sum_{i=1}^{n} p_i \left( \beta_0 + \beta_1 p_i + \varepsilon_i - \left( \hat{\beta}_0 + \hat{\beta}_1 p_i \right) \right) = 0$$

$$\sum_{i=1}^{n} \left( \beta_0 - \hat{\beta}_0 \right) p_i + \sum_{i=1}^{n} \left( \beta_1 - \hat{\beta}_1 \right) p_i^2 + \sum_{i=1}^{n} \varepsilon_i p_i = 0$$

$$\left( \beta_0 - \hat{\beta}_0 \right) \bar{p} + \left( \beta_1 - \hat{\beta}_1 \right) \frac{1}{n} \sum_{i=1}^{n} p_i^2 + \frac{1}{n} \sum_{i=1}^{n} \varepsilon_i p_i = 0$$

Although outside the Price equation literature, it is unusual to apply minimization of squared errors in a modeling context – where we know what the true model is, and therefore what $\beta_0$ and $\beta_1$ are – we can nonetheless still do it. In an infinite population, we can use that the errors have mean 0, which implies that $\lim_{n \to \infty} \frac{1}{n} \sum_{i=1}^{n} \varepsilon_i = 0$, and we can use that the errors are uncorrelated with the $p$-scores, which implies that $\lim_{n \to \infty} \frac{1}{n} \sum_{i=1}^{n} \varepsilon_i p_i = 0$. Then we have two equations:

$$\beta_0 - \hat{\beta}_0 + \left(\beta_1 - \hat{\beta}_1\right)\bar{p} = 0$$

$$\left(\beta_0 - \hat{\beta}_0\right)\bar{p} + \left(\beta_1 - \hat{\beta}_1\right)\frac{1}{n}\sum_{i=1}^{n}p_i^2 = 0$$

It is clear that both of these hold if $\hat{\beta}_0 = \beta_0$ and $\hat{\beta}_1 = \beta_1$. In a statistical context, one cannot do this, because there we do not know the $\beta_0$ and $\beta_1$, and the purpose of the estimation procedure is to get an estimate of those. Here, however, we do know what the actual model is, and we can simply pick $\hat{\beta}_0 = \beta_0$ and $\hat{\beta}_1 = \beta_1$.

Of course, on its own, this is a futile exercise, as it unveils something that is not veiled. It does, however, illustrate that Price equation 1 applied to Model 1 gets the model right. This is an instructive benchmark for when we go on to allow for Price-like equations that are applied to non-matching models. An observation that may also help comparisons with this benchmark is that the errors that are found by the minimization coincide with the actual errors. Here we use that we chose $\hat{\beta}_0 = \beta_0$ and $\hat{\beta}_1 = \beta_1$.

$$\hat{\varepsilon}_i = w_i - \left(\hat{\beta}_0 + \hat{\beta}_1 p_i\right) = \beta_0 + \beta_1 p_i + \varepsilon_i - \left(\hat{\beta}_0 + \hat{\beta}_1 p_i\right) = \varepsilon_i$$

for $i \ldots n$.

If we would leave out the actual errors $\varepsilon_i$ at the very beginning, then the calculations come out the same. This would amount to assuming that $\varepsilon_i = 0$ for all $i$, which is more than just assuming that errors have mean 0 (which implies that $\lim_{n\to\infty}\frac{1}{n}\sum_{i=1}^{n}\varepsilon_i = 0$) and that they are uncorrelated with $p$-scores (which implies that $\lim_{n\to\infty}\frac{1}{n}\sum_{i=1}^{n}\varepsilon_i p_i = 0$). It can be seen, though, as a shortcut for assuming infinitely large populations, as we do here. In that case, we would get $\hat{\varepsilon}_i = 0$ for $i = 1, \ldots, n$.

I will not repeat this for Price equation 2 applied to Model 2, or Price equation 3 applied to Model 3, as I hope it is clear that all minimizations will result in the estimators coinciding with the actual coefficients. This is also verified in the shortcuts provided below in Detailed Calculations 2.4.

## 2.2 Price equation 1, applied to Model 2
We now apply Price-like equation 1 to Model 2. The sum of squared errors is

$$\sum_{i=1}^{n}\hat{\varepsilon}_i^2 = \sum_{i=1}^{n}\left(w_i - \left(\hat{\beta}_0 + \hat{\beta}_1 p_i\right)\right)^2 = \sum_{i=1}^{n}\left(\gamma_{0,0} + \gamma_{1,0}p_i + \gamma_{0,1}q_i + \varepsilon_i - \left(\hat{\beta}_0 + \hat{\beta}_1 p_i\right)\right)^2$$

If we minimize this, we begin with setting the derivative to $\hat{\beta}_0$ to 0.

$$-2\sum_{i=1}^{n}\left(\gamma_{0,0} + \gamma_{1,0}p_i + \gamma_{0,1}q_i + \varepsilon_i - \left(\hat{\beta}_0 + \hat{\beta}_1 p_i\right)\right) = 0$$

$$n\left(\gamma_{0,0} - \hat{\beta}_0\right) + \left(\gamma_{1,0} - \hat{\beta}_1\right)\sum_{i=1}^{n}p_i + \gamma_{0,1}\sum_{i=1}^{n}q_i + \sum_{i=1}^{n}\varepsilon_i = 0$$

$$\left(\gamma_{0,0} - \hat{\beta}_0\right) + \left(\gamma_{1,0} + \gamma_{0,1} - \hat{\beta}_1\right)\bar{p} + \frac{1}{n}\sum_{i=1}^{n}\varepsilon_i = 0$$

In the last step, we use that $\bar{p} = \bar{q}$. Then we set the derivative to $\hat{\beta}_1$ to 0

$$-2\sum_{i=1}^{n}p_i\left(\gamma_{0,0} + \gamma_{1,0}p_i + \gamma_{0,1}q_i + \varepsilon_i - \left(\hat{\beta}_0 + \hat{\beta}_1 p_i\right)\right) = 0$$

$$\left(\gamma_{0,0} - \hat{\beta}_0\right)\sum_{i=1}^{n}p_i + \left(\gamma_{1,0} - \hat{\beta}_1\right)\sum_{i=1}^{n}p_i^2 + \gamma_{0,1}\sum_{i=1}^{n}p_i q_i + \sum_{i=1}^{n}\varepsilon_i p_i = 0$$

$$\left(\gamma_{0,0} - \hat{\beta}_0\right) \bar{p} + \left(\gamma_{1,0} - \hat{\beta}_1\right) \frac{1}{n} \sum_{i=1}^{n} p_i^2 + \gamma_{0,1} \frac{1}{n} \sum_{i=1}^{n} p_i q_i + \frac{1}{n} \sum_{i=1}^{n} \varepsilon_i p_i = 0$$

In an infinite population, we can use that the errors are assumed to have mean 0, which implies that $\frac{1}{n} \sum_{i=1}^{n} \varepsilon_i$ becomes 0 for large $n$. We also use that the errors are uncorrelated with the $p$-scores, which implies that $\frac{1}{n} \sum_{i=1}^{n} \varepsilon_i p_i$ also becomes 0 for large $n$. The we multiply the first equation by $\bar{p}$, which gives us two equations:

$$\left(\gamma_{0,0} - \hat{\beta}_0\right) \bar{p} + \left(\gamma_{1,0} + \gamma_{0,1} - \hat{\beta}_1\right) \bar{p}^2 = 0$$

and

$$\left(\gamma_{0,0} - \hat{\beta}_0\right) \bar{p} + \left(\gamma_{1,0} - \hat{\beta}_1\right) \frac{1}{n} \sum_{i=1}^{n} p_i^2 + \gamma_{0,1} \frac{1}{n} \sum_{i=1}^{n} p_i q_i = 0$$

These combine to

$$\left(\gamma_{1,0} + \gamma_{0,1} - \hat{\beta}_1\right) \bar{p}^2 = \left(\gamma_{1,0} - \hat{\beta}_1\right) \frac{1}{n} \sum_{i=1}^{n} p_i^2 + \gamma_{0,1} \frac{1}{n} \sum_{i=1}^{n} p_i q_i$$

$$0 = \left(\gamma_{1,0} - \hat{\beta}_1\right) \left( \frac{1}{n} \sum_{i=1}^{n} p_i^2 - \frac{1}{n^2} \left( \sum_{i=1}^{n} p_i \right)^2 \right) + \gamma_{0,1} \left( \frac{1}{n} \sum_{i=1}^{n} p_i q_i - \frac{1}{n^2} \left( \sum_{i=1}^{n} p_i \right)^2 \right)$$

With $\sum_{i=1}^{n} p_i = \sum_{i=1}^{n} q_i$, this is also

$$0 = \left(\gamma_{1,0} - \hat{\beta}_1\right) \text{Var}\left(p\right) + \gamma_{0,1} \text{Cov}\left(p, q\right)$$

or

$$\hat{\beta}_1 = \gamma_{1,0} + \gamma_{0,1} \frac{\text{Cov}\left(p, q\right)}{\text{Var}\left(p\right)} = \gamma_{1,0} + r\gamma_{0,1}$$

Then combining with the first equation gives

$$\left(\gamma_{0,0} - \hat{\beta}_0\right) + \left(\gamma_{1,0} + \gamma_{0,1} - \left(\gamma_{1,0} + \gamma_{0,1} r\right)\right) \bar{p} = 0$$

$$\hat{\beta}_0 = \gamma_{0,0} + \left(1 - r\right) \bar{p} \gamma_{0,1}$$

If we now go back to the errors, we find that

$$\hat{\varepsilon}_i = w_i - \left(\hat{\beta}_0 + \hat{\beta}_1 p_i\right)$$

$$= \gamma_{0,0} + \gamma_{1,0} p_i + \gamma_{0,1} q_i + \varepsilon_i - \left(\gamma_{0,0} + \left(1 - r\right) \bar{p} \gamma_{0,1} + \left(\gamma_{1,0} + r\gamma_{0,1}\right) p_i\right)$$

$$= \gamma_{0,1} \left(q_i - \left(\left(1 - r\right) \bar{p} + r p_i\right)\right) + \varepsilon_i$$

for all $i = 1, \ldots, n$. The errors, the squared sum of which is minimized, therefore do not coincide with the actual errors and include $\gamma_{0,1}$ times the gap between $q_i$ and $\left(1 - r\right) \bar{p} + r p_i$. The first is the $q$-score of individual $i$, the second would be a predictor for $q_i$ based on $\bar{p}$ and $p_i$. This part represents the misspecification-induced part of the estimated errors.

If we again assume $\varepsilon_i = 0$ for all $i$ as a shortcut for assuming an infinitely large population, then $\hat{\varepsilon}_i = \gamma_{0,1} \left(q_i - \left(\left(1 - r\right) \bar{p} + r p_i\right)\right)$ only reflects the misspecification.

## 2.3 Price equation 2, applied to Model 3

We now apply Price-like equation 2 to Model 3. The derivation is rather boring, but there is no way around it. The sum of squared errors is

$$\sum_{i=1}^{n} \hat{\varepsilon}_i^2 = \sum_{i=1}^{n} \left( \delta_{0,0} + \delta_{1,0}p_i + \delta_{0,1}q_i + \delta_{1,1}p_iq_i + \varepsilon_i - \left( \hat{\gamma}_{0,0} + \hat{\gamma}_{1,0}p_i + \hat{\gamma}_{0,1}q_i \right) \right)^2$$

If we minimize this, we begin with setting the derivative to $\hat{\gamma}_{0,0}$ to 0.

$$-2\sum_{i=1}^{n} \left( \delta_{0,0} + \delta_{1,0}p_i + \delta_{0,1}q_i + \delta_{1,1}p_iq_i + \varepsilon_i - \left( \hat{\gamma}_{0,0} + \hat{\gamma}_{1,0}p_i + \hat{\gamma}_{0,1}q_i \right) \right) = 0$$

$$n\left( \delta_{0,0} - \hat{\gamma}_{0,0} \right) + \left( \delta_{1,0} - \hat{\gamma}_{1,0} \right) \sum_{i=1}^{n} p_i + \left( \delta_{0,1} - \hat{\gamma}_{0,1} \right) \sum_{i=1}^{n} q_i + \delta_{1,1} \sum_{i=1}^{n} p_iq_i + \sum_{i=1}^{n} \varepsilon_i = 0$$

Then we set the derivative to $\hat{\gamma}_{1,0}$ to 0

$$-2\sum_{i=1}^{n} p_i \left( \delta_{0,0} + \delta_{1,0}p_i + \delta_{0,1}q_i + \delta_{1,1}p_iq_i + \varepsilon_i - \left( \hat{\gamma}_{0,0} + \hat{\gamma}_{1,0}p_i + \hat{\gamma}_{0,1}q_i \right) \right) = 0$$

$$\left( \delta_{0,0} - \hat{\gamma}_{0,0} \right) \sum_{i=1}^{n} p_i + \left( \delta_{1,0} - \hat{\gamma}_{1,0} \right) \sum_{i=1}^{n} p_i^2 + \left( \delta_{0,1} - \hat{\gamma}_{0,1} \right) \sum_{i=1}^{n} p_iq_i + \delta_{1,1} \sum_{i=1}^{n} p_i^2q_i + \sum_{i=1}^{n} \varepsilon_ip_i = 0$$

Finally, we set the derivative to $\hat{\gamma}_{0,1}$ to 0

$$-2\sum_{i=1}^{n} q_i \left( \delta_{0,0} + \delta_{1,0}p_i + \delta_{0,1}q_i + \delta_{1,1}p_iq_i + \varepsilon_i - \left( \gamma_{0,0} + \hat{\gamma}_{1,0}p_i + \hat{\gamma}_{0,1}q_i \right) \right) = 0$$

$$\left( \delta_{0,0} - \hat{\gamma}_{0,0} \right) \sum_{i=1}^{n} q_i + \left( \delta_{1,0} - \hat{\gamma}_{1,0} \right) \sum_{i=1}^{n} p_iq_i + \left( \delta_{0,1} - \hat{\gamma}_{0,1} \right) \sum_{i=1}^{n} q_i^2 + \delta_{1,1} \sum_{i=1}^{n} p_iq_i^2 + \sum_{i=1}^{n} \varepsilon_iq_i = 0$$

If we then use that $\overset{lim}{\underset{n\to\infty}{}}\frac{1}{n}\sum_{i=1}^{n}\varepsilon_i = \overset{lim}{\underset{n\to\infty}{}}\frac{1}{n}\sum_{i=1}^{n}\varepsilon_ip_i = \overset{lim}{\underset{n\to\infty}{}}\frac{1}{n}\sum_{i=1}^{n}\varepsilon_iq_i = 0$, we can rewrite these three equations as

$$\left( \delta_{0,0} - \hat{\gamma}_{0,0} \right) + \left( \delta_{1,0} - \hat{\gamma}_{1,0} \right) \bar{p} + \left( \delta_{0,1} - \hat{\gamma}_{0,1} \right) \bar{q} + \delta_{1,1}\overline{pq} = 0$$

$$\left( \delta_{0,0} - \hat{\gamma}_{0,0} \right) \bar{p} + \left( \delta_{1,0} - \hat{\gamma}_{1,0} \right) \overline{p^2} + \left( \delta_{0,1} - \hat{\gamma}_{0,1} \right) \overline{pq} + \delta_{1,1}\overline{p^2q} = 0$$

$$\left( \delta_{0,0} - \hat{\gamma}_{0,0} \right) \bar{q} + \left( \delta_{1,0} - \hat{\gamma}_{1,0} \right) \overline{pq} + \left( \delta_{0,1} - \hat{\gamma}_{0,1} \right) \overline{q^2} + \delta_{1,1}\overline{pq^2} = 0$$

If we multiply the first equation by $\bar{p}$, it becomes

$$\left( \delta_{0,0} - \hat{\gamma}_{0,0} \right) \bar{p} + \left( \delta_{1,0} - \hat{\gamma}_{1,0} \right) \left( \bar{p} \right)^2 + \left( \delta_{0,1} - \hat{\gamma}_{0,1} \right) \left( \bar{p} \right)\left( \bar{q} \right) + \delta_{1,1}\left( \bar{p} \right)\left( \overline{pq} \right) = 0$$

Combining it with the second equation, we get

$$\left( \delta_{1,0} - \hat{\gamma}_{1,0} \right) \left( \left( \bar{p} \right)^2 - \overline{p^2} \right) + \left( \delta_{0,1} - \hat{\gamma}_{0,1} \right) \left( \left( \bar{p} \right)\left( \bar{q} \right) - \overline{pq} \right) + \delta_{1,1}\left( \left( \bar{p} \right)\left( \overline{pq} \right) - \overline{p^2q} \right) = 0$$

If we multiply the first equation by $\bar{q}$, and combine it with the third equation, we get

$$\left( \delta_{1,0} - \hat{\gamma}_{1,0} \right) \left( \left( \bar{p} \right)\left( \bar{q} \right) - \overline{pq} \right) + \left( \delta_{0,1} - \hat{\gamma}_{0,1} \right) \left( \left( \bar{q} \right)\left( \bar{q} \right) - \overline{q^2} \right) + \delta_{1,1}\left( \left( \bar{q} \right)\left( \overline{pq} \right) - \overline{pq^2} \right) = 0$$

This can be written more succinctly as

$$\left( \delta_{1,0} - \hat{\gamma}_{1,0} \right) \text{Var}\left( p \right) + \left( \delta_{0,1} - \hat{\gamma}_{0,1} \right) \text{Cov}\left( p,q \right) + \delta_{1,1}\text{Cov}\left( p,pq \right) = 0$$

$$\left( \delta_{1,0} - \hat{\gamma}_{1,0} \right) \text{Cov}\left( p,q \right) + \left( \delta_{0,1} - \hat{\gamma}_{0,1} \right) \text{Var}\left( q \right) + \delta_{1,1}\text{Cov}\left( q,pq \right) = 0$$

If we multiply the second equation with $\frac{\text{Var}(p)}{\text{Cov}(p,q)}$, it becomes

$$\left(\delta_{1,0} - \hat{\gamma}_{1,0}\right) \text{Var}\left(p\right) + \left(\delta_{0,1} - \hat{\gamma}_{0,1}\right) \frac{\text{Var}\left(q\right)\text{Var}\left(p\right)}{\text{Cov}\left(p,q\right)} + \delta_{1,1} \frac{\text{Cov}\left(q,pq\right)\text{Var}\left(p\right)}{\text{Cov}\left(p,q\right)} = 0$$

Combining that with the first equation gives us

$$\left(\delta_{0,1} - \hat{\gamma}_{0,1}\right)\text{Cov}\left(p,q\right) + \delta_{1,1}\text{Cov}\left(p,pq\right) = \left(\delta_{0,1} - \hat{\gamma}_{0,1}\right)\frac{\text{Var}\left(q\right)\text{Var}\left(p\right)}{\text{Cov}\left(p,q\right)} + \delta_{1,1}\frac{\text{Cov}\left(q,pq\right)\text{Var}\left(p\right)}{\text{Cov}\left(p,q\right)}$$

which we rewrite as

$$\left(\delta_{0,1} - \hat{\gamma}_{0,1}\right)\left(\text{Cov}\left(p,q\right) - \frac{\text{Var}\left(q\right)\text{Var}\left(p\right)}{\text{Cov}\left(p,q\right)}\right) + \delta_{1,1}\left(\text{Cov}\left(p,pq\right) - \frac{\text{Cov}\left(q,pq\right)\text{Var}\left(p\right)}{\text{Cov}\left(p,q\right)}\right) = 0$$

$$\left(\delta_{0,1} - \hat{\gamma}_{0,1}\right) + \delta_{1,1}\left(\frac{\text{Cov}\left(p,pq\right) - \dfrac{\text{Cov}\left(q,pq\right)\text{Var}\left(p\right)}{\text{Cov}\left(p,q\right)}}{\text{Cov}\left(p,q\right) - \dfrac{\text{Var}\left(q\right)\text{Var}\left(p\right)}{\text{Cov}\left(p,q\right)}}\right) = 0$$

$$\delta_{0,1} + \delta_{1,1}\left(\frac{\text{Cov}\left(p,pq\right) - \dfrac{\text{Cov}\left(q,pq\right)\text{Var}\left(p\right)}{\text{Cov}\left(p,q\right)}}{\text{Cov}\left(p,q\right) - \dfrac{\text{Var}\left(q\right)\text{Var}\left(p\right)}{\text{Cov}\left(p,q\right)}}\right) = \hat{\gamma}_{0,1}$$

$$\hat{\gamma}_{0,1} = \delta_{0,1} + \delta_{1,1}\left(\frac{\text{Cov}\left(p,q\right)\text{Cov}\left(p,pq\right) - \text{Cov}\left(q,pq\right)\text{Var}\left(p\right)}{\left(\text{Cov}\left(p,q\right)\right)^2 - \text{Var}\left(p\right)\text{Var}\left(q\right)}\right)$$

This means we found $\hat{\gamma}_{0,1}$. In order to also find $\hat{\gamma}_{1,0}$, we fill this in in the equation

$$\left(\delta_{1,0} - \hat{\gamma}_{1,0}\right)\text{Var}\left(p\right) + \left(\delta_{0,1} - \hat{\gamma}_{0,1}\right)\text{Cov}\left(p,q\right) + \delta_{1,1}\text{Cov}\left(p,pq\right) = 0$$

which gives us

$$\left(\delta_{1,0} - \hat{\gamma}_{1,0}\right)\text{Var}\left(p\right) - \delta_{1,1}\left(\frac{\text{Cov}\left(p,q\right)\text{Cov}\left(p,pq\right) - \text{Cov}\left(q,pq\right)\text{Var}\left(p\right)}{\left(\text{Cov}\left(p,q\right)\right)^2 - \text{Var}\left(p\right)\text{Var}\left(q\right)}\right)\text{Cov}\left(p,q\right)$$

$$+ \delta_{1,1}\text{Cov}\left(p,pq\right) = 0$$

This we can rewrite as follows

$$\left(\delta_{1,0} - \hat{\gamma}_{1,0}\right)\text{Var}\left(p\right) - \delta_{1,1}\left(\frac{\left(\text{Cov}\left(p,q\right)\right)^2\text{Cov}\left(p,pq\right) - \text{Cov}\left(p,q\right)\text{Cov}\left(q,pq\right)\text{Var}\left(p\right)}{\left(\text{Cov}\left(p,q\right)\right)^2 - \text{Var}\left(p\right)\text{Var}\left(q\right)}\right)$$

$$+ \delta_{1,1}\left(\frac{\left(\text{Cov}\left(p,q\right)\right)^2\text{Cov}\left(p,pq\right) - \text{Cov}\left(p,pq\right)\text{Var}\left(p\right)\text{Var}\left(q\right)}{\left(\text{Cov}\left(p,q\right)\right)^2 - \text{Var}\left(p\right)\text{Var}\left(q\right)}\right) = 0$$

$$\left(\delta_{1,0} - \hat{\gamma}_{1,0}\right)\text{Var}\left(p\right) - \delta_{1,1}\left(\frac{\text{Cov}\left(p,pq\right)\text{Var}\left(p\right)\text{Var}\left(q\right) - \text{Cov}\left(p,q\right)\text{Cov}\left(q,pq\right)\text{Var}\left(p\right)}{\left(\text{Cov}\left(p,q\right)\right)^2 - \text{Var}\left(p\right)\text{Var}\left(q\right)}\right) = 0$$

$$\left(\delta_{1,0} - \hat{\gamma}_{1,0}\right) - \delta_{1,1}\left(\frac{\text{Cov}\left(p,pq\right)\text{Var}\left(q\right) - \text{Cov}\left(p,q\right)\text{Cov}\left(q,pq\right)}{\left(\text{Cov}\left(p,q\right)\right)^2 - \text{Var}\left(p\right)\text{Var}\left(q\right)}\right) = 0$$

And thereby we found

$$\hat{\gamma}_{1,0} = \delta_{1,0} - \delta_{1,1}\left(\frac{\text{Cov}\left(p,pq\right)\text{Var}\left(q\right) - \text{Cov}\left(p,q\right)\text{Cov}\left(q,pq\right)}{\left(\text{Cov}\left(p,q\right)\right)^2 - \text{Var}\left(p\right)\text{Var}\left(q\right)}\right)$$

## 2.4. Shortcuts and simple checks

Given that we know that for every choice for a model, the Generalized Price equation has $\bar{w}\Delta\bar{p}$ on the left-hand side, and something times $\mathrm{Var}\left(p\right)$ on the right-hand side, we can also, for an infinitely large population, fill in the model to find an expression for $\Delta\bar{p}$, multiply it by $\bar{w}$ and divide by $\mathrm{Var}\left(p\right)$ to recover what the core term in the Price-like equation would have to amount to. This is what we will do here, and it works for Price-like equations that are applied to the matching model. We begin with some additional details in describing states of the infinite population for the simple setting with asexual reproduction and a binary $p$-score.

### 2.4.1 Population states

With asexual reproduction and a binary $p$-score, a population state in an infinite population model is characterized by the share of individuals that has a $p$-score of 1, which is $\bar{p}$. Every individual is matched with one partner, and since the $q$-score represents the $p$-score of the partner, the first observation is that these averages must be the same; $\bar{p} = \bar{q}$.

The second observation is that, with a binary $p$-score $E\left(p^2\right) = E\left(p\right)$, and hence

$$\mathrm{Var}\left(p\right) = E\left(p^2\right) - E^2\left(p\right) = E\left(p\right) - E^2\left(p\right) = \bar{p}\left(1 - \bar{p}\right)$$

Similarly, $\mathrm{Var}\left(p\right) = \mathrm{Var}\left(q\right)$ if every individual is matched with one partner. We will also refer to those with a $p$-score of 1 as cooperators and those with a $p$-score of 0 as defectors. This does not describe the trait very well in Model 1, but it is consistent across models.

The population structure is characterized by relatedness $r$, which determines the probabilities with which the two types are matched with themselves and each other.

$$P\left(q_i = 1 \mid p_i = 1\right) = \left(1 - r\right)\bar{p} + r$$

$$P\left(q_i = 1 \mid p_i = 0\right) = \left(1 - r\right)\bar{p}$$

$$P\left(q_i = 0 \mid p_i = 0\right) = \left(1 - r\right)\left(1 - \bar{p}\right) + r$$

$$P\left(q_i = 0 \mid p_i = 1\right) = \left(1 - r\right)\left(1 - \bar{p}\right)$$

Another way to write this is $r = P\left(q_i = 1 \mid p_i = 1\right) - P\left(q_i = 1 \mid p_i = 0\right)$, which justifies thinking of $r$ as the additional probability of being matched to a type, if it is the same type as one is oneself (see *van Veelen et al., 2017*).

In an infinite population, these probabilities determine the shares of different types of pairs in a straightforward way. If $f_0$ is the share of pairs in which both have a $p$-score of 0, $f_1$ is the share of pairs in which one has a $p$-score of 0, and the other a $p$-score of 1, and $f_2$ is the share of pairs in which both have a $p$-score of 1, then these shares are given by

$$f_0 = \left(1 - r\right)\left(1 - \bar{p}\right)^2 + r\left(1 - \bar{p}\right)$$

$$f_1 = \left(1 - r\right)2\bar{p}\left(1 - \bar{p}\right)$$

$$f_2 = \left(1 - r\right)\bar{p}^2 + r\bar{p}$$

This implies that, in this simple setting, with a binary $p$-score, and every individual being matched with one other individual, there is a straightforward expression for $\mathrm{Cov}\left(p, q\right)$.

$$\mathrm{Cov}\left(p, q\right) = E\left(pq\right) - E\left(p\right)E\left(q\right) = f_2 - \bar{p}\bar{q} = f_2 - \bar{p}^2 = \left(1 - r\right)\bar{p}^2 + r\bar{p} - \bar{p}^2$$
$$= r\bar{p} - r\bar{p}^2 = r\bar{p}\left(1 - \bar{p}\right)$$

This is useful for calculating $r$ in Model 2, which is also $r_{0,1}$ in Model 3:

$$r = r_{0,1} = \frac{\mathrm{Cov}\left(p, q\right)}{\mathrm{Var}\left(p\right)} = \frac{r\bar{p}\left(1 - \bar{p}\right)}{\bar{p}\left(1 - \bar{p}\right)}$$

For calculating $r_{1,1}$ in Model 3, and for calculating $s_b$ and $s_c$, which feature in the condition if we apply Price-like equation 2 to Model 3, we will also need the following covariances. Here we again use the fact that they are binary, which means that, for instance $p^2q$ is 1 if $p$ and $q$ are both 1, and 0 in all other cases.

$$\text{Cov}(p, pq) = E\left(p^2 q\right) - E(p) E(pq) = f_2 - \bar{p} f_2 = (1 - \bar{p}) f_2 = (1 - \bar{p}) \left((1 - r)\bar{p}^2 + r\bar{p}\right)$$

$$= \bar{p}(1 - \bar{p})\left((1 - r)\bar{p} + r\right)$$

with $\bar{p} = \bar{q}$, also

$$\text{Cov}(q, pq) = E\left(pq^2\right) - E(q) E(pq) = f_2 - \bar{q} f_2 = f_2 - \bar{p} f_2 = \text{Cov}(p, pq)$$

These are useful for computing the $r_{1,1}$, which features in Model 3.

$$r_{1,1} = \frac{\text{Cov}(p, pq)}{\text{Var}(p)} = \frac{\bar{p}(1 - \bar{p})\left((1 - r)\bar{p} + r\right)}{\bar{p}(1 - \bar{p})} = \left((1 - r)\bar{p} + r\right)$$

There are two more bits of algebra that we will need below. This is not particularly exciting, but later we will be happy to have them available.

$$s_b = \frac{\text{Cov}(p, q)\,\text{Cov}(p, pq) - \text{Cov}(q, pq)\,\text{Var}(p)}{\left(\text{Cov}(p, q)\right)^2 - \text{Var}(p)\,\text{Var}(q)}$$

$$= \frac{\bar{p}(1 - \bar{p})\left((1 - r)\bar{p} + r\right)\left(r\bar{p}(1 - \bar{p}) - \bar{p}(1 - \bar{p})\right)}{\left(r\bar{p}(1 - \bar{p})\right)^2 - \left(\bar{p}(1 - \bar{p})\right)^2}$$

$$= \frac{\left((1 - r)\bar{p} + r\right)(r - 1)}{\left(r^2 - 1\right)} = \frac{\left((1 - r)\bar{p} + r\right)(1 - r)}{1 - r^2} = \frac{(1 - r)\bar{p} + r}{1 + r}$$

Because $\text{Cov}(q, pq) = \text{Cov}(p, pq)$, we find that

$$s_c = \frac{\text{Cov}(p, pq)\,\text{Var}(q) - \text{Cov}(p, q)\,\text{Cov}(q, pq)}{\left(\text{Cov}(p, q)\right)^2 - \text{Var}(p)\,\text{Var}(q)} = -s_b$$

### 2.4.2 Model 1
Model 1 is

$$w_i = \beta_0 + \beta_1 p_i + \varepsilon_i$$

In an infinite population, this means that the average fitness of "cooperators" and "defectors" is

$$\bar{w}_C = \beta_0 + \beta_1$$

$$\bar{w}_D = \beta_0$$

The overall average fitness is

$$\bar{w} = \bar{p}\left(\beta_0 + \beta_1\right) + (1 - \bar{p})\beta_0 = \beta_0 + \bar{p}\beta_1$$

This makes

$$\Delta\bar{p} = \frac{\bar{p}\left(\beta_0 + \beta_1\right)}{\bar{w}} - \bar{p} = \frac{\bar{p}\left(\beta_0 + \beta_1\right)}{\beta_0 + \bar{p}\beta_1} - \frac{\bar{p}\left(\beta_0 + \bar{p}\beta_1\right)}{\beta_0 + \bar{p}\beta_1} = \frac{\beta_1 \cdot \bar{p}(1 - \bar{p})}{\bar{w}}$$

This implies that

$$\bar{w}\Delta\bar{p} = \beta_1 \cdot \bar{p}(1 - \bar{p}) = \beta_1 \cdot \text{Var}(p)$$

The Generalized Price equation in regression form implies that if we take statistical Model 1, then

$$\bar{w}\Delta\bar{p} = \hat{\beta}_1 \cdot \text{Var}(p)$$

That implies that we do not have to do the actual minimizing of the sum of squared errors that we have done in Detailed Calculations 2.4.1 to conclude that $\hat{\beta}_1 = \beta_1$ if we apply Price-like equation 1 to Model 1.

### 2.4.3 Model 2
Model 2 is

$$w_i = \gamma_{0,0} + \gamma_{1,0}p_i + \gamma_{0,1}q_i + \varepsilon_i$$

In an infinite population, this means that the average fitness of cooperators and defectors is

$$\bar{w}_C = \gamma_{0,0} + \gamma_{1,0} + \left((1-r)\bar{p} + r\right)\gamma_{0,1}$$

$$\bar{w}_D = \gamma_{0,0} + (1-r)\bar{p}\gamma_{0,1}$$

The overall average fitness is

$$\bar{w} = \bar{p}\left(\gamma_{0,0} + \gamma_{1,0} + \left((1-r)\bar{p} + r\right)\gamma_{0,1}\right) + (1-\bar{p})\left(\gamma_{0,0} + (1-r)\bar{p}\gamma_{0,1}\right)$$

$$= \gamma_{0,0} + \bar{p}\left(\gamma_{1,0} + \gamma_{0,1}\right)$$

This makes

$$\Delta\bar{p} = \frac{\bar{p}\bar{w}_C}{\bar{w}} - \bar{p} = \frac{\bar{p}\left(\gamma_{0,0} + \gamma_{1,0} + \left((1-r)\bar{p} + r\right)\gamma_{0,1}\right)}{\gamma_{0,0} + \bar{p}\left(\gamma_{1,0} + \gamma_{0,1}\right)} - \frac{\bar{p}\left(\gamma_{0,0} + \bar{p}\left(\gamma_{1,0} + \gamma_{0,1}\right)\right)}{\gamma_{0,0} + \bar{p}\left(\gamma_{1,0} + \gamma_{0,1}\right)}$$

$$= \frac{\bar{p}\left(1-\bar{p}\right)\left(\gamma_{1,0} + r\gamma_{0,1}\right)}{\bar{w}}$$

This implies that

$$\bar{w}\Delta\bar{p} = \left(\gamma_{1,0} + r\gamma_{0,1}\right) \cdot \bar{p}\left(1-\bar{p}\right) = \left(\gamma_{1,0} + r\gamma_{0,1}\right) \cdot \text{Var}(p)$$

The Generalized Price equation in regression form implies that if we take statistical Model 2, then

$$\bar{w}\Delta\bar{p} = \left(\hat{\gamma}_{1,0} + r\hat{\gamma}_{0,1}\right) \cdot \text{Var}(p)$$

That implies that we do not have to do the actual minimizing of the sum of squared errors to conclude that $\hat{\gamma}_{1,0} = \gamma_{1,0}$ and $\hat{\gamma}_{0,1} = \gamma_{0,1}$ if we apply Price-like equation 2 to Model 2.

With $b = \gamma_{0,1}$ and $-c = \gamma_{1,0}$, here we of course recognize Hamilton's original rule.

The Generalized Price equation also implies that if we take statistical Model 1, then still

$$\bar{w}\Delta\bar{p} = \hat{\beta}_1 \cdot \text{Var}(p)$$

That implies that also here, we do not have to do the actual minimizing of the sum of squared errors that we did in Detailed Calculations 2.4.2 above to conclude that, if we apply the Generalized Price equation using statistical Model 1 to Model 2, we get $\hat{\beta}_1 = \gamma_{1,0} + r\gamma_{0,1}$. For other mismatches, this shortcut does not work, and we will have to do the actual minimization.

### 2.4.4 Model 3
Model 3 is

$$w_i = \delta_{0,0} + \delta_{1,0}p_i + \delta_{0,1}q_i + \delta_{1,1}p_iq_i + \varepsilon_i$$

In an infinite population, this means that the average fitness of cooperators and defectors is

$$\bar{w}_C = \delta_{0,0} + \delta_{1,0} + \left((1-r)\bar{p} + r\right)\left(\delta_{0,1} + \delta_{1,1}\right)$$

$$\bar{w}_D = \delta_{0,0} + (1-r)\,\bar{p}\delta_{0,1}$$

The overall average fitness is

$$\bar{w} = \delta_{0,0} + \bar{p}\left(\delta_{0,1} + \delta_{1,0}\right) + \bar{p}\left((1-r)\,\bar{p} + r\right)\delta_{1,1}$$

This makes

$$\begin{aligned}
\Delta\bar{p} &= \frac{\bar{p}\bar{w}_C}{\bar{w}} - \bar{p}\\
&= \frac{\bar{p}\left(\delta_{0,0} + \delta_{1,0} + \left((1-r)\,\bar{p} + r\right)\left(\delta_{0,1} + \delta_{1,1}\right)\right)}{1 + \bar{p}\left(\delta_{0,1} + \delta_{1,0}\right) + \bar{p}\left((1-r)\,\bar{p} + r\right)\delta_{1,1}}\\
&\quad - \frac{\bar{p}\left(\delta_{0,0} + \bar{p}\left(\delta_{0,1} + \delta_{1,0}\right) + \bar{p}\left((1-r)\,\bar{p} + r\right)\delta_{1,1}\right)}{1 + \bar{p}\left(\delta_{0,1} + \delta_{1,0}\right) + \bar{p}\left((1-r)\,\bar{p} + r\right)\delta_{1,1}}\\
&= \frac{\delta_{1,0}\bar{p}\left(1-\bar{p}\right) + r\delta_{0,1}\bar{p}\left(1-\bar{p}\right) + \left((1-r)\,\bar{p} + r\right)\delta_{1,1}\bar{p}\left(1-\bar{p}\right)}{\bar{w}}
\end{aligned}$$

This implies that

$$\begin{aligned}
\bar{w}\Delta\bar{p} &= \left(\delta_{1,0} + r\delta_{0,1} + \left((1-r)\,\bar{p} + r\right)\delta_{1,1}\right)\cdot\bar{p}\left(1-\bar{p}\right)\\
&= \left(\delta_{1,0} + r\delta_{0,1} + \left((1-r)\,\bar{p} + r\right)\delta_{1,1}\right)\cdot\mathrm{Var}\left(p\right)
\end{aligned}$$

The Generalized Price equation in regression form implies that if we take statistical Model 3, then

$$\bar{w}\Delta\bar{p} = \left(\hat{\delta}_{1,0} + r\hat{\delta}_{0,1} + \left((1-r)\,\bar{p} + r\right)\hat{\delta}_{1,1}\right)\cdot\mathrm{Var}\left(p\right)$$

That implies that we do not have to do the actual minimizing of the sum of squared errors to conclude that $\hat{\delta}_{1,0} = \delta_{1,0}$, $\hat{\delta}_{0,1} = \delta_{0,1}$, and $\hat{\delta}_{1,1} = \delta_{1,1}$ if we apply Price-like equation 3 to Model 3.

## 2.4.5 Price-like equation 2, applied to Model 3
The Generalized Price equation in regression form for Model 2 is

$$\bar{w}\Delta\bar{p} = \left(\hat{\gamma}_{1,0} + r\hat{\gamma}_{0,1}\right)\cdot\mathrm{Var}\left(p\right)$$

If we apply this to Model 3, then this does not allow for a straightforward computation of $\hat{\gamma}_{1,0}$ and $\hat{\gamma}_{0,1}$ without having to do the actual minimization of the sum of squared errors. The reason for that is that now we have one equation, that is, we have $\hat{\gamma}_{1,0} + \hat{\gamma}_{0,1} = \delta_{1,0} + r\delta_{0,1} + \left((1-r)\,\bar{p} + r\right)\delta_{1,1}$, and two variables, $\hat{\gamma}_{1,0}$ and $\hat{\gamma}_{0,1}$. What we can do, however, is just verify that the $\hat{\gamma}_{1,0}$ and $\hat{\gamma}_{0,1}$ that we found in B4.3, work. There, we found that

$$\hat{\gamma}_{0,0} = \delta_{0,0} - (1-r)\,\bar{p}s_b\delta_{1,1}$$

$$\hat{\gamma}_{1,0} = \delta_{1,0} - s_c\delta_{1,1}$$

$$\hat{\gamma}_{0,1} = \delta_{0,1} + s_b\delta_{1,1}$$

with

$$s_b = \frac{(1-r)\,\bar{p} + r}{1+r}$$

$$s_c = \frac{-(1-r)\,\bar{p} + r}{1+r}$$

If we then fill those in, we get

$$\bar{w}\Delta\bar{p} = \left(\hat{\gamma}_{1,0} + r\hat{\gamma}_{0,1}\right) \cdot \text{Var}\left(p\right) = \left(\delta_{1,0} - s_c\delta_{1,1} + r\left(\delta_{0,1} + s_b\delta_{1,1}\right)\right) \cdot \text{Var}\left(p\right)$$

$$= \left(\delta_{1,0} + r\delta_{0,1} + \left(rs_b - s_c\right)\delta_{1,1}\right) \cdot \text{Var}\left(p\right)$$

$$= \left(\delta_{1,0} + r\delta_{0,1} + \left(\left(1-r\right)\bar{p} + r\right)\delta_{1,1}\right) \cdot \text{Var}\left(p\right)$$

This is indeed what we saw above.

### Relation to the claim in *Grafen, 1985a*

If the true model were Model 2, one would interpret $\hat{\gamma}_{1,0}$ as the effect of having the gene on oneself. The Generalized Price equation in regression form for Model 2 applied to Model 3 gives, as we saw above

$$\hat{\gamma}_{1,0} = \delta_{1,0} + \frac{\left(1-r\right)\bar{p} + r}{1+r}\delta_{1,1},$$

This is the lower red line in *Appendix 2—figure 4* below. It is not a constant and cannot be interpreted as the effect of having the gene on oneself. We can search for a possible interpretation, though, by calculating the *average* effects of having the gene. We will begin by taking the perspective of those with the gene and calculating the difference between their actual fitness and what their fitness would have been, had they not had the gene (this is also referred to as the counterfactual method in *van Veelen et al., 2017*; *van Veelen, 2018a*). This amounts to $\delta_{1,0}$ plus $\delta_{1,1}$ times the probability that carriers are matched with another individual that has the gene.

$$\delta_{1,0} + \left(\left(1-r\right)\bar{p} + r\right)\delta_{1,1}$$

In *Appendix 2—figure 4* below, this is the second dashed blue line from the bottom.

We can also take the perspective of those without the gene and calculate the difference between what their fitness would have been if they had the gene and their actual fitness. This is $\delta_{1,0}$ plus $\delta_{1,1}$ times the probability that non-carriers are matched with an individual that has the gene.

$$\delta_{1,0} + \left(1-r\right)\bar{p}\delta_{1,1}$$

In *Appendix 2—figure 4* below, this is the dashed blue line at the bottom.

Finally, one could also average these average effects over carriers and non-carriers. That would give

$$\bar{p}\left(\delta_{1,0} + \left(\left(1-r\right)\bar{p} + r\right)\delta_{1,1}\right) + \left(1-\bar{p}\right)\left(\delta_{1,0} + \left(1-r\right)\bar{p}\delta_{1,1}\right)$$

$$= \delta_{1,0} + \left(1-r\right)\bar{p}\delta_{1,1} + r\bar{p}\delta_{1,1} = \delta_{1,0} + \bar{p}\delta_{1,1}$$

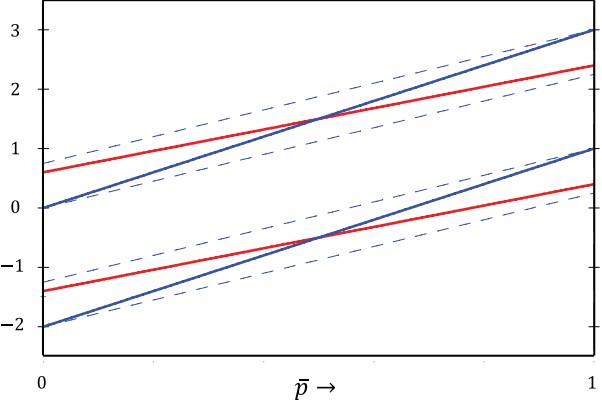

**Appendix 2—figure 4.** Applying the Generalized Price equation for Model 2 to Model 3. For this figure, we chose Model 3 with $\delta_{1,0} = -2$, $\delta_{1,1} = 0$, and $\delta_{1,1} = 3$. The red lines are $\hat{\gamma}_{1,0}$ (the lower one), and $\hat{\gamma}_{0,1}$ (the higher one), which, when applied to Model 2, would have been the effect on the individual itself and the effect on the partner. The average effect on the individual itself is the lower unbroken blue line, which is accompanied by the

*Appendix 2—figure 4 continued*

average effect for the defectors (the dashed line below it) and the average effect for cooperators (the dashed line immediately above it). The average effect on the partner is the higher unbroken blue line, which is accompanied by the average effect for the defectors (the dashed line immediately below it) and the average effect for cooperators (the dashed line above it).

In *Appendix 2—figure 4* above, this is the lower unbroken blue line. We see that neither of these three matches the that we get from applying the Generalized Price equation for Model 2 to Model 3.

We can do the same for $\hat{\gamma}_{0,1}$. If the true model were Model 2, one would interpret $\hat{\gamma}_{0,1}$ as the effect of having the gene on one's partner. The Generalized Price equation in regression form for Model 2 applied to Model 3 gives

$$\hat{\gamma}_{0,1} = \delta_{0,1} + \frac{(1-r)\,\bar{p} + r}{1+r}\delta_{1,1},$$

This is the higher up red line in *Appendix 2—figure 4* above. This is not a constant either and cannot be interpreted as the effect of having the gene on the interaction partner. We will begin by taking the perspective of those with the gene and calculating the difference between the actual fitness of their partner and what their partner's fitness would have been, had they not had the gene. This amounts to $\delta_{0,1}$ plus $\delta_{1,1}$ times the probability that carriers are matched with another individual that has the gene.

$$\delta_{0,1} + \left((1-r)\,\bar{p} + r\right)\delta_{1,1}$$

In *Appendix 2—figure 4* above, this is the dashed blue line at the top.

We can also take the perspective of those without the gene, and calculate the difference between what their partner's fitness would have been, if they themselves would have had the gene, and their actual fitness. This is $\delta_{0,1}$ plus $\delta_{1,1}$ times the probability that non-carriers are matched with an individual that has the gene.

$$\delta_{0,1} + \left(1-r\right)\,\bar{p}\delta_{1,1}$$

In *Appendix 2—figure 4* above, this is the second dashed blue line from the top.

Finally, one could also average these average effects over carriers and non-carriers. That would give

$$\delta_{0,1} + \left(1-r\right)\,\bar{p}\delta_{1,1} + r\bar{p}\delta_{1,1} = \delta_{0,1} + \bar{p}\delta_{1,1}$$

In *Appendix 2—figure 4* above, this is the higher up unbroken blue line. We see that also here, neither of these three matches the $\hat{\gamma}_{0,1}$ that we get from applying the Generalized Price equation in regression form for Model 2 to Model 3.

*Appendix 2—figure 4* illustrates this and visualizes that trying to understand the working of Model 3 through applying the Generalized Price equation for Model 2 on it is misleading. The values for $\hat{\gamma}_{1,0}$ and $\hat{\gamma}_{0,1}$ that this results in do not coincide with the average effect on the individual itself and the average effect on the partner.

