## [Editor Report · eLife Assessment]

Kin selection and inclusive fitness have generated significant controversy. This paper reconsiders the general form of Hamilton's rule in which benefits and costs are defined as regression coefficients, with higher-order coefficients being added to accommodate non-linear interactions. The paper is a **landmark** contribution to the field with **compelling**, systematic analysis, giving clarity to long-standing debates.

---

## [Referee Report · Joint Public Review]

This manuscript reconsiders the "general form" of Hamilton's rule, in which "benefit" and "cost" are defined as regression coefficients. It points out that there is no reason to insist on Hamilton's rule of the form -c+br>0, and that, in fact, arbitrarily many terms (i.e. higher-order regression coefficients) can be added to Hamilton's rule to reflect nonlinear interactions. Furthermore, it argues that insisting on a rule of the form -c+br>0 can result in conditions that are true but meaningless and that statistical considerations should be employed to determine which form of Hamilton's rule is meaningful for a given dataset or model.

Comments on latest version:

The authors have provided a robust, valuable and detailed response to the previous reviews.

Comments from Reviewer #1: I have nothing further to add.

Comments from Reviewer #2: I appreciate the clarifications the author has made to the manuscript regarding (i) "sample covariance" terminology, (ii) the generality of the "generalized Price equation", and (iii) the distinction between the covariance and regression forms of the Price equation. I also appreciate that the ms now engages more deeply with some of the previous literature on regression-based Hamilton's rules (e.g. Smith et al., 2010; Rousset 2015). I feel these revisions make this contribution more valuable, and also more technically sound, since the term "sample covariance" is no longer used incorrectly.

I also add that I agree with the substance of the authors' response to Reviewer #3. That is, the original submission was very clear that the regression-based Hamilton's rule is already completely general in the range of situations to which it applies, and that the added "generality" in the present ms refers to the variety of regression models that can be applied to these situations. In this way, the original ms already anticipates and addresses the criticism that Reviewer #3 raises.

Reviewer #3 did not provide comments on the revised version.

---

## [Author Response]

The following is the authors’ response to the original reviews

**Reviewer #1 (Public review):**
Summary:There has been intense controversy over the generality of Hamilton's inclusive fitness rule for how evolution works on social behaviors. All generally agree that relatedness can be a game changer, for example allowing for otherwise unselectable altruistic behaviors when 𝑐 < 𝑟𝑏, where 𝑐 is the fitness cost to the altruism, 𝑏 is the fitness benefit to another, and 𝑟 their relatedness. Many complications have been successfully incorporated into the theory, including different reproductive values and viscous population structures.

I agree, especially if by incorporating viscous population structures, the reviewer means the discovery of the cancellation effect (Wilson, Pollock, and Dugatkin, 1992, Taylor, 1992).

The controversy has centered on another dimension; Hamilton's original model was for additive fitness, but how does his result hold when fitnesses are non-additive? One approach has been not to worry about a general result but just find results for particular cases. A consistent finding is that the results depend on the frequency of the social allele - nonadditivity causes frequency dependence that was absent in Hamilton's approach.

Just to be extra precise: Hamilton’s (1964) original model did not use the Price equation nor the regression approach to define costs and benefits, and it did indeed simply presuppose fixed, additive fitness effects.

Also for extra precision on terminology: many researchers will describe all fitnesses in social evolution as frequency dependent. The reason they do, is that with or without additivity, both the fitness of cooperators (with the social allele) and the fitness of defectors (without the social alle) typically increase in the frequency of cooperators in the population; the more cooperators there are, the more individuals run into them, which increases average fitness. The result depending on the frequency I take to mean that which of those two fitnesses is larger flips at a certain frequency, which automatically implies that the difference between them is depending on the frequency of the social allele. This is indeed the result of non-additivity. We will return to this in more detail in the response to Reviewer #3. Also at the end of Appendix 2 I have added a bit to be extra precise regarding frequency dependence.

Two other approaches derive from Queller via the Price equation. Queller 1 is to find forms like Hamilton's rule, but with additional terms that deal with non-additive interaction, each with an r-like population structure variable multiplied by a b-like fitness effect (Queller, 1985). Queller 2 redefines the fitness effects c and b as partial regressions of the actor's and recipient's genes on fitness. This leaves Hamilton's rule intact, just with new definitions of c and b that depend on frequency (Queller, 1992a).Queller 2 is the version that has been most adopted by the inclusive fitness community along with assertions that Hamilton's rule in completely general. In this paper, van Veelen argues that Queller 1 is the correct approach. He derives a general form that Queller only hinted at. He does so within a more rigorous framework that puts both Price's equation and Hamilton's rule on firmer statistical ground. Within that framework, the Queller 2 approach is seen to be a statistical misspecification - it employs a model without interaction in cases that actually do have interaction. If we accept that this is a fatal flaw, the original version of Hamilton's rule is limited to linear fitness models, which might not be common.

I totally agree.

Strengths:While the approach is not entirely new, this paper provides a more rigorous approach and a more general result. It shows that both Queller 1 and Queller 2 are identities and give accurate results, because both are derived from the Price equation, which is an identity. So why prefer Queller 1? It identifies the misspecification issue with the Queller 2 approach and points out its consequences. For example, it will not give the minimum squared differences between the model and data. It does not separate the behavioral effects of the individuals from the population state (𝑏 and 𝑐 become dependent on 𝑟 and the population frequency).

Just to be precise on a detail: in the data domain, as long as the number of parameters in a statistical model is lower than the number of data points, adding parameters typically (generically) lowers the sum of squared errors. That is to say, for an underspecified statistical model, the sum of squared errors goes down if a parameter is added, but for an already overspecified statistical model, the same is still true (although, typically, by how much the sum of squared errors is reduced will differ). The model specification task for a statistician includes knowing when to keep adding parameters, because the data suggest that the model is still underspecified, and when to stop adding parameters, because the model is well-specified, even if adding parameters still reduces the sum of squared errors.

In a modeling context, on the other hand, one can say that sum of squared differences will stop decreasing at the point where the statistical model is well-specified, that is: when it matches the model we are considering.

The paper also shows how the same problems can apply to non-social traits. Epistasis is the non-additivity of effects of two genes within the individual. (So one wonders why have we not had a similarly fierce controversy over how we should treat epistasis?)The paper is clearly written. Though somewhat repetitive, particularly in the long supplement, most of that repetition has the purpose of underscoring how the same points apply equally to a variety of different models.Finally, this may be a big step towards reconciliation in the inclusive fitness wars. Van Veelen has been one of the harshest critics of inclusive fitness, and now he is proposing a version of it.

I am very happy to hear this, because I am indeed hopeful for reconciliation. I would like to add a comment, though. The debate on Hamilton’s rule/inclusive fitness is regularly thought of as a battle between two partizan camps, where both sides care at least as much about winning as they do about getting things right. This is totally understandable, because to some degree that is true. Also, I agree that it is fair to position me in the camp that is critical of the inclusive fitness literature. However, I would like to think that I have not been taking random shots at Hamilton’s rule. I have pointed to problems with the typical use of the Price equation and Hamilton’s rule, and I think I did for very good reasons. I am obviously very happy that finding the Generalized Price equation, and the general version of Hamilton’s rule, allowed me to go beyond this, and (finally) offer a correct alternative, and I totally appreciate that this opens the door for reconciliation, as this reviewer points out. But I would not describe this as a road-to-Damascus moment. In order to illustrate the continuity in my work, I would like to point to three papers.

In van Veelen (2007), I pointed to the missing link between the central result in Hamilton’s (1964) famous paper (which states that selection dynamics take the population to a state where **mean** inclusive fitness is maximized), and Hamilton’s actual rule (which states that selection will lead to individuals maximizing their **individual** inclusive fitness). My repair stated the additional assumptions that were necessary to make the latter follow from the former. I would say that this can hardly be characterized as an attack on Hamilton’s rule. Reading Hamilton (1964) with enough care to notice something is missing, and then repairing it, I think is a sign of respect, and not an attack.

Van Veelen (2011) is about the replicator dynamics for n-player games, with the possibility of assortment. This puts the paper in a domain that does not assume weak selection, and that is typically not much oriented towards inclusive fitness. I included a theorem that implies that, under the condition of linearity, inclusive fitness not only gets the direction of selection right, but \begin{document}$rb -c$\end{document} becomes a parameter that also determines the speed of selection. This I think is representative, in the sense that in many of my papers, I carefully stake out when the classic version of Hamilton’s rule does work.

In Akdeniz and van Veelen (2020), we moreover take a totally standard inclusive fitness approach in a model of the cancellation effect at the group level.

I would say that this does not line up with the image of a harsh critic that takes random shots at Hamilton’s rule or inclusive fitness.

Weaknesses:van Veelen argues that the field essentially abandoned the Queller 1 approach after its publication. I think this is putting it too strongly - there have been a number of theoretical studies that incorporate extra terms with higher-order relatednesses. It is probably accurate to say that there has been relative neglect. But perhaps this is partly due to a perception that this approach is difficult to apply.

I can imagine that the perceived difficulty in application may have played a role in the neglect of the Queller 1 approach. What for sure has played a role, and I would think a much bigger one, is that the literature has been pretty outspoken that the Queller 1 approach is the wrong way to go. The main text cites a number of papers that hold this position very emphatically. The first one of those was a News and Views by Alan Grafen [1985] that accompanied the paper in which Queller presented his Queller 1 approach and I am very happy that Appendix 2 shows on how many levels this News and Views was wrong. There is only a handful of papers that follow the Queller 1 example.

The model in this paper is quite elegant and helps clarify conceptual issues, but I wonder how practical it will turn out to be. In terms of modeling complicated cases, I suspect most practitioners will continue doing what they have been doing, for example using population genetics or adaptive dynamics, without worrying about neatly separating out a series of terms multiplying fitness coefficients and population structure coefficients.

I am not sure if I see what the reviewer envisions practitioners that use population genetics will keep on doing. I would think that the Generalized Price equation in regression form is a description of population genetic dynamics, and therefore, if practitioners will not make an effort to “neatly separate out a series of terms multiplying fitness coefficients and population structure coefficients”, then all I can say is that they should. I cannot do more than explain why, if they do not, they are at risk of mischaracterizing what gets selected and why.

Regarding those that use adaptive dynamics, I would say that this is a whole different approach. Within this approach, one can also apply inclusive fitness; see Section 6 and Appendix D of van Veelen et al. (2017). Appendix D is full of deep technical results and was done by Benjamin Allen.

For empirical studies, it is going to be hard to even try to estimate all those additional parameters. In reality, even the standard Hamilton's rule is rarely tested by trying to estimate all its parameters. Instead, it is commonly tested more indirectly, for example by comparative tests of the importance of relatedness. That of course would not distinguish between additive and non-additive models that both depend on relatedness, but it does test the core idea of kin selection. It will be interesting to see if van Veelen's approach stimulates new ways of exploring the real world.

Regarding the impact on empirical studies, there are a few things that I would like to say. The first is that I would just like to repeat, maybe a bit more elaborately, what I wrote at the end of the main text. Given that the generalized version of Hamilton’s rule produces a host of Hamilton-like rules, and given the fact that all of them by construction indicate the direction of selection accurately, the question whether or not Hamilton’s rule holds turns out to be ill-posed. That means that we can stop doing empirical tests of Hamilton’s rule, which are predicated on the idea that Hamilton’s rule, with benefits and costs being determined by the regression method, could be violated – which it cannot. Instead, we should do empirical studies to find out which version of Hamilton’s rule applies to which behaviour in which species. Side note: it is possible to violate Hamilton’s rule, if costs and benefits are defined according to the counterfactual method; see van Veelen et al. (2017) and van Veelen (2018). This way of defining costs and benefits is less common, although there are authors that find this definition natural enough to assume that this is the way in which everybody defines costs and benefits (Karlin and Matessi, 1983, Matessi and Karlin, 1984).

I would like to not understate what a step forward this is. The size of the step forwards is of course also due to the dismal point of departure. As theorists, we have failed our empiricists, because all 12 studies included in the review by Bourke (2014) of papers that explicitly test Hamilton’s rule are based on the misguided idea that the traditional Hamilton’s rule, with costs and benefits defined according to the regression method, can be violated. While the field does sometimes have disdain for mathematical nit-picking, this is a point where a little more attention to detail would have really helped. If the hypothesis is that Hamilton’s rule holds, and the null is that it does not, then trying to specify how the empirical quantity that reflects inclusive fitness would be distributed under the null hypothesis (in order to do the right statistical tests) would have forced researchers to do something with the information that this quantity is not distributed at all, because Hamilton’s rule is general (in the sense that it holds for any way in which the world works). If one would prefer to reverse the null and the alternative hypothesis, one would run into similar problems. Understanding that the question is ill-posed therefore is a big step forwards from the terrible state of statistics and the waste of research time, attention and money on the empirical side of this field (see also Section 8 of van Veelen et al., 2017).

I would agree that doing comparative statics may not be much affected by this. Section 5 of van Veelen et al. (2017) indicates that there can be a large set of circumstances under which the general idea “relatedness up → cooperation up” still applies. But that may be a bit unambitious, and Section 8 of van Veelen et al. (2017), and the final section of van Veelen (2018) contain some reflections on empirical testing that may allow us to go beyond that. As long as there is change happening in the Generalized Price equation, the population is not in equilibrium. For empirical tests, one can either aim to capture selection as it happens, or assume that what we observe reflects properties of an equilibrium. This leads to interesting reflections on how to do empirics, which may differ between traits that are continuous and traits that are discrete again: see van Veelen et al. (2017), and van Veelen (2018).

**Reviewer #2 (Public review):**
Summary:This manuscript reconsiders the "general form" of Hamilton's rule, in which "benefit" and "cost" are defined as regression coefficients. It points out that there is no reason to insist on Hamilton's rule of the form \begin{document}$- c + br \gt0$\end{document}, and that, in fact, arbitrarily many terms (i.e. higherorder regression coefficients) can be added to Hamilton's rule to reflect nonlinear interactions. Furthermore, it argues that insisting on a rule of the form \begin{document}$- c + br \gt0$\end{document} can result in conditions that are true but meaningless and that statistical considerations should be employed to determine which form of Hamilton's rule is meaningful for a given dataset or model.

Totally right. I cannot help to want to be extra precise, though, by distinguishing between the data domain and the modelling domain. In the data domain, statistical considerations apply in order to avoid misspecification. In this domain, avoiding misspecification can be complicated, because we do not know the underlying data generating process, and we depend on noisy data to make a best guess. In the modeling domain, however, there is no excuse for misspecification, as the model is postulated by the modeler. I therefore would think that in this domain, it does not really require “statistical considerations” to minimize the probability of misspecification; we can get the probability of misspecification all the way down to 0 by just choosing not to do it.

Strengths:The point is an important one. While it is not entirely novel-the idea of adding extra terms to Hamilton's rule has arisen sporadically (Queller, 1985, 2011; Fletcher et al., 2006; van Veelen et al., 2017)--it is very useful to have a systematic treatment of this point. I think the manuscript can make an important contribution by helping to clarify a number of debates in the literature. I particularly appreciate the heterozygote advantage example in the SI.

Me too, and I really hope the readers make it this far! I have thought of putting it in the main text, but did not know where that would fit.

Weaknesses:Although the mathematical analysis is rigorously done and I largely agree with the conclusions, I feel there are some issues regarding terminology, some regarding the state of the field, and the practice of statistics that need to be clarified if the manuscript is truly to resolve the outstanding issues of the field. Otherwise, I worry that it will in some ways add to the confusion.(1) The "generalized" Price equation: I agree that the equations labeled (PE.C) and (GPE.C) are different in a subtle yet meaningful way. But I do not see any way in which (GPE.C) is more general than (PE.C). That is, I cannot envision any circumstance in which (GPE.C) applies but (PE.C) does not. A term other than "generalized" should be used.

This is a great point! Just to make sure that those that read the reports online understand this point, let me add some detail. The equation labeled (PE.C) – which is short for Price equation in covariance form – is\begin{document}$$\displaystyle \bar{w} \Delta \bar{p}=\mathrm{Cov}(w, p)+E(w \Delta p)$$\end{document}

The derivation in Appendix 1 then assumes that we have a statistical model that includes a constant and a linear term for the p-score. It then defines the model-estimated fitness of individual \begin{document}$i$\end{document} as \begin{document}$\hat{w}_i = w_i - \varepsilon_{i}$\end{document}, where \begin{document}$w_i$\end{document} is the realized number of offspring of individual \begin{document}$i$\end{document}, and \begin{document}$\varepsilon_{i}$\end{document} is the error term – and it is the sum over all individuals of this error term-squared that is minimized. The vector of model-estimated fitnesses \begin{document}$\hat{w}=\left[\hat{w}_{1}, \ldots, \hat{w}_{n}\right]$\end{document} will typically be different for different choices of the statistical model. Appendix 1 then goes on to show that, whatever the statistical model is that is used, for all of them \begin{document}$\mathrm{Cov}(\hat{w}, p)=\mathrm{Cov}(w, p)$\end{document}, as long as the statistical model includes a constant and a linear term for the p-score. That means that we can rewrite (PE.C) as\begin{document}$$\displaystyle \bar{w} \Delta \bar{p}=\mathrm{Cov}(\hat{w}, p)+E(w \Delta p)$$\end{document}

The point that the reviewer is making, is that this is not really a generalization. For a given dataset (or, more generally, for a given population transition, whether empirical or in a model), \begin{document}$\mathrm{Cov}(w, p)$\end{document} is just a number, and it happens to be the case that \begin{document}$\mathrm{Cov}(\hat{w}, p)$\end{document} returns the same number, whatever statistical model we use for determining what the model-estimated fitnesses \begin{document}$\hat{w}_i$\end{document} are (as long as the statistical model includes a constant and a linear term for the p-score). In other words, (PE.C) is not really nested in (GPE.C), so (GPE.C) is not a proper generalization of (PE.C).

This is a totally correct point, and I had actually struggled a bit with the question what terminology to use here. Equation (GPE.C) is definitely **general**, in the sense that we can change the statistical model, and thereby change the vector of model-estimated fitnesses \begin{document}$\hat{w}=\left[\hat{w}_{1}, \ldots, \hat{w}_{n}\right]$\end{document}, but as long as we keep the constant and the linear term in the statistical model, the equation still applies. But it is **not a generalization of (PE.C)**.

I do however have a hard time coming up with a better label. The General Price equation may be a bit better, but it still suggests generalization. The Statistical Model-based Price equation does not suggest or imply generalization, but it does not convey how general it is, and it suggests that it could be an alternative to the normal Price equation that one may or may not choose to use – while this version really is the one we should use. It may moreover create the impression that this is only for doing statistics, and one might use the traditional Price equation for anything that is not statistics. I cannot really think of other good alternatives, but I am of course open to suggestions.

So, by lack of a better label, I called this the Generalized Price equation in covariance form. Though clearly imperfect, there are still a few good things about this label. The first is that, as mentioned above, this equation is general, in the sense that it holds, regardless of the statistical model. The second reason is that this is Step 1 in a sequence of three steps, the other two of which do produce proper generalizations. Step 2 goes from this equation in covariance form to the Generalized Price Equation in regression form, which is a proper generalization of the traditional Price equation in regression form. Step 3 goes from the Generalized Price Equation in regression form to the general version of Hamilton’s rule, which is also a proper generalization of the classical Hamilton’s rule. Since I would suggest that Step 1 on its own is kind of useless, and therefore Step 1 and Step 2 will typically come as a package, I would be tempted to think that this justifies the abuse of terminology for the Price Equation in covariance form. I did however add the observation made by the reviewer at the point where the Generalized Price equation (in both forms) is derived, so I hope this at least partly addresses this concern.

(2) Regression vs covariance forms of the Price equation: I think the author uses "generalized" in reference to what Price called the "regression form" of his equation. But to almost everyone in the field, the "Price Equation" refers to the covariance form. For this reason, it is very confusing when the manuscript refers to the regression form as simply "the Price Equation".As an example, in the box on p. 15, the manuscript states "The Price equation can be generalized, in the sense that one can write a variety of Price-like equations for a variety of possible true models, that may have generated the data." But it is not the Price equation (covariance form) that is being generalized here. It is only the regression that Price used that is being generalized.To be consistent with the field, I suggest the term "Price Equation" be used only to refer to the covariance form unless it is otherwise specified as in "regression form of the Price equation".

I am not sure about the level of confusion induced here, but I totally see that it can be helpful to avoid all ambiguity. I therefore went over everything, and whenever I wrote “Price equation”, I tried to make sure it comes either with “in covariance form” or with “in regression form”. At some places, it is a bit over the top to keep repeating “in regression form”, when it is abundantly clear which form is being discussed. Also, I added no qualifiers if a statement is true for both forms of the Price equation, or if the claim refers to the whole package of going through Step 1 and Step 2 mentioned above.

(3) Sample covariance: The author refers to the covariance in the Price equation as “sample covariance”. This is not correct, since sample covariance has a denominator of N-1 rather than N (Bessel’s correction). The correct term, when summing over an entire population, is “population covariance”. Price (1972) was clear about this: “In this paper we will be concerned with population functions and make no use of sample functions”. This point is elaborated on by Frank (2012), in the subsection “Interpretation of Covariance”.

I totally agree. On page 418 of van Veelen (2005), I wrote:

“Another possibility is that we think of \begin{document}$z_i$\end{document} and \begin{document}$q_i$\end{document}, \begin{document}$i = 1, \dots, N$\end{document} as realizations of a jointly distributed random variable. […] In that case the expression between square brackets is a good approximation for what statisticians […] call a sample covariance. A sample covariance is defined as \begin{document}$\frac{\sum^{N}_{i=1}{(z_i - \overline{z})(q_i - \overline{q})}}{N-1}$\end{document}, but in large samples it is OK to replace \begin{document}$N-1$\end{document} by \begin{document}$N$\end{document},and then this formula reduces to Price’s \begin{document}$\mathrm{Cov}(z,q)$\end{document}.”

In van Veelen et al. (2012), I slid a little, because in Box 1 on page 66, I wrote that \begin{document}$\left[ \frac{\sum^{N}_{i=1} z_i q_i}{N} - \left( \frac{\sum^{N}_{i=1} z_i}{N} \right) \left( \frac{\sum^{N}_{i=1} q_i}{N} \right) \right]$\end{document} is the sample covariance, and only in footnote 1 on the same page did I include Bessel’s correction, when I wrote:

“To be perfectly precise, the sample covariance is defined as \begin{document}$\frac{N}{N-1} \left[ \frac{\sum^{N}_{i=1} z_i q_i}{N} - \left( \frac{\sum^{N}_{i=1} z_i}{N} \right) \left( \frac{\sum^{N}_{i=1} q_i}{N} \right) \right]$\end{document}.”

In this manuscript, I slid a little further, and left Bessel’s correction out altogether. I am happy that the reviewer pointed this out, so I can make this maximally precise again.

The reviewer also quotes Price (1972), page 485:

“In this paper we will be concerned with population functions and make no use of sample functions”.

Below, the reviewer will return to the issue of distinguishing between the sample covariance with Bessel’s correction, and the sample covariance without Bessel’s correction, where the latter is regularly also referred to as the population covariance. A natural interpretation of the quote from Price (1972), if we read a bit around this quote in the paper, is that the difference between his “population functions” and his “sample functions” is indeed Bessel’s correction.

The reviewer also states that Frank (2012) elaborates on this in the subsection “Interpretation of Covariance”. What is interesting, though, is that, when Frank (2012) writes, on page 1017, that “It is important to distinguish between population measures and sample measures”, the difference between those is not that one does, and the other does not include Bessel’s correction. The difference between “population measures” and “sample measures” in Frank (2012), page 1017, is that “In many statistical applications, one only has data on a subset of the full population, that subset forming a sample.”

The distinction between a population covariance and a sample covariance in Frank (2012) therefore is that they are “covariances” of different things (where the word covariances is in quotation marks, because, again, they are not really covariances). Besides just making sure that Price (1972) and Frank (2012) are not using these terms in the same way, this also **perfectly illustrates** the mix-up between statistical populations (or data generating processes) and biological populations that I discuss in the subsection “Being careful with labels” in Section 2 of Appendix 1. I will return to this below, when I explain why I want to avoid using the word “population covariance” for the sample covariance without Bessel’s correction.

Of course, the difference is negligible when the population is large. However, the author applies the covariance formula to populations as small as 𝑁 = 2, for which the correction factor is significant.

Absolutely right.

The author objects to using the term "population covariance" (SI, pp. 8-9) on the grounds that it might be misleading if the covariance, regression coefficients, etc. are used for inference because in this case, what is being inferred is not a population statistic but an underlying relationship. However, I am not convinced that statistical inference is or should be the primary use of the Price equation (see next point). At any rate, avoiding potential confusion is not a sufficient reason to use incorrect terminology.

There are a few related, but separate issues. One is what to call the \begin{document}$\mathrm{Cov}(w,p)$\end{document}-term. Another, somewhat broader, is to avoid mixing up statistical populations and biological populations. A third is what the primary use of the Price equation is. The third issue I will respond to below, where it reappears. Here I will focus on the first two, which can be discussed without addressing the third.

In a data context, I now call the \begin{document}$\mathrm{Cov}(w,p)$\end{document}-term “\begin{document}$\frac{n-1}{n}$\end{document} times the sample covariance, or, in other words, the sample covariance without Bessel’s correction”. This should be unambiguous. In a modeling context I refer to \begin{document}$\mathrm{Cov}(w,p)$\end{document}-term as “the -term” and describe it as a summary statistic or a notational convention. There are two reasons for this choice.

The first is that neither of these use the word “population”. I like this, because there is a persistent scope for confusion between statistical populations and biological populations (as exemplified by Frank, 2012). This leads to an incorrect, but widespread intuition that if we “know the entire (biological) population” in a data context, there is nothing that can be estimated. This is what the subsection “Being careful with labels” in section 2 of Appendix 1 is all about.

The second reason is that by using two labels, I also differentiate between the data context and the modeling context. This is important for reasons I will return to later.

Relatedly, I suggest avoiding using 𝐸 for the second term in the Price equation, since (as the ms points out), it is not the expectation of any random variable. It is a population mean. There is no reason not to use something like Avg or bar notation to indicate population mean. Price (1972) uses "ave" for average.

I totally agree that the second term in the Price equation is not an expectation. I made this point in van Veelen (2005), and I repeated this in the manuscript. This remark by the reviewer prompted me to spell this out a bit more emphatically in Appendix 1. That still leaves me with the choice what notation to use.

I therefore looked up all contributions to the Theme issue “Fifty years of the Price equation” in the Philosophical Transactions of the Royal Society B, and found that almost all contributions use 𝐸, sometimes saying that this refers to an expectation or an average. Of course, this is wrong. However (and this is another argument), it is equally wrong as using 𝐶𝑜𝑣 or 𝑉𝑎𝑟. The terms abbreviated as 𝐶𝑜𝑣 and 𝑉𝑎𝑟 are equally much not a covariance and a variance as the term abbreviated as 𝐸 is not an expectation. So I would think that there are a few reasons for sticking with 𝐸 here; (1) consistency with the literature; (2) consistency with the treatment of other terms; and (3) the fact that this term is not really of any importance in this manuscript. I do however totally understand the reviewer’s reasons, which I suppose include that for using 𝐸, there are relatively unproblematic alternatives (ave or upper bar) that are not available for the other terms. I hope therefore that being a bit more emphatic in the manuscript about 𝐸 not being an expectation at least partly addresses this concern.

I should add, however, that the distinction between population statistics vs sample statistics goes away for regression coefficients (e.g. b, c, and r in Hamilton's rule) since in this case, Bessel's correction cancels out.

Totally correct.

(4) Descriptive vs. inferential statistics: When discussing the statistical quantities in the Price Equation, the author appears to treat them all as inferential statistics. That is, he takes the position that the population data are all generated by some probabilistic model and that the goal of computing the statistical quantities in the Price Equation is to correctly infer this model.

Before I respond to this, I would like to point out that this literature has started going off the rails right from the very beginning. One of the initial construction errors was to use the ungeneralized Price equation in regression form. The other one is that the paper in which Price (1970) presented his equation is inconsistent, and suggests that the equation can be used for **constructing** hypotheses and for **testing** them **at the same time** (see van Veelen (2005), page 416). That, of course, is not possible; the first happens in the theory/modeling domain, and the second in the empirical testing/statistics domain, and they are separate exercises.

These construction errors have warped the literature based on it, and have resulted in a lot of mental gymnastics and esoteric statements, which are needed if we are not willing to consider the possibility that there could be anything amiss with the original paper by Price (1970).

In this paper, I undo both of these construction errors. Undoing the second one means exploring both domains separately. In Sections 2-4 of Appendix 1 I explore the possibility that the Price equation is applied to data. In Section 5 of Appendix 1 I explore the possibility that it is used in a modelling context. The primary effort here is just to do it right, and I have not read anything to suggest that I did not succeed in doing this. Secondarily, of course, I also want to contrast this to what happens in the existing literature. That is what this point by the reviewer is about. It is therefore important to be aware that seeing the contrast accurately is complicated by the apologetic warp in the existing literature.

As a first effort to unwarp, I would like to point to the fact that I am not taking any position on what the Price equation should be used for. All I do here is explore (and find) possibilities, both in the statistical inference domain and in the modeling domain. I also find that there is scope for misspecification in both, and that, in both domains, we should want to avoid misspecification. The thing that I criticize in the existing literature therefore is not the choice of domain. The thing that I criticize is the insistence on, and celebrating of what is most accurately described as misspecification. This typically happens in the modeling domain.

It is worth pointing out that those who argue in favor of the Price Equation do not see it this way: "it is a mistake to assume that it must be the evolutionary theorist, writing out covariances, who is performing the equivalent of a statistical analysis." (Gardner, West, and Wild, 2011); "Neither data nor inferences are considered here" (Rousset, 2015). From what I can tell, to the supporters of the Price equation and the regression form of Hamilton's rule, the statistical quantities involved are either population-level *descriptive* statistics (in an empirical context), or else are statistics of random variables (in a stochastic modeling context).

Again, this description of the friction between my paper and the existing literature is predicated on the suggestion that I have only one domain in mind where the Price equation can be applied. That is not the case; I consider both.

In the previous paragraph, the reviewer states that I “treat statistical quantities as inferential statistics”, and in this paragraph the reviewer contrasts that with the supporters of the (ungeneralized) Price equation that supposedly treat the same quantities as “descriptive statistics”. This is also beside the point, but it will take some effort to sort out the spaghetti of entangled arguments (where the spaghetti is the result of the history in this field, as indicated earlier).

First of all, it is not unimportant to point out that the way most people use the terms “inferential statistics” and “descriptive statistics” is that the first refers to an activity, and the second to a function of a bunch of numbers, typically data. Inferential statistics is a combination of parameter estimation and model specification (those are activities). Descriptive statistics are for instance the average values of variables of interest (which makes them a function of a set of numbers). When doing inferential statistics (or statistical inference), looking at the descriptive statistics of the dataset is just a routine before the real work begins. It is important to remember that.

Now I suppose that this reviewer uses these words a little differently. When he or she writes that I “treat statistical quantities as inferential statistics”, I assume that the reviewer means that I want to use a term like \begin{document}$\beta=\frac{\mathrm{Cov}(w, p)}{\mathrm{Var}(p)}$\end{document} for doing statistical inference, or that, when I want to interpret such a term, I include considerations typical of statistical inference. Within the data domain, that is totally correct. In the paper I argue that there are very good reasons for this. We would like to know what the data can tell us about the actual fitness function, and if we do our statistical inference right, and choose our Price-like equation accordingly, then that means that we would be able to give a meaningful interpretation to a term like \begin{document}$\beta=\frac{\mathrm{Cov}(w, p)}{\mathrm{Var}(p)}$\end{document}. It also means that we then have an equation that describes the genetic population dynamics accurately.

When the reviewer states that other papers treat them as “population level descriptive statistics” in an empirical context, I have a hard time coming up with papers for which that is the case. Most papers apply the Price equation in the modeling domain (that is to say: this is true in evolution; in ecology the Price equation is often applied to data – see Pillai and Gouhier (2019) and Bourrat et al. (2023)). But even if there are researchers that apply the Price equation to data, then considering these statistical quantities as “descriptive statistics” would not make sense. Looking at the descriptive statistics alone is not an empirical exercise; it is just a routine that happens before the actual statistical inference starts. In a data context, saying that considerations that are standard in statistical inference do not apply, because one is just not doing statistical inference, is the equivalent of an admission of guilt. If you do not consider statistical significance, and never mention that sample size could matter, because you are using these terms as “descriptive statistics, not inferential statistics”, then you’re basically admitting to not doing a serious empirical study.

Besides treating statistical quantities as descriptive statistics in a data context, the reviewer also states that, in a stochastic modeling context, other researchers treat the same statistical quantities as “statistics of random variables”. This is first of all very generous to the existing literature. I imagine that the reviewer is imagining a modeling exercise where for instance the covariance between two variables is postulated. A theory exercise would then take that as a starting point for the derivation of some theoretical result. This, however, is not what happens in most of the literature.

There are two things that I would like to point out. First of all, postulating covariances and deriving results from assumptions regarding those covariances is not an activity that requires using the Price equation. There are many stochastic models that function perfectly fine without the Price equation. This is maybe a detail, but it is important to realize that what the reviewer probably thinks of as a legitimate theoretical exercise may be something that can very well be done without the Price equation.

Secondly, I would like to repeat something that I have pointed out before, which is that the Price equation can be written for any transition, whether this transition is likely or unlikely, given a model, and even for transitions that are impossible. For all of those transitions, one can write the (ungeneralized) Price equation, and for all of those, the Price equation will be an identity, and it will contain the things that the reviewer refers to as “statistical quantities”. It is important to realize that these “statistical quantities”, therefore, are properties of a transition, and that every transition comes with its own ”statistical quantity”. That implies that they are not properties of random variables; they reflect something regarding one transition. What one could imagine, though, is the following. To fix ideas, let’s take the Price equation in regression form, and focus on \begin{document}$\beta=\frac{\mathrm{Cov}(w, p)}{\mathrm{Var}(p)}$\end{document}. A meaningful modeling exercise starts with assumptions about the likelihood of all different transitions, and therefore the likelihood of different values of 𝛽 materializing – or it starts with assumptions that imply those probabilities. In a theoretical exercise, one could then derive statements about the expectation and variance of those “statistical quantities”. For instance, one can calculate the expected value \begin{document}$E[\beta]= E\left[\frac{\mathrm{Cov}(w, p)}{\mathrm{Var}(p)}\right]$\end{document}, and the variance \begin{document}$\mathrm{Var}[\beta]= \mathrm{Var}\left[\frac{\mathrm{Cov}(w, p)}{\mathrm{Var}(p)}\right]$\end{document}, where this expectation is a proper expectation (taken over the probabilities with which these transitions materialize) and this variance is a proper variance, for the same reason.

This is what I do on page 416 of van Veelen (2005) and in Section 5 of Appendix 1. I think something like this is what the reviewer may have in mind, but it is worth pointing out that this still does not mean that the \begin{document}$\beta=\frac{\mathrm{Cov}(w, p)}{\mathrm{Var}(p)}$\end{document} from the Price equation for any given transition is now a property of a random variable. Much of the literature, however, is not at the level of sophistication that I imagine the reviewer has in mind – although there are papers that are; see the discussion below of Rousset and Billiard (2000) and Van Cleve (2015).

In the appendix to this reply, I will address the quotes from Gardner, West, and Wild (2011) and Rousset (2015). This takes up some space, so that is why it is at the end of this reply.

In short, the manuscript seems to argue that Price equation users are performing statistical inference incorrectly, whereas the users insist that they are not doing statistical inference at all.

That is not what the manuscript argues, but I am happy to clarify. The manuscript explores both the use of the Price equation when applied to data (and therefore for statistical inference) and when applied to transitions in a model. The criticism on the existing literature is not that it performs statistical inference incorrectly. The criticism is that the literature insists on misspecification, which typically happens in a modelling context.

The problem (and here I think the author would agree with me) arises when users of the Price equation go on to make predictive or causal claims that would require the kind of statistical analysis they claim not to be doing. Claims of the form "Hamilton's rule predicts.." or use of terms like "benefit" and "cost" suggest that one has inferred a predictive or causal relationship in the given data, while somehow bypassing the entire theory of statistical inference.

I do not really know how to interpret this paragraph. The use of the word “data” suggests that this pertains to a data context, but I do not know what would qualify as a “predictive claim” in that domain, or how any study would go from data to a claim of the form “Hamilton’s rule predicts …”. Again, I do not really know papers that apply the Price equation to data. None of the empirical papers reviewed in Bourke (2014) for instance do. I would however agree that it is close to obvious that an approach that does indeed bypass the entire theory of statistical inference cannot identify causal relations in datasets. I think the examples in Section 2 of Appendix 1 also clearly illustrate that a literature in which the word “sample size” is absent, cannot be doing statistical inference.

There is also a third way to use the Price equation which is entirely unobjectionable: as a way to express the relationship between individual-level fitness and population-level gene frequency change in a form that is convenient for further algebraic manipulation. I suspect that this is actually the most common use of the Price equation in practice.

I am not sure if I understand what it means for the Price equation to “express the relationship between individual-level fitness and population-level gene frequency change”. That is a bit reminiscent of how John Maynard Smith saw the Price equation (Okasha, 2005), but he also emphasized that he was unable to follow George Price and his equation. For sure, it cannot be that one side of the Price equation reflects something at the individual level and the other something at the population level, because both sides of the Price equation are equally aggregated over the population. Just to be safe, and to avoid unwarranted associative thinking, I would therefore choose to be minimalistic, and say that the Price equation is an identity for a transition between a parent population and an offspring population.

Regardless of the words we choose, however, the question how harmless or objectionable the use of the Price equation is in the literature is absolutely relevant. In earlier papers I have tried to cover a spectrum of examples of different ways to use (or misuse) the Price equation. In van Veelen (2005) I cover Grafen (1985a), Taylor (1989), Price (1972), and Sober and Wilson (2007). The main paper that is discussed in van Veelen et al. (2012) is Queller (1992b), but Section 7 of that paper also discusses the way the Price equation is used in Rousset and Billiard (2000), Taylor (1989), Queller (1985), and Page and Nowak (2002). These discussions also come with a description of how much it takes to repair them, and this varies all the way from nothing, or a bit of minor rewording, to being beyond repair.

What is good to observe, is that the papers in which the use of the Price equation is the least problematic, are also the papers in which, if the reference to the Price equation would be taken out, nothing really changes. These are papers that start with a model, or a collection of models, and that, at some point in the derivation of their results, point to a step that can, but does not have to be described as using the Price equation. An example of this is Rousset and Billiard (2000); see the detailed description in Section 7 of van Veelen et al. (2012).

I am happy to point to a few more papers on the no harm, no foul end of the spectrum here.

Allen and Tarnita (2012) discuss properties of the dynamics in a well-defined set of models. Towards the end of the paper, a version of the Price equation more or less naturally appears. This is more of an interesting aside, though, and does not really play a role in derivation of the core results of the paper. Van Cleve (2015) is similar to Rousset and Billiard (2000), in that the “application of the Price equation” there is a minor ingredient of the derivation of the results. (A detail that this reviewer may find worth mentioning, given earlier comments, is that Van Cleve (2015) writes the left-hand side of the Price equation as \begin{document}$E(w \Delta p | \mathbf{p} )$\end{document}, instead of \begin{document}$\bar{w} \Delta \bar{p}$\end{document}. First two very unimportant things. Van Cleve (2015) uses \begin{document}$w$\end{document} for mean fitness, for which \begin{document}$\bar{w}$\end{document} is a more common symbol. Another detail of lesser importance is that it includes the vector of parent p-scores in the notation, which in their notation is \begin{document}$\mathbf{p}$\end{document}. More importantly, however, is that Van Cleve (2015) writes \begin{document}$E(\Delta p)$\end{document} for \begin{document}$\Delta \bar{p}$\end{document}, which extends the (mis)use of the symbol \begin{document}$E$\end{document} for what really is just an average. This is consistent within the Price equation, in the sense that it now denotes the average with \begin{document}$E$\end{document}, both on the right-hand side and on the left-hand side of the Price equation. It can however be a little bit confusing, because when Rousset and Billiard (2000) write \begin{document}$E_{p}(\bar{W} \Delta p)$\end{document}, then this is a proper expectation. In their case, this summarizes all possible transitions out of a given state, and weighs them by their probabilities of happening, given a state summarized by \begin{document}$p$\end{document}.) I am also happy to extend the spectrum a bit here. Some papers on inclusive fitness do not use the Price equation at all, even though one could imagine places where it could be inserted. A nice example of such a paper is Taylor et al. (2007).

In this paper, I hope I can be excused from taking a complete inventory of this literature, and I hope that I do not have to count how many papers fall into the different categories. This would help assess the veracity of the suspicion the reviewer has, which is that the most common use of the Price equation is entirely unobjectionable, but I just do not have the time. I would however not want to underestimate the aggregate damage done in this field. The spectrum spanned in my earlier papers does include a fair amount of nonsense results. This typically happens in papers that do not study a specific model or set of models, but that take the Price equation as their point of departure for their theorizing. Also there seems to be a positive correlation between how exalted and venerating the language is that is used when describing the wonders and depths of the Price equation, and how little sense the claims make that are “derived” with it.

We also should not set the bar too low. This is a literature that, at the starting point, has a few construction errors in it, as described in the paper. That is reason for concern. Moreover, one of the main end products of this literature is what we send our empiricists to the field with. As Section 8 of van Veelen et al. (2017) indicates, what we have supplied to our empiricists to work with is nothing short of terrible. I would therefore want to maintain that the damage done is enormous, and if there are also a few papers around that may use the ungeneralized Price equation in an innocuous way, then that is not enough redemption for my taste. We are still facing a literature in which, at every instance where the Price equation is used, we still need to check in which category it falls.

For a paper that aims to clarify these thorny concepts in the literature, I think it is worth pointing out these different interpretations of statistical quantities in the Price equation (descriptive statistics vs inferential statistics vs algebraic manipulation). One can then critique the conclusions that are inappropriately drawn from the Price equation, which would require rigorous statistical inference to draw. Without these clarifications, supporters of the Price equation will again argue that this manuscript has misunderstood the purpose of the equation and that they never claimed to do inference in the first place.

I would like to return to the point that I made at the beginning of my response to point (4), which is that the “thorniness” of these concepts is the result of the warp in the literature, resulting from the construction errors in Price (1970). If people want to understand how to apply the Price equation right, I think that reading Appendix 1 and 2 would work just fine. Again, I have not read anything that suggests that there is anything incorrect in there, so if the literature contains “thorny” concepts, it might just be that this is the result of the mental gymnastics necessitated by the unwillingness to accept that there might be something not completely right with Price (1970). Moreover, given my experiences in the field, I am not sure that there is anything that I could say that would convince the supporters of the ungeneralized Price equation.

(5) "True" models: Even if one accepts that the statistical quantities in the Price equation are inferential in nature, the author appears to go a step further by asserting that, even in empirical populations, there is a specific "true" model which it is our goal to infer. This assumption manifests at many points in the SI when the author refers to the "true model" or "true, underlying population structure" in the context of an empirical population.

Again, in Appendix 1 I explore both a data context and a modeling context. In the modeling context none of this applies, because in such a context, there is only the model that we postulate. In the part in which I explore what the Price equation can do in a data context, I do indeed use words like “true model” or "true underlying population structure".

I do not think it is necessary or appropriate, in empirical contexts, to posit the existence of a Platonic "true" model that is generating the data. Real populations are not governed by mathematical models. Moreover, the goal of statistical inference is not to determine the "true model" for given data but to say whether a given statistical model is justified based on this data. Fitting a linear model, for example, does not rule out the possibility there may be higher-order interactions - it just means we do not have a statistical basis to infer these higher-order interactions from the data (say, because their p-scores are insignificant), and so we leave them out.

This remark suggests that the statistical approach in Sections 2-4 of Appendix 1 is more naïve than it should be, and that I would overlook the possibility of, for instance, interaction effects that are really nonzero, but that are statistically not significant. Now first of all, at a superficial level, I would like to say that this strikes me as somewhat inconsistent. In the remarks further back, the reviewer seems to excuse those that use the Price equation on data without any statistical considerations whatsoever. The reason why the reviewer is giving them a pass, is that they are “just not doing statistical inference”. Instead, they are doing this whole other thing with, you know, descriptive statistics. As I indicated above, that is just a fancy way of saying that they are not doing serious statistics – or serious empirics, for that matter.

In this comment, on the other hand, the reviewer also suggests that the statistics that I use to replace the total absence of any statistical considerations with, is not quite up to snuff. Below, I will indicate why that is not the case at all, but I think it is also worth registering a touch of irony there.

In order to address this issue, it is worth first observing that the whole of classical statistics is based on probability theory in the following sense. We are always asking ourselves the question: if the data generating process works like this, what would the likelihood be of certain outcomes (datasets); and if the data generating process works some other way (sometimes: the complement of whatever “this” is), what would the likelihood then be of the same outcomes. By comparing those, we draw inferences about the underlying data generating process (which is a word suggestive of a “Platonic” world view that the reviewer seems to reject). Therefore, if one would impose a ban on using Platonic words like “true data generating process”; “actual fitness function”; or “the population structure that is out there”, it would be impossible to teach any course in statistics, basic or advanced. Also it would be impossible to practice, and talk about, applied statistics.

Now the reviewer claims that “Real populations are not governed by mathematical models”. I do not really know if I agree or disagree with that statement, but the example that the reviewer gives does not fit that claim. The reviewer suggests that if we find a higher order term not to be statistically significant (and therefore we reject the hypothesis that it is nonzero), then that would not necessarily mean that it is not there. That is totally true, and statisticians tend to be fully aware of that. But that does not imply that there is no true data-generating process; the whole premise of this example is that there is, but that the sample size is not large enough to determine it in a detailed enough way so as to include this interaction effect, that apparently is small relative to the sample size.

The third thing to reflect on here, is that the reviewer seems to suggest that the Generalized Price equation in regression form, as presented in my paper, comes with a specific statistical approach, that he or she classifies as philosophically naïve or unsophisticated. That, however, is not the case, and I am very grateful that this remark by this reviewer allows me to make a point that I think shines a light on how the Generalized Price equation puts the train that started going off the rails in 1970 back on track, and reconnects it with the statistics it borrows its terminology from. To see that, it is good to be aware that statistics never gives certainty. The whole discipline is built around the awareness that it is possible to draw the wrong inference, and the aim is to determine, minimize, and balance, the likelihoods of making different wrong inferences. So, statistics produces statements about the confidence with which one can say that something works one way or the other. In some instances, the data are not enough to say anything with any confidence. In other cases, the data are rich enough so that it is really unlikely that we incorrectly infer that for instance a certain gene matters for fitness.

The nice thing about the setup with the Generalized Price equation, is that those statistical considerations translate one-to-one to considerations regarding which Price-like equation to choose. If the data do not allow us to pick any model with confidence, then we should be equally agnostic about which Price-like equation describes the population genetic dynamics accurately. If the statistics gives us high confidence that a certain model matches the data, then we should pick the matching Price-like equation with the same confidence. This also carries over to higher level statistical considerations.

If we think about terms that, if we would gather a gargantuan amount of data, might be statistically significant, but very small, then economists call those statistically significant, but economically insignificant. When rejecting the statistical significance on the basis of a not gargantuan dataset, statisticians are aware that terms that really have a zero effect, as well as terms, the effect of which is really small, are rejected with the same statistical test – and that we should be fine with that. All such considerations carry over to what we think of regarding the choice of a Price-like equation to describe the population genetic dynamics. Even if people disagree about whether or not to include a term that is statistically significant, but relatively small, such a disagreement can still happen within this setup, and just translates to a disagreement on which Price-like equation to choose.

Similarly, people could also disagree about whether it is justified to use polynomials to characterize a fitness function. If we decide that we can, because of Taylor expansions, then the core result of the paper implies that the population genetic dynamics can be summarized by a generalized Hamilton’s rule (as long as the fitness function includes a constant and a linear term regarding the p-score). On the other hand, if we do not believe this is justified, and prefer to use an altogether different family of fitness functions, then we can no longer do this. All of this leaves space for all kinds of statistical considerations and disagreements, that just carry over to the choice for one or the other Price-like equation as an accurate description of the population genetic dynamics. Or, if one does not believe polynomials should be used, then this leads to not picking any Price-like equation at all.

So, this is a long way of saying that the Generalized Price equation creates space for all statistical considerations to regain their place, and does not hinge on one approach to statistics or another.

What we can say is that if we apply the statistical model to data generated by a probabilistic model, and if these models match, then as the number of observations grows to infinity, the estimators in the statistical model converge to the parameters of the data-generating one. But this is a mathematical statement, not a statement about real-world populations.

Again, I do not know if I agree or disagree with the last sentence. However, that does not really matter, because either option only has implications for how we are to think of the relation between a Price-like equation describing a population genetic dynamics and real-world populations. It is not relevant for the question which Price-like equation to pick, or whether to pick one at all.

A resolution I suggest to points 3, 4, and 5 above is:*A priori, the statistical quantities in the Price Equation are descriptive statistics, pertaining only to the specific population data given.*If one wishes to impute any predictive power, generalizability, or causal meaning to these statistics, all the standard considerations of inferential statistics apply. In particular, one must choose a statistical model that is justified based on the given data. In this case, one is not guaranteed to obtain the standard (linear) Hamilton's rule and may obtain any of an infinite family of rules.*If one uses a model that is not justified based on the given data, the results will still be correct for the given population data but will lack any meaning or generalizability beyond that.*In particular, if one considers data generated by a probabilistic model, and applies a statistical model that does not match the data-generating one, the results will be misleading, and will not generalize beyond the randomly generated realization one uses.Of course, the author may propose a different resolution to points 3-5, but they should be resolved somehow. Otherwise, the terminology in the manuscript will be incorrect and the ms will not resolve confusion in the field.

I have outlined my solutions extensively above. I really appreciate that Reviewers #1 and #2 have spent time and attention on the manuscript and on the long appendices.

**Appendix to the response to reviewer #2: Some remarks on Gardner, West & Wild (2011), Frank (2012), and Rousset (2015)**

An accurate response to the quote from Gardner, West, and Wild (2011) in the review report takes up space. I therefore wanted to put that in an appendix to the response to reviewer #2. I also include a few paragraphs regarding Frank (2012) and Rousset (2015), both of which are also mentioned by reviewer #2. All of this might also be of interest to people that are curious about how what I find in my paper relates to the existing literature.

**Gardner, West & Wild (2011)** The quote I am responding to is “it is a mistake to assume that it must be the evolutionary theorist, writing out covariances, who is performing the equivalent of a statistical analysis” I want to put that into context, so I will go over the whole paragraph that surrounds the quote. The paragraph is called ‘**Statistics and Evolutionary Theory**’ and can be found on page 1038 of the paper. I think that it is worth pointing out that it is not easy to respond to their somewhat impressionistic collages of words and formulas. I will therefore cut the paragraph up in a few smaller bits and try to make sense of it bit by bit. The paragraph begins with:

“Our account of the general theory of kin selection has been framed in statistical terms.”

Based on what they write two sentences down, the best match between those words and what they do in the paper would be: “our account uses words like “covariance”, “variance” and “expectation” for things that are not what “covariance”, “variance” and “expectation” mean in probability theory and statistics.” I would be totally open to an argument why that is nonetheless OK to do, but the way Gardner, West, and Wild (2011) phrase it obscures the fact that this needs any justification or reflection at all. “Framing something in statistical terms” is unspecific enough to sound completely harmless.

“The use of statistical methods in the mathematical development of Darwinian theory has itself been subjected to recent criticism (van Veelen, 2005; Nowak et al., 2010b), so we address this criticism here.”

Also here, specifics would be helpful. The “use of statistical methods” sounds like it is more than just using terms from statistics, so this might refer to the minimizing of the sum of squared differences, which is also mentioned a sentence down in Gardner, West, and Wild (2011). If it does, then it is worth observing that in statistics, the minimizing of the sum of squared differences (or residuals, or errors) comes with theorems that point very clearly to what is being achieved by doing this. The **Gauss–Markov theorem** states that the ordinary least squares (OLS) estimator has the lowest variance within the class of linear unbiased estimators. This implies that minimizing the sum of squared errors helps answering a well-defined question in statistics; under certain conditions, an OLS estimator is our best shot at uncovering an unknown relation between variables. To also minimize a sum of squared differences, but now in the modeling domain, qualifies as “use of statistical methods” only in a very shallow way. It means that a similar minimization is performed. Without an equivalent of the **Gauss–Markov theorem** that would shine a light on what it is that is being achieved by doing so, that does not carry the same weight as it does in the statistics domain – in that it does not carry any weight at all.

“The concern is that statistical terms – such as covariances and least-squares regressions – should properly be reserved for conventional statistical analyses, where hypotheses are tested against explicit data, and that they are out of place in the foundations of evolutionary theory (van Veelen, 2005; Nowak et al., 2010b).”

Again, a few things are a bit vague. What are “explicit data”? Are there data that are not explicit? Why the generic “foundations of evolutionary theory”, instead of a more specific description of what these statistical terms are used for? But either way, this is a misrepresentation of what I wrote in van Veelen (2005). I did not suggest to “reserve statistical terms for conventional statistical analysis” just because. As I do here in the current paper, what I did there was explore the possibilities for the Price equation to help with what I then called Type I and Type II questions. Type I questions find themselves in the modeling domain and Type II questions find themselves in the statistical domain. I was not arguing for a ban on applying statistical concepts outside of the domain of statistical inference. All that I said is that in its current practice, it does not really help answering questions of either type.

“However, this concern is misplaced. First, natural selection is a statistical process, and it is therefore natural that this should be defined in terms of aggregate statistics, even if only strictly by analogy (Frank, 1997a, 1998).”

This is a vague non-argument. Almost nothing is well-defined here. What does it mean for natural selection to be a statistical process? Is that just an unusual term for a random process? If so, then I suppose I agree, but that has nothing to do with what I state or claim. And what does it mean to be defined in terms of aggregate statistics? What is the alternative? I have no idea how any of this relates to anything that I claim or state in my papers.

“Second, Fisher (1930, p198) coined the term ‘covariance’ in the context of his exposition of the genetical theory of natural selection, so the evolutionary usage of this term has precedent over the way the term is used in other fields.”

This is what I would call a “historic fallacy”. The fact that Fisher coined the term “covariance” in a book on genetics and natural selection does not mean that any “evolutionary usage” of the term “covariance”, however nonsensical, now has precedent over the way the term is used in other fields. Irrespective of the path that the history of science, genetics, or statistics took, right now we are in a place where about every student at every university anywhere in the world that takes a course in probability theory and/or statistics, learns that covariance is a property of a random variable (see also Wikipedia). And they do for a very good reason; it is essential in recognizing the relation between probability theory on the one hand and statistics on the other. Being curious how this “evolutionary usage” of the term covariance works, if covariance turns out not to be a property of a random variable, is therefore perfectly justified, and “Fisher coined the term” is not a safe word that exempts it from scrutiny.

“Third, it is a mistake to assume that it must be the evolutionary theorist, writing out covariances, who is performing the equivalent of a statistical analysis.”

Again, that is just not what anyone is saying. Nobody is suggesting that an evolutionary theorist should perform the equivalent of statistical analysis. All I did was point to how little is being achieved by transferring formulas from statistics to a modeling context.

“A better analogy is to regard Mother Nature in the role of statistician, analysing fitness effects of genes by the method of least-squares, and driving genetic change according to the results of her analyses (cf. Crow, 2008).”

I have no idea what any of this means. Mother Nature is a personification of something that is not a person, and that does not have cognition. Without sentience, “Mother Nature” cannot assume the role of statistician, and cannot analyse fitness effects.

“More generally, analogy is the basis of all understanding, so when isomorphisms arise unexpectedly between different branches of mathematics (in this case, theoretical population genetics and statistical least-squares analysis) this represents an opportunity for advancing scientific progress and not an anomaly that is to be avoided.”

This is a strawman argument, puffed up with platitudes. Nobody is arguing against analogies. But what is the analogy supposed to be here? Just taking least squares from statistical inference and performing it in a modeling context does not make it an analogy. The **Gauss–Markov theorem**, which is the basis for why least squares helps answering questions in statistics, just does not mean anything in a modeling context. OLS in modeling is just willful misspecification, and nothing that it does in statistics translates to anything meaningful in modeling. Again, declaring it an analogy, or an isomorphism, does not make it one.

**Frank (2012)** Because the reviewer also mentions Frank (2012), I would like to include a small remark on this paper too. “Natural Selection. IV. The Price equation” by Frank (2012) is partly a response to my earlier criticism of the use of the Price equation. Much like Gardner, West, and Wild (2011), I would describe this paper as what is called a ”flight forwards” in Dutch. While the questions I ask are relatively prosaic (such as: how does the Price equation help derive a prediction from model assumptions?), Frank (2012) pivots to suggesting that there is a profound philosophy-of-science disagreement that I am on the wrong side of. It is close to impossible to respond to Frank (2012), because it is a labyrinth of arguments that sound deep and impressive, but that are just not specific enough to know how they relate to points that I made – or even just what they mean in general. Just to pick a random paragraph:

“Is there some reorientation for the expression of natural selection that may provide subtle perspective, from which we can understand our subject more deeply and analyse our problems with greater ease and greater insight? My answer is, as I have mentioned, that the Price equation provides that sort of reorientation. To argue the point, I will have to keep at the distinction between the concrete and the abstract, and the relative roles of those two endpoints in mature theoretical understanding.”

For many of those terms, I have no real idea what they mean, and also reading the rest of the paper does not help understanding what this has to do with the more prosaic questions that are waiting for an answer. What is “reorientation”? What does “concrete” versus “abstract” have to do with the question what is being achieved by doing least squares regressions in modeling? What would be an example of a mature and an immature theoretical understanding?

**Rousset (2015)** is also mentioned by the reviewer. This paper is not esoteric. It states, as reviewer #2 points out, that "neither data nor inferences are considered". This paper therefore finds itself in the modeling domain, and not in the data domain. It does however still dodge the question what the benefits are of misspecification in the modeling domain. As a matter of fact, it denies that there is misspecification at all.

“In the presence of synergies, the residuals have zero mean and are uncorrelated to the predictors. No further assumption is made about the distribution of the residuals. Thus, there is no sense in which the regression is misspecified.”

This is a remarkable quote, and testament to the lasting impact of the construction errors in Price (1970). Misspecification is literally defined as getting the model wrong. In statistics, avoiding misspecification can be complicated, because of the noise in the data. The real datagenerating process is unknown, and because of the noise, there is always the possibility that data that are generated by one model look like they could also have been generated by another. The challenge is to reduce the odds of getting the model wrong to acceptable proportions, which is what statistical tests are for. But in modeling, we know what the model is; it is postulated by the modeler. Therefore, misspecification can be avoided by just not replacing it with a different model.

What is being discussed in this part of Rousset (2015) is replacing what in this manuscript is called Model 3 (\begin{document}$w_i = \alpha + \beta_{1,0} p_i + \beta_{0,1}q_i + \beta_{1,1}p_i q_i + \varepsilon_i$\end{document}) with Model 2 (\begin{document}$w_i = \alpha + \beta_{1,0}p_i + \beta_{0,1}q_i + \varepsilon_i$\end{document}), and choosing the parameters in Model 2 so that it is as close as it can be to Model 3. This is just the definition of misspecification. That is to say: the misspecification part is the choosing of Model 2 as a reference model. The minimizing of the sum of squared residuals one could consider as minimizing the damage.

While Rousset (2015) finds itself in the modeling domain, it does nonetheless point to the field of statistics here, by stating that “the residuals have zero mean and are uncorrelated to the predictors”. From this, the paper concludes that “there is no sense in which the regression is misspecified”. That is just plain wrong. Minimizing the sum of the squared residuals guarantees that the residuals are uncorrelated with the variables that are included in the reference model, with respect to which the squared sum of residuals is minimized. The criterion that Rousset (2015) uses is that the model is well-specified if there is no correlation between the residuals (here: \begin{document}$\hat{w}_{i}-w_{i}$\end{document}) and the variables included in the reference model (here: \begin{document}$p_i$\end{document} and \begin{document}$q_i$\end{document}). But according to this criterion, all models would always be well-specified, and no model could ever be misspecified. The correct criterion, however, also requires that the residuals are not correlated with **variables not included in the reference model**. And here, the residuals are in fact correlated with \begin{document}$p_i q_i$\end{document}, which is the variable that is included in Model 3, but not in Model 2. Therefore, according to the correct version of this criterion, this model is in fact misspecified – as it should be, because getting the model wrong is the definition of misspecification.

In order to make sure that there can be no misunderstanding, I have added subsections at the end of Section 2 and Section 4 of Appendix 1, and at the end of Section 2 of Appendix 2. These subsections show that the algebra of minimizing the sum of squared errors implies that there is no correlation between the errors, or the residuals, and the variables that are included in the model. This is by no means something new; it is the reason why we do OLS to begin with. For additional details about misspecification, I would refer to Section 1b (viii) in van Veelen (2020).

Finally, there is a detail worth noticing. In the main text, as well as in Appendix B, I use an analogy (and, unlike what Gardner, West, and Wild, 2011, refer to as an analogy, this actually is one). This is an analogy between two choices. On the one hand, there is the choice between Price-like equation 1 (based on Model 1 as a reference model) and Price-like equation 2 (based on Model 2 as a reference model) both applied to Model 2. On the other hand, there is the choice between Price-like equation 2 (based on Model 2 as a reference model) and Price-like equation 3 (based on Model 3 as a reference model) both applied to Model 3. Model 1 is the non-social model, Model 2 is the social model without interaction term, and Model 3 is the social model with interaction term. That makes the first choice a choice between treating a social model as a social model, or as a non-social model. The second choice is between treating a social model with interaction term as a social model with interaction term, or as a social model without interaction term. The power of this analogy is that every argument against treating the social model as if it is a non-social model is also an argument against treating the social model with interaction term as if it is a social model without interaction term.

This ties in with the incorrect criterion for when a model is well-specified from Rousset (2015) as follows. His criterion (that there should be no correlation between the residuals and the variables in the model) declares the social model without interaction term well-specified as a reference model, when we are considering a social model with interaction term. According to the same criterion, however, the non-social model would also have to be declared to be wellspecified as a reference model, when the model we are considering is a social model. The reason is that also here, there is no correlation between the residuals and the variables that are included in this model. This is clearly not what anyone is advocating for, and for good reasons. The residuals here would, after all, be correlated with the p-score of the partner, which is a variable that is not included in the non-social model. This is a good indication that we should not use the non-social model for a social trait.

**Reviewer #3 (Public review):**

Before responding to this review, I would like to express that I appreciate the fact that the reviews and the responses are public at eLife. Besides just being useful in general, this also allows readers to get a behind the scenes glimpse into the state of the field, and the level of the reviewing. While the reports by Reviewers #1 and #2 show openness and an interest in getting things right, the report by Reviewer #3 is representative of the many review reports that I have received from the inclusive fitness community in the past. These reports tend to be rhetorically strong, and to those who do not have the time to dig deeper in the details, these reports are probably also convincing. I will therefore go through this review line by line to show how little there is behind the confident off-hand dismissal.

There is an interesting mathematical connection - an "isomorphism"-between Price's equation and least-squares linear regression.

This is esoteric and needlessly vague. Why is the word “isomorphism” used? In mathematics, an isomorphism is a structure-preserving mapping. The Price equation is an equation, or an identity, which makes it a bit difficult to imagine what the set of objects is on one end of the mapping. Least-squares linear regression can perhaps be seen as a function of a dataset, which would make it a single object (one function). This complicates things at the other end of the mapping too, if that set is a singleton set. The only isomorphism that I can think of is a trivial isomorphism where one equation is mapped onto one function and vice versa. It seems unlikely that this is what the reviewer means. The word isomorphism moreover is in quotes, so maybe this is supposed to be figurative. But what would it be that is being suggested here by this figure of speech? Just saying that there is, as the reviewer puts it, an “interesting mathematical connection”, does not make it so. It would already be a start to just specify what the mathematical connection is, because I have a hard time seeing what that would be. Is it just that, if you divide the Cov(𝑤, 𝑝)-term by the Var(𝑝)-term, then you get a regression coefficient? If that is what the reviewer has in mind, that would be a rather shallow observation.

Some people have misinterpreted this connection as meaning that there is a generalitylimiting assumption of linearity within Price's equation, and hence that Hamilton's rule-which is derived from Price's equation-provides only an approximation of the action of natural selection.

Here, the reviewer pulls a switcheroo. The use of the word “general”, or “generality”, here refers to the fact that the classical Price equation is an identity for all possible transitions between a parent and an offspring population. This is the sense in which the inclusive fitness literature uses the word general, and so do I in the relevant places in the manuscript. When I do, I make sure to add phrases like “in the sense that whatever the true model is, it always gets the direction of selection right”. As a consequence, the classical Hamilton’s rule is also totally general, in the same sense.

One of the core points of the paper is that this is not unique to the classical Price equation. As a matter of fact, there is a large set of Price-like equations and Hamilton-like rules that are equally much identities, and equally much general (in the sense that they get the direction of selection right for all possible transitions). The being an identity and being completely general (in this sense) therefore cannot be a decisive criterion in favour of the classical Price equation and the classical Hamilton’s rule.

On the other hand, the way in which my Generalized Price equation and my generalized version of Hamilton’s rule are general, is that they do not restrict the statistical model with respect to which errors are squared, summed and minimized to one linear statistical model. This generalization generates the variety of Price-like equations and Hamilton-like rules mentioned above (all of which are general in the sense of always getting the direction of selection right) and it gives us the flexibility to pick one that separates terms that reflect the fitness function from terms that reflect the population state.

In response to my generalizing the Price equation and Hamilton’s rule in this second sense, the criticism of the reviewer comes down to saying that the Price equation and Hamilton’s rule do not need generalizing, because they already are general – the switcheroo being that this refers to generality in the first sense. That makes it sound like this could be an honest mistake, confusing one way in which these can be described as general with another. However, I really hammered this point home in the manuscript. Even a cursory reading of the manuscript reveals that I am fully aware that the classical Price equation and the classical Hamilton’s rule are general in the first sense.

It is also not helpful that, as a description of what I supposedly claim, this is impressionistic, and lacks specificity. The Price equation is an equation, or an identity. What does it mean for there to be an “assumption of linearity” within it? For the classical Price equation in covariance form (which Reviewer #2 argues is what most people think of as “the Price equation”) there is no way in which one can transform this into a meaningful statement. There is just nothing in there to which the adjective “linear” can be applied. Linearity only becomes a thing when we ask ourselves how we can interpret the regression coefficient in the classical Price equation in regression form. That would be the linearity of the statistical model the differences with which are squared, summed and minimized in the regression.

This is in contrast to the majority view that Hamilton's rule is a fully general and exact result.

Again, in this manuscript, I write, time and again, that the classical Hamilton’s rule is fully general (in the sense that it is applies to any transition), and exact (if that means that it always gets the direction of selection right). So, this is clearly not where the contrast with the majority view lies. The contrast with the majority view is that the majority insist on misspecification, and I suggest not to do that.

To briefly give some mathematical details: Price's equation defines the action of natural selection in relation to a trait of interest as the covariance between fitness 𝑤 and the genetic breeding value 𝑔 for the trait, i.e. \begin{document}$\mathrm{Cov}(w,g)$\end{document};

The Price equation is an identity, not a definition. When deciding on a definition, there is some freedom. We can choose to define \begin{document}$\subset $\end{document} so that \begin{document}$A \subset B$\end{document} means that \begin{document}$A$\end{document} is a strict subset of \begin{document}$B$\end{document}; or we can choose to define \begin{document}$\subset $\end{document} so that \begin{document}$A \subset B$\end{document} means that \begin{document}$A$\end{document} is a (not necessarily strict) subset of \begin{document}$B$\end{document}. The Price equation does not “define the action of natural selection”, because it is an identity. There is no freedom to “define” \begin{document}$\bar{w} \Delta \bar{p}$\end{document} any other way.

The more serious reason why this is conceptually also a little dangerous, is the following. Imagine a locus with two alleles. Both of them are non-coding bits of DNA. Selection therefore does not act on either of them. Now imagine a parent population with an average p-score of 0.5, or, in other words, the frequency of these alleles in the parent population is 50-50. That makes the expected value of the p-score in the offspring population 0.5 too. In finite populations, however, randomness can make the p-score grow a bit larger or a bit smaller than 0.5. If the parent population is small, the variance (the expected squared deviation from 0.5) can actually be sizeable. If the p-score in the offspring population lands above 0.5, then the Price equation has a \begin{document}$\Delta \bar{p} > 0$\end{document} and a \begin{document}$\mathrm{Cov}(w,p) > 0$\end{document}. Describing the Price equation as “defining the action of natural selection” now suggests that higher p-scores have been selected for (or, in other words, that “the action of natural selection in relation to a trait of interest” is positive). With equal probability, however, \begin{document}$\Delta \bar{p} \lt0$\end{document} and therefore also \begin{document}$\mathrm{Cov}(w,p) < 0$\end{document}, and this would then make us draw the opposite conclusion, that natural selection has acted to lower the p-scores in the population. Both of those would be wrong, because in this situation, it would have been randomness that changed the average p-score.

this is a fully general result that applies exactly to any arbitrary set of \begin{document}$(g,w)$\end{document} data; without any loss of generality this covariance can be expressed as the product of genetic variance \begin{document}$\mathrm{Var}(p)$\end{document} and a coefficient \begin{document}$b(g,w)$\end{document}, the coefficient simply being defined as \begin{document}$b(g,w) = \left[\frac{\mathrm{Cov}(w, p)}{\mathrm{Var}(p)}\right]$\end{document} for all \begin{document}$\mathrm{Var}(p) > 0$\end{document}; it happens that if one fits a straight line to the same \begin{document}$(g,w)$\end{document} data by means of least-squares regression then the slope of that line is equal to \begin{document}$b(g,w)$\end{document}.

Why this needs to be explained is a bit of a mystery. These “mathematical details” are in almost all Price equation papers, and they are the point of departure of my Appendix 1 (it is on page 7 of a more than 90 page long set of appendices). Seeing the need to explain this suggests that the reviewer thinks that there is a chance that I or anyone reading this paper would have missed this. I have not, and, more importantly, none of this invalidates the point I make in the paper.

All of this has already been discussed, repeatedly, in the literature.

All of this has already been discussed, repeatedly, in the literature indeed. It is just that it does not engage with anything I write in the manuscript, or that I wrote in my other papers.

Now turn to the present paper: the first sentence of the Abstract says "The generality of Hamilton's rule is much debated", and then the next sentence says "In this paper, I show that this debate can be resolved by constructing a general version of Hamilton's rule".

This is correct.

But immediately it's clear that this isn't really resolving the debate, what this paper is actually doing is asserting the correctness of the minority view (i.e. that Hamilton's rule as it currently stands is not a general result)

It seems to me that the reason why this is “immediately clear” to this reviewer is that the reviewer has not processed the contents of the paper. I am not sure if I have to repeat this, but I am not saying that “Hamilton’s rule as it currently stands” is not general (in the sense that it always gets the direction of selection right). It is, and I say that it is a bunch of times. But so are other rules.

and then attempting to build a more general form of Hamilton's rule upon that shaky foundation.

I am not just “attempting to build a more general form of Hamilton's rule”. I did in fact build a more general form of Hamilton’s rule (where the generality refers to the richer set of reference statistical models).

Predictably, the paper erroneously interprets the standard formulation of Hamilton's rule as a linear approximation and develops non-linear extensions to improve the goodness of fit for a result that is already exactly correct.

Nowhere in the paper or the appendices do I describe the standard formulation of Hamilton’s rule (or, for that matter, any formulation of Hamilton’s rule) as an “approximation”. It is just not a word that has anything to do with this. If we are doing statistical inference, and the sum of squared errors that is minimized decreases by adding a variable in the statistical model with regard to which the sum of squared errors is minimized, then that will typically improve the goodness of fit. In statistics this is not described that as an improvement in how well the statistical model “approximates” the data, or whatever it is that the reviewer would suggest is being approximated here.

This is not a convincing contribution. It will not change minds or improve understanding of the topic.

There is indeed plenty of scope for this not to change minds or improve understanding of the topic. It will not change the minds or improve the understanding of those that are not really interested in getting this right. Obviously, it will also not convince those that do not read it.

Nor is it particularly novel. Smith et al (2010, "A generalisation of Hamilton's rule for the evolution of microbial cooperation" Science 328, 1700-1703) similarly interpreted Hamilton's rule as a linear model and provided a corresponding polynomial expansion - usefully fitting the model to microbial data so as to learn something about the costs and benefits of cooperation in an empirical setting. it's odd that this paper isn't cited here.

Let me begin by pointing to what I agree with. Given that Smith et al. (2010) and my manuscript are both in the business of generalizing Hamilton’s rule, it would be helpful to the reader if my paper includes more information about how the two efforts relate. I will discuss the relation below, and I will also include that in Appendix B, and point to it in the main text. Before I do, however, I would like to point to two details in the review report that fit a pattern.

The first is that the reviewer describes what Smith et al. (2010) do as “useful”, and seems to think of fitting polynomial expansions as a legitimate way to “learn something about the costs and benefits of cooperation in an empirical setting”. That sounds quite positive. My paper, in which I supposedly repeat this, however, is characterized as misguided. This fits a pattern; all of the reviews I received from the inclusive fitness community include a “done before”, and regularly the done before is described approvingly, while my paper is described as fundamentally flawed.

Also customary is the lack of detail. What would be really useful here, is something like “equation A.14 in this manuscript is the same as equation 6 in Smith et al. (2010) if we choose \begin{document}$r=\frac{1}{N}$\end{document}. This kind of statement would pin down the way in which what I do has been done before. That, however, would require going into detail, at the risk of finding out that what is done in my manuscript is actually quite different from what happens in Smith et al. (2010). That is also a recurrent thing. When I look up the done before, I typically find something that is not quite the same.

Now on to the paper. What Smith et al. (2010) try to do is something that I wholeheartedly support. It is an empirical study that tries to capture non-linearity. A first point of order is that it is worth asking ourselves: linear or non-linear in what? For that, I would like to go back to the setup of my manuscript. Model 2 from the Main Text is\begin{document}$$\displaystyle w_{i}=\alpha+\beta_{1,0} p_{i}+\beta_{0,1} q_{i}+\varepsilon_{i}$$\end{document}

In this fitness function, \begin{document}$p_i$\end{document} is the p-score of individual \begin{document}$i$\end{document} and \begin{document}$q_i$\end{document} is the p-score of the partner that individual \begin{document}$i$\end{document} is matched with. This is a standard model of social behaviour if \begin{document}$\beta_{1,0} < 0$\end{document} and \begin{document}$\beta_{0,1} > 0$\end{document}. Such choices for \begin{document}$\beta_{1,0}$\end{document} and \begin{document}$\beta_{0,1}$\end{document} indicate that having a higher p-score decreases the fitness of individual 𝑖 and increases the fitness of its partner. Here we assume that \begin{document}$\alpha = 1$\end{document}, \begin{document}$\beta_{1,0} = -1$\end{document}, and \begin{document}$\beta_{0,1} = 2$\end{document}. We assume that p-scores can only be 0 or 1, or, in other words, we assume that there are only cooperators and defectors in the population (or, in terms of Smith et al., 2010: cooperators and cheaters).

For a well-mixed population, where the likelihood of being matched with a cooperator is the same for cooperators and defectors (it is equal to the frequency of cooperators for both), we can now plot the fitnesses of cooperators (red) and defectors (blue) as a function of the frequency of cooperators (Appendix 1-figure 6 left).

We can do the same for a population with relatedness \begin{document}$\frac{1}{4}$\end{document} where the probability of being matched with a cooperator is \begin{document}$\frac{1}{4} + \frac{3}{4}f_c$\end{document} for cooperators, and \begin{document}$\frac{3}{4}f_c$\end{document} for defectors, where \begin{document}$f_c$\end{document} is the frequency of cooperators (Appendix 1-figure 6 right). For relatedness \begin{document}$r=0$\end{document} and \begin{document}$r= \frac{1}{4}$\end{document}, cooperation is selected against at every frequency.

Increasing relatedness further, we would find that for \begin{document}$r=\frac{1}{2}$\end{document} the lines coincide, which implies that at every frequency, cooperation is neither selected for nor against. For \begin{document}$r > \frac{1}{2}$\end{document} cooperation will be selected for at every frequency. This pattern implies that, as we have seen in the manuscript, the classical Hamilton’s rule works perfectly fine for Model 2; with \begin{document}$c= -\beta_{1,0} = 1$\end{document} and \begin{document}$b=\beta_{0,1}=2$\end{document}, cooperation is selected for if and only if \begin{document}$rb\gtc$\end{document}. The fitnesses of cooperators and defectors as functions of the frequency of cooperators, moreover, are always parallel lines, regardless of relatedness.

Model 3 in the main text extends Model 2 by adding an interaction term:\begin{document}$$\displaystyle w_{i}=\alpha+\beta_{1,0} p_{i}+\beta_{0,1} q_{i}+\beta_{1,1} p_{i} q_{i}+\varepsilon_{i}$$\end{document}

Now we choose \begin{document}$\alpha = 1$\end{document}, \begin{document}$\beta_{1,0} = -1$\end{document}, \begin{document}$\beta_{0,1} = 1$\end{document}, and \begin{document}$\beta_{1,1} = 1$\end{document}. We again draw the fitnesses of cooperators and defectors, both at relatedness \begin{document}$r=0$\end{document} (Appendix 1-figure 7 left) and at relatedness \begin{document}$r=\frac{1}{4}$\end{document} (Appendix 1-figure 7 right). In the manuscript, I argue that the appropriate version of Hamilton’s rule here is Queller’s rule: \begin{document}$r_{0,1} b_{0,1} + r_{1,1}b_{1,1} > c$\end{document} with \begin{document}$c = - \beta_{1,0}=1$\end{document}, \begin{document}$b_{0,1} = \beta_{0,1} = 1$\end{document}, and \begin{document}$b_{1,1} = \beta_{1,1} = 1$\end{document}. The fitnesses of cooperators and defectors as functions of the frequency of cooperators are still straight lines, but they are no longer parallel.

The first thing to observe, therefore, is that a model with synergy, in which the classic version of Hamilton’s rule would be misspecified, and Queller’s rule would be well-specified, does not require the fitnesses as functions of the frequencies of cooperators to be non-linear. All that changes with the addition of the interaction term, is that they stop being parallel.

The paper by Smith et al. (2010) is an effort to capture non-linearities in the way fitnesses depend on the frequency of cooperators. That, therefore, goes beyond the step from Model 2 to Model 3. Whether it uses the right method to capture those non-linearities, we will come back to in a second, but it is important to realize that also without these non-linearities, the classic version of Hamilton’s rule can be too limiting to accurately describe selection. (Here, I should add that this implies that we were wrong in Wu et al. [2013], when we suggested that “for this experiment, it seems unnecessary to use the generalized Hamilton’s rule, if instead the Malthusian fitness is adopted. In other words, the Wrightian fitness approach calls for a generalization of Hamilton’s rule, whereas the Malthusian fitness approach does not (or at least not in a drastic way, as Malthusian fitnesses are almost linear in the frequency of cooperators).” Using Malthusian fitnesses, the functions were close to linear, but not close to parallel, and therefore also here, Hamilton’s rule needs generalizing - albeit in a different way than Smith et al. [2010] did.)

The cooperation that is observed in the Myxococcus xanthus studied by Smith et al. (2010) is not a good match with a model where individuals are matched in pairs for an interaction that determines their fitnesses. These microbes cooperate in large groups, and a better match would therefore be the n-player public goods games studied in van Veelen (2018). There, we see that simple, straightforward ways to describe synergies (or anti-synergies) can easily lead to fitnesses not being linear in the frequency of cooperators.

The way Smith et al. (2010) try to capture those non-linearities, however, is not free of complications. We addressed those in Wu et al. (2013), and I summarized them, shortly, in van Veelen (2018). One of the issues is that most of the non-linearity Smith et al. (2010) pick up is the result of considering Wrightian fitness rather than Malthusian fitness. In a continuous time model with a constant growth rate, the population size at time 𝑡 is \begin{document}$N(t) = e^{mt} N(0)$\end{document}, where \begin{document}$m$\end{document} is the Malthusian fitness. In a discrete time model with a constant average number of offspring per individual, the population at time \begin{document}$t$\end{document} is \begin{document}$N(t) = w^t N(0)$\end{document}, where \begin{document}$w$\end{document} is the Wrightian fitness. If we take \begin{document}$m = \ln w$\end{document}, these are the same, and if \begin{document}$w$\end{document} is close to 1, then \begin{document}$m$\end{document} can be approximated by \begin{document}$w-1$\end{document}. That also implies that if \begin{document}$w$\end{document} is close to 1 (or, equivalently, if \begin{document}$m$\end{document} is close to 0) one is locally linear if the other is too. However, in the experiment by Smith et al. (2010) the aggregate fitness effects are not small, and what is highly nonlinear in terms of Wrightian fitness is close to linear in Malthusian fitness.

Another complication is that the Taylor coefficients that Smith et al. (2010) find are the result of a combination of the data and the choice of a functional form they choose to first apply to their data. That means that a different choice of a functional form would have given different Taylor coefficients, while the in-between transformation can also be skipped. Also, the number of Taylor coefficients is larger than the dimensionality of the data, which are based on averages for 6 frequencies. For more details on these complications, I would like to refer to Wu et al. (2013) and van Veelen (2018). A nice detail is that if we consider the way the fitnesses of cooperators and defectors compare when using Malthusian fitnesses, then a comparison of the slopes actually suggests anti-synergies, which leads to a stable mix of cooperators and cheaters, already in the absence of population structure. This matches what is suggested by Archetti and Scheuring, (2011, 2012) and Archetti (2018).

Besides these technical complications, Smith et al. (2010) is also different, in the sense that it is an empirical paper. It does not contain the Generalized Price equation, it contains no insights regarding how to derive population genetic dynamics from the Generalized Price equation, or how to derive the appropriate rules from those, and it has a very different approach to separating fitness effects and population structure.

To end on a positive note, I would like to quote a bit out of Wu et al. (2013):

“While we criticise these mathematical issues, we are convinced that Smith et al. (2010) aim into the right direction: to incorporate the nonlinearities characteristic of biology into social evolution, we may have to extend and generalize the approach of inclusive fitness. It would be beautiful if such a generalization would ultimately include Hamilton’s original rule as a special case […].”

I like to think that this is exactly what I have done in this paper.

**References**

Akdeniz, A., & van Veelen, M. (2020). The cancellation effect at the group level. Evolution, 74(7), 1246–1254. doi: 10.1111/evo.13995

Allen, B., & Tarnita, C. E. (2012). Measures of success in a class of evolutionary models with fixed population size and structure. Journal of Mathematical Biology, 68, 109–143. doi: 10.1007/s00285-012-0622-x

Archetti, M. (2018). How to Analyze Models of Nonlinear Public Goods. Games 2018, Vol. 9, Page 17, 9(2), 17. doi: 10.3390/g9020017

Archetti, M., & Scheuring, I. (2011). Coexistence of cooperation and defection in public goods games. Evolution, 65(4), 1140–1148. doi: 10.1111/j.1558-5646.2010.01185.x

Archetti, M., & Scheuring, I. (2012). Review: Game theory of public goods in one-shot social dilemmas without assortment. Journal of Theoretical Biology, 299, 9–20. doi: 10.1016/j.jtbi.2011.06.018

Bourke, A. F. G. (2014). Hamilton’s rule and the causes of social evolution. Philosophical Transactions of the Royal Society B: Biological Sciences, 369(1642), 20130362. doi: 10.1098/rstb.2013.0362

Bourrat, P., Godsoe, W., Pillai, P., Gouhier, T. C., Ulrich, W., Gotelli, N. J., & van Veelen, M. (2023). What is the price of using the Price equation in ecology? Oikos, 2023(8). doi: 10.1111/oik.10024

Crow, J. F. (2008). Commentary: Haldane and beanbag genetics. International Journal of Epidemiology, 37(3), 442–445. doi: 10.1093/ije/dyn048

Fisher, R. (1930). The genetical theory of natural selection. Retrieved from https://www.cabidigitallibrary.org/doi/full/10.5555/19601600934

Fletcher, J. A., & Zwick, M. (2006). Unifying the theories of inclusive fitness and reciprocal altruism. American Naturalist, 168(2), 252–262. doi: 10.1086/506529

Frank, S. A. (1997). The Price equation, Fisher’s fundamental theorem, kin selection, and causal analysis. Evolution, 51(6), 1712–1729. doi: 10.1111/j.1558-5646.1997.tb05096.x

Frank, S. A. (1998). Foundations of social evolution. Princeton: Princeton University Press.

Frank, S. A. (2012). Natural selection. IV. The Price equation*. Journal of Evolutionary Biology, 25(6), 1002–1019. doi: 10.1111/j.1420-9101.2012.02498.x

Gardner, A., West, S. A., & Wild, G. (2011). The genetical theory of kin selection. Journal of Evolutionary Biology, 24(5), 1020–1043. doi: 10.1111/j.1420-9101.2011.02236.x

Grafen, A. (1985a). A geometric view of relatedness. Oxford Surveys in Evolutionary Biology, 2(2), 28-89.

Grafen, A. (1985b). News and Views. Evolutionary theory: Hamilton’s rule OK. Nature, 318(6044), 310–311. doi: 10.1038/318310a0

Hamilton, W. D. (1964). The genetical evolution of social behaviour. I. Journal of Theoretical Biology, 7(1), 1–16. doi: 10.1016/0022-5193(64)90038-4

Karlin, S., & Matessi, C. (1983). The eleventh R. A. Fisher Memorial Lecture - Kin selection and altruism. Proceedings of the Royal Society of London. Series B. Biological Sciences, 219(1216), 327–353. doi: 10.1098/rspb.1983.0077

Matessi, C., & Karlin, S. (1984). On the evolution of altruism by kin selection. Proceedings of the National Academy of Sciences, 81(6), 1754–1758. doi: 10.1073/pnas.81.6.1754

Nowak, M. A., Tarnita, C. E., & Wilson, E. O. (2010). The evolution of eusociality. Nature, 466(7310), 1057–1062. doi: 10.1038/nature09205

Okasha, S. (2005). Maynard Smith on the levels of selection question. Biology and Philosophy, 20(5), 989–1010. doi: 10.1007/S10539-005-9019-1

Page, K. M., & Nowak, M. A. (2002). Unifying evolutionary dynamics. Journal of Theoretical Biology, 219(1). doi: 10.1016/S0022-5193(02)93112-7

Pillai, P., & Gouhier, T. C. (2019). Not even wrong: the spurious measurement of biodiversity’s effects on ecosystem functioning. Ecology, 100(7), e02645. doi: 10.1002/ecy.2645

Price, G. R. (1970). Selection and Covariance. Nature, 227(5257), 520–521. doi: 10.1038/227520a0

Price, G. R. (1972). Extension of covariance selection mathematics. Annals of Human Genetics, 35(4), 485-490.

Queller, D. C. (1985). Kinship, reciprocity and synergism in the evolution of social behaviour. Nature, 318(6044), 366–367. doi: 10.1038/318366a0

Queller, D. C. (1992a). A general model for kin selection. Evolution, 46(2), 376–380. doi: 10.1111/j.1558-5646.1992.tb02045.x

Queller, D. C. (1992b). Quantitative Genetics, Inclusive Fitness, and Group Selection. The American Naturalist, 139(3), 540–558. doi: 10.1086/285343

Queller, D. C. (2011). Expanded social fitness and Hamilton’s rule for kin, kith, and kind. Proceedings of the National Academy of Sciences, 108(supplement_2), 10792–10799. doi: 10.1073/pnas.1100298108

Rousset, & Billiard. (2000). A theoretical basis for measures of kin selection in subdivided populations: Finite populations and localized dispersal. Journal of Evolutionary Biology, 13(5). doi: 10.1046/j.1420-9101.2000.00219.x

Rousset, F. (2015). Regression, least squares, and the general version of inclusive fitness. Evolution, 69(11), 2963–2970. doi: 10.1111/evo.12791

Smith, J., Van Dyken, J. D., & Zee, P. C. (2010). A generalization of hamilton’s rule for the evolution of microbial cooperation. Science, 328(5986), 1700–1703. doi: 10.1126/science.1189675

Sober, Elliott., & Wilson, D. Sloan. (2007). Unto others : the evolution and psychology of unselfish behavior. 394. Retrieved from https://www.hup.harvard.edu/books/9780674930476

Taylor, P. D. (1992). Altruism in viscous populations - an inclusive fitness model. Evolutionary Ecology, 6(4), 352–356. doi: 10.1007/bf02270971

Taylor, Peter D. (1989). Evolutionary stability in one-parameter models under weak selection. Theoretical Population Biology, 36(2), 125–143. doi: 10.1016/00405809(89)90025-7

Taylor, Peter D., Day, T., & Wild, G. (2007). Evolution of cooperation in a finite homogeneous graph. Nature, 447(7143), 469–472. doi: 10.1038/nature05784

Van Cleve, J. (2015). Social evolution and genetic interactions in the short and long term. Theoretical Population Biology, 103. doi: 10.1016/j.tpb.2015.05.002

van Veelen, M. (2005). On the use of the Price equation. Journal of Theoretical Biology, 237(4). doi: 10.1016/j.jtbi.2005.04.026

van Veelen, M. (2007). Hamilton’s missing link. Journal of Theoretical Biology, 246(3). doi: 10.1016/j.jtbi.2007.01.001

van Veelen, M. (2011). The replicator dynamics with n players and population structure. Journal of Theoretical Biology, 276(1). doi: 10.1016/j.jtbi.2011.01.044

van Veelen, M. (2018). Can Hamilton’s rule be violated? ELife, 7. doi: 10.7554/eLife.41901

van Veelen, M. (2020). The problem with the Price equation. Philosophical Transactions of the Royal Society B: Biological Sciences, 375(1797), 20190355. doi: 10.1098/rstb.2019.0355

van Veelen, M., Allen, B., Hoffman, M., Simon, B., & Veller, C. (2017). Hamilton’s rule. Journal of Theoretical Biology, 414. doi: 10.1016/j.jtbi.2016.08.019

van Veelen, M., García, J., Sabelis, M. W., & Egas, M. (2012). Group selection and inclusive fitness are not equivalent; the Price equation vs. models and statistics. Journal of Theoretical Biology, 299. doi: 10.1016/j.jtbi.2011.07.025

Wilson, D. S., Pollock, G. B., & Dugatkin, L. A. (1992). Can altruism evolve in purely viscous populations? Evolutionary Ecology, 6(4), 331–341. doi: 10.1007/bf02270969

Wu, B., Gokhale, C. S., van Veelen, M., Wang, L., & Traulsen, A. (2013). Interpretations arising from Wrightian and Malthusian fitness under strong frequency dependent selection. Ecology and Evolution, 3(5). doi: 10.1002/ece3.500